# Efficiently Escaping Saddle Points under Generalized Smoothness via Self-Bounding Regularity

**Daniel Yiming Cao**[*]     **August Y. Chen**[*]     **Karthik Sridharan**[*]     **Benjamin Tang**[*]
Department of Computer Science, Cornell University
{dyc33, ayc74, ks999, bt283}@cornell.edu

## Abstract

We study the optimization of non-convex functions that are not necessarily smooth (gradient and/or Hessian are Lipschitz) using first order methods. Smoothness is a restrictive assumption in machine learning in both theory and practice, motivating significant recent work on finding first order stationary points of functions satisfying generalizations of smoothness with first order methods. We develop a novel framework that lets us systematically study the convergence of a large class of first-order optimization algorithms (which we call decrease procedures) under generalizations of smoothness. We instantiate our framework to analyze the convergence of first order optimization algorithms to first and *second* order stationary points under generalizations of smoothness. As a consequence, we establish the first convergence guarantees for first order methods to second order stationary points under generalizations of smoothness. We demonstrate that several canonical examples fall under our framework, and highlight practical implications.

## 1 Introduction

A widely studied problem in machine learning (ML) and optimization is finding a First Order Stationary Point (FOSP) of a generic function $F$ with domain $\mathbb{R}^d$, defined as follows:

$$\text{Given a tolerance } \varepsilon > 0, \text{ find } \boldsymbol{w} \text{ such that } \|\nabla F(\boldsymbol{w})\| \le \varepsilon. \tag{1}$$

The methods of choice in theory and practice for this task are Gradient Descent (GD), Stochastic Gradient Descent (SGD), and variants thereof. Under the additional assumption of (second-order) *smoothness* on $F$, i.e. that the gradient $\nabla F$ is Lipschitz with parameter $L > 0$, this task is well-understood. In several settings – such as with access to exact gradients, stochastic gradients, Hessian-Vector Products, and the exact Hessian – we have matching upper and lower bounds. The literature on this problem is extensive; for a subset see e.g. Ghadimi and Lan (2013); Johnson and Zhang (2013); Fang et al. (2018, 2019); Foster et al. (2019); Arjevani et al. (2020); Carmon et al. (2020, 2021).

However, for many non-convex functions $F$, FOSPs are uninformative. A significant and difficult problem established in the literature for over a decade – which carries strong theoretical and practical implications in optimization for machine learning – is establishing efficient rates for finding a Second Order Stationary Point (SOSP). In many non-convex optimization problems such as Phase Retrieval and Matrix Square Root (Ge et al., 2015; Jin et al., 2017; Ge et al., 2017; Sun et al., 2018), SOSPs are global minima. Finding a SOSP is defined as follows:

$$\text{Given a tolerance } \varepsilon > 0, \text{ find } \boldsymbol{w} \text{ such that } \|\nabla F(\boldsymbol{w})\| \le \varepsilon, \nabla^2 F(\boldsymbol{w}) \succeq -\sqrt{\varepsilon}\boldsymbol{I}, \tag{2}$$

where $\succeq$ denotes the PSD order, $\boldsymbol{I}$ is the $d \times d$ identity matrix, and $\nabla^2 F(\boldsymbol{w})$ is the Hessian of $F$.[2]

Under the additional *Hessian Lipschitz* assumption, that the operator norm of the Hessian $\nabla^2 F$ in addition to the gradient $\nabla F$ is Lipschitz, this task is also well-understood. Under these regularity

---

[*]Authors are listed in alphabetical order.

[2]There are several definitions of a SOSP; see Remark 5 for why we use this definition here.

assumptions, finding SOSPs is classical under exact oracle access to the full Hessian $\nabla^2 F$. Decades ago, it was shown that cubic regularization and trust region methods succeed (Nesterov and Polyak, 2006; Conn et al., 2000), with a matching lower bound in Arjevani et al. (2020). Motivated by the success of non-convex optimization in ML via first order methods, solving this problem (2) with first order methods has seen much recent study (Ge et al., 2015; Jin et al., 2017; Fang et al., 2019; Arjevani et al., 2020; Jin et al., 2021a). We have matching upper and lower bounds in several cases, such as for SGD which is perhaps most relevant to ML (Fang et al., 2019; Arjevani et al., 2020).

However, in many optimization problems in ML, the gradient and Hessian of the loss function is not Lipschitz. This was observed empirically through extensive experiments of Zhang et al. (2019) on LSTMs and of Crawshaw et al. (2022) on transformers. We provide theoretical examples in Subsection 3.6. As such, a line of work began in Zhang et al. (2019) on studying finding FOSPs under weaker regularity assumptions, see e.g. (Zhang et al., 2020; Jin et al., 2021b; Crawshaw et al., 2022; Reisizadeh et al., 2023; Li et al., 2023b; Wang et al., 2024; Hong and Lin, 2024; Gaash et al., 2025; Yu et al., 2025). The regularity assumption generally made is $(L_0, L_1)$-smoothness: $\left\| \nabla^2 F(\boldsymbol{w}) \right\|_{\mathrm{op}} \leq L_0 + L_1 \|\nabla F(\boldsymbol{w})\|$ for all $\boldsymbol{w} \in \mathbb{R}^d$ for some $L_0, L_1 \geq 0$. This allows for arbitrarily polynomial growth rates of $F$ in $\|\boldsymbol{w}\|$. The guarantees in Zhang et al. (2019) and follow-up works generally hold for adaptive methods, presented as theoretical justification for gradient clipping.

The authors of Li et al. (2023a), under a milder regularity assumption than Zhang et al. (2019), studied finding FOSPs via *fixed-step-size* GD and SGD rather than adaptive methods. In particular, *Li et al. (2023a) demonstrated clipping is not necessary for $(L_0, L_1)$-smooth functions.* Related works extended this analysis to Nesterov's Accelerated Gradient Descent (Li et al., 2023b; Hong and Lin, 2024). Xie et al. (2024) studied finding SOSPs under $(L_0, L_1)$-smoothness and a similar assumption that for all $\boldsymbol{w}$, in a small neighborhood of $\boldsymbol{w}$, the Hessian of $F$ is Lipschitz with parameter $M_0 + M_1 \|\nabla F(\boldsymbol{w})\|$. However, their algorithm is *second-order* and requires the *full* Hessian, analogous to classical work (Nesterov and Polyak, 2006; Conn et al., 2000). This contrasts with recent developments of finding SOSPs using first order methods when $F$ has Lipschitz gradient and Hessian, which are more pertinent to ML where first-order algorithms are the only tractable method (Ge et al., 2015; Jin et al., 2017; Fang et al., 2019; Arjevani et al., 2020; Jin et al., 2021a).

## 1.1 Our Contributions

In this work, we develop a novel framework to study **non-asymptotic** guarantees finding FOSPs and SOSPs via first-order methods, for functions whose gradient and/or Hessian are not Lipschitz. Central to our work is the following regularity assumption:

**Assumption 1.1** (Second-Order Self-Bounding Regularity). *$F$ is twice differentiable, and there exists a non-decreasing function $\rho_1 : \mathbb{R}_{\geq 0} \mapsto \mathbb{R}_{\geq 0}$ such that $\left\| \nabla^2 F(\boldsymbol{w}) \right\|_{\mathrm{op}} \leq \rho_1(F(\boldsymbol{w}))$ for all $\boldsymbol{w} \in \mathbb{R}^d$.*

This assumption implies the relevant Hessian operator norm is upper bounded by a function of the function value. It was also made in De Sa et al. (2022) for the different task of studying global convergence of GD/SGD, where it was shown that Assumption 1.1 holds for many canonical non-convex optimization problems. Some quantitative control of the Hessian is necessary for non-asymptotic guarantees of finding FOSPs (Kornowski et al., 2024). In Example 1, we show these prior assumptions are not satisfied by a natural univariate function. We show in Proposition A.1 that Assumption 1.1 **generalizes** $(L_0, L_1)$-smoothness and its extension from Li et al. (2023a), and that

$$(L_0, L_1)\text{-smoothness } (\left\| \nabla^2 F \right\|_{\mathrm{op}} \leq L_0 + L_1 \|\nabla F\|) \implies \text{ Assumption 1.1 with } \rho_1(x) = \tfrac{3}{2}L_0 + 4L_1^2 x.$$

For finding SOSPs, we impose the following additional regularity assumption:

**Assumption 1.2** (Third-Order Self-Bounding Regularity). *$F$ satisfies Assumption 1.1, and either:*

- *$F$ is three-times differentiable everywhere, and for some non-decreasing function $\rho_2 : \mathbb{R}_{\geq 0} \to \mathbb{R}_{\geq 0}$, $\left\| \nabla^3 F(\boldsymbol{w}) \right\|_{\mathrm{op}} \leq \rho_2(F(\boldsymbol{w}))$ for all $\boldsymbol{w} \in \mathbb{R}^d$.*

- *Or for some constant $\delta > 0$ and some non-decreasing function $\rho_2 : \mathbb{R}_{\geq 0} \to \mathbb{R}_{\geq 0}$, for all $\boldsymbol{w}, \boldsymbol{w}' \in \mathbb{R}^d$ with $\|\boldsymbol{w} - \boldsymbol{w}'\| \leq \delta$, we have $\left\| \nabla^2 F(\boldsymbol{w}) - \nabla^2 F(\boldsymbol{w}') \right\|_{\mathrm{op}} \leq \rho_2(F(\boldsymbol{w}))\|\boldsymbol{w} - \boldsymbol{w}'\|$.*

Assumption 1.2 naturally extends Assumption 1.1, and generalizes the Hessian Lipschitz assumption ubiquitous in the literature on *non-asymptotic* rates for finding SOSPs. (We note that the works Lee et al. (2016, 2019) established *asymptotic* guarantees for GD finding SOSPs without the Hessian

Lipschitz assumption, and note their proof strategy uses Lipschitzness of the gradient in a crucial way.) In Subsection 3.6, we show several canonical non-convex losses with non-Lipschitz gradient and Hessian satisfy Assumption 1.2. Assumption 1.2 covers several growth rates of interest (e.g. univariate self-concordant functions satisfying Assumption 1.1). It also subsumes that of Xie et al. (2024), which to our knowledge is the only other result on finding SOSPs under generalized smoothness (but uses the full Hessian). Under the assumptions of Xie et al. (2024), an explicit, simple form for $\rho_2(\cdot)$ can be found. We detail all of this in Example 2.

Furthermore, Assumption 1.2 encompasses several examples of Distributionally Robust Optimization (DRO) problems. Xie et al. (2024) very interestingly demonstrates that under mild assumptions, the objective of DRO satisfies their Assumption 3, see Theorem 3 therein. Assumption 3 of Xie et al. (2024) is subsumed by Assumption 1.2 as per our Example 2. Thus our results apply to DRO. DRO is a general optimization problem that has significant applications in fairness in machine learning and in learning under distribution shifts; see Xie et al. (2024) for more discussion.

We now introduce the following standard definition, which, when combined with Assumption 1.1 and Assumption 1.2, forms the core of our argument, as we explain in Subsection 2.1.

**Definition 1.1.** *For a function $F$ and threshold $\alpha$, the $\alpha$-sublevel set of $F$ is $\mathcal{L}_{F,\alpha} = \{\boldsymbol{w} : F(\boldsymbol{w}) \le \alpha\}$.*

Now, our contributions are as follows:

1. **We develop a novel, systematic framework** detailed in Section 2 and Theorem 2.1 to study the convergence of first order methods to FOSPs and SOSPs under Assumption 1.1 and Assumption 1.2 respectively. The core idea is in Subsection 2.1. **Our framework lets us systematically analyze existing practical, and widely used first-order optimization algorithms in the challenging generalized smooth setting.**

2. **Main Results, non-asymptotic convergence to SOSPs:** Under Assumption 1.2, we establish efficient rates for first-order optimization algorithms finding SOSPs. See Theorem 3.4 for Perturbed GD (Jin et al., 2017) and Theorem 3.5 for Restarted SGD (Fang et al., 2019). The dependence on $\varepsilon, d$ matches that in the smooth setting, and in particular is polylogarithmic in $d$. This is particularly pertinent for ML applications, where the ambient dimension is so large that the second-order methods of Xie et al. (2024) are not feasible.

3. Non-asymptotic convergence to FOSPs: Under Assumption 1.1, we establish efficient rates for GD, Adaptive GD, and SGD finding FOSPs. See Theorem 3.1, Theorem 3.2, and Theorem 3.3 respectively. The dependence on $\varepsilon, d$ again matches that in the smooth setting.

4. We provide examples and practical implications in Subsection 3.6. Our examples are direct corollaries of Theorem 3.4, Theorem 3.5. They show variants of GD/SGD globally optimize non-convex 'strict-saddle' losses from ML with non-Lipschitz gradient and Hessian.

**Notation:** $\mathbb{B}(\boldsymbol{p}, R)$ denotes the Euclidean $l_2$ ball centered at $\boldsymbol{p} \in \mathbb{R}^d$ with radius $R \ge 0$, with boundary. By shifting, we assume WLOG that $F$ attains a minimum value of $0$. We follow the convention that $F$ is smooth, specifically $L$-smooth, if $\left\|\nabla^2 F\right\| \le L$ holds globally. We always let $\boldsymbol{w}_0$ denote the initialization of a given algorithm (which is clear from context) unless stated otherwise.

## 2 Main Idea

### 2.1 High Level Idea

One classic analysis of GD on smooth functions to converge to a FOSP goes by establishing decrease per iterate, via the so-called 'Descent Lemma' (Bubeck et al., 2015). For $L$-smooth functions, setting the step size $\eta = \frac{1}{L}$ in GD,

$$F(\boldsymbol{w}_{t+1}) \le F(\boldsymbol{w}_t) - \eta\left(1 - \frac{1}{2}L\eta\right)\|\nabla F(\boldsymbol{w}_t)\|^2 = F(\boldsymbol{w}_t) - \frac{1}{2L}\|\nabla F(\boldsymbol{w}_t)\|^2. \tag{3}$$

Such an analysis fails if $F$ is not $L$-smooth. Following the above recipe under Assumption 1.1, as such a bound $L < \infty$ need not exist, one must set $\eta = 0$ and does not obtain any convergence rate.

**Core Insight 1:** The first simple but powerful insight in our work is that many optimization algorithms such as GD decrease the function value at each iterate (with high probability) when $\eta$ is appropriately chosen as a function of the smoothness (Hessian operator norm) at the *current* iterate.

Specifically, consider iterates of GD initialized at some $\boldsymbol{w}_0$. For step size $\eta$ small enough in terms of $\|\nabla^2 F(\boldsymbol{w}_0)\|$, the next iterate $\boldsymbol{w}_1$ of GD is sufficiently 'local' (see Corollary 1). This lets us upper bound $\|\nabla^2 F\|$ along the segment $\overline{\boldsymbol{w}_0 \boldsymbol{w}_1}$ by an increasing function $L_1(F(\boldsymbol{w}_0))$ of $F(\boldsymbol{w}_0)$ (see Lemma 3.2). Thus, for appropriate $\eta$ in terms of $F(\boldsymbol{w}_0)$, we obtain $F(\boldsymbol{w}_1) \le F(\boldsymbol{w}_0)$, and so $\boldsymbol{w}_1$ lies in the $F(\boldsymbol{w}_0)$-sublevel set $\mathcal{L}_{F,F(\boldsymbol{w}_0)}$.

**Core Insight 2:** Crucially, we can 'chain together' this decrease. By Assumption 1.1, the afore-mentioned argument goes through at any $\boldsymbol{w}$ in the $F(\boldsymbol{w}_0)$-sublevel set $\mathcal{L}_{F,F(\boldsymbol{w}_0)}$ – in particular, at $\boldsymbol{w}_1$. Consequently, this *same* step size $\eta$ is small enough to ensure $F(\boldsymbol{w}_2) \le F(\boldsymbol{w}_1) \le F(\boldsymbol{w}_0)$, and so forth through all the iterates of GD. Moreover, this argument yields a convergence rate. As each iterate is in $\mathcal{L}_{F,F(\boldsymbol{w}_0)}$, if the gradient norm is at least $\varepsilon$ at each iterate, we obtain decrease of at least $\frac{\varepsilon^2}{2L_1(F(\boldsymbol{w}_0))}$ per iterate analogously to (3). Too many iterations contradicts that $F$ is lower bounded by 0 (recall Notation), so we must reach an iterate $\boldsymbol{w}_t$ which is a FOSP within $\frac{2L_1(F(\boldsymbol{w}_0))F(\boldsymbol{w}_0)}{\varepsilon^2}$ iterates.

**Generalizing the argument:** This idea is powerful enough to readily analyze SGD and variants of GD/SGD which find SOSPs. Rather than a single iterate where decrease need not hold, we consider a sequence of consecutive $t_{\text{thres}}$ iterates. We show with high probability, the last iterate in this sequence decreases function value for $\boldsymbol{w} \in \mathcal{L}_{F,F(\boldsymbol{w}_0)}$. To do so, recall the analyses of first-order optimization algorithms often establish decrease by considering 'local' behavior. Locally around $\boldsymbol{w} \in \mathcal{L}_{F,F(\boldsymbol{w}_0)}$, Assumption 1.1 and Assumption 1.2 give enough control over the relevant derivatives to do so.

Then the above argument still goes through, with a fixed step size defined in terms of $F(\boldsymbol{w}_0)$. We group the iterates of the algorithm into 'blocks' of length $t_{\text{thres}}$, and establish $F(\boldsymbol{w}_{t_{\text{thres}}}) \le F(\boldsymbol{w}_0)$ and so forth (rather than establishing $F(\boldsymbol{w}_2) \le F(\boldsymbol{w}_1) \le F(\boldsymbol{w}_0)$ for consecutive iterates).

### 2.2   The Formal Framework

Consider a set of interest $\mathcal{S}$, e.g. FOSPs or SOSPs with tolerance $\varepsilon$. We begin by presenting a simpler version of our formal framework. Consider a deterministic update procedure $\mathcal{A} : \mathbb{R}^d \to \mathbb{R}^d$, where the output of $\mathcal{A}$ denotes the future iterate of the algorithm. For example, $\mathcal{A}(\boldsymbol{w}) = \boldsymbol{w} - \eta \nabla F(\boldsymbol{w})$ for GD. Following Subsection 2.1, we consider algorithms that decrease function value in the $F(\boldsymbol{w}_0)$-sublevel set $\mathcal{L}_{F,F(\boldsymbol{w}_0)}$ if they have not reached $\mathcal{S}$. The following definition formalizes this property:

**Definition 2.1** (Special case of Decrease Procedure in Definition 2.2). *Consider a set of interest $\mathcal{S}$, a decrease threshold $\Delta > 0$, a point $\boldsymbol{u}_0$, and a deterministic procedure $\mathcal{A}$ to compute the next iteration. We say $\mathcal{A}$ forms a $(\mathcal{S}, t_{\text{oracle}}(\boldsymbol{u}_0), \Delta(\boldsymbol{u}_0), \boldsymbol{u}_0)$-decrease procedure if computing $\mathcal{A}(\boldsymbol{u}_0)$ takes at most $t_{\text{oracle}}(\boldsymbol{u}_0)$ oracle calls, and one of the following holds:*

$$1)\ F(\mathcal{A}(\boldsymbol{u}_0)) < F(\boldsymbol{u}_0) - \Delta(\boldsymbol{u}_0), \qquad or \qquad 2)\ \mathcal{A}(\boldsymbol{u}_0) \cap \mathcal{S} \ne \{\}.$$

Here 1) means that the subsequent iterate has smaller function value, and 2) means that the rule of output $\mathcal{A}_2$ outputs a sequence of candidate vectors, one of which is in $\mathcal{S}$.

Then, Theorem 2.1 states that if $\mathcal{A}$ is a decrease procedure for all $\boldsymbol{u}_0$ in $\mathcal{L}_{F,F(\boldsymbol{w}_0)}$, we can bound the number of oracle calls for $\mathcal{A}$ to output a candidate vector in $\mathcal{S}$, e.g. for GD to output a FOSP. We prove it arguing as in Subsection 2.1, 'chaining together' the decrease per iterate in $\mathcal{L}_{F,F(\boldsymbol{w}_0)}$. Then as $F$ is lower bounded, 1) in Definition 2.2 cannot occur too often, so 2) must occur at some point.

We now generalize this to randomized procedures $\mathcal{A}$ which can output several candidate vectors.

**Framework in full generality.**   Consider an update procedure $\mathcal{A} : \mathbb{R}^d \to \mathbb{R}^d \times \bigcup_{n=0}^{\infty} (\mathbb{R}^d)^n$ (possibly randomized). We now consider a map $\mathcal{A} = (\mathcal{A}_1, \mathcal{A}_2)$, $\mathcal{A} : \mathbb{R}^d \to \mathbb{R}^d \times \bigcup_{n=0}^{\infty} (\mathbb{R}^d)^n$ defined as follows:

For all $\boldsymbol{u} \in \mathbb{R}^d$, $\mathcal{A}(\boldsymbol{u}) = (\boldsymbol{p}_1, \boldsymbol{p}_2)$ for $\boldsymbol{p}_1 \in \mathbb{R}^d, \boldsymbol{p}_2 \in \bigcup_{n=0}^{\infty} (\mathbb{R}^d)^n$, and define $\mathcal{A}_1(\boldsymbol{u}) := \boldsymbol{p}_1, \mathcal{A}_2(\boldsymbol{u}) := \boldsymbol{p}_2$.

Intuitively, $\mathcal{A}_1$ computes a future iterate $\mathcal{A}_1(\boldsymbol{u})$. $\mathcal{A}_2$ outputs a sequence of candidate vectors in $\mathbb{R}^d$, among which we hope one lies in $\mathcal{S}$ (e.g. different candidate models in statistical learning).

However, the output of $\mathcal{A}_1$ need not correspond to the 'next iterate' in the traditional sense. For SGD, $\mathcal{A}_1$ does *not* output the next iterate of SGD, but rather the iterate produced by SGD after $K_0 > 1$ steps. This is necessary to guarantee decrease; a single step of SGD need not decrease the value of $F$, but with high probability and large enough $K_0$, a consecutive 'block' of $K_0$ iterates will. We will lay this out concretely next in Subsection 2.3.

**Remark 1.** Often $\mathcal{A}_2$ will output a single vector in $\mathbb{R}^d$, which we hope lies in $\mathcal{S}$, but this is not always the case. Consider guarantees for GD or SGD, which upper bound $\frac{1}{T}\sum_{t=1}^T \|\nabla F(\boldsymbol{w}_t)\|^2 \le \varepsilon^2$ or $\frac{1}{T}\sum_{t=1}^T \|\nabla F(\boldsymbol{w}_t)\| \le \varepsilon$. This only ensures a single $\boldsymbol{w}_t \in \mathcal{S}, 1 \le t \le T$ where $\mathcal{S}$ is the set of FOSPs to tolerance $\varepsilon$ (e.g. Zhang et al. (2019), Jin et al. (2021b), Li et al. (2023b), Xie et al. (2024) and many others). Consequently $(\boldsymbol{w}_1, \ldots, \boldsymbol{w}_T)$ is our sequence of candidate vectors, and the guarantee obtained is that $\boldsymbol{w}_t \in \mathcal{S}$ for some $1 \le t \le T$. We thus allow for $\mathcal{A}_2$ to output multiple candidate vectors.

The following definition formalizes a common property of optimization algorithms we study:

**Definition 2.2** (Decrease Procedure). *Consider a set of interest $\mathcal{S}$, a confidence parameter $\delta > 0$, a decrease threshold $\Delta > 0$, a point $\boldsymbol{u}_0$, and a procedure $\mathcal{A}$ to compute the next iteration. We say $\mathcal{A}$ forms a $(\mathcal{S}, t_{\mathrm{oracle}}(\boldsymbol{u}_0), \Delta(\boldsymbol{u}_0), \delta(\boldsymbol{u}_0), \boldsymbol{u}_0)$-decrease procedure if with probability at least $1 - \delta(\boldsymbol{u}_0)$ over the randomness in $\mathcal{A}$ to compute $\mathcal{A}(\boldsymbol{u}_0)$ from $\boldsymbol{u}_0$, computing $\mathcal{A}(\boldsymbol{u}_0)$ takes at most $t_{\mathrm{oracle}}(\boldsymbol{u}_0)$ oracle calls, and one of the following holds:*

$$1)\ F(\mathcal{A}_1(\boldsymbol{u}_0)) < F(\boldsymbol{u}_0) - \Delta(\boldsymbol{u}_0), \qquad or \qquad 2)\ \mathcal{A}_2(\boldsymbol{u}_0) \cap \mathcal{S} \ne \{\}.$$

Here 1) means that the subsequent iterate has smaller function value, and 2) means that the rule of output $\mathcal{A}_2$ outputs a sequence of candidate vectors, one of which is in $\mathcal{S}$. $\mathcal{A}$ forms a $(\mathcal{S}, t_{\mathrm{oracle}}(\boldsymbol{u}_0), \Delta(\boldsymbol{u}_0), \delta(\boldsymbol{u}_0), \boldsymbol{u}_0)$-decrease procedure if 1) or 2) occurs with high probability.

**Informal Theorem:** For analogous reasons as before, we will establish that if $\mathcal{A}$ is a decrease procedure for all $\boldsymbol{u}_0$ in $\mathcal{L}_{F, F(\boldsymbol{w}_0)}$, we can bound the number of oracle calls for $\mathcal{A}_2$ to output a candidate vector lying in $\mathcal{S}$. Formally, this is Theorem 2.1.

### 2.3 Examples Subsumed by Framework

We demonstrate that a host of first-order optimization algorithms are covered in our framework, and highlight the general recipe for using our framework.

**GD:** Starting from $\boldsymbol{u}$, the next iterate of GD with step size $\eta > 0$ is $\boldsymbol{u} - \eta \nabla F(\boldsymbol{u})$.

1. For $\varepsilon > 0$, let $\mathcal{S} = \{\boldsymbol{w} : \|\nabla F(\boldsymbol{w})\| \le \varepsilon\}$, the set of FOSPs.

2. For all $\boldsymbol{u}_0 \in \mathbb{R}^d$, let $\mathcal{A}(\boldsymbol{u}_0) = (\boldsymbol{u}_0 - \eta \nabla F(\boldsymbol{u}_0), \boldsymbol{u}_0)$. Hence, $\mathcal{A}_1(\boldsymbol{u}_0) = \boldsymbol{u}_0 - \eta \nabla F(\boldsymbol{u}_0)$, $\mathcal{A}_2(\boldsymbol{u}_0) = \boldsymbol{u}_0$, and $t_{\mathrm{oracle}}(\boldsymbol{u}_0) = 1$.

3. **In Claim 1, we establish that if $F$ is satisfies Assumption 1.1, then $\mathcal{A}$ is a decrease procedure for all $\boldsymbol{u}_0 \in \mathcal{L}_{F, F(\boldsymbol{w}_0)}$, for suitable $\eta$ depending on $F(\boldsymbol{w}_0)$.** Our result for GD, Theorem 3.1, subsequently follows by our general framework Theorem 2.1.

**Adaptive GD:** Starting from $\boldsymbol{u}$, the next iterate of Adaptive GD is $\boldsymbol{u} - \eta_{\boldsymbol{u}} \nabla F(\boldsymbol{u})$, where $\eta_{\boldsymbol{u}} > 0$ is an adaptive step size that depends on $\boldsymbol{u}$.

1. For $\varepsilon > 0$, let $\mathcal{S} = \{\boldsymbol{w} : \|\nabla F(\boldsymbol{w})\| \le \varepsilon\}$, the set of FOSPs.

2. For all $\boldsymbol{u}_0 \in \mathbb{R}^d$, let $\mathcal{A}(\boldsymbol{u}_0) = (\boldsymbol{u}_0 - \eta_{\boldsymbol{u}_0} \nabla F(\boldsymbol{u}_0), \boldsymbol{u}_0)$. Hence, $\mathcal{A}_1(\boldsymbol{u}_0) = \boldsymbol{u}_0 - \eta \nabla F(\boldsymbol{u}_0)$, $\mathcal{A}_2(\boldsymbol{u}_0) = \boldsymbol{u}_0$, and $t_{\mathrm{oracle}}(\boldsymbol{u}_0) = 1$.

3. **In Claim 4, we establish that if $F$ is satisfies Assumption 1.1, then $\mathcal{A}$ is a decrease procedure for all $\boldsymbol{u}_0 \in \mathcal{L}_{F, F(\boldsymbol{w}_0)}$, for suitable $\eta_{\boldsymbol{u}}$ depending on $F(\boldsymbol{w}_0)$ and $\|\nabla F(\boldsymbol{u})\|$.** Our result for Adaptive GD, Theorem 3.2, then follows by Theorem 2.1.

However, for SGD and other randomized algorithms involving randomness, 1) in Definition 2.2 does not hold deterministically. This is where the generality in our framework is powerful. For SGD, by concentration inequalities we show that 1) is true *with high probability over a long enough 'block' of subsequent iterates*, as long as none of the iterates in the block have small gradient. We then define $\mathcal{A}$ so that $\mathcal{A}_1$ outputs the composition of the SGD steps in the block, and $\mathcal{A}_2$ outputs all the iterates of the block. The resulting guarantee is that one of the points among all the blocks lies in $\mathcal{S}$.

**SGD:** Starting from $\boldsymbol{u}$, letting $\nabla f(\boldsymbol{u}; \zeta)$ be a stochastic gradient oracle where $\zeta$ is a minibatch sample, the next iterate of SGD is $\boldsymbol{u} - \eta \nabla f(\boldsymbol{u}; \zeta)$ where $\eta > 0$ is the step size.

1. For $\varepsilon > 0$, let $\mathcal{S} = \{\boldsymbol{w} : \|\nabla F(\boldsymbol{w})\| \le \varepsilon\}$, the set of FOSPs.

2. Consider any $K_0 \geq 1$. For all $\boldsymbol{u}_0 \in \mathbb{R}^d$, let $\boldsymbol{p}_0 = \boldsymbol{u}_0$, and define a sequence $(\boldsymbol{p}_i)_{0 \leq i \leq K_0}$ via $\boldsymbol{p}_i = \boldsymbol{p}_{i-1} - \eta \nabla f(\boldsymbol{p}_{i-1}; \boldsymbol{\zeta}_i)$, where the $\boldsymbol{\zeta}_i$ are i.i.d. minibatch samples. Note this sequence can be equivalently defined by repeatedly composing the function $\boldsymbol{u} \to \boldsymbol{u} - \eta \nabla f(\boldsymbol{u}; \boldsymbol{\zeta})$. We then define $\mathcal{A}(\boldsymbol{u}_0) = (\boldsymbol{p}_{K_0}, (\boldsymbol{p}_i)_{0 \leq i \leq K_0 - 1})$, hence $\mathcal{A}_1(\boldsymbol{u}_0) = \boldsymbol{p}_{K_0}$, $\mathcal{A}_2(\boldsymbol{u}_0) = (\boldsymbol{p}_i)_{0 \leq i \leq K_0 - 1}$. Note all the $\boldsymbol{p}_i$ are a function of $\boldsymbol{u}_0$ and the randomness in the stochastic gradient oracle $\nabla f(\cdot; \cdot)$. We let $t_{\text{oracle}}(\boldsymbol{u}_0) = K_0$, which need not equal 1. This procedure is clearly SGD, with its iterates divided into blocks of length $K_0$.

3. **In Claim 5, we establish that if $F$ is satisfies Assumption 1.1 and $\nabla f(\cdot; \cdot)$ satisfies Assumption 3.1, then $\mathcal{A}$ is a decrease procedure for all $\boldsymbol{u}_0 \in \mathcal{L}_{F, F(\boldsymbol{w}_0)}$ for suitable algorithm parameters.** Our result for SGD, Theorem 3.3, then follows by Theorem 2.1.

**SOSP-finding algorithms:** We now study finding SOSPs using first order methods under our regularity assumptions. We analyze two algorithms to achieve this under exact and stochastic gradients, respectively Perturbed GD (Algorithm 1, Jin et al. (2017)) and Restarted SGD (Algorithm 2, Fang et al. (2019)). We remark that our framework likely subsumes many other algorithms.

**Perturbed GD:** This algorithm, formally written in Algorithm 1, Section D, is as follows. At $\boldsymbol{u}$,

- If $\|\nabla F(\boldsymbol{u})\| > g_{\text{thres}}$ for some appropriate $g_{\text{thres}}$, the algorithm simply runs a step of GD.
- Else, Algorithm 1 adds uniform noise from a ball with particular radius and runs GD for $t_{\text{thres}}$ iterations for suitably chosen $t_{\text{thres}}$, yielding $\boldsymbol{u}'$. We check if $F(\boldsymbol{u}') - F(\boldsymbol{u}) \leq -f_{\text{thres}}$ for some appropriate $f_{\text{thres}}$. If decrease does not occur, we return $\boldsymbol{u}$; if decrease occurred, we go back to the If/Else with $\boldsymbol{u}'$ in place of $\boldsymbol{u}$.

Notice now that the oracle complexity $t_{\text{oracle}}$, probability $\delta$, and amount of decrease $\Delta$ depend on the location $\boldsymbol{u}$. Our framework readily subsumes this example as follows.

1. For $\varepsilon > 0$, let $\mathcal{S} = \{\boldsymbol{w} : \|\nabla F(\boldsymbol{w})\| \leq \varepsilon, \nabla^2 F(\boldsymbol{w}) \succeq -\sqrt{\varepsilon} \boldsymbol{I}\}$, the set of SOSPs.

2. For all $\boldsymbol{u}_0 \in \mathbb{R}^d$, if $\|\nabla F(\boldsymbol{u}_0)\| > g_{\text{thres}}$, we let

$$\mathcal{A}(\boldsymbol{u}_0) = (\boldsymbol{u}_0 - \eta \nabla F(\boldsymbol{u}_0), \boldsymbol{u}_0), \text{ hence } \mathcal{A}_1(\boldsymbol{u}_0) = \boldsymbol{u}_0 - \eta \nabla F(\boldsymbol{u}_0), \mathcal{A}_2(\boldsymbol{u}_0) = \boldsymbol{u}_0.$$

Otherwise if $\|\nabla F(\boldsymbol{u}_0)\| \leq g_{\text{thres}}$, we let $\boldsymbol{p}_0 = \boldsymbol{u}_0 + \boldsymbol{\xi}$ where $\boldsymbol{\xi}$ is uniform from $\mathbb{B}(\vec{\boldsymbol{0}}, r)$, and define a sequence $(\boldsymbol{p}_i)_{0 \leq i \leq t_{\text{thres}}}$ via $\boldsymbol{p}_i = \boldsymbol{p}_{i-1} - \eta \nabla F(\boldsymbol{p}_{i-1})$. We then define

$$\mathcal{A}(\boldsymbol{u}_0) = (\boldsymbol{p}_{t_{\text{thres}}}, \boldsymbol{u}_0), \text{ hence } \mathcal{A}_1(\boldsymbol{u}_0) = \boldsymbol{p}_{t_{\text{thres}}}, \mathcal{A}_2(\boldsymbol{u}_0) = \boldsymbol{u}_0.$$

Thus

$$t_{\text{oracle}}(\boldsymbol{u}_0) = \begin{cases} t_{\text{thres}} & : \|\nabla F(\boldsymbol{u}_0)\| \leq g_{\text{thres}} \\ 1 & : \|\nabla F(\boldsymbol{u}_0)\| > g_{\text{thres}}. \end{cases}$$

This is identical to Algorithm 1, and highlights why $t_{\text{oracle}}, \delta, \Delta$ need to depend on $\boldsymbol{u}_0$.

3. **In Claim 2, we establish that if $F$ satisfies Assumption 1.2, then $\mathcal{A}$ is a decrease procedure for all $\boldsymbol{u}_0 \in \mathcal{L}_{F, F(\boldsymbol{w}_0)}$ for suitable algorithm parameters.** Our result for Perturbed GD, Theorem 3.4, then follows by Theorem 2.1.

**Restarted SGD:** This algorithm, formally written in Algorithm 2, Section E, works as follows. Take $B = \tilde{\Theta}(\varepsilon^{0.5})$, $K_0 = \tilde{\Theta}(\varepsilon^{-2})$. Consider an anchor point $\boldsymbol{u}$, first taken to be the initialization $\boldsymbol{w}_0$. The algorithm runs SGD until its iterates first escape the ball $\mathbb{B}(\boldsymbol{u}, B)$, tracking at most $K_0$ iterations.

- If an escape occurs within $K_0$ iterations, letting $\boldsymbol{u}'$ be the first iterate that escaped $\mathbb{B}(\boldsymbol{u}, B)$, the algorithm sets $\boldsymbol{u}'$ to be the anchor point and runs the same procedure.
- If these $K_0$ iterates do not escape within $K_0$ iterations, return their *average*.

We cover Restarted SGD in our framework as follows.

1. For $\varepsilon > 0$, let $\mathcal{S} = \{\boldsymbol{w} : \|\nabla F(\boldsymbol{w})\| \leq \varepsilon, \nabla^2 F(\boldsymbol{w}) \succeq -\sqrt{\varepsilon} \boldsymbol{I}\}$, the set of SOSPs.

2. For all $\boldsymbol{u}_0 \in \mathbb{R}^d$, let $\boldsymbol{p}_0 = \boldsymbol{u}_0$. We define a sequence $(\boldsymbol{p}_i)_{0 \leq i \leq K_0}$ via $\boldsymbol{p}_i = \boldsymbol{p}_{i-1} - \eta(\nabla f(\boldsymbol{p}_{i-1}; \boldsymbol{\zeta}_i) + \tilde{\sigma} \Lambda^i)$, where $\nabla f(\cdot; \cdot)$ is our stochastic gradient oracle, the $\boldsymbol{\zeta}_i$ are i.i.d. minibatch samples, the $\Lambda^i \sim \mathbb{B}(\vec{\boldsymbol{0}}, 1)$ are i.i.d., and $\tilde{\sigma}$ is a parameter governing the noise level. Note this sequence can be equivalently defined by repeatedly composing the function

$\boldsymbol{u} \to \boldsymbol{u} - \eta(\nabla f(\boldsymbol{u}; \boldsymbol{\zeta}) + \tilde{\sigma}\Lambda)$. If it exists, let $i, 1 \leq i \leq K_0$ be the minimal index such that $\|\boldsymbol{p}_i - \boldsymbol{p}_0\| > B$. Otherwise let $i = K_0$. In either case, we define

$$\mathcal{A}(\boldsymbol{u}_0) = \left(\boldsymbol{p}_i, \frac{1}{i}\sum_{t=0}^{i-1}\boldsymbol{p}_t\right), \text{ hence } \mathcal{A}_1(\boldsymbol{u}_0) = \boldsymbol{p}_i, \mathcal{A}_2(\boldsymbol{u}_0) = \frac{1}{i}\sum_{t=0}^{i-1}\boldsymbol{p}_t.$$

We let $t_{\text{oracle}}(\boldsymbol{u}_0) = K_0$.[3] This is clearly identical to Algorithm 2.

3. **In Claim 7, we establish that if $F$ satisfies Assumption 1.2 and $\nabla f(\cdot; \cdot)$ satisfies Assumption 3.1 and Assumption 3.2, then $\mathcal{A}$ is a decrease procedure for all $\boldsymbol{u}_0 \in \mathcal{L}_{F,F(\boldsymbol{w}_0)}$ for suitable algorithm parameters.** Our result for Restarted GD, Theorem 3.5, then follows by Theorem 2.1.

**Theorem 2.1** (General Framework). *Consider a given initialization $\boldsymbol{w}_0$ of $\mathcal{A}$ and a desired set $\mathcal{S}$. Define a sequence $(\boldsymbol{w}_t)_{t\geq0}$ recursively by $\boldsymbol{w}_{t+1} = \mathcal{A}_1(\boldsymbol{w}_t)$. Suppose that for all $\boldsymbol{u}_0 \in \mathcal{L}_{F,F(\boldsymbol{w}_0)}$, $\mathcal{A}$ forms a $(\mathcal{S}, t_{\text{oracle}}(\boldsymbol{u}_0), \Delta(\boldsymbol{u}_0), \delta(\boldsymbol{u}_0), \boldsymbol{u}_0)$-decrease procedure. Define $\overline{\Delta} = \inf_{\boldsymbol{u}\in\mathcal{L}_{F,F(\boldsymbol{w}_0)}} \frac{\Delta(\boldsymbol{u})}{t_{\text{oracle}}(\boldsymbol{u})}$. Then with probability at least*

$$1 - \sup_{\boldsymbol{u}\in\mathcal{L}_{F,F(\boldsymbol{w}_0)}} \delta(\boldsymbol{u}) \cdot \sup_{\boldsymbol{u}\in\mathcal{L}_{F,F(\boldsymbol{w}_0)}} \left\{\frac{F(\boldsymbol{w}_0)}{\Delta(\boldsymbol{u})}\right\}, \text{ upon making } N = \frac{F(\boldsymbol{w}_0)}{\overline{\Delta}} + \sup_{\boldsymbol{u}\in\mathcal{L}_{F,F(\boldsymbol{w}_0)}} t_{\text{oracle}}(\boldsymbol{u})$$

*oracle calls, there exists $\boldsymbol{w}_t \in (\boldsymbol{w}_t)_{t\geq0}$ such that $\mathcal{A}_2(\boldsymbol{w}_t) \cap \mathcal{S} \neq \{\}$. I.e. for some $\boldsymbol{w}_t$, $\mathcal{A}_2(\boldsymbol{w}_t)$ will output a sequence of candidate vectors, one of which is in $\mathcal{S}$. Furthermore, if the output of $\mathcal{A}_2$ has length at most $S$, then the number of candidate vectors outputted is at most $S \cdot \sup_{\boldsymbol{u}\in\mathcal{L}_{F,F(\boldsymbol{w}_0)}} \left\{\frac{F(\boldsymbol{w}_0)}{\Delta(\boldsymbol{u})}\right\}$.*

Our full proof is in Section B.[4] The proof formalizes the main idea from Subsection 2.1, by 'chaining together' the decrease per iterate in $\mathcal{L}_{F,F(\boldsymbol{w}_0)}$. Then as $F$ is lower bounded, 1) in Definition 2.2 cannot occur too many times, so 2) must occur at some point.

**Remark 2.** To verify $\mathcal{A}$ is a decrease procedure in $\mathcal{L}_{F,F(\boldsymbol{w}_0)}$, we can systematically port over analyses in the literature. As discussed in Subsection 2.1, $\boldsymbol{u}_0$ being in $\mathcal{L}_{F,F(\boldsymbol{w}_0)}$ allows us to show the algorithm is 'local', crucially giving us quantitative control over the relevant derivatives. We view this as a core *strength* of our work; our framework allows us to *systematically* extend results from the smooth setting to generalizations of smoothness.

# 3 Convergence Results

Here we systematically obtain our convergence results for the algorithms listed in Subsection 2.3, by formally showing that they are decrease procedures. **Our main results are Theorem 3.4, Theorem 3.5: that under Assumption 1.2, variants of GD/SGD can find SOSPs.** We note our dependence on $\varepsilon, d$ for Theorem 3.1, Theorem 3.2, Theorem 3.3, and Theorem 3.5 match lower bounds for smooth functions (Carmon et al., 2020, 2021; Arjevani et al., 2020), and hence are optimal in this setting too.[5] We present examples and implications of our results in Subsection 3.6.

**Remark 3** (Dependence on Initialization). In our results, the step size $\eta$ here depends only on $\rho_1(F(\boldsymbol{w}_0))$, a fixed value depending only on initialization. Moreover, the expressions on $\eta$ depending on $\rho_1(F(\boldsymbol{w}_0))$ in our results and proofs to follow are only an **upper bound** for working step sizes. We do not need to know these exact values. Therefore, all that is needed is an upper bound on fixed quantities such as $\rho_1(F(\boldsymbol{w}_0))$; hence a working step size $\eta$ for our algorithms in practice and theory can be found using cross validation or binary search.

Letting $\eta(\boldsymbol{w}_0)$ be an upper bound on the step size $\eta$ needed to guarantee convergence, we note by searching over $\log(\eta(\boldsymbol{w}_0))$ with binary search, we will find an $\eta$ with a constant factor 2 of $\eta(\boldsymbol{w}_0)$. This $\log$ factor will be logarithmic in $\varepsilon, d$, and will only change the claimed iteration complexity by a universal constant factor. The latter is because the amount of decrease in the definition of Decrease Procedure will in turn only change by a universal constant multiple.

---

[3]Defining $i$ as above, note that we can compute $\mathcal{A}(\boldsymbol{u}_0)$ using $i$ rather than $K_0$ oracle calls, but this change does not affect runtime beyond constant factors.

[4]The extra second term in the sum defining $N$ occurs as $t_{\text{oracle}}, \Delta, \delta$ have $\boldsymbol{u}_0$-dependence.

[5]Dependence on $\varepsilon$ in Theorem 3.3 and on $\varepsilon, d$ in Theorem 3.5 are tight up to log factors.

**Remark 4** (On Adaptivity). Our results hold for non-adaptive versions of GD/SGD and their variants. That said, one can interpret cross validation or binary search over $\eta$ as adaptive algorithms in their own right. As mentioned above, it is relatively straightforward to obtain analogous results to our current ones for cross validation or binary search. In the learning from data setting, one can make the cross validation result formal using classic techniques.

## 3.1 Gradient Descent

**Theorem 3.1** (GD for FOSP). *Suppose $F$ satisfies Assumption 1.1. Run GD initialized at $\boldsymbol{w}_0$, with step size $\eta = \frac{1}{L_1(\boldsymbol{w}_0)}$ where $L_1(\boldsymbol{w}_0)$ is defined in (4). Then letting*

$$T = \frac{2F(\boldsymbol{w}_0)L_1(\boldsymbol{w}_0)}{\varepsilon^2}, \text{ within } T+1 \text{ oracle calls to } \nabla F(\cdot),$$

*GD will output $T$ candidate vectors $(\boldsymbol{p}_1, \ldots, \boldsymbol{p}_T)$, one of which satisfies $\|\nabla F(\boldsymbol{p}_t)\| \le \varepsilon$.*

We prove Theorem 3.1 here to show our strategy's simplicity. The following Lemmas, proved in Subsection A.3, help show GD is 'local' for $\boldsymbol{w} \in \mathcal{L}_{F,F(\boldsymbol{w}_0)}$.

**Corollary 1.** *For $F$ satisfying Assumption 1.1, we have $\|\nabla F(\boldsymbol{w})\| \le \rho_0(F(\boldsymbol{w}))$, where $\rho_0 : \mathbb{R}_{\ge 0} \to \mathbb{R}_{\ge 0}$ is a non-decreasing function given by $\rho_0(x) = \rho_1(x)\sqrt{2\theta(x)}$, where $\theta(x) = \int_0^x \frac{1}{\rho_1(v)}\mathrm{d}v$.*

**Lemma 3.1.** *Under Assumption 1.1, for $\boldsymbol{x}, \boldsymbol{y}$ with $\|\boldsymbol{y}-\boldsymbol{x}\| \le \frac{1}{\rho_0(F(\boldsymbol{x})+1)}$, $F(\boldsymbol{y}) - F(\boldsymbol{x}) \le 1$.*

Combining the above with Assumption 1.1 immediately gives:

**Lemma 3.2.** *Suppose $F$ satisfies Assumption 1.1. Defining $\rho_0$ as in Corollary 1, let*

$$L_1(\boldsymbol{w}_0) = \max\{1, \rho_0(F(\boldsymbol{w}_0)+1), \rho_0(F(\boldsymbol{w}_0))\rho_0(F(\boldsymbol{w}_0)+1), \rho_1(F(\boldsymbol{w}_0)+1)\}. \qquad (4)$$

*Then for all $\boldsymbol{w} \in \mathcal{L}_{F,F(\boldsymbol{w}_0)}$, $\|\nabla^2 F(\boldsymbol{u})\|_{\mathrm{op}} \le L_1(\boldsymbol{w}_0)$ for all $\boldsymbol{u} \in \mathbb{B}(\boldsymbol{w}, \rho_0(F(\boldsymbol{w}_0)+1)^{-1})$.*

**Proof of Theorem 3.1.** Use Theorem 2.1 with $\mathcal{S} = \{\boldsymbol{w} : \|\nabla F(\boldsymbol{w})\| \le \varepsilon\}$, defining $\mathcal{A}$ as in Subsection 2.3. Upon applying Theorem 2.1, the following Claim directly proves Theorem 3.1:

**Claim 1.** *For any $\boldsymbol{u}_0$ in $\mathcal{L}_{F,F(\boldsymbol{w}_0)}$, $\mathcal{A}$ is a $(\mathcal{S}, 1, \frac{\varepsilon^2}{2L_1(\boldsymbol{w}_0)}, 0, \boldsymbol{u}_0)$-decrease procedure.*

To prove Claim 1, note for $\boldsymbol{u}_0 \in \mathcal{S}$, by definition of $\mathcal{A}_2$ that $\mathcal{A}_2(\boldsymbol{u}_0) = (\boldsymbol{u}_0) \in \mathcal{S}$. Now if $\boldsymbol{u}_0 \notin \mathcal{S}$ (i.e. $\|\nabla F(\boldsymbol{u}_0)\| > \varepsilon$), consider $\boldsymbol{u}_1 = \mathcal{A}_1(\boldsymbol{u}_0) = \boldsymbol{u}_0 - \eta\nabla F(\boldsymbol{u}_0)$. By Corollary 1 and as $F(\boldsymbol{u}_0) \le F(\boldsymbol{w}_0)$, $\|\nabla F(\boldsymbol{u}_0)\| \le \rho_0(F(\boldsymbol{u}_0)) \le \rho_0(F(\boldsymbol{w}_0))$, so by choice of $\eta$,

$$\|\boldsymbol{u}_1 - \boldsymbol{u}_0\| = \eta\|\nabla F(\boldsymbol{u}_0)\| \le \eta\rho_0(F(\boldsymbol{w}_0)) \le \rho_0(F(\boldsymbol{w}_0)+1)^{-1}.$$

By Lemma 3.2, for all $\boldsymbol{p}$ in the line segment $\overline{\boldsymbol{u}_0\boldsymbol{u}_1}$, $\|\nabla^2 F(\boldsymbol{p})\|_{\mathrm{op}} \le L_1(\boldsymbol{w}_0)$. By Lemma A.1, which only depends on the smoothness constant in the segment between the two iterates (see Subsection A.1),

$$F(\boldsymbol{u}_1) \le F(\boldsymbol{u}_0) - \eta\|\nabla F(\boldsymbol{u}_0)\|^2 + \frac{L_1(\boldsymbol{w}_0)\eta^2}{2} \cdot \|\nabla F(\boldsymbol{u}_0)\|^2 < F(\boldsymbol{u}_0) - \frac{\varepsilon^2}{2L_1(\boldsymbol{w}_0)},$$

as $\|\nabla F(\boldsymbol{u}_0)\| > \varepsilon$ and by our choice of $\eta$. This proves Claim 1, completing the proof. $\qquad \square$

Note it is critical here that $\boldsymbol{u}_0$ is in the $F(\boldsymbol{w}_0)$-sublevel set. Also, to satisfy Corollary 1, $\rho_0(x)$ just needs to be a non-decreasing pointwise upper bound of $\rho_1(x)\sqrt{2\theta(x)}$. For example when $F$ is $(L_0, L_1)$-smooth, we show in Proposition A.2 that we can take $\rho_0(x) = 2L_0^{1/2}x^{1/2} + 5L_1^2 L_0^{-1/2}x^{3/2}$.

## 3.2 Adaptive Gradient Descent

Our proof and framework readily adapt to Adaptive GD, as discussed Subsection 2.3. It is even easier as Adaptive GD is automatically 'local' via gradient clipping. Our proof is in Subsection C.1.

**Theorem 3.2** (GD for FOSP). *Suppose $F$ satisfies Assumption 1.1. Run Adaptive GD initialized at $\boldsymbol{w}_0$, with adaptive step size $\eta_{\boldsymbol{w}_t} = \min\left\{\frac{1}{L_1'(\boldsymbol{w}_0)}, \frac{1}{\rho_0(F(\boldsymbol{w}_0)+1)\|\nabla F(\boldsymbol{w}_t)\|}\right\}$ where $L_1'(\boldsymbol{w}_0) = \rho_1(F(\boldsymbol{w}_0)+1)$. Let $T = \frac{2F(\boldsymbol{w}_0)}{\min\left\{\frac{L_1'(\boldsymbol{w}_0)}{\rho_0(F(\boldsymbol{w}_0)+1)^2}, \frac{\varepsilon^2}{L_1'(\boldsymbol{w}_0)}\right\}}$. Within $T+1$ oracle calls to $\nabla F(\cdot)$, Adaptive GD will output $T$ candidate vectors $(\boldsymbol{p}_1, \ldots, \boldsymbol{p}_T)$, one of which satisfies $\|\nabla F(\boldsymbol{p}_t)\| \le \varepsilon$.*

## 3.3 Stochastic Gradient Descent

We make the following assumption on the stochastic gradient oracle:

**Assumption 3.1.** *The stochastic gradient oracle $\nabla f(\cdot;\cdot)$ is unbiased (i.e. $\mathbb{E}_{\boldsymbol{\zeta}}[\nabla f(\cdot;\boldsymbol{\zeta})] = \nabla F(\cdot)$), and for a non-decreasing function $\sigma : \mathbb{R}^+ \mapsto \mathbb{R}^+$ and all $\boldsymbol{w}, \boldsymbol{\zeta}$, $\|\nabla f(\boldsymbol{w};\boldsymbol{\zeta}) - \nabla F(\boldsymbol{w})\|^2 \leq \sigma(F(\boldsymbol{w}))^2$.*

In many problems of interest in ML, noise scales with function value (Wojtowytsch, 2023, 2024); Assumption 3.1 captures this setting. Note we do not assume a global bound on $\|\nabla F\|$ or $F$, thus noise is *unbounded*. We show in Remark 7 that one can extend Theorem 3.3 to when $\|\nabla f(\boldsymbol{w};\boldsymbol{\zeta}) - \nabla F(\boldsymbol{w})\|$ is sub-Gaussian with parameter $\sigma(F(\boldsymbol{w}))$ with a longer technical argument. We also note that bounding $L_2$ gradient error in terms of function value has been studied – denoted by the expected smoothness assumption – in Gower et al. (2019, 2021).

**Theorem 3.3** (SGD for FOSP). *Suppose $F$ satisfies Assumption 1.1 and that the stochastic gradient oracle $\nabla f(\cdot;\cdot)$ satisfies Assumption 3.1. For any $\delta \in (0,1)$, run SGD initialized at $\boldsymbol{w}_0$, for a given fixed step size $\eta \leq \tilde{O}(\varepsilon^2)$ depending on $\varepsilon$, $\delta$, and $F(\boldsymbol{w}_0)$. Then with probability at least $1 - \delta$, within*

$$T = \tilde{O}\left(\frac{1}{\varepsilon^4} \cdot \mathrm{polylog}(1/\varepsilon, 1/\delta)\right) \text{ oracle calls to } \nabla f(\cdot;\cdot),$$

*SGD will output $T$ candidate vectors $\boldsymbol{w}$, one of which satisfies $\|\nabla F(\boldsymbol{w})\| \leq \varepsilon$.*

Here $\tilde{O}(\cdot)$ hides additional $F(\boldsymbol{w}_0)$-dependence. Our full proof is in Subsection C.2. As discussed in Subsection 2.3, the idea is similar to the proof of Theorem 3.1, except we now establish high-probability decrease over blocks of consecutive iterates using concentration inequalities.

## 3.4 Perturbed Gradient Descent

**Theorem 3.4** (Perturbed GD for SOSP). *Suppose $F$ satisfies Assumption 1.2. For any $\delta \in (0,1)$, run Perturbed GD (Algorithm 1, from Jin et al. (2017)) initialized at $\boldsymbol{w}_0$, with appropriate step size $\eta$ and other parameters depending on $\varepsilon, \delta, d$, and $F(\boldsymbol{w}_0)$. Then with probability at least $1 - \delta$, within*

$$T = O\left(\frac{1}{\varepsilon^2} \log^4\left(\frac{d}{\varepsilon\delta}\right)\right) \text{ oracle calls to } \nabla F(\cdot),$$

*Perturbed GD outputs $T$ candidates $\boldsymbol{w}$, one of which satisfies $\|\nabla F(\boldsymbol{w})\| \leq \varepsilon, \nabla^2 F(\boldsymbol{w}) \succeq -\sqrt{\varepsilon}\boldsymbol{I}$.*

**Remark 5.** Here we find $\boldsymbol{w}$ with $\nabla^2 F(\boldsymbol{w}) \succeq -\sqrt{\varepsilon}\boldsymbol{I}$, which is most sensible without Lipschitz Hessian.

For Perturbed GD here in Subsection 3.4, asymptotic notation hides universal constants and dependence on $F(\boldsymbol{w}_0)$. The full proof is in Section D; here we give the main ideas. Define $\mathcal{A}, t_{\mathrm{oracle}}(\boldsymbol{u}_0), \mathcal{S}$ as in Subsection 2.3 for Perturbed GD. Consider $g_{\mathrm{thres}} = \tilde{\Theta}(\varepsilon)$, $f_{\mathrm{thres}} = \tilde{\Theta}(\varepsilon^{1.5})$ defined in Algorithm 1. Let

$$\Delta(\boldsymbol{u}_0) = \begin{cases} f_{\mathrm{thres}} & : \|\nabla F(\boldsymbol{u}_0)\| \leq g_{\mathrm{thres}} \\ \frac{\eta}{2} \cdot g_{\mathrm{thres}}^2 & : \|\nabla F(\boldsymbol{u}_0)\| > g_{\mathrm{thres}}. \end{cases}$$

The central Claim is as follows, from which Theorem 3.4 follows directly via Theorem 2.1:

**Claim 2.** *For all $\boldsymbol{u}_0 \in \mathcal{L}_{F,F(\boldsymbol{w}_0)}$, $\mathcal{A}$ is a $\left(\mathcal{S}, t_{\mathrm{oracle}}(\boldsymbol{u}_0), \Delta(\boldsymbol{u}_0), \frac{dL_1(\boldsymbol{w}_0)}{\sqrt{\varepsilon}} e^{-\chi}, \boldsymbol{u}_0\right)$-decrease procedure, where $\chi = \Theta\left(\log\left(\frac{d}{\varepsilon^{2.5}\delta}\right)\right)$ and $L_1(\boldsymbol{w}_0)$ is defined in (4).*

Perturbed GD is a decrease procedure only in $\mathcal{L}_{F,F(\boldsymbol{w}_0)}$ where we have quantitative control on $F$ and its derivatives – *using our framework is crucial*. To prove Claim 2, we note the analysis of Perturbed GD in Jin et al. (2017) only considers 'local' points close to the current iterate the algorithm. Thus we can apply similar analysis, using Lemma 3.1, Lemma 3.2, and the similar Lemma D.1 to give enough control over the derivatives of $F$ between these 'local' points close to $\boldsymbol{u}_0 \in \mathcal{L}_{F,F(\boldsymbol{w}_0)}$.

## 3.5 Restarted Stochastic Gradient Descent

In addition to Assumption 3.1, we will make the following mild assumption on the error of the stochastic gradient oracle, a relaxation of Assumption 1 of Fang et al. (2019).

**Assumption 3.2.** *For every $\boldsymbol{w}, \boldsymbol{\zeta}$, $\left\|\nabla^2 f(\boldsymbol{w};\boldsymbol{\zeta})\right\|_{\mathrm{op}} \leq \rho_3(\|\nabla f(\boldsymbol{w};\boldsymbol{\zeta})\|, F(\boldsymbol{w}))$, where $\rho_3(\cdot,\cdot) : \mathbb{R}_{\geq 0} \times \mathbb{R}_{\geq 0} \to \mathbb{R}_{\geq 0}$ is non-decreasing in both arguments.*

Note if $f(\cdot; \boldsymbol{\zeta})$ satisfies the regularity assumptions of Zhang et al. (2019) or Li et al. (2023a) for every $\boldsymbol{\zeta}$, then Assumption 3.2 is satisfied. However, Assumption 3.2 goes well beyond these assumptions, allowing for the operator norm of $\nabla^2 f(\cdot; \boldsymbol{\zeta})$ to also diverge in $F(\boldsymbol{w})$.[6]

**Theorem 3.5** (Restarted SGD for SOSP). *Suppose $F$ satisfies Assumption 1.2 and $\nabla f(\cdot; \cdot)$ satisfies Assumption 3.1 and Assumption 3.2. For any $\delta \in (0, 1)$, run Restarted SGD (Algorithm 2, the same algorithm from Fang et al. (2019)) initialized at $\boldsymbol{w}_0$, with appropriate step size $\eta$ and other parameters depending on $\varepsilon$, $\delta$, $d$, and $F(\boldsymbol{w}_0)$. Then with probability at least $1 - \delta$, upon making*

$$T = \tilde{O}\left(\frac{1}{\varepsilon^{3.5}}\right) \text{ oracle calls to } \nabla f(\cdot; \cdot),$$

*Restarted SGD outputs $T$ candidates $\boldsymbol{w}$, one of which satisfies $\|\nabla F(\boldsymbol{w})\| \le \varepsilon, \nabla^2 F(\boldsymbol{w}) \succeq -\sqrt{\varepsilon}\boldsymbol{I}$.*

Here $\tilde{O}(\cdot)$ only hides constant factors, $F(\boldsymbol{w}_0)$-dependent constants, and logarithmic factors in $d, 1/\varepsilon, 1/\delta$. We specify the exact parameters and detail the proof in Section E. The proof follows our framework instantiated for Restarted GD as in Subsection 2.3. The crux again is establishing that the algorithm is a decrease procedure in the $F(\boldsymbol{w}_0)$-sublevel set, done in Claim 7.

### 3.6 Examples

Several interesting problems in ML and optimization, such as Phase Retrieval and Matrix PCA, can be globally optimized by finding a SOSP (but not a FOSP), and satisfy Assumption 1.2. See Section F for these verifications. Thus Theorem 3.4 and Theorem 3.5 immediately imply we can solve the following problems, with no customized analysis required.

**Phase Retrieval:** We reconstruct a hidden vector $\boldsymbol{w}^* \in \mathbb{R}^d$ with $\|\boldsymbol{w}^*\| = 1$ using phaseless observations $\mathcal{S} = \{(\boldsymbol{a}_j, y_j)\}$ where $y_j = \langle \boldsymbol{a}_j, \boldsymbol{w}^* \rangle^2$, $\boldsymbol{a}_j \sim \mathcal{N}(\vec{\boldsymbol{0}}, \boldsymbol{I}_d)$. The population loss is $F_{\text{pr}}(\boldsymbol{w}) = \mathbb{E}_{\boldsymbol{a} \sim \mathcal{N}(\vec{\boldsymbol{0}}, \boldsymbol{I}_d)}\left[\left(\langle \boldsymbol{a}, \boldsymbol{w} \rangle^2 - \langle \boldsymbol{a}, \boldsymbol{w}^* \rangle^2\right)^2\right]$.

**Matrix PCA:** Given a $d \times d$ symmetric positive definite (PD) matrix $\boldsymbol{M}$, we aim to find $\boldsymbol{w} \in \mathbb{R}^d$ (the first principal component) minimizing $F_{\text{pca}}(\boldsymbol{w}) = \frac{1}{2}\|\boldsymbol{w}\boldsymbol{w}^\top - \boldsymbol{M}\|_F^2$.

### 3.7 Practical Implications and Simulations

Our results show under generalizations of smoothness, unlike with Lipschitz gradient/Hessian, the larger the loss is at initialization (larger $F(\boldsymbol{w}_0)$) and larger self-bounding functions $\rho_1(\cdot)$ shrink the 'window' for choosing a working $\eta$. Specifically, with larger loss at initialization, the smaller the largest working step size is, in contrast to optimizing smooth functions. *This implies in practice, for losses with non-Lipschitz gradient/Hessian, one should tune $\eta$ based on suboptimality at initialization.*

In Section G, we validate this finding through simulations with GD and SGD on several natural smooth and generalized smooth functions, namely $F(\boldsymbol{w}) = \|\boldsymbol{A}\boldsymbol{w}\|^p$ for $p = 2, 3, 4, 5, 6$. Our simulations show the above theoretical conclusions match behavior in practice, validating the practical implications of our theoretical results on which step sizes successfully optimize generalized smooth functions.

## 4 Conclusion

We present a systematic framework to analyze the convergence of first order methods to FOSPs and SOSPs under generalizations of smoothness, extending key results in finding SOSPs via first-order methods to this setting. Our work *elucidates fundamental behavior of first-order optimization algorithms*, showing that 'chaining together high-probability decrease' enables their success under generalizations of smoothness. Our framework applies for many other algorithms (e.g. Langevin Dynamics) and sets of interest $\mathcal{S}$ (e.g. higher order stationary points, or minima with good generalization properties). It can also inform the design of new optimization algorithms, by designing procedures which are decrease procedures. These promising directions are left for future research.

## 5 Acknowledgments

We thank Dylan J. Foster and Ayush Sekhari for discussions, and Anthony Bao, Fan Chen, and Albert Gong for useful suggestions on the presentation of our manuscript.

---

[6]While the above assumes that $f(\cdot; \boldsymbol{\zeta})$ is twice differentiable, it can be easily phrased in terms of $\nabla f(\cdot; \boldsymbol{\zeta})$.

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

# Contents

# A   Technical Preliminaries

## A.1   Helpful Background Lemmas

We will use the following classical inequalities from optimization to show we still have some notion of control if we have local bounds on the relevant derivatives.

**Lemma A.1.** *Suppose $F$ is twice differentiable, and for all $\boldsymbol{u} \in \overline{\boldsymbol{xy}}$ (the line segment) we have $\left\|\nabla^2 F(\boldsymbol{u})\right\|_{\mathrm{op}} \le L$. Then, we have*

$$F(\boldsymbol{y}) \le F(\boldsymbol{x}) + \langle \nabla F(\boldsymbol{x}), \boldsymbol{y} - \boldsymbol{x} \rangle + \frac{L}{2} \|\boldsymbol{y} - \boldsymbol{x}\|^2.$$

**Proof.** This follows by the proof of Lemma 3.4 in Bubeck et al. (2015). In particular, one can readily verify that $\boldsymbol{x} + t(\boldsymbol{y} - \boldsymbol{x}) \in \overline{\boldsymbol{xy}}$ for all $t \in [0, 1]$. Hence for all $t \in [0, 1]$ and $\boldsymbol{u}$ in the line segment between $\boldsymbol{x}$ and $\boldsymbol{x} + t(\boldsymbol{y} - \boldsymbol{x})$, $\left\|\nabla^2 F(\boldsymbol{u})\right\|_{\mathrm{op}} \le L$. Thus,

$$
\begin{aligned}
|F(\boldsymbol{y}) - F(\boldsymbol{x}) - \langle \nabla F(\boldsymbol{x}), \boldsymbol{y} - \boldsymbol{x} \rangle| &= \left| \int_0^1 \langle \nabla F(\boldsymbol{x} + t(\boldsymbol{y} - \boldsymbol{x})), \boldsymbol{y} - \boldsymbol{x} \rangle \mathrm{d}t - \langle \nabla F(\boldsymbol{x}), \boldsymbol{y} - \boldsymbol{x} \rangle \right| \\
&= \left| \int_0^1 \langle \nabla F(\boldsymbol{x} + t(\boldsymbol{y} - \boldsymbol{x})) - \nabla F(\boldsymbol{x}), \boldsymbol{y} - \boldsymbol{x} \rangle \mathrm{d}t \right| \\
&\le \left| \int_0^1 Lt \|\boldsymbol{y} - \boldsymbol{x}\|^2 \mathrm{d}t \right| = \frac{L}{2} \|\boldsymbol{y} - \boldsymbol{x}\|^2.
\end{aligned}
$$

This gives the desired result. $\qquad\square$

Analogously, one can show the following by considering the local second-order approximation around $\boldsymbol{x}$.

**Lemma A.2.** *Suppose $F$ is twice differentiable, and for all $\boldsymbol{u} \in \overline{\boldsymbol{xy}}$ (again the line segment), we have*

$$\left\|\nabla^2 F(\boldsymbol{u}) - \nabla^2 F(\boldsymbol{x})\right\|_{\mathrm{op}} \le L \|\boldsymbol{u} - \boldsymbol{x}\|.$$

*Then,*

$$F(\boldsymbol{y}) \le F(\boldsymbol{x}) + \langle \nabla F(\boldsymbol{x}), \boldsymbol{y} - \boldsymbol{x} \rangle + \frac{1}{2}(\boldsymbol{y} - \boldsymbol{x})^\top \nabla^2 F(\boldsymbol{x})(\boldsymbol{y} - \boldsymbol{x}) + \frac{L}{6} \|\boldsymbol{y} - \boldsymbol{x}\|^3.$$

**Proof.** Similarly to the proof of Lemma A.1, we show this via the proof of Lemma 1 in Nesterov and Polyak (2006). Analogously as in the proof of Lemma A.1, one can readily verify that for any $\boldsymbol{y}' \in \overline{\boldsymbol{xy}}$, $\boldsymbol{x} + t(\boldsymbol{y}' - \boldsymbol{x}) \in \overline{\boldsymbol{xy}}$ holds for all $t \in [0, 1]$. Hence for all $t \in [0, 1]$, applying the condition of this Lemma,

$$\left\|\nabla^2 F(\boldsymbol{x} + t(\boldsymbol{y}' - \boldsymbol{x})) - \nabla^2 F(\boldsymbol{x})\right\|_{\mathrm{op}} \le Lt \|\boldsymbol{y}' - \boldsymbol{x}\|.$$

Thus for any $\boldsymbol{y}' \in \overline{\boldsymbol{xy}}$, by Cauchy-Schwartz and the above, we obtain

$$
\begin{aligned}
\left\|\nabla F(\boldsymbol{y}') - \nabla F(\boldsymbol{x}) - \langle \nabla^2 F(\boldsymbol{x}), \boldsymbol{y}' - \boldsymbol{x} \rangle\right\| &= \left\| \int_0^1 \langle \nabla^2 F(\boldsymbol{x} + t(\boldsymbol{y}' - \boldsymbol{x})), \boldsymbol{y}' - \boldsymbol{x} \rangle \mathrm{d}t - \langle \nabla^2 F(\boldsymbol{x}), \boldsymbol{y}' - \boldsymbol{x} \rangle \right\| \\
&= \left\| \int_0^1 \langle \nabla^2 F(\boldsymbol{x} + t(\boldsymbol{y}' - \boldsymbol{x})) - \nabla^2 F(\boldsymbol{x}), \boldsymbol{y}' - \boldsymbol{x} \rangle \mathrm{d}t \right\| \\
&\le \left| \int_0^1 Lt \|\boldsymbol{y}' - \boldsymbol{x}\|^2 \mathrm{d}t \right| = \frac{L}{2} \|\boldsymbol{y}' - \boldsymbol{x}\|^2.
\end{aligned}
$$

Applying the above relation for $\boldsymbol{y}' = \boldsymbol{x} + t(\boldsymbol{y} - \boldsymbol{x})$ which is in $\overline{\boldsymbol{xy}}$ for all $t \in [0, 1]$, we obtain

$$
\begin{aligned}
&\left| F(\boldsymbol{y}) - F(\boldsymbol{x}) - \langle \nabla F(\boldsymbol{x}), \boldsymbol{y} - \boldsymbol{x} \rangle - \frac{1}{2}\langle \nabla^2 F(\boldsymbol{x})(\boldsymbol{y} - \boldsymbol{x}), \boldsymbol{y} - \boldsymbol{x} \rangle \right| \\
&= \left| \int_0^1 \langle \nabla F(\boldsymbol{x} + t(\boldsymbol{y} - \boldsymbol{x})) - \nabla F(\boldsymbol{x}) - t\nabla^2 F(\boldsymbol{x})(\boldsymbol{y} - \boldsymbol{x}), \boldsymbol{y} - \boldsymbol{x} \rangle \mathrm{d}t \right|
\end{aligned}
$$

$$= \left| \int_0^1 \langle \nabla F(\boldsymbol{x} + t(\boldsymbol{y} - \boldsymbol{x})) - \nabla F(\boldsymbol{x}) - \nabla^2 F(\boldsymbol{x}) \cdot t(\boldsymbol{y} - \boldsymbol{x}), \boldsymbol{y} - \boldsymbol{x} \rangle \mathrm{d}t \right|$$

$$\leq \int_0^1 \|\boldsymbol{y} - \boldsymbol{x}\| \cdot \frac{L}{2} \|t(\boldsymbol{y} - \boldsymbol{x})\|^2 \mathrm{d}t = \frac{L}{6} \|\boldsymbol{y} - \boldsymbol{x}\|^3.$$

This gives the desired result. $\qquad\qquad\square$

We will also use the following Lemmas.

**Lemma A.3.** *For vectors $\boldsymbol{a}$, $\boldsymbol{b}$, the matrix operator norm $\|\boldsymbol{a}\boldsymbol{b}^\top\|_{\mathrm{op}} \leq \|\boldsymbol{a}\|\|\boldsymbol{b}\|$.*

**Proof.** Consider any unit vector $\boldsymbol{x}$. By Cauchy-Schwartz and associativity, we have

$$\boldsymbol{x}^\top (\boldsymbol{a}\boldsymbol{b}^\top)\boldsymbol{x} \leq \langle \boldsymbol{x}, \boldsymbol{a}\rangle\langle \boldsymbol{x}, \boldsymbol{b}\rangle \leq \|\boldsymbol{x}\|^2 \|\boldsymbol{a}\|\|\boldsymbol{b}\| = \|\boldsymbol{a}\|\|\boldsymbol{b}\|.$$

The conclusion follows by definition of operator norm. $\qquad\qquad\square$

**Lemma A.4.** *Consider any non-negative, continuous function $g(x)$ such that $\lim_{x\to\infty} g(x) = \infty$ and such that $g(x) > 0$ on $[1, \infty)$. Then on $[1, \infty)$, $g(x)$ can be lower bounded by a strictly positive, infinitely differentiable, strictly increasing function $\tilde{g}(x)$, where $\tilde{g}$ has domain $[1, \infty)$.*

**Proof.** We will explicitly construct such a $\tilde{g}$ in terms of $g$. First, since $\lim_{x\to\infty} g(x) = \infty$, for all $i \geq 1$, there exists $t_i \in [1, \infty)$ such that $g(x) \geq i + 1$ for all $x \geq t_i$. We furthermore can clearly assume $2 \leq t_1 < t_2 < \cdots$, by increasing each $t_N$ if necessary. Also let $t_0 = 1$. Thus $\bigcup_{i \geq 0}[t_i, t_{i+1})$ forms a disjoint union of $[1, \infty)$.

Now, let $c = \min\left(1, \inf_{x \in [1, t_1]} g(x)\right) > 0$; the strict inequality here holds as $t_1 < \infty$ and as $g$ is continuous. Define a sequence $\{b_i\}_{i \geq 0}$ by $b_0 = c/2, b_1 = c$, and $b_i = i$ for all $i \geq 2$. Thus $b_0 < b_1 < \cdots$. Furthermore, this construction of $\{b_i\}_{i \geq 0}$ implies for all $i \geq 0$, we have $g(x) \geq b_{i+1}$ for all $x \in [t_i, t_{i+1}]$.

Now construct $\tilde{g}(x)$ as follows. For all $i \geq 0$, we let $\tilde{g}(x)$ equal a function $h_i(x)$ defined on $[t_i, t_{i+1}]$ such that $h_i(t_i) = b_i$, $h_i(t_{i+1}) = b_{i+1}$, where we define $h_i$ as follows. We first define $h : [0, 1] \to [0, 1]$ such that $h$ is infinitely differentiable, $h(0) = 0$, $h(1) = 1$, $h^{(n)}(0) = h^{(n)}(1) = 0$ for all $n \geq 1$ where $h^{(n)}$ denotes the $n$-th derivative, and $h'(x) > 0$ for all $x \in (0, 1)$. To this end we use a construction from Chen and Sridharan (2025): let

$$h(x) = \frac{e^{-\frac{1}{x^2}}}{e^{-\frac{1}{x^2}} + e^{-\frac{1}{1-x^2}}} \text{ on } (0, 1),$$

and extend $h$ to $[0, 1]$ by $h(0) = 0, h(1) = 1$. We justify these claims about $h$ shortly below. Now we let

$$h_i(x) = (b_{i+1} - b_i) \cdot h\left(\frac{x - t_i}{t_{i+1} - t_i}\right) + b_i \text{ for all } i \geq 0.$$

We now check $h$ satisfies the claimed properties.

- In Chen and Sridharan (2025), it is argued that $h$ maps to $[0, 1]$, $h(0) = 0$, $h(1) = 1$, and that $h$ is infinitely differentiable. It is also argued in Chen and Sridharan (2025), Lemma 11.5, that $h'(x)$ (which is called $\tilde{p}(x)$ there) is non-negative on $[0, 1]$.

- Next, we check $h^{(n)}(0) = h^{(n)}(1) = 0$ for all $n \geq 1$. Via a straightforward induction outlined in Chen and Sridharan (2025), one can check that $\left(e^{-\frac{1}{x^2}}\right)^{(n)} = 0$, $\left(e^{-\frac{1}{1-x^2}}\right)^{(n)} = 0$ for all $n \geq 1$ (following the standard convention in analysis that $0 \cdot \infty = 0$, see e.g. Folland (1999)). Now let $f(x) = e^{-\frac{1}{x^2}}$, $g(x) = e^{-\frac{1}{x^2}} + e^{-\frac{1}{1-x^2}}$, thus $h = f/g$. Consequently $f^{(n)}(0) = 0$, $f^{(n)}(1) = 0$, $g^{(n)}(0) = 0$, $g^{(n)}(1) = 0$ for all $n \geq 1$. As $g > 0$ always holds in $[0, 1]$ as shown in Chen and Sridharan (2025) and can be easily checked, we have $f = gh$. A straightforward induction gives $f^{(n)} = \sum_{k=0}^n \binom{n}{k} g^{(k)} h^{(n-k)}$ where $\binom{n}{k}$ is the binomial coefficient. We thus obtain $gh^{(n)} = f^{(n)} - \sum_{k=0}^{n-1} \binom{n}{k} g^{(k)} h^{(n-k)}$. For any $n \geq 1$,

taking $x = 0, 1$ in this expression for $h(x)$ and noting at least one of $k, n - k \geq 1$ for $0 \leq k \leq n - 1$ implies $g(0)h^{(n)}(0) = g(1)h^{(n)}(1) = 0$. Recalling $g(x) > 0$ on $[0, 1]$ proves $h^{(n)}(0) = h^{(n)}(1) = 0$ for $n \geq 1$, as requested.

- Finally, we check that $h'(x) > 0$ for all $x \in (0, 1)$. Consider any $x \in (0, 1)$. By a calculation in Lemma 11.5, Chen and Sridharan (2025), we have $h'(x) > 0$ if and only if $q(x) = \frac{2}{x^3}\left(e^{-\frac{1}{x^2}} + e^{-\frac{1}{1-x^2}}\right) + e^{\frac{-1}{x^2}} \cdot \frac{2}{x^3} + e^{-\frac{1}{1-x^2}} \cdot \frac{-2x}{(1-x^2)^2} > 0$. If $x \in [\frac{\sqrt{2}}{2}, 1)$, directly following the proof of Lemma 11.5 in Chen and Sridharan (2025) establishes that $q(x) > 0$. Otherwise if $x \in (0, \frac{\sqrt{2}}{2})$, note the strict inequality $\frac{1}{x^3} > \frac{x}{(1-x^2)^2}$, which in turn implies $q(x) > 0$.

By the above properties of $h$, it follows from the Chain Rule that for all $i \geq 0$, $h_i$ satisfies the following properties:

- $h_i(t_i) = b_i$, $h_i(t_{i+1}) = b_{i+1}$, and $h_i(x) \in [b_i, b_{i+1}]$ for all $x \in [t_i, t_{i+1}]$.

- $h_i$ is infinitely differentiable.

- $h_i'(x) > 0$ for $x \in (t_i, t_{i+1})$, and for all $x \in [t_i, t_{i+1}], h_i'(x) \geq 0$.

- For all $n \geq 1$, $h_i^{(n)}(t_i) = h_i^{(n)}(t_{i+1}) = 0$, where again $h_i^{(n)}$ denotes the $n$-th derivative.

Finally, we check that $\tilde{g}$ has the desired properties:

- $\tilde{g}$ is well-defined: This follows because for all $i \geq 1$, we have $h_i(t_i) = h_{i-1}(t_i) = b_i$.

- $\tilde{g}$ is strictly positive: This follows because $h_i(x) \in [b_i, b_{i+1}] \subseteq (0, \infty)$ for all $x \in [t_i, t_{i+1}]$.

- $\tilde{g}$ is continuous, and moreover is infinitely differentiable: Continuity of $\tilde{g}$ follows because each $h_i$ is infinitely differentiable, and hence continuous, combined with the fact that for all $i \geq 1$, we have $h_i(t_i) = h_{i-1}(t_i) = b_i$. Infinite differentiability of $\tilde{g}$ follows because each $h_i$ is infinitely differentiable, and because for all $n \geq 1$ and all $i \geq 0$, $h_i^{(n)}(t_i) = h_i^{(n)}(t_{i+1}) = 0$.

- $\tilde{g}(x) \leq g(x)$ always holds for $x \in [1, \infty)$: Recall for all $i \geq 0$, we have $g(x) \geq b_{i+1}$ for all $x \in [t_i, t_{i+1}]$. Since we have $\tilde{g}(x) = h_i(x) \leq b_{i+1}$ for all $x \in [t_i, t_{i+1}]$, it follows that for all $x \in [t_i, t_{i+1}]$, $\tilde{g}(x) \leq g(x)$. The result follows upon recalling that $\bigcup_{i \geq 0}[t_i, t_{i+1})$ forms a disjoint union of $[1, \infty)$.

- $\tilde{g}$ is strictly increasing: Consider any $x_1 < x_2, x_1, x_2 \in [1, \infty)$. Since $x_1 < x_2$, and recalling that $\bigcup_{i \geq 0}[t_i, t_{i+1})$ forms a disjoint union of $[1, \infty)$, it follows that for some $j \geq 0$, $(x_1, x_2) \cap (t_j, t_{j+1}) \neq \varnothing$. This intersection is open, and therefore contains some open interval $(a, b) \subseteq (t_j, t_{j+1})$. Let $c' = \inf_{x \in [\frac{2a+b}{3}, \frac{a+2b}{3}]} h_j'(x) > 0$, where the strict inequality follows as $[\frac{2a+b}{3}, \frac{a+2b}{3}] \subseteq (t_j, t_{j+1})$, and by continuity of $h_j'$ on the compact $[\frac{2a+b}{3}, \frac{a+2b}{3}]$. Since we have $h_i'(x) \geq 0$ for all $x \in [t_i, t_{i+1}]$ for all $i \geq 0$, we obtain

$$\tilde{g}(x_2) \geq 0 + c' \cdot \frac{b-a}{3} + \tilde{g}(x_1) > \tilde{g}(x_1).$$

This proves that $\tilde{g}$ is strictly increasing as claimed.

Thus, we have constructed a function $\tilde{g}$ that satisfies the requested properties. $\qquad\square$

## A.2 Comparison of Assumptions with Literature

Here, we establish that our regularity conditions are more general than those of literature.

**Proposition A.1.** *If $\left\|\nabla^2 F(\boldsymbol{w})\right\| \leq l(\nabla F(\boldsymbol{w}))$ for non-decreasing, differentiable sub-quadratic $l$ (where sub-quadratic means that $\lim_{x \to \infty} \frac{l(x)}{x^2} = 0$), then our Assumption 1.1 is satisfied for some non-decreasing $\rho_1(x)$. In this generality, $\rho_1(x)$ depends on $l(x)$, and can be found explicitly from the construction from Lemma A.4.*

*Furthermore, suppose $F$ is $(L_0, L_1)$-smooth, that $\left\|\nabla^2 F(\boldsymbol{w})\right\| \leq L_0 + L_1\|\nabla F(\boldsymbol{w})\|$ for $L_0, L_1 \geq 0$. Then Assumption 1.1 is satisfied with $\rho_1(x) = \frac{3}{2}L_0 + 4L_1^2 x$.*

**Proof.** Essentially this follows from Lemma 3.5, Li et al. (2023a), where it is shown that these assumptions of Zhang et al. (2019), Li et al. (2023a) imply an upper bound on $\|\nabla F(\boldsymbol{w})\|$ in terms of an increasing function of $F(\boldsymbol{w})$; combining with the assumptions of Zhang et al. (2019); Li et al. (2023a) implies that $\|\nabla^2 F(\boldsymbol{w})\|$ is upper bounded in terms of an increasing function of $F(\boldsymbol{w})$.

**Proof for general $l$:** Consider any $\boldsymbol{w} \in \mathbb{R}^d$. By Lemma 3.5 of Li et al. (2023a),

$$\|\nabla F(\boldsymbol{w})\|^2 \le 2\ell(2\|\nabla F(\boldsymbol{w})\|) \cdot F(\boldsymbol{w}).$$

This implies

$$\frac{4\|\nabla F(\boldsymbol{w})\|^2}{\ell(2\|\nabla F(\boldsymbol{w})\|)} \le 8F(\boldsymbol{w}).$$

Let $2\|\nabla F(\boldsymbol{w})\| = t$. Consider when $t \ge 2$. Then the left hand side equals $\frac{t^2}{l(t)}$. Note that WLOG, we can add 1 to $l(\cdot)$ so that $l(t) \ge 1$ for $t \ge 1$. Thus $\frac{t^2}{l(t)}$ is continuous on $[1, \infty)$, and furthermore is positive on this interval. Now note $\lim_{x \to \infty} \frac{x^2}{l(x)} = \infty$ by the condition (including after adding 1 WLOG), and thus by Lemma A.4, $\frac{x^2}{l(x)}$ is lower bounded by some strictly increasing function $\tilde{g}(x)$ on $[2, \infty)$. Therefore, $\tilde{g}$ is invertible and so we have

$$\tilde{g}(2\|\nabla F(\boldsymbol{w})\|) \le \frac{4\|\nabla F(\boldsymbol{w})\|^2}{\ell(2\|\nabla F(\boldsymbol{w})\|)} \le 8F(\boldsymbol{w}) \implies \|\nabla F(\boldsymbol{w})\| \le \frac{1}{2}\tilde{g}^{-1}(8F(\boldsymbol{w})).$$

Then by the assumptions of Li et al. (2023b), it holds that

$$\left\|\nabla^2 F(\boldsymbol{w})\right\| \le l\left(\frac{1}{2}\tilde{g}^{-1}(8F(\boldsymbol{w}))\right).$$

Else when $t < 2$, we have $\|\nabla F(\boldsymbol{w})\| \le 1$, and by the assumptions of Li et al. (2023b), we have $\left\|\nabla^2 F(\boldsymbol{w})\right\| \le l(1)$.

Thus the assumptions of Li et al. (2023b) imply that the following always holds:

$$\left\|\nabla^2 F(\boldsymbol{w})\right\| \le l\left(\frac{1}{2}\tilde{g}^{-1}(8F(\boldsymbol{w}))\right) + l(1).$$

We thus can take $\rho_1(x) = l\left(\frac{1}{2}\tilde{g}^{-1}(8x)\right) + l(1)$, which is clearly non-negative. It remains to check that $l\left(\frac{1}{2}\tilde{g}^{-1}(8x)\right)$ is non-decreasing. As $l$ is non-decreasing, as compositions of non-decreasing functions are non-decreasing, it remains to check that $\frac{1}{2}\tilde{g}^{-1}(8x)$ is non-decreasing. Since $\tilde{g}$ is non-decreasing, $\tilde{g}^{-1}$ is non-decreasing as well, and this completes the proof.

**Proof for $(L_0, L_1)$-smoothness:** First, when $L_1 = 0$ the result is immediate, so from here on out suppose $L_1 > 0$. By Lemma 3.5 from Li et al. (2023a) we have for all $\boldsymbol{w} \in \mathbb{R}^d$,

$$\|\nabla F(\boldsymbol{w})\|^2 \le 2\ell(2\|\nabla F(\boldsymbol{w})\|) \cdot F(\boldsymbol{w}),$$

where $\ell(x) = L_0 + L_1(x)$ for $L_0, L_1 \ge 0$. We thus obtain:

$$\|\nabla F(\boldsymbol{w})\|^2 \le 2(L_0 + 2L_1\|\nabla F(\boldsymbol{w})\|) \cdot F(\boldsymbol{w})$$
$$= 2L_0 F(\boldsymbol{w}) + 4L_1\|\nabla F(\boldsymbol{w})\|F(\boldsymbol{w}).$$

Rewriting this inequality, we get

$$\|\nabla F(\boldsymbol{w})\|^2 - 4L_1\|\nabla F(\boldsymbol{w})\|F(\boldsymbol{w}) - 2L_0 F(\boldsymbol{w}) \le 0.$$

Consider the quadratic $x^2 - 4L_1 F(\boldsymbol{w}) \cdot x - 2L_0 F(\boldsymbol{w})$. The coefficient on the quadratic term is positive, and the quadratic is non-negative when $x = \|\nabla F(\boldsymbol{w})\|$. Thus $\|\nabla F(\boldsymbol{w})\|$ must be no larger than the largest root of $x^2 - 4L_1 F(\boldsymbol{w}) \cdot x - 2L_0 F(\boldsymbol{w})$, and we obtain

$$\|\nabla F(\boldsymbol{w})\| \le \frac{1}{2}\left(4L_1 F(\boldsymbol{w}) + \sqrt{16L_1^2 F(\boldsymbol{w})^2 + 8L_0 F(\boldsymbol{w})}\right)$$

$$\le 2L_1 F(\boldsymbol{w}) + \sqrt{(2L_1 F(\boldsymbol{w}))^2 + 2L_0 F(\boldsymbol{w})} \qquad (5)$$

If $F(\boldsymbol{w}) = 0$, the above immediately implies $\|\nabla F(\boldsymbol{w})\| = 0$. Otherwise, recall by shifting (in Notation) that $F(\boldsymbol{w}) \geq 0$ always holds, so suppose $F(\boldsymbol{w}) > 0$. Recall also from earlier that it suffices to show the result for $L_1 > 0$. Applying the inequality $\sqrt{a^2 + b} \leq a + \frac{b}{2a}$, valid for all $a > 0, b \geq 0$ with $a = 2L_1 F(\boldsymbol{w}) > 0$, $b = 2L_0 F(\boldsymbol{w}) \geq 0$, we obtain

$$\sqrt{(2L_1 F(\boldsymbol{w}))^2 + 2L_0 F(\boldsymbol{w})} \ \leq \ 2L_1 F(\boldsymbol{w}) + \frac{L_0}{2L_1}.$$

Substituting into (5) gives that for all $\boldsymbol{w}$ with $F(\boldsymbol{w}) > 0$, we have

$$\|\nabla F(\boldsymbol{w})\| \leq \frac{L_0}{2L_1} + 4L_1 F(\boldsymbol{w}). \tag{6}$$

By the argument earlier, if $F(\boldsymbol{w}) = 0$, the above bound (6) holds too. Thus (6) holds for all $\boldsymbol{w} \in \mathbb{R}^d$. Now inserting (6) into the definition of $(L_0, L_1)$-smoothness gives

$$\left\|\nabla^2 F(\boldsymbol{w})\right\| \ \leq \ L_0 + L_1\left(\frac{L_0}{2L_1} + 4L_1 F(\boldsymbol{w})\right) = \tfrac{3}{2}L_0 + 4L_1^2 F(\boldsymbol{w}).$$

Hence Assumption 1.1 is satisfied with the increasing function $\rho_1(x) = \tfrac{3}{2}L_0 + 4L_1^2 x$. $\qquad\square$

**Proposition A.2.** *When $F$ is $(L_0, L_1)$-smooth, letting $\rho_0(x) = 2L_0^{1/2}x^{1/2} + \frac{5L_1^2}{L_0^{1/2}}x^{3/2}$, we have* $\|\nabla F(\boldsymbol{w})\| \leq \rho_0(F(\boldsymbol{w}))$.

**Proof.** By Proposition A.1, we can take $\rho_1(x) = \tfrac{3}{2}L_0 + 4L_1^2 x$ in this case. As noted in Subsection 3.1, we need to show that $2L_0^{1/2}x^{1/2} + \frac{5L_1^2}{L_0^{1/2}}x^{3/2}$ is a pointwise upper bound on

$$\rho_1(x)\sqrt{2\theta(x)} \text{ where } \theta(x) = \int_0^x \frac{1}{\rho_1(v)}\mathrm{d}v.$$

To this end note for each $x \geq 0$ that $\theta(x) \leq x \cdot \frac{1}{\frac{3}{2}L_0} = \frac{2}{3L_0}x$, thus for each $x \geq 0$,

$$\rho_1(x)\sqrt{2\theta(x)} \leq \left(\frac{3}{2}L_0 + 4L_1^2 x\right)\sqrt{\frac{4}{3L_0}x} \leq 2L_0^{1/2}x^{1/2} + \frac{5L_1^2}{L_0^{1/2}}x^{3/2}.$$

This completes the proof. $\qquad\square$

**Example 1.** We now provide a natural example of a univariate function that satisfies our regularity assumptions but does not necessarily satisfy those of Li et al. (2023b) for non-convex optimization. Namely, consider the univariate function:

$$F(x) = 1 - \log(\cos(1 + x)), 0 \leq x < \frac{\pi}{2} - 1.$$

The argument here is in radians. The first derivative is:

$$F'(x) = \tan(1 + x).$$

The second derivative is:

$$F''(x) = \sec^2(1 + x).$$

Thus as $\tan^2(\theta) + 1 = \sec^2(\theta)$, $F$ satisfies the ODE:

$$F''(x) = F'(x)^2 + 1. \tag{7}$$

Suppose that $F$ satisfied the conditions of Li et al. (2023b) for non-convex optimization on the relevant domain, thus for all $0 \leq x < \frac{\pi}{2} - 1$, we would have

$$F''(x) \leq \ell(F'(x)),$$

for some sub-quadratic $l(\cdot)$.

Then by (7) and noting $F'(x) > 0$ on the domain, we obtain for all $0 \le x < \frac{\pi}{2} - 1$

$$1 \le 1 + \frac{1}{F'(x)^2} = \frac{F'(x)^2 + 1}{F'(x)^2} = \frac{F''(x)}{F'(x)^2} \le \frac{\ell(F'(x))}{F'(x)^2}.$$

As $l$ is subquadratic, there exists $x' < \infty$ such that $l(x)/x^2 < 1$ for all $x > x'$. Noting $F'(x) \to \infty$ for $x \to \frac{\pi}{2} - 1$ yields a contradiction.

Consequently $F$ does not satisfy the conditions of Li et al. (2023b) for non-convex optimization. However, we show that $F$ satisfies Assumption 1.1. Rewriting $F''(x)$ in terms of $F(x)$, note that:

$$\cos(1 + x) = e^{1 - F(x)},$$

and thus:

$$F''(x) = \sec^2(1 + x) = \frac{1}{\cos^2(1 + x)} = e^{2(F(x) - 1)}.$$

Hence we can define the increasing, non-negative function

$$\rho_1(t) = e^{2(t-1)},$$

which satisfies:

$$F''(x) \le \rho_1(F(x)).$$

Thus $F$ satisfies Assumption 1.1 (in the relevant domain).

We now discuss Assumption 1.2.

**Example 2.** First, we show that Assumption 1.2 captures several univariate functions of interest. Notice also if $F(\boldsymbol{w})$ is a sum of functions satisfying Assumption 1.2, Triangle Inequality implies that $F(\boldsymbol{w})$ also satisfies Assumption 1.2.

- Polynomials: Consider whenever $F(x)$ is a linear combination of monomials $x^p$ for $p \ge 1$, combined with a constant term. We claim $F(x)$ satisfies Assumption 1.2. By linearity of derivative and Triangle Inequality, it suffices to prove this whenever $F(x) = x^p$ for $p \ge 1$ as the constant term vanishes, and then add up all the non-decreasing, non-negative functions on the right hand side to form $\rho_1$ and $\rho_2$. To this end note $F''(x) = p(p-1)x^{p-2}$, thus

$$|F''(x)| = p(p-1)x^{p-2} \le p(p-1)(x^p + 1) = p(p-1)(F(x) + 1).$$

Similarly, $F'''(x) = p(p-1)(p-2)x^{p-3}$, thus

$$|F'''(x)| = p(p-1)(p-2)x^{p-3} \le p(p-1)(p-2)(1 + F(x)).$$

Noting $p(p-1)(1+t)$ and $p(p-1)(p-2)(1+t)$ are non-decreasing and non-negative for $t \ge 0$, combined with our earlier remarks that it suffices to prove this result when $F(x) = x^p$, completes the proof.

- Single-exponential functions: Consider when $F(x) = a^x = e^{x \ln a}$ for $a > 1$. Then $F''(x) = (\ln a)^2 e^{x \ln a}$, $F'''(x) = (\ln a)^3 e^{x \ln a}$, and so we can take $\rho_1(t) = (\ln a)^2 t, \rho_2(t) = (\ln a)^3 t$.

- Doubly-exponential functions: Consider when $F(x) = a^{b^x} = e^{\ln a e^{x \ln b}}$ for $a, b > 1$. Thus

$$F'(x) = e^{\ln a e^{x \ln b}} \cdot \ln a e^{x \ln b} \cdot \ln b = \ln a \ln b F(x) e^{x \ln b}.$$

It follows that

$$F''(x) = \ln a \ln b \big(F'(x) e^{x \ln b} + \ln b F(x) e^{x \ln b}\big) = (\ln a)(\ln b)^2 F(x)(e^{2x \ln b} \ln a + e^{x \ln b}).$$

This then implies

$$
\begin{aligned}
F'''(x) &= (\ln a)(\ln b)^2 F(x)(e^{2x \ln b} 2 \ln a \ln b + e^{x \ln b} \ln b) \\
&\quad + (\ln a)(\ln b)^2 (e^{2x \ln b} \ln a + e^{x \ln b}) \ln a \ln b F(x) e^{x \ln b} \\
&= (\ln a)(\ln b)^3 F(x)\big(2e^{2x \ln b} \ln a + e^{x \ln b} + e^{3x \ln b}(\ln a)^2 + e^{2x \ln b} \ln a\big).
\end{aligned}
$$

Notice

$$e^{x \ln b} \le e^{\ln a e^{x \ln b}} - 1 < F(x),$$

therefore we have

$$F''(x) \le (\ln a)(\ln b)^2 F(x)\big(F(x)^2 \ln a + F(x)\big),$$
$$F'''(x) \le (\ln a)(\ln b)^3 F(x)\big(F(x)^3(\ln a)^2 + 3F(x)^2 \ln a + F(x)\big).$$

We thus can take

$$\rho_1(t) = (\ln a)(\ln b)^2 t\big(t^2 \ln a + t\big),$$
$$\rho_2(t) = (\ln a)(\ln b)^3 t(t^3(\ln a)^2 + 3t^2 \ln a + t),$$

which are clearly non-negative and non-decreasing on $[0, \infty)$.

- Next we highlight the natural example of any self-concordant function $F : \mathbb{R} \to \mathbb{R}$. Thus

$$|F'''(x)| \le 2F''(x)^{3/2} \le 2|F''(x)|^{3/2}.$$

Suppose $F$ satisfies Assumption 1.1. Then there exists a non-negative, non-decreasing $\rho_1$ such that $|F''(x)| \le \rho_1(F(x))$. Thus,

$$|F'''(x)| \le 2\rho_1(F(x))^{3/2}.$$

Since $\rho_1$ is non-negative and non-decreasing, $\rho_2(t) := 2\rho_1(t)^{3/2}$ is as well, and thus Assumption 1.2 is satisfied.

Next, we show that the regularity assumptions Assumptions 1 and 3 of Xie et al. (2024), which they need for their guarantees finding SOSPs, are less general than Assumption 1.2 when $F$ is twice-differentiable. To do so we show they imply Assumption 1.2, and are hence subsumed by Assumption 1.2.

When $F$ is twice-differentiable, their Assumption 1 implies $(L_0, L_1)$-smoothness. As shown in Proposition A.2, this means that

$$\|\nabla F(\boldsymbol{w})\| \le \rho_0(F(\boldsymbol{w})) \text{ where } \rho_0(x) = 2L_0^{1/2}x^{1/2} + \frac{5L_1^2}{L_0^{1/2}}x^{3/2}.$$

Their Assumption 3 implies for $M_0, M_1 \ge 0$ and some $\delta > 0$ that for all $\boldsymbol{w}, \boldsymbol{w}'$ with $\|\boldsymbol{w} - \boldsymbol{w}'\| \le \delta$,

$$\left\|\nabla^2 F(\boldsymbol{w}) - \nabla^2 F(\boldsymbol{w}')\right\|_{\mathrm{op}} \le \|\boldsymbol{w} - \boldsymbol{w}'\|(M_0 + M_1\|\nabla F(\boldsymbol{w})\|).$$

Combining this with the earlier display gives for all $\boldsymbol{w}, \boldsymbol{w}'$ with $\|\boldsymbol{w} - \boldsymbol{w}'\| \le \delta$,

$$\left\|\nabla^2 F(\boldsymbol{w}) - \nabla^2 F(\boldsymbol{w}')\right\|_{\mathrm{op}} \le \|\boldsymbol{w} - \boldsymbol{w}'\|(M_0 + M_1\rho_0(F(\boldsymbol{w}))),$$

where $\rho_0(x) = 2L_0^{1/2}x^{1/2} + \frac{5L_1^2}{L_0^{1/2}}x^{3/2}$. We thus see that $F$ satisfies Assumption 1.2 with the non-decreasing, non-negative function $\rho_2(x) = M_0 + M_1\left(2L_0^{1/2}x^{1/2} + \frac{5L_1^2}{L_0^{1/2}}x^{3/2}\right)$, where the latter two properties are evident as $\rho_0(\cdot)$ is non-decreasing and non-negative.

### A.3 Proofs of Technical Results

Now, we prove general results used throughout our work. We prove Corollary 1, which gives us control over the gradient:

**Proof of Corollary 1.** Applying Lemma 11, De Sa et al. (2022) with $\Phi$ in place of $F$, we obtain

$$\|\nabla F(\boldsymbol{w})\| \le \rho(F(\boldsymbol{w}))\sqrt{2\theta(F(\boldsymbol{w}))} = \rho_0(F(\boldsymbol{w})),$$

where $\theta(\cdot)$ is defined as in the statement of Corollary 1. To prove $\rho_0(x)$ is increasing, simply note $\theta$ and thus $\sqrt{\theta}$ are clearly increasing, and are both non-negative. $\rho_1$ is non-decreasing and non-negative as well, thus $\rho_0$ is non-decreasing and non-negative. $\square$

We also prove the central Lemma 3.1, which is very important to our results: it lets us control the change in function value under our regularity assumptions. We first state the following Lemma from Li et al. (2023a), a generalization of Gronwall's Inequality:

**Lemma A.5** (Lemma A.3, Li et al. (2023a))**.** *Let $\alpha : [a,b] \to [0,\infty)$ and $\beta : [0,\infty) \to [0,\infty)$ be two continuous functions. Suppose $\alpha'(t) \leq \beta(\alpha(t))$ almost everywhere over $(a,b)$. Let $\phi(u) = \int_0^u \frac{1}{\beta(v)} dv$. Then for all all $t \in [a,b]$,*

$$\phi(\alpha(t)) \leq \phi(\alpha(a)) - a + t.$$

This allows us to prove Lemma 3.1, which is an extension of Lemma A.4, Li et al. (2023a):

**Proof of Lemma 3.1.** The proof is essentially identical to the proof of Lemma A.4, Li et al. (2023a). Let $\boldsymbol{z}(t) = (1-t)\boldsymbol{x} + t\boldsymbol{y}$, $\alpha(t) = F(\boldsymbol{z}(t))$. Then for all $t \in (0,1)$, we obtain

$$
\begin{aligned}
\alpha'(t) &= \lim_{s \to t} \frac{\alpha(s) - \alpha(t)}{s - t} \\
&\leq \lim_{s \to t} \frac{|F(\boldsymbol{z}(s)) - F(\boldsymbol{z}(t))|}{s - t} \\
&= \left| \lim_{s \to t} \frac{F(\boldsymbol{z}(s)) - F(\boldsymbol{z}(t))}{s - t} \right| \\
&= \left| \frac{\mathrm{d}}{\mathrm{d}t} F(\boldsymbol{z}(t)) \right| \\
&= \left| \nabla F(\boldsymbol{z}(t))^\top (\boldsymbol{y} - \boldsymbol{x}) \right| \\
&\leq \rho_0(F(\boldsymbol{z}(t))) \|\boldsymbol{y} - \boldsymbol{x}\|,
\end{aligned}
$$

the last step using $\|\nabla F(\boldsymbol{w})\| \leq \rho_0(F(\boldsymbol{w}))$. Let $\beta(x) = \|\boldsymbol{y} - \boldsymbol{x}\|\rho_0(x)$ and let $\phi(u) = \int_0^u \frac{1}{\beta(v)} dv$. Thus, $\alpha'(t) \leq \beta(\alpha(t))$ almost everywhere. Applying Lemma A.5 gives

$$\phi(F(\boldsymbol{y})) = \phi(\alpha(1)) \leq \phi(\alpha(0)) + 1 = \phi(F(\boldsymbol{x})) + 1.$$

Let $\psi(u) = \|\boldsymbol{y} - \boldsymbol{x}\|\phi(u) = \int_0^u \frac{1}{\rho_0(v)} dv$, which is clearly strictly increasing. Consequently we obtain from the above and assumption on $\boldsymbol{y}$ that

$$
\begin{aligned}
\psi(F(\boldsymbol{y})) &\leq \psi(F(\boldsymbol{x})) + \|\boldsymbol{y} - \boldsymbol{x}\| \\
&\leq \psi(F(\boldsymbol{x})) + \frac{1}{\rho_0(F(\boldsymbol{x}) + 1)} \\
&\leq \int_0^{F(\boldsymbol{x})} \frac{1}{\rho_0(v)} dv + \int_{F(\boldsymbol{x})}^{F(\boldsymbol{x})+1} \frac{1}{\rho_0(v)} dv \\
&= \int_0^{F(\boldsymbol{x})+1} \frac{1}{\rho_0(v)} dv = \psi(F(\boldsymbol{x}) + 1).
\end{aligned}
$$

Since $\psi$ is strictly increasing, taking inverses implies

$$F(\boldsymbol{y}) \leq F(\boldsymbol{x}) + 1,$$

as desired. $\qquad\square$

We also introduce the following Lemma, which lets us exploit Assumption 1.2 to control the Lipschitz constant of the Hessian of $F$.

**Lemma A.6.** *Suppose $F$ satisfies Assumption 1.2. Suppose $\boldsymbol{x}, \boldsymbol{y} \in \mathbb{R}^d$ are such that $\|\boldsymbol{y} - \boldsymbol{x}\| \leq r$ for some $r > 0$. Then*

$$\left\| \nabla^2 F(\boldsymbol{x}) - \nabla^2 F(\boldsymbol{y}) \right\|_{\mathrm{op}} \leq \|\boldsymbol{x} - \boldsymbol{y}\| \cdot \sup_{\boldsymbol{u} \in \overline{\boldsymbol{x}\boldsymbol{y}}} \rho_2(F(\boldsymbol{u})).$$

*In particular, we have*

$$\left\| \nabla^2 F(\boldsymbol{x}) - \nabla^2 F(\boldsymbol{y}) \right\|_{\mathrm{op}} \leq \|\boldsymbol{x} - \boldsymbol{y}\| \cdot \sup_{\boldsymbol{u} \in \mathbb{B}(\boldsymbol{y},r)} \rho_2(F(\boldsymbol{u})).$$

**Proof.** Consider $\delta > 0$, either from Assumption 1.2 if the second case of Assumption 1.2 holds, and otherwise set to some arbitrary positive real. Similar to the proof of Lemma 3.1, divide the line segment between $\boldsymbol{x}, \boldsymbol{y}$ into $N = \frac{\|\boldsymbol{x} - \boldsymbol{y}\|}{\delta}$ equally spaced segments of length $\delta$ between points $\boldsymbol{x}_i$, where we define $\boldsymbol{x}_0 = \boldsymbol{x}, \boldsymbol{x}_1, \ldots, \boldsymbol{x}_{N-1}, \boldsymbol{x}_N = \boldsymbol{y}$. Thus $\|\boldsymbol{x} - \boldsymbol{y}\| = N\delta$.

Suppose for all $\boldsymbol{u} \in \overline{\boldsymbol{xy}}$ we have $\left\|\nabla^3 F(\boldsymbol{u})\right\|_{\mathrm{op}} \le L$. Consider any $\boldsymbol{x}', \boldsymbol{y}'$ in the line segment $\overline{\boldsymbol{xy}}$. Applying this for $\boldsymbol{x}' + t(\boldsymbol{y}' - \boldsymbol{x}')$ for $t \in [0, 1]$, which always lies in the line segment $\overline{\boldsymbol{xy}}$, we obtain

$$\left\|\nabla^2 F(\boldsymbol{y}') - \nabla^2 F(\boldsymbol{x}')\right\|_{\mathrm{op}} \le \left\|\int_0^1 \langle \nabla^3 F(\boldsymbol{x}' + t(\boldsymbol{y}' - \boldsymbol{x}')), \boldsymbol{y}' - \boldsymbol{x}' \rangle \mathrm{d}t\right\| \le L\|\boldsymbol{y}' - \boldsymbol{x}'\|.$$

Consequently irrespective of which case of Assumption 1.2 holds, because $\|\boldsymbol{x}_i - \boldsymbol{x}_{i-1}\| \le \delta$, we have for each $i, 1 \le i \le N$ that

$$\left\|\nabla^2 F(\boldsymbol{x}_i) - \nabla^2 F(\boldsymbol{x}_{i-1})\right\|_{\mathrm{op}} \le \|\boldsymbol{x}_i - \boldsymbol{x}_{i-1}\| \sup_{\boldsymbol{u} \in \overline{\boldsymbol{xy}}} \rho_2(F(\boldsymbol{u})).$$

Now Triangle Inequality gives

$$\begin{aligned}
\left\|\nabla^2 F(\boldsymbol{x}) - \nabla^2 F(\boldsymbol{y})\right\|_{\mathrm{op}} &\le \sum_{i=1}^N \left\|\nabla^2 F(\boldsymbol{x}_i) - \nabla^2 F(\boldsymbol{x}_{i-1})\right\|_{\mathrm{op}} \\
&\le \sum_{i=1}^N \|\boldsymbol{x}_i - \boldsymbol{x}_{i-1}\| \sup_{\boldsymbol{u} \in \overline{\boldsymbol{xy}}} \rho_2(F(\boldsymbol{u})) \\
&\le N\delta \cdot \sup_{\boldsymbol{u} \in \overline{\boldsymbol{xy}}} \rho_2(F(\boldsymbol{u})) \\
&= \|\boldsymbol{x} - \boldsymbol{y}\| \sup_{\boldsymbol{u} \in \overline{\boldsymbol{xy}}} \rho_2(F(\boldsymbol{u})),
\end{aligned}$$

as desired. $\qquad\square$

We will also generalize the proof of Theorem 3.1 to show that GD, when initialized in the $F(\boldsymbol{w}_0)$-sublevel set $\mathcal{L}_{F,F(\boldsymbol{w}_0)}$ with appropriate step size defined in terms of $F(\boldsymbol{w}_0)$, never increases function value.

**Lemma A.7.** *Consider any $\boldsymbol{w}_0 \in \mathbb{R}^d$, and consider iterates $\{\boldsymbol{u}_t\}_{t \ge 0}$ of GD initialized at any $\boldsymbol{u}_0 \in \mathcal{L}_{F,F(\boldsymbol{w}_0)}$, the $F(\boldsymbol{w}_0)$-sublevel set. If the step size $\eta$ of GD is at most $\frac{1}{L_1(\boldsymbol{w}_0)}$ where $L_1(\cdot)$ is defined as per (4), then $F(\boldsymbol{u}_t) \le F(\boldsymbol{u}_0)$ for all $t \ge 0$.*

**Proof.** It suffices to prove this for $t = 1$; a simple inductive argument then establishes this for all $t \ge 0$. We have $\boldsymbol{u}_1 = \boldsymbol{u}_0 - \eta \nabla F(\boldsymbol{u}_0)$. By Corollary 1 and because $\boldsymbol{u}_0 \in \mathcal{L}_{F,F(\boldsymbol{w}_0)}$, $\|\nabla F(\boldsymbol{u}_0)\| \le \rho_0(F(\boldsymbol{u}_0)) \le \rho_0(F(\boldsymbol{w}_0))$. Thus by choice of $\eta$ and definition of $L_1(\boldsymbol{w}_0)$,

$$\|\boldsymbol{u}_1 - \boldsymbol{u}_0\| = \eta\|\nabla F(\boldsymbol{u}_0)\| \le \eta\rho_0(F(\boldsymbol{w}_0)) \le \frac{1}{\rho_0(F(\boldsymbol{w}_0) + 1)}.$$

By Lemma 3.2, because $\boldsymbol{u}_0 \in \mathcal{L}_{F,F(\boldsymbol{w}_0)}$, for all $\boldsymbol{p}$ in the line segment $\overline{\boldsymbol{u}_0 \boldsymbol{u}_1}$ we have $\left\|\nabla^2 F(\boldsymbol{p})\right\|_{\mathrm{op}} \le L_1(\boldsymbol{w}_0)$. By Lemma A.1, it follows that

$$\begin{aligned}
F(\boldsymbol{u}_1) &\le F(\boldsymbol{u}_0) - \eta\|\nabla F(\boldsymbol{u}_0)\|^2 + \frac{L_1(\boldsymbol{w}_0)\eta^2}{2} \cdot \|\nabla F(\boldsymbol{u}_0)\|^2 \\
&\le F(\boldsymbol{u}_0) + \|\nabla F(\boldsymbol{u}_0)\|^2 \cdot \left(-\eta + \frac{L_1(\boldsymbol{w}_0)\eta^2}{2}\right).
\end{aligned}$$

Noting $-\eta + \frac{L_1(\boldsymbol{w}_0)\eta^2}{2} \le 0$ for $\eta \in \left[0, \frac{2}{L_1(\boldsymbol{w}_0)}\right]$, the conclusion follows. $\qquad\square$

# B   Proof of Framework

**Proof of Theorem 2.1.** For convenience, for all $n \ge 0$, define $p_n := 1 - n \cdot \sup_{\boldsymbol{u} \in \mathcal{L}_{F,F(\boldsymbol{w}_0)}} \delta(\boldsymbol{u})$. Also let $T = \sup_{\boldsymbol{u} \in \mathcal{L}_{F,F(\boldsymbol{w}_0)}} \left\{\frac{F(\boldsymbol{w}_0)}{\Delta(\boldsymbol{u})}\right\}$.

**Lemma B.1.** *For any $n \ge 0$, let $\mathcal{E}_n$ be the event that the sequence of iterates $(\boldsymbol{w}_t)_{0 \le t \le n-1}$ satisfies either:*

    *1. The event $\mathcal{E}_{n,1}$: For all $0 \le t \le n - 1$, $F(\mathcal{A}_1(\boldsymbol{w}_t)) < F(\boldsymbol{w}_t) - \Delta(\boldsymbol{w}_t)$.*

2. *The event $\mathcal{E}_{n,2}$: There exists $\boldsymbol{w}_t \in (\boldsymbol{w}_t)_{0 \le t \le n-1}$ such that $\mathcal{A}_2(\boldsymbol{w}_t) \cap \mathcal{S} \ne \{\}$, and for all $\boldsymbol{w}_s$ with $0 \le s < t$, we have $F(\mathcal{A}_1(\boldsymbol{w}_s)) < F(\boldsymbol{w}_s) - \Delta(\boldsymbol{w}_s)$.*

*That is, $\mathcal{E}_n = \mathcal{E}_{n,1} \cup \mathcal{E}_{n,2}$. Then over the randomness in $\mathcal{A}$, we have $\mathbb{P}(\mathcal{E}_n) \ge p_n$ for all $n \ge 0$.*

**Proof.** We proceed by induction on $n$. The base case $n = 0$ is vacuously evident, and the case $n = 1$ follows immediately by the definition of a decrease procedure from Definition 2.2 and hypotheses of Theorem 2.1.

For the inductive step, suppose Lemma B.1 is true for some $n \ge 1$; we show it is for $n + 1$. By the inductive hypothesis, we know that $\mathbb{P}(\mathcal{E}_n) \ge p_n$. We aim to show $\mathbb{P}(\mathcal{E}_n) \ge p_{n+1}$. If $p_n \le 0$ there is nothing to prove, so suppose now that $n$ is such that $p_n > 0$.

1. Let $p = \mathbb{P}(\mathcal{E}_{n,2}|\mathcal{E}_n)$. Note $\mathcal{E}_{n,2} \subseteq \mathcal{E}_{n+1,2} \subseteq \mathcal{E}_{n+1}$.

2. Let $\mathcal{B} := \mathcal{E}_{n,1} \cap \mathcal{E}_{n,2}^c$. Thus, if $\mathcal{B}$ occurs, then all the $(\boldsymbol{w}_t)_{0 \le t \le n-1}$ are such that $F(\mathcal{A}_1(\boldsymbol{w}_t)) < F(\boldsymbol{w}_t) - \Delta(\boldsymbol{w}_t)$, but $\mathcal{E}_{n,2}$ did not occur. Note $\mathcal{E}_n$ is the disjoint union $\mathcal{E}_{n,2} \sqcup \mathcal{B}$, so $\mathbb{P}(\mathcal{B}|\mathcal{E}_n) = 1 - p$.

   Under $\mathcal{B}$, we know $\boldsymbol{w}_n = \mathcal{A}(\boldsymbol{w}_{n-1})$ is such that $F(\boldsymbol{w}_n) \le F(\boldsymbol{w}_0)$. Hence $\boldsymbol{w}_n \in \mathcal{L}_{F,F(\boldsymbol{w}_0)}$. Therefore, conditioned on $\mathcal{B}$, by the hypotheses of Theorem 2.1 we have with probability at least $p_0$ that either $F(\mathcal{A}_1(\boldsymbol{w}_n)) < F(\boldsymbol{w}_n) - \Delta(\boldsymbol{w}_n)$ or $\mathcal{A}_2(\boldsymbol{w}_n) \cap \mathcal{S} \ne \{\}$.

   Let $\mathcal{C}$ be the event that $F(\mathcal{A}_1(\boldsymbol{w}_n)) < F(\boldsymbol{w}_n) - \Delta(\boldsymbol{w}_n)$ occurs. Let $\mathcal{D}$ be the event that $\mathcal{A}_2(\boldsymbol{w}_n) \cap \mathcal{S} \ne \{\}$ occurs but $\mathcal{C}$ does not occur. Recall that $\boldsymbol{w}_n \in \mathcal{L}_{F,F(\boldsymbol{w}_0)}$ conditioned on $\mathcal{B}$. Furthermore recall that $\mathcal{A}(\boldsymbol{w}_n)$ is only a function of $\boldsymbol{w}_n$, and none of the $(\boldsymbol{w}_t)_{0 \le t \le n-1}$. Thus the definition of decrease procedure, Definition 2.2, implies that

   $$\mathbb{P}(\mathcal{C} \sqcup \mathcal{D}|\mathcal{B}) \ge p_0.$$

Now since $\mathbb{P}(\mathcal{B}) = \mathbb{P}(\mathcal{B}|\mathcal{E}_n)\mathbb{P}(\mathcal{E}_n) \ge (1-p)p_n > 0$, Bayes' Rule implies

$$\mathbb{P}((\mathcal{B} \cap \mathcal{C}) \sqcup (\mathcal{B} \cap \mathcal{D})|\mathcal{B}) = \frac{\mathbb{P}(\mathcal{B} \cap ((\mathcal{B} \cap \mathcal{C}) \sqcup (\mathcal{B} \cap \mathcal{D})))}{\mathbb{P}(\mathcal{B})}$$
$$= \frac{\mathbb{P}(\mathcal{B} \cap (\mathcal{C} \sqcup \mathcal{D}))}{\mathbb{P}(\mathcal{B})} = \mathbb{P}(\mathcal{C} \sqcup \mathcal{D}|\mathcal{B}) \ge p_0.$$

Note $\mathcal{B} \cap \mathcal{C}$ implies that $\mathcal{E}_{n+1,1}$ occurs, since under $\mathcal{B} \cap \mathcal{C}$ we have $F(\mathcal{A}_1(\boldsymbol{w}_t)) < F(\boldsymbol{w}_t) - \Delta(\boldsymbol{w}_t)$ for all $0 \le t \le n$. Similarly, $\mathcal{B} \cap \mathcal{D}$ implies that $\mathcal{E}_{n+1,2}$ occurs, since under $\mathcal{B} \cap \mathcal{D}$ we have $F(\mathcal{A}_1(\boldsymbol{w}_t)) < F(\boldsymbol{w}_t) - \Delta(\boldsymbol{w}_t)$ for $0 \le t \le n-1$ and $\mathcal{A}_2(\boldsymbol{w}_n) \cap \mathcal{S} \ne \{\}$.

Thus recalling $\mathcal{E}_{n,2}, \mathcal{B}$ are disjoint, we see that $\mathcal{E}_{n+1}$ contains the following disjoint union of events:

$$\mathcal{E}_{n+1} \supseteq \mathcal{E}_{n,2} \sqcup (\mathcal{B} \cap \mathcal{C}) \sqcup (\mathcal{B} \cap \mathcal{D}).$$

The above observations imply via Bayes' Rule that

$$\begin{aligned}
\mathbb{P}(\mathcal{E}_{n+1}) &\ge \mathbb{P}(\mathcal{E}_{n,2} \sqcup (\mathcal{B} \cap \mathcal{C}) \sqcup (\mathcal{B} \cap \mathcal{D})) \\
&= \mathbb{P}(\mathcal{E}_{n,2}) + \mathbb{P}((\mathcal{B} \cap \mathcal{C}) \sqcup (\mathcal{B} \cap \mathcal{D})) \\
&= \mathbb{P}(\mathcal{E}_{n,2}|\mathcal{E}_n)\mathbb{P}(\mathcal{E}_n) + \mathbb{P}((\mathcal{B} \cap \mathcal{C}) \sqcup (\mathcal{B} \cap \mathcal{D})|\mathcal{B})\mathbb{P}(\mathcal{B}|\mathcal{E}_n)\mathbb{P}(\mathcal{E}_n) \\
&= \mathbb{P}(\mathcal{E}_n)(p + \mathbb{P}((\mathcal{B} \cap \mathcal{C}) \sqcup (\mathcal{B} \cap \mathcal{D})|\mathcal{B}) \cdot (1-p)) \\
&\ge p_n(p + p_0(1-p)) \\
&\ge p_n(p_0 p + p_0(1-p)) = p_n p_0 \ge p_{n+1}.
\end{aligned}$$

Here we used that $\mathbb{P}(\mathcal{E}_n) \ge p_n$, $p_n p_0 \ge p_{n+1}$ which follows immediately from the definition of $p_n$, $p_0 \le 1$, and simple manipulations. The inductive step, and hence the proof, is thus complete. $\square$

Using Lemma B.1 now readily proves the following:

**Claim 3.** *Let $\mathcal{E}$ be the event that there exists $\boldsymbol{w}_t$ with $\boldsymbol{w}_t \in (\boldsymbol{w}_t)_{0 \le t \le T-1}$ such that $\mathcal{A}_2(\boldsymbol{w}_t) \cap \mathcal{S} \ne \{\}$, and for all $\boldsymbol{w}_s$ with $0 \le s < t$, we have $F(\mathcal{A}_1(\boldsymbol{w}_s)) < F(\boldsymbol{w}_s) - \Delta(\boldsymbol{w}_s)$. Then $\mathbb{P}(\mathcal{E}) \ge p_T$.*

**Proof of Claim 3.** Apply Lemma B.1 with $n = T$. Following the notation from there, we have that the event $\mathcal{E}_T = \mathcal{E}_{T,1} \sqcup \mathcal{E}_{T,2}$ has probability at least $p_T$.

Suppose that $\mathcal{E}_{T,1}$ occurs. Note $\mathcal{E}_{T,1}$ implies that $\boldsymbol{w}_t \in \mathcal{L}_{F,F(\boldsymbol{w}_0)}$ for all $0 \le t \le T$. Therefore

$$\Delta(\boldsymbol{w}_t) \ge \inf_{\boldsymbol{u} \in \mathcal{L}_{F,F(\boldsymbol{w}_0)}} \Delta(\boldsymbol{u}) \text{ for all } 0 \le t \le T. \tag{8}$$

Moreover, telescoping the direct implication of $\mathcal{E}_{T,1}$ gives that

$$F(\boldsymbol{w}_T) < F(\boldsymbol{w}_0) - \sum_{t=0}^{T-1} \Delta(\boldsymbol{w}_t). \tag{9}$$

Combining (8) and (9) and recalling that we shifted WLOG so $F$ has minimum value 0 (see Notation) gives

$$T \inf_{\boldsymbol{u} \in \mathcal{L}_{F,F(\boldsymbol{w}_0)}} \Delta(\boldsymbol{u}) \le \sum_{t=0}^{T-1} \Delta(\boldsymbol{w}_t) < F(\boldsymbol{w}_0) - F(\boldsymbol{w}_T) \le F(\boldsymbol{w}_0).$$

This contradicts our choice of $T$.

Thus $\mathcal{E}_{T,1}$ cannot occur, and so $\mathcal{E}_{T,2}$ must occur, i.e. $\mathcal{E}_T = \mathcal{E}_{T,2}$. Note $\mathcal{E}_{T,2}$ is exactly the event $\mathcal{E}$. Thus

$$\mathbb{P}(\mathcal{E}) = \mathbb{P}(\mathcal{E}_{T,2}) = \mathbb{P}(\mathcal{E}_T) \ge p_T,$$

as desired. $\qquad\square$

Conditioning on the event $\mathcal{E}$ from Claim 3, by Claim 3, we immediately recover the desired guarantee on the output, probability, and number of candidate vectors stated in Theorem 2.1. The only part remaining to prove Theorem 2.1 is to establish the bound $N = \frac{F(\boldsymbol{w}_0)}{\overline{\Delta}} + \sup_{\boldsymbol{u} \in \mathcal{L}_{F,F(\boldsymbol{w}_0)}} t_{\text{oracle}}(\boldsymbol{u})$ on the number of oracle calls.

To this end, condition on $\mathcal{E}$ from Claim 3 in all of the following, and follow the notation from there, in particular the definition of $\boldsymbol{w}_t$. Directly, we obtain that the number of oracle calls is at most $\sum_{i=0}^{t} t_{\text{oracle}}(\boldsymbol{w}_i)$ (the last term $t_{\text{oracle}}(\boldsymbol{w}_t)$ in the sum appears since computing $\mathcal{A}(\boldsymbol{w}_t)$ and $\mathcal{A}(\boldsymbol{w}_t)$ takes at most $t_{\text{oracle}}(\boldsymbol{w}_t)$ oracle calls). We now upper bound this sum.

As we are conditioning on $\mathcal{E}$ and since we assumed WLOG by shifting that $F$ has minimum value 0, we have

$$F(\boldsymbol{w}_{i+1}) - F(\boldsymbol{w}_i) < -\Delta(\boldsymbol{w}_i) < 0 \text{ for all } 0 \le i \le t-1 \implies \sum_{i=0}^{t-1} \Delta(\boldsymbol{w}_i) < F(\boldsymbol{w}_0) - F(\boldsymbol{w}_t) \le F(\boldsymbol{w}_0). \tag{10}$$

The above also implies $F(\boldsymbol{w}_i) \le F(\boldsymbol{w}_0)$, i.e. $\boldsymbol{w}_i \in \mathcal{L}_{F,F(\boldsymbol{w}_0)}$, for all $0 \le i \le t$. Therefore, $t_{\text{oracle}}(\boldsymbol{w}_i) \le \sup_{\boldsymbol{u} \in \mathcal{L}_{F,F(\boldsymbol{w}_0)}} t_{\text{oracle}}(\boldsymbol{u})$ for all $0 \le i \le t$. Thus (10) gives

$$\frac{F(\boldsymbol{w}_0)}{\sum_{i=0}^{t-1} t_{\text{oracle}}(\boldsymbol{w}_i)} > \frac{\sum_{i=0}^{t-1} \Delta(\boldsymbol{w}_i)}{\sum_{i=0}^{t-1} t_{\text{oracle}}(\boldsymbol{w}_i)} \ge \min_{0 \le i \le t-1} \frac{\Delta(\boldsymbol{w}_i)}{t_{\text{oracle}}(\boldsymbol{w}_i)} \ge \overline{\Delta},$$

where the last inequality uses the elementary inequality $\frac{\sum_{i=1}^{k'} a_i}{\sum_{i=1}^{k'} b_i} \ge \min_i \frac{a_i}{b_i}$ for $a_i \ge 0, b_i > 0$, that $\boldsymbol{w}_i \in \mathcal{L}_{F,F(\boldsymbol{w}_0)}$ for all $0 \le i \le t-1$, and the definition of $\overline{\Delta}$. Rearranging and recalling $t_{\text{oracle}}(\boldsymbol{w}_t) \le \sup_{\boldsymbol{u} \in \mathcal{L}_{F,F(\boldsymbol{w}_0)}} t_{\text{oracle}}(\boldsymbol{u})$ as justified above, we obtain

$$\sum_{i=0}^{t} t_{\text{oracle}}(\boldsymbol{w}_i) \le \sup_{\boldsymbol{u} \in \mathcal{L}_{F,F(\boldsymbol{w}_0)}} t_{\text{oracle}}(\boldsymbol{u}) + \sum_{i=0}^{t-1} t_{\text{oracle}}(\boldsymbol{w}_i) \le \sup_{\boldsymbol{u} \in \mathcal{L}_{F,F(\boldsymbol{w}_0)}} t_{\text{oracle}}(\boldsymbol{u}) + \frac{F(\boldsymbol{w}_0)}{\overline{\Delta}}.$$

This yields the desired conclusion on oracle complexity, completing the proof. $\qquad\square$

## C First Order Convergence Proofs

### C.1 Proofs for Adaptive GD

**Proof.** As with the proof of Theorem 3.1, we use Theorem 2.1. We again have $\mathcal{S} = \{\boldsymbol{w} : \|\nabla F(\boldsymbol{w})\| \le \varepsilon\}$, and recall the choice of $\eta$ from Theorem 3.2. Now we let $\mathcal{A}(\boldsymbol{u}_0) = (\boldsymbol{u}_0 - \eta_{\boldsymbol{u}_0} \nabla F(\boldsymbol{u}_0), \boldsymbol{u}_0)$. Thus $\mathcal{A}_1(\boldsymbol{u}_0) = \boldsymbol{u}_0 - \eta \nabla F(\boldsymbol{u}_0)$, $\mathcal{A}_2(\boldsymbol{u}_0) = \boldsymbol{u}_0$, and $t_{\text{oracle}}(\boldsymbol{u}_0) = 1$.

**Claim 4.** *For any $\boldsymbol{u}_0$ in the $F(\boldsymbol{w}_0)$-sublevel set $\mathcal{L}_{F,F(\boldsymbol{w}_0)}$, $\mathcal{A}$ is a $(\mathcal{S}, 1, \min\left\{\frac{L_1'(\boldsymbol{w}_0)}{2\rho_0(F(\boldsymbol{w}_0)+1)^2}, \frac{\varepsilon^2}{2L_1'(\boldsymbol{w}_0)}\right\}, 0, \boldsymbol{u}_0)$-decrease procedure.*

To show this, analogously to the proof of Theorem 3.1, for any $\boldsymbol{u}_0 \notin \mathcal{S}$ in the $F(\boldsymbol{w}_0)$-sublevel set $\mathcal{L}_{F,F(\boldsymbol{w}_0)}$, we will show that the function will deterministically decrease by strictly greater than $\min\left\{\frac{L_1'(\boldsymbol{w}_0)}{\rho_0(F(\boldsymbol{w}_0)+1)^2}, \frac{\varepsilon^2}{2L_1'(\boldsymbol{w}_0)}\right\}$ at the next iterate. By definition of $\mathcal{A}_2$, exactly as with the proof of Theorem 3.1, we conclude via Theorem 2.1 upon showing Claim 4.

To show Claim 4, by choice of step size, we have $\eta_{\boldsymbol{u}_0}\|\nabla F(\boldsymbol{u}_0)\| \leq \frac{1}{\rho_0(F(\boldsymbol{w}_0)+1)}$. Thus

$$\|\boldsymbol{u}_1 - \boldsymbol{u}_0\| \leq \frac{1}{\rho_0(F(\boldsymbol{w}_0)+1)} \leq \frac{1}{\rho_0(F(\boldsymbol{u}_0)+1)}.$$

Now combining Lemma 3.1 with Assumption 1.1, and because $\boldsymbol{u}_0 \in \mathcal{L}_{F,F(\boldsymbol{w}_0)}$, we see for all $\boldsymbol{p} \in \overline{\boldsymbol{u}_0\boldsymbol{u}_1}$, $\left\|\nabla^2 F(\boldsymbol{p})\right\|_{\mathrm{op}} \leq L_1'(\boldsymbol{w}_0)$ where $L_1'(\boldsymbol{w}_0)$ is defined as in the statement of Theorem 3.2. We thus obtain by Lemma A.1,

$$F(\boldsymbol{u}_1) \leq F(\boldsymbol{u}_0) - \eta\|\nabla F(\boldsymbol{u}_0)\|^2 + \frac{L_1'(\boldsymbol{w}_0)\eta^2}{2} \cdot \|\nabla F(\boldsymbol{u}_0)\|^2. \tag{11}$$

Recall that $\boldsymbol{u}_0 \notin \mathcal{S}$, so $\|\nabla F(\boldsymbol{u}_0)\| > \varepsilon$. We break into cases:

1. If $\|\nabla F(\boldsymbol{u}_0)\| > \frac{L_1'(\boldsymbol{w}_0)}{\rho_0(F(\boldsymbol{w}_0)+1)}$, then $\eta_{\boldsymbol{u}_0} = \frac{1}{\rho_0(F(\boldsymbol{w}_0)+1)\|\nabla F(\boldsymbol{u}_0)\|}$. In this case, substituting into (11) gives

$$\begin{aligned}
F(\boldsymbol{u}_1) &\leq F(\boldsymbol{u}_0) - \eta\|\nabla F(\boldsymbol{u}_0)\|^2 + \frac{L_1'(\boldsymbol{w}_0)\eta^2}{2} \cdot \|\nabla F(\boldsymbol{u}_0)\|^2 \\
&= F(\boldsymbol{u}_0) - \frac{1}{\rho_0(F(\boldsymbol{w}_0)+1)}\|\nabla F(\boldsymbol{u}_0)\| + \frac{L_1'(\boldsymbol{w}_0)}{2\rho_0(F(\boldsymbol{w}_0)+1)^2} \\
&< F(\boldsymbol{u}_0) - \frac{1}{2} \cdot \frac{L_1'(\boldsymbol{w}_0)}{\rho_0(F(\boldsymbol{w}_0)+1)^2}.
\end{aligned}$$

2. Else if $\|\nabla F(\boldsymbol{u}_0)\| \leq L_1'(\boldsymbol{w}_0)$, then $\eta_{\boldsymbol{u}_0} = \frac{1}{L_1'(\boldsymbol{w}_0)}$. In this case, substituting into (11) gives

$$\begin{aligned}
F(\boldsymbol{u}_1) &\leq F(\boldsymbol{u}_0) - \eta\|\nabla F(\boldsymbol{u}_0)\|^2 + \frac{L_1'(\boldsymbol{w}_0)\eta^2}{2} \cdot \|\nabla F(\boldsymbol{u}_0)\|^2 \\
&\leq F(\boldsymbol{u}_0) - \frac{\|\nabla F(\boldsymbol{u}_0)\|^2}{2L_1'(\boldsymbol{w}_0)} < F(\boldsymbol{u}_0) - \frac{\varepsilon^2}{2L_1'(\boldsymbol{w}_0)},
\end{aligned}$$

   where we used that $\|\nabla F(\boldsymbol{u}_0)\| > \varepsilon$.

In either case, for $\|\nabla F(\boldsymbol{u}_0)\| > \varepsilon$ we have that

$$F(\boldsymbol{u}_1) < F(\boldsymbol{u}_0) - \min\left\{\frac{L_1'(\boldsymbol{w}_0)}{2\rho_0(F(\boldsymbol{w}_0)+1)^2}, \frac{\varepsilon^2}{2L_1'(\boldsymbol{w}_0)}\right\}.$$

This proves Claim 4. By our framework Theorem 2.1, the proof is complete. $\qquad\square$

## C.2  Proofs for SGD for FOSPs

Here, we prove Theorem 3.3. We first introduce technical preliminaries, which will also be used in Section E.

**Theorem C.1** (Vector-Valued Azuma-Hoeffding, Theorem 3.5 in Pinelis (1994))**.** *Let $\boldsymbol{\varepsilon}_1, \ldots, \boldsymbol{\varepsilon}_K \in \mathbb{R}^d$ be such that for all $k$, $\mathbb{E}\left[\boldsymbol{\varepsilon}_k|\mathfrak{F}^{k-1}\right] = 0$, $\|\boldsymbol{\varepsilon}_k\|^2 \leq \sigma_k^2$. Then for any $\lambda > 0$,*

$$\mathbb{P}\left(\left\|\sum_{k=1}^K \boldsymbol{\varepsilon}_k\right\| \geq \lambda\right) \leq 4\exp\left(-\frac{\lambda^2}{4\sum_{k=1}^K \sigma_k^2}\right).$$

Note the bound here is dimension free, so this result does not follow directly from standard Azuma-Hoeffding. Such a result can also be found in Kallenberg and Sztencel (1991); Zhang (2005); Fang et al. (2019).

**Theorem C.2** (Data-Dependent Concentration Inequality, Lemma 3 in Rakhlin et al. (2012)). *Let $\varepsilon_1, \ldots, \varepsilon_K \in \mathbb{R}$ be such that for all $k$, $\mathbb{E}[\varepsilon_k | \mathfrak{F}^{k-1}] = 0$, $\mathbb{E}[\varepsilon_k^2 | \mathfrak{F}^{k-1}] \le \sigma_k^2$. Furthermore suppose that $\mathbb{P}(\varepsilon_k \le b | \mathfrak{F}^{k-1}) = 1$. Letting $V_K = \sum_{k=1}^{K} \sigma_k^2$, for any $\delta < 1/e$, $K \ge 4$, we have*

$$\mathbb{P}\left( \sum_{k=1}^{K} \varepsilon_k > 2 \max\left\{ 2\sqrt{V_k}, b\sqrt{\log(1/\delta)} \right\} \sqrt{\log(1/\delta)} \right) \le \delta \log(K).$$

Such a result is also presented in Zhang (2005); Bartlett et al. (2008); Fang et al. (2019).

We will first prove Theorem 3.3 in the case where $\|\nabla f(\boldsymbol{w}; \boldsymbol{\zeta}) - \nabla F(\boldsymbol{w})\|$ is bounded by $\sigma(F(\boldsymbol{w}))$. As noted in Fang et al. (2019), these same inequalities hold when the martingale difference is not bounded or almost-surely bounded but rather the norms are sub-Gaussian with parameter $\sigma_k$. Thus after the proof, we remark how to straightforwardly generalize Theorem 3.3 to the case when $\|\nabla f(\boldsymbol{w}; \boldsymbol{\zeta}) - \nabla F(\boldsymbol{w})\|$ is sub-Gaussian with parameter $\sigma(F(\boldsymbol{w}))$ in Remark 7.

Now, we prove Theorem 3.3.

**Proof.** We use our framework Theorem 2.1 with $\mathcal{S} = \{\boldsymbol{w} : \|\nabla F(\boldsymbol{w})\| \le \varepsilon\}$. Recall as per the discussion of SGD in our framework in Subsection 2.3, we let $\boldsymbol{p}_0 = \boldsymbol{u}_0$, and define a sequence $(\boldsymbol{p}_i)_{0 \le i \le K_0}$ via

$$\boldsymbol{p}_i = \boldsymbol{p}_{i-1} - \eta \nabla f(\boldsymbol{p}_{i-1}; \boldsymbol{\zeta}_i),$$

where the $\boldsymbol{\zeta}_i$ are minibatch samples i.i.d. across different $i$. Note this sequence can be equivalently defined by repeated compositions of the function $\boldsymbol{u} \to \boldsymbol{u} - \eta \nabla f(\boldsymbol{u}; \boldsymbol{\zeta})$.

We now let $\mathcal{A}(\boldsymbol{u}_0) = (\boldsymbol{p}_{K_0}, (\boldsymbol{p}_i)_{0 \le i \le K_0 - 1})$, hence $\mathcal{A}_1(\boldsymbol{u}_0) = \boldsymbol{p}_{K_0}$, $\mathcal{A}_2(\boldsymbol{u}_0) = (\boldsymbol{p}_i)_{0 \le i \le K_0 - 1}$. Thus $t_{\text{oracle}}(\boldsymbol{u}_0) = K_0$. Also note the noise $\boldsymbol{\xi}_t$ used defining $\mathcal{A}$ are independent across different $t$.

For appropriate $\eta = \tilde{\Theta}(\varepsilon^2)$, $K_0 = \tilde{\Theta}(\varepsilon^{-2})$ depending only on $\varepsilon, \delta, F(\boldsymbol{w}_0)$ and polylogarithmically in $1/\delta$, which we define below, we establish the following Claim 5:

**Claim 5.** *For any $\boldsymbol{u}_0$ in the $F(\boldsymbol{w}_0)$-sublevel set $\mathcal{L}_{F, F(\boldsymbol{w}_0)}$, $\mathcal{A}$ is a $(\mathcal{S}, K_0, \frac{\eta K_0 \varepsilon^2}{4}, p, \boldsymbol{u}_0)$-decrease procedure, where $p = \frac{\delta \eta K_0 \varepsilon^2}{4(F(\boldsymbol{w}_0) + 1)}$.*

Then using Theorem 2.1, we then directly conclude Theorem 3.3.

To show Claim 5, consider any $\boldsymbol{u}_0$ in the $F(\boldsymbol{w}_0)$-sublevel set but not in $\mathcal{S}$. Following the notation from above, consider a 'block' of $K_0$ consecutive iterates of SGD starting at $\boldsymbol{p}_0 = \boldsymbol{u}_0$. We establish that with probability at least $1 - p$, if none of the iterates $\{\boldsymbol{p}_0 = \boldsymbol{u}_0, \ldots, \boldsymbol{p}_{K_0 - 1}\}$ lie in $\mathcal{S}$, then $F(\boldsymbol{p}_{K_0}) < F(\boldsymbol{p}_0) - \Delta$ where $\Delta = \frac{\eta K_0 \varepsilon^2}{4}$. Then recalling the definitions of $\mathcal{A}_2$, we immediately conclude Claim 5.

**Definitions and Parameters:** For convenience, define

$$L_0(\boldsymbol{w}_0) = \rho_0(F(\boldsymbol{w}_0) + 1),$$
$$L_1(\boldsymbol{w}_0) = \rho_1(F(\boldsymbol{w}_0) + 1),$$
$$\sigma_1(\boldsymbol{w}_0) = \sigma(F(\boldsymbol{w}_0) + 1),$$
$$B(\boldsymbol{w}_0) = \sigma_1(\boldsymbol{w}_0)^2 + \frac{1}{8}\sigma_1(\boldsymbol{w}_0)L_0(\boldsymbol{w}_0).$$

Also define

$$\boldsymbol{\xi}_{t+1} = \nabla f(\boldsymbol{p}_t; \boldsymbol{\zeta}_{t+1}) - \nabla F(\boldsymbol{p}_t),$$

where $\boldsymbol{\zeta}_{t+1}$ denotes the i.i.d. minibatch samples. Note by Assumption 3.1 that $\mathbb{E}[\boldsymbol{\xi}_{t+1}] = 0$, where expectation is with respect to $\boldsymbol{\zeta}_{t+1}$.

In particular, we choose these parameters as follows:

$$\tilde{\eta} = \frac{\varepsilon^2}{\tilde{L}(\boldsymbol{w}_0) \log(1/\varepsilon)^6 \log(1/\delta)^6}$$

$$K_0 = \frac{C(\boldsymbol{w}_0)}{\varepsilon^2} \log(1/\tilde{\eta})^2 \log(1/\delta)^2 \log(1/\varepsilon)^2,$$

$$\eta = \frac{1}{\max\{1, \rho_0(F(\boldsymbol{w}_0)+1)\}} \cdot \tilde{\eta},$$

where

$$C(\boldsymbol{w}_0) = 128 B(\boldsymbol{w}_0) \vee 64 (F(\boldsymbol{w}_0)+1)^2,$$
$$\tilde{L}'(\boldsymbol{w}_0) = 8 L_1(\boldsymbol{w}_0)(L_0(\boldsymbol{w}_0)^2 + \sigma_1(\boldsymbol{w}_0)^2) \vee 2 L_0(\boldsymbol{w}_0) \vee 4\sigma_1(\boldsymbol{w}_0),$$
$$\tilde{L}(\boldsymbol{w}_0) = \tilde{L}'(\boldsymbol{w}_0)^2 C(\boldsymbol{w}_0)^2 \vee (3\sqrt{2} \log(\tilde{L}(\boldsymbol{w}_0)))^8 \vee (3\sqrt{2})^8.$$

**Remark 6.** Note that $C, \tilde{L}', \tilde{L}$ depend only polynomially in terms of the self-bounding functions $\rho_0, \rho_1, \sigma$, and $F(\boldsymbol{w}_0)$.

Note we can assume WLOG that $\varepsilon$ and the desired probability $\delta$ are at most some small enough *universal* constants in $(0, 1)$; by doing so, the result does not change up to universal constant, and hence is identical under the $O(\cdot)$. Consequently we may assume WLOG that $\tilde{\eta}$ and $\eta$ are at most some small enough universal constant in $(0, 1)$ and that $K_0 \geq 4$.

**Claim 6.** *For $\varepsilon, \delta$ small enough universal constants, the above choice of parameters satisfies the following properties:*

$$\max\{1, \rho_0(F(\boldsymbol{w}_0)+1)\}\eta$$
$$= \tilde{\eta} \leq \min\left\{ \frac{\varepsilon^2}{8L_1(\boldsymbol{w}_0)(L_0(\boldsymbol{w}_0)^2 + \sigma_1(\boldsymbol{w}_0)^2)}, \frac{1}{2K_0 L_0(\boldsymbol{w}_0)}, \frac{1}{4\sigma_1(\boldsymbol{w}_0)\sqrt{K_0 \log(4K_0/p)}} \right\}, \tag{12}$$

$$K_0 \varepsilon^2 \geq 128 B(\boldsymbol{w}_0) \log\left(\frac{2\log K_0}{p}\right). \tag{13}$$

For the sake of brevity, we prove Claim 6 after the our main proof. Checking this is a matter of elementary, albeit tedious, univariate inequalities.

Again, our plan is to apply Theorem 2.1 by showing decrease with high probability for a block of $K_0$ iterates starting at $\boldsymbol{p}_0$.

**Notation:** Let $\mathfrak{F}^t$ denote the filtration of all information up through $\boldsymbol{p}_t$, but *not* including the mini-batch sample $\boldsymbol{\zeta}_{t+1}$. Let $\mathcal{K}$ be a stopping time denoting the first $t$ such that $\boldsymbol{p}_t \notin \mathbb{B}\left(\boldsymbol{p}_0, \frac{1}{\rho_0(F(\boldsymbol{w}_0)+1)}\right)$, i.e. the escape time of the iterates beginning at $\boldsymbol{p}_0$ from $\mathbb{B}\left(\boldsymbol{p}_0, \frac{1}{\rho_0(F(\boldsymbol{w}_0)+1)}\right) = \mathbb{B}\left(\boldsymbol{u}_0, \frac{1}{\rho_0(F(\boldsymbol{w}_0)+1)}\right)$.

We first detail two high probability events we will condition on for the remainder of the proof:

- By Vector-Valued Azuma Hoeffding Theorem C.1, for a given $1 \leq t \leq K_0$ we have with probability at least $1 - \frac{p}{2K_0}$,

$$\left\| \eta \sum_{k=1}^{t} \boldsymbol{\xi}_k \right\| \leq 2\eta \sqrt{\log(48K_0/p) \sum_{k=1}^{t} \sigma(F(\boldsymbol{p}_{k-1}))^2} = 2\eta \sqrt{\log(4K_0/p) \sum_{k=0}^{t-1} \sigma(F(\boldsymbol{p}_{k-1}))^2}.$$

This follows since each $\mathbb{E}[\boldsymbol{\xi}_k | \mathfrak{F}^{k-1}] = 0$ as the stochastic gradient oracle is unbiased, and as $\|\boldsymbol{\xi}_k\| \leq \sigma(F(\boldsymbol{p}_{k-1}))$ by Assumption 3.1.

Thus by Union Bound, with probability at least $1 - p/2$, we have for all $1 \leq t \leq K_0$ that

$$\left\| \eta \sum_{k=1}^{t} \boldsymbol{\xi}_k \right\| \leq 2\eta \sqrt{\log(4K_0/p) \sum_{k=0}^{t-1} \sigma(F(\boldsymbol{p}_k))^2}. \tag{14}$$

Denote this event by $\mathcal{E}_1$, so $\mathbb{P}(\mathcal{E}_1) \geq 1 - p/2$.

- We define a stochastic process with the following trick to derive uniform bounds. Define the following sequence of real numbers:

$$Y_t := -\eta \langle \nabla F(\boldsymbol{p}_t), \boldsymbol{\xi}_{t+1} \rangle 1_{t < \mathcal{K}}.$$

Notice $1_{t < \mathcal{K}}$ is $\mathfrak{F}^t$-measurable, as $\{t < \mathcal{K}\}$ holds if and only if $\boldsymbol{p}_1, \ldots, \boldsymbol{p}_t \in \mathbb{B}\left(\boldsymbol{p}_0, \frac{1}{\rho_0(F(\boldsymbol{w}_0)+1)}\right)$.

Clearly $\nabla F(\boldsymbol{p}_t)$ is also $\mathfrak{F}^t$-measurable. Thus as the stochastic gradient oracle is unbiased (i.e. $\mathbb{E}\left[\boldsymbol{\xi}_{t+1} | \mathfrak{F}^t\right] = 0$),

$$\mathbb{E}[Y_t] = \mathbb{E}\left[\langle \nabla F(\boldsymbol{p}_t), \boldsymbol{\xi}_{t+1} \rangle 1_{t < \mathcal{K}} | \mathfrak{F}^t\right] = 0.$$

For $t \geq \mathcal{K}$ we have $Y_t \equiv 0$. For $t < \mathcal{K}$, we have $\boldsymbol{p}_t \in \mathbb{B}\left(\boldsymbol{p}_0, \frac{1}{\rho_0(F(\boldsymbol{w}_0)+1)}\right)$. Consequently by Lemma 3.1 and Corollary 1 we have

$$|Y_t| \leq \eta |\langle \nabla F(\boldsymbol{p}_t), \boldsymbol{\xi}_{t+1} \rangle| \leq \eta \|\nabla F(\boldsymbol{p}_t)\| \|\boldsymbol{\xi}_{t+1}\| \leq \eta \rho_0(F(\boldsymbol{w}_0)+1) \|\boldsymbol{\xi}_{t+1}\|.$$

Moreover by Assumption 3.1 and Lemma 3.1,

$$\|\boldsymbol{\xi}_{t+1}\| \leq \sigma(F(\boldsymbol{p}_t)) \leq \sigma(F(\boldsymbol{w}_0)+1) = \sigma_1(\boldsymbol{w}_0).$$

In particular, recall that $\boldsymbol{\xi}_{t+1}$ is the difference between the gradient oracle and actual gradient at $\boldsymbol{p}_t$.

By the above arguments, both of the following inequalities hold deterministically:

$$|Y_t| \leq \eta \|\nabla F(\boldsymbol{p}_t)\| \sigma_1(\boldsymbol{w}_0),$$
$$|Y_t| \leq \eta \rho_0(F(\boldsymbol{w}_0)+1)\sigma_1(\boldsymbol{w}_0) = \eta L_0(\boldsymbol{w}_0)\sigma_1(\boldsymbol{w}_0).$$

We now apply both of these bounds in Data-Dependent Concentration Inequality, Theorem C.2 (whose conditions hold because we can assume $\delta, \varepsilon$ are at most given universal constants, so $K_0 \geq 4, 2\log K_0/p > e$). Consequently we obtain with probability at least $1 - \frac{p}{2}$ that

$$-\eta \sum_{t=0}^{K_0-1} \langle \nabla F(\boldsymbol{p}_t), \boldsymbol{\xi}_{t+1} \rangle 1_{t < \mathcal{K}} \leq 2\eta L_0(\boldsymbol{w}_0)\sigma_1(\boldsymbol{w}_0) \log\left(\frac{2\log K_0}{p}\right) \bigvee$$

$$4\sqrt{\eta^2 \sigma_1(\boldsymbol{w}_0)^2 \sum_{t=0}^{K_0-1} \|\nabla F(\boldsymbol{p}_t)\|^2} \sqrt{\log\left(\frac{2\log K_0}{p}\right)}. \quad (15)$$

Denote this event by $\mathcal{E}_2$, so $\mathbb{P}(\mathcal{E}_2) \geq 1 - p/2$.

For the rest of this proof, we condition on $\mathcal{E}_1 \cap \mathcal{E}_2$. By the above, $\mathcal{E}_1 \cap \mathcal{E}$ occurs with probability at least $1 - p$. Denote $\mathcal{E} = \mathcal{E}_1 \cap \mathcal{E}_2$.

A-priori, these bounds are not particularly useful, especially in our more challenging setting under Assumption 3.2 where noise can depend on function value. However conditioned on $\mathcal{E}$, we prove that SGD is sufficiently 'local', in particular that $\|\boldsymbol{p}_t - \boldsymbol{u}_0\| \leq 1$ for all $t, 1 \leq t \leq K_0$. This will then give us control over function value via Lemma 3.1, which then allow us to make use of these bounds in a more standard way.

**Lemma C.1.** *Conditioned on $\mathcal{E}_1$ (and hence conditioned on $\mathcal{E}$), for all $t, 1 \leq t \leq K_0$, we have*

$$\|\boldsymbol{p}_t - \boldsymbol{p}_0\| = \|\boldsymbol{p}_t - \boldsymbol{u}_0\| \leq \frac{1}{\rho_0(F(\boldsymbol{w}_0)+1)}.$$

**Proof.** We go by induction on $t$. Notice after $t$ iterates,

$$\boldsymbol{p}_t = \boldsymbol{w}_0 - \eta \sum_{k=0}^{t-1} \nabla F(\boldsymbol{p}_k) - \eta \sum_{k=1}^{t} \boldsymbol{\xi}_k.$$

For the base case $t = 1$, we have from Corollary 1 that $\|\nabla F(\boldsymbol{w}_0)\| \leq \rho_0(F(\boldsymbol{w}_0)) \leq L_0(\boldsymbol{w}_0)$. From the definition of the high-probability event $\mathcal{E}_1$ and properties of $\eta$ from Claim 6, and as $\sigma_1(\boldsymbol{w}_0) \geq \sigma(\boldsymbol{w}_0)$), it follows that

$$\|\eta \boldsymbol{\xi}_1\| \leq 2\eta \sigma(F(\boldsymbol{w}_0))\sqrt{K_0 \log(4K_0/p)} \leq \frac{1}{2\rho_0(F(\boldsymbol{w}_0)+1)}.$$

Consequently by properties of $\eta$ from Claim 6,

$$\|\boldsymbol{p}_1 - \boldsymbol{p}_0\| \le \|\eta\nabla F(\boldsymbol{w}_0)\| + \|\eta\boldsymbol{\xi}_0\| \le \frac{1}{\rho_0(F(\boldsymbol{w}_0)+1)}.$$

This finishes the proof of the base case.

Now suppose Lemma C.1 holds for all $1 \le k \le t-1$; we will show it for $t$. From Lemma 3.1, for all $k \le t-1$, we have

$$\|\nabla F(\boldsymbol{p}_k)\| \le \rho_0(F(\boldsymbol{w}_0)+1) \le L_0(\boldsymbol{w}_0).$$

Thus for each $k$, we have

$$\sigma(F(\boldsymbol{p}_k)) \le \sigma(F(\boldsymbol{w}_0)+1) = \sigma_1(\boldsymbol{w}_0).$$

Thus conditioned on $\mathcal{E}_1$ we obtain

$$\begin{aligned}
\|\boldsymbol{p}_t - \boldsymbol{p}_0\| &\le \left\|\eta\sum_{k=0}^{t-1}\nabla F(\boldsymbol{p}_k)\right\| + \left\|\eta\sum_{k=1}^{t}\boldsymbol{\xi}_k\right\| \\
&\le \eta K_0 L_0(\boldsymbol{w}_0) + 2\eta\sqrt{\log(4K_0/p)\sum_{k=0}^{K_0-1}\sigma_1(\boldsymbol{w}_0)^2} \\
&= \eta K_0 L_0(\boldsymbol{w}_0) + 2\eta\sigma_1(\boldsymbol{w}_0)\sqrt{K_0\log(4K_0/p)} \\
&\le \frac{1}{2\rho_0(F(\boldsymbol{w}_0)+1)} + \frac{1}{2\rho_0(F(\boldsymbol{w}_0)+1)} = \frac{1}{\rho_0(F(\boldsymbol{w}_0)+1)}.
\end{aligned}$$

Here we used the choice of $\eta$ from Claim 6 and the upper bound (14) on $\left\|\eta\sum_{k=1}^{t}\boldsymbol{\xi}_k\right\|$ implied by $\mathcal{E}_1$. This completes the induction. $\qquad\square$

Now that we know the iterates of SGD are 'sufficiently local' for $K_0$ iterations via Lemma C.1, the finish is straightforward. Condition on $\mathcal{E}$ for the rest of the proof. Consider any $0 \le t \le K_0 - 1$. $\mathcal{E}$ implies for all $\boldsymbol{p} \in \overline{\boldsymbol{p}_{t-1}\boldsymbol{p}_t}$, writing $\boldsymbol{p} = \theta\boldsymbol{p}_{t-1} + (1-\theta)\boldsymbol{p}_t$ for $\theta \in [0,1]$, that we have

$$\|\boldsymbol{p} - \boldsymbol{p}_0\| \le \theta\|\boldsymbol{p}_{t-1} - \boldsymbol{p}_0\| + (1-\theta)\|\boldsymbol{p}_t - \boldsymbol{p}_0\| \le (1-\theta+\theta)\cdot\frac{1}{\rho_0(F(\boldsymbol{w}_0)+1)} = \frac{1}{\rho_0(F(\boldsymbol{w}_0)+1)}.$$

Consequently $F(\boldsymbol{p}) \le \rho_0(F(\boldsymbol{w}_0)+1)$, so the above combined with Assumption 1.1 gives

$$\left\|\nabla^2 F(\boldsymbol{p})\right\| \le L_1(\boldsymbol{w}_0). \tag{16}$$

We also obtain from Lemma C.1 together with Corollary 1 and Assumption 3.1 that for all $0 \le t \le K_0$,

$$\begin{aligned}
\|\boldsymbol{\xi}_t\| &\le \sigma(F(\boldsymbol{w}_0)+1) = \sigma_1(\boldsymbol{w}_0), \\
\|\nabla F(\boldsymbol{p}_t)\| &\le \rho_0(F(\boldsymbol{w}_0)+1) = L_0(\boldsymbol{w}_0). \tag{17}
\end{aligned}$$

Now by Lemma A.1 and (16),

$$\begin{aligned}
F(\boldsymbol{p}_{t+1}) &\le F(\boldsymbol{p}_t) - \eta\langle\nabla F(\boldsymbol{p}_t), \nabla f(\boldsymbol{p}_t; \boldsymbol{\zeta}_{t+1})\rangle + \frac{\eta^2 L_1(\boldsymbol{w}_0)}{2}\|\nabla f(\boldsymbol{p}_t; \boldsymbol{\zeta}_{t+1})\|^2 \\
&\le F(\boldsymbol{p}_t) - \eta\|\nabla F(\boldsymbol{p}_t)\|^2 - \eta\langle\nabla F(\boldsymbol{p}_t), \boldsymbol{\xi}_{t+1}\rangle + \eta^2 L_1(\boldsymbol{w}_0)\Big(\|\nabla F(\boldsymbol{p}_t)\|^2 + \|\boldsymbol{\xi}_{t+1}\|^2\Big).
\end{aligned}$$

The last step uses the definition of $\boldsymbol{\xi}_{t+1}$ and Young's Inequality.

Summing and telescoping the above for $0 \le t \le K_0 - 1$, and applying (17), gives

$$\begin{aligned}
F(\boldsymbol{p}_{K_0}) \le F(\boldsymbol{p}_0) &- \eta\sum_{t=0}^{K_0-1}\|\nabla F(\boldsymbol{p}_t)\|^2 - \eta\sum_{t=0}^{K_0-1}\langle\nabla F(\boldsymbol{p}_t), \boldsymbol{\xi}_{t+1}\rangle \\
&+ \eta^2 K_0 L_0(\boldsymbol{w}_0)^2 L_1(\boldsymbol{w}_0) + \eta^2 K_0 \sigma_1^2(\boldsymbol{w}_0) L_1(\boldsymbol{w}_0). \tag{18}
\end{aligned}$$

Now, conditioned on $\mathcal{E}$, we upper bound

$$-\eta\sum_{t=0}^{K_0-1}\langle\nabla F(\boldsymbol{p}_t), \boldsymbol{\xi}_{t+1}\rangle$$

using (15). Under $\mathcal{E}$, by Lemma C.1 and Lemma 3.1, we have $\boldsymbol{p}_t \in \mathbb{B}\left(\boldsymbol{p}_0, \frac{1}{\rho_0(F(\boldsymbol{w}_0)+1)}\right)$ for all $1 \le t \le K_0$, which implies that $t < \mathcal{K}$ for all $1 \le t \le K_0$. Therefore

$$-\eta \sum_{t=0}^{K_0-1} \langle \nabla F(\boldsymbol{p}_t), \boldsymbol{\xi}_{t+1} \rangle = -\eta \sum_{t=0}^{K_0-1} \langle \nabla F(\boldsymbol{p}_t), \boldsymbol{\xi}_{t+1} \rangle 1_{t<\mathcal{K}}.$$

Now AM-GM gives

$$4\sqrt{\eta^2 \sigma_1(\boldsymbol{w}_0)^2 \sum_{t=0}^{K_0-1} \|\nabla F(\boldsymbol{p}_t)\|^2} \sqrt{\log\left(\frac{2\log K_0}{p}\right)}$$

$$\le 2\eta\left(\frac{1}{4} \sum_{t=0}^{K_0-1} \|\nabla F(\boldsymbol{p}_t)\|^2 + 8\sigma_1(\boldsymbol{w}_0)^2 \log\left(\frac{2\log K_0}{p}\right)\right).$$

Combining with (15), we obtain

$$-\eta \sum_{t=0}^{K_0-1} \langle \nabla F(\boldsymbol{p}_t), \boldsymbol{\xi}_{t+1} \rangle = -\eta \sum_{t=0}^{K_0-1} \langle \nabla F(\boldsymbol{p}_t), \boldsymbol{\xi}_{t+1} \rangle 1_{t<\mathcal{K}}$$

$$\le \frac{\eta}{2} \sum_{t=0}^{K_0-1} \|\nabla F(\boldsymbol{p}_t)\|^2 + 16\eta B(\boldsymbol{w}_0) \log\left(\frac{2\log K_0}{p}\right).$$

Combining with (18) gives

$$F(\boldsymbol{p}_{K_0}) \le F(\boldsymbol{p}_0) - \frac{\eta}{2} \sum_{t=0}^{K_0-1} \|\nabla F(\boldsymbol{p}_t)\|^2 + 16\eta B(\boldsymbol{w}_0) \log\left(\frac{2\log K_0}{p}\right) + \eta^2 K_0 L_0(\boldsymbol{w}_0)^2 L_1(\boldsymbol{w}_0)$$

$$+ \eta^2 K_0 \sigma_1^2(\boldsymbol{w}_0) L_1(\boldsymbol{w}_0). \tag{19}$$

Suppose that $\|\nabla F(\boldsymbol{p}_t)\| > \varepsilon$ for all $0 \le t \le K_0 - 1$. Then the above gives

$$F(\boldsymbol{p}_{K_0}) < F(\boldsymbol{p}_0) - \frac{\eta K_0 \varepsilon^2}{2} + 16\eta B(\boldsymbol{w}_0) \log\left(\frac{2\log K_0}{p}\right)$$

$$+ \eta^2 K_0 L_0(\boldsymbol{w}_0)^2 L_1(\boldsymbol{w}_0) + \eta^2 K_0 \sigma_1^2(\boldsymbol{w}_0) L_1(\boldsymbol{w}_0).$$

To make use of this bound, by our choice of $\eta$, Claim 6 implies that

$$\eta^2 K_0 L_0(\boldsymbol{w}_0)^2 L_1(\boldsymbol{w}_0) + \eta^2 K_0 \sigma_1^2(\boldsymbol{w}_0) L_1(\boldsymbol{w}_0) \le \frac{\eta K_0 \varepsilon^2}{8}.$$

By choice of $K_0$, Claim 6 implies that

$$16\eta B(\boldsymbol{w}_0) \log\left(\frac{2\log K_0}{p}\right) \le \frac{\eta K_0 \varepsilon^2}{8}.$$

The above was all conditioned on $\mathcal{E}$, which occurred with probability at least $1 - p$. Thus by (19), we obtain that with this same probability which is at least $1 - p$, if none of $\boldsymbol{p}_0, \ldots, \boldsymbol{p}_{K_0-1}$ have gradient norm larger than $\varepsilon$, we have

$$F(\boldsymbol{p}_{K_0}) < F(\boldsymbol{p}_0) - \frac{\eta K_0 \varepsilon^2}{4} = F(\boldsymbol{u}_0) - \frac{\eta K_0 \varepsilon^2}{4}.$$

This establishes that $\mathcal{A}$ is a $(\mathcal{S}, K_0 + 1, \frac{\eta K_0 \varepsilon^2}{4}, p, \boldsymbol{u}_0)$-decrease procedure. Following our initial observations, we conclude via Theorem 2.1. $\qquad\square$

Now we prove Claim 6.

**Proof of Claim 6.** We first prove (13). Recall we chose

$$K_0 = \frac{C(\boldsymbol{w}_0)}{\varepsilon^2} \log(1/\tilde{\eta})^2 \log(1/\delta)^2 \log(1/\varepsilon)^2.$$

Furthermore recall $p = \frac{\delta \tilde{\eta} K_0 \varepsilon^2}{4(F(\boldsymbol{w}_0)+1)}$. Thus, (13) holds if and only if

$$C(\boldsymbol{w}_0) \log(1/\tilde{\eta})^2 \log(1/\delta)^2 \log(1/\varepsilon)^2 \ge 128 B(\boldsymbol{w}_0) \log\left(\frac{8\log K_0 \cdot (F(\boldsymbol{w}_0)+1)}{\delta \tilde{\eta} K_0 \varepsilon^2}\right).$$

As $C(\boldsymbol{w}_0) \geq 128 B(\boldsymbol{w}_0) \vee 64(F(\boldsymbol{w}_0)+1)^2$, again using the expression for $K_0$, it suffices to prove

$$\log(1/\tilde{\eta})^2 \log(1/\delta)^2 \log(1/\varepsilon)^2 \geq \log\left(\frac{\log K_0}{C(\boldsymbol{w}_0)^{1/2}\delta\tilde{\eta}\log(1/\tilde{\eta})^2\log(1/\delta)^2}\right).$$

As $\log(1/\delta), \log(1/\tilde{\eta})$ are both larger than 1, it suffices to prove

$$
\begin{aligned}
&\log(1/\tilde{\eta})^2 \log(1/\delta)^2 \log(1/\varepsilon)^2 \\
&\geq \log(1/\tilde{\eta}) + \log(1/\delta) \\
&\quad + \log\left(\frac{\log C(\boldsymbol{w}_0) + \log(1/\varepsilon^2) + 2\log\log(1/\tilde{\eta}) + 2\log\log(1/\delta) + 2\log\log(1/\varepsilon)}{C(\boldsymbol{w}_0)^{1/2}}\right).
\end{aligned}
$$

Since $C(\boldsymbol{w}_0) \geq 64$, it satisfies $\log C(\boldsymbol{w}_0) < C(\boldsymbol{w}_0)^{1/2}$, so it suffices to prove

$$
\begin{aligned}
&\log(1/\tilde{\eta})^2 \log(1/\delta)^2 \log(1/\varepsilon)^2 \\
&\geq \log(1/\tilde{\eta}) + \log(1/\delta) \\
&\quad + \log(1 + 2\log(1/\varepsilon) + 2\log\log(1/\tilde{\eta}) + 2\log\log(1/\delta) + 2\log\log(1/\varepsilon)).
\end{aligned}
$$

By comparing 'degrees', we conclude recalling we can assume WLOG that $\delta, \varepsilon, \tilde{\eta}$ are smaller than some universal constant.

Now we prove (12). We will prove that

$$\tilde{\eta} \leq \frac{1}{\tilde{L}'(\boldsymbol{w}_0)K_0\sqrt{\log(4K_0/p)}}. \tag{20}$$

After proving (20), recalling our choice of $K_0 > 1/\varepsilon^2$ directly implies (12). To show (20), equivalently, we want to show

$$\tilde{\eta}\log(1/\tilde{\eta})^2\sqrt{\log(4K_0/p)} \leq \frac{\varepsilon^2}{\tilde{L}'(\boldsymbol{w}_0)C(\boldsymbol{w}_0)\log(1/\delta)^2\log(1/\varepsilon)^2}.$$

Recalling the definition of $p$, this holds if and only if

$$\tilde{\eta}\log(1/\tilde{\eta})^2\sqrt{\log\left(\frac{16(F(\boldsymbol{w}_0)+1)}{\delta\tilde{\eta}\varepsilon^2}\right)} \leq \frac{\varepsilon^2}{\tilde{L}'(\boldsymbol{w}_0)C(\boldsymbol{w}_0)\log(1/\delta)^2\log(1/\varepsilon)^2}.$$

Now we explicitly recall our expression for $\tilde{\eta} = \frac{\varepsilon^2}{\tilde{L}(\boldsymbol{w}_0)\log(1/\varepsilon)^6\log(1/\delta)^6}$. Plugging this in and recalling $\tilde{L}(\boldsymbol{w}_0) \geq \tilde{L}'(\boldsymbol{w}_0)^2 C(\boldsymbol{w}_0)^2$, it suffices to prove

$$
\begin{aligned}
&\frac{1}{\tilde{L}(\boldsymbol{w}_0)^{1/2}\log(1/\varepsilon)^6\log(1/\delta)^6}\log\left(\frac{\tilde{L}(\boldsymbol{w}_0)\log(1/\varepsilon)^6\log(1/\delta)^6}{\varepsilon^2}\right)^2 \\
&\quad \cdot \sqrt{\log\left(\frac{16(F(\boldsymbol{w}_0)+1)\tilde{L}(\boldsymbol{w}_0)\log(1/\varepsilon)^6\log(1/\delta)^6}{\delta\varepsilon^4}\right)} \\
&\leq \frac{1}{\log(1/\delta)^2\log(1/\varepsilon)^2}.
\end{aligned}
$$

Thus it suffices to prove:

$$
\begin{aligned}
&\frac{18}{\tilde{L}(\boldsymbol{w}_0)^{1/2}}\log\left(\frac{\tilde{L}(\boldsymbol{w}_0)\log(1/\varepsilon)\log(1/\delta)}{\varepsilon}\right)^2\sqrt{\log\left(\frac{16(F(\boldsymbol{w}_0)+1)\tilde{L}(\boldsymbol{w}_0)\log(1/\varepsilon)\log(1/\delta)}{\delta\varepsilon}\right)} \\
&\leq \log(1/\delta)^4\log(1/\varepsilon)^4.
\end{aligned}
$$

Recall $\tilde{L}(\boldsymbol{w}_0)^{1/8} \geq 3\sqrt{2}\log(\tilde{L}(\boldsymbol{w}_0)) \vee 3\sqrt{2}$ and so

$$\frac{3\sqrt{2}}{\tilde{L}(\boldsymbol{w}_0)^{1/4}}\log\left(\frac{\tilde{L}(\boldsymbol{w}_0)\log(1/\varepsilon)\log(1/\delta)}{\varepsilon}\right)$$

$$\leq \frac{3\sqrt{2}}{\tilde{L}(\boldsymbol{w}_0)^{1/4}}\big(\log(1/\varepsilon) + \log\log(1/\varepsilon) + \log\log(1/\delta) + \log\tilde{L}(\boldsymbol{w}_0)\big)$$

$$\leq \frac{1}{\tilde{L}(\boldsymbol{w}_0)^{1/8}}\big(1 + \log(1/\varepsilon) + \log\log(1/\varepsilon) + \log\log(1/\delta)\big).$$

Thus it suffices to show

$$\frac{1}{\tilde{L}(\boldsymbol{w}_0)^{1/4}}\big(1 + \log(1/\varepsilon) + \log\log(1/\varepsilon) + \log\log(1/\delta)\big)^2$$

$$\cdot\sqrt{\log\left(\frac{16(F(\boldsymbol{w}_0)+1)\tilde{L}(\boldsymbol{w}_0)\log(1/\varepsilon)\log(1/\delta)}{\delta\varepsilon}\right)}$$

$$\leq \log(1/\delta)^4\log(1/\varepsilon)^4.$$

To this end recall $\tilde{L}(\boldsymbol{w}_0)^{1/8} \geq \log(16(F(\boldsymbol{w}_0)+1)\tilde{L}(\boldsymbol{w}_0))$, thus

$$\frac{1}{\tilde{L}(\boldsymbol{w}_0)^{1/8}}\log\left(\frac{16(F(\boldsymbol{w}_0)+1)\tilde{L}(\boldsymbol{w}_0)\log(1/\varepsilon)\log(1/\delta)}{\delta\varepsilon}\right)$$

$$= \frac{1}{\tilde{L}(\boldsymbol{w}_0)^{1/8}}\big(\log(16(F(\boldsymbol{w}_0)+1)\tilde{L}(\boldsymbol{w}_0)) + \log(1/\delta) + \log(1/\varepsilon) + \log\log(1/\delta)) + \log\log(1/\varepsilon)\big)$$

$$\leq 1 + \log(1/\delta) + \log(1/\varepsilon) + \log\log(1/\delta)) + \log\log(1/\varepsilon).$$

Therefore it suffices to show

$$\big(1 + \log(1/\varepsilon) + \log\log(1/\varepsilon) + \log\log(1/\delta)\big)^2$$

$$\cdot\big(1 + \log(1/\delta) + \log(1/\varepsilon) + \log\log(1/\delta)) + \log\log(1/\varepsilon)\big)^{1/2}$$

$$\leq \log(1/\delta)^4\log(1/\varepsilon)^4.$$

Evidently the above holds for small enough universal constants $\delta, \varepsilon$ (compare 'degrees'), so we conclude the proof. $\qquad\square$

**Remark 7.** We also discuss how to extend this result to when the $\|\boldsymbol{\xi}_t\|$ has sub-Gaussianity parameter $\sigma(F(\boldsymbol{p}_t))$. The extension is straightforward. Again, we aim to prove Claim 5. For the rest of this remark, follow the notation from the proof for SGD above. Besides applying Theorem C.1, Theorem C.2 when the relevant random variables are sub-Gaussian, which still hold true as mentioned in Fang et al. (2019), the only other time we used that $\|\boldsymbol{\xi}_t\| \leq \sigma(F(\boldsymbol{p}_t))$ holds deterministically is to derive (18).

We apply Theorem C.1, Theorem C.2 identically to the proof earlier. This time, we have for $t < \mathcal{K}$ that $\boldsymbol{\xi}_{t+1}$ is sub-Gaussian with parameter $\sigma_1(\boldsymbol{w}_0)$, thanks to the same trick of multiplying with $1_{t<\mathcal{K}}$ when applying Theorem C.2.

The only change is as follows: in the definition $\mathcal{E}$, add in the intersection the event $\mathcal{E}_3$ that for all $1 \leq t \leq K_0$, $\|\boldsymbol{\xi}_t\|^2 \leq \sigma(F(\boldsymbol{p}_t))^2\log(K_0/p)$, where $p$ is defined the same as before. We control the probability of $\mathcal{E}_3$ via the following Lemma:

**Lemma C.2** (Equivalent of Lemma 12, De Sa et al. (2022)). *With probability at least $1 - p$, we have for all $1 \leq t \leq K_0$,*

$$\|\boldsymbol{\xi}_t\|^2 \leq \sigma(F(\boldsymbol{p}_t))^2\log(K_0/p).$$

**Proof.** By Assumption 3.1, with probability $1 - \frac{p}{K_0}$, we have

$$\frac{\|\boldsymbol{\xi}_t\|^2}{\sigma(F(\boldsymbol{p}_t))^2} \leq \log(K_0/p).$$

A Union Bound finishes the proof. $\qquad\square$

Now we condition on $\mathcal{E} = \mathcal{E}_1 \cap \mathcal{E}_2 \cap \mathcal{E}_3$, which has probability at least $1 - 2p$ by combining

our earlier argument with Lemma C.2. Note this only changes the resulting guarantee by a universal constant. We still have Lemma C.1, which does not require an upper bound on *each* $\|\boldsymbol{\xi}_t\|$ in its proof but simply uses concentration from event $\mathcal{E}_1$.

Thus, conditioned on $\mathcal{E}$, we still have $F(\boldsymbol{p}_t) \le F(\boldsymbol{w}_0) + 1$ by Lemma C.1, Lemma 3.1, and as $\boldsymbol{u}_0 \in \mathcal{L}_{F,F(\boldsymbol{w}_0)}$. Now conditioned on $\mathcal{E}$, by Lemma C.2, we still have the following upper bound for all $1 \le t \le K_0$:

$$\|\boldsymbol{\xi}_t\|^2 \le \sigma(F(\boldsymbol{w}_0) + 1)^2 \log(K_0/p) = \sigma_1(\boldsymbol{w}_0)^2 \log(K_0/p).$$

Therefore conditioned on $\mathcal{E}$, we can still derive a bound analogous to (18). This resulting bound changes by only a $\log(K_0/p)$ factor (from Lemma C.2, see the above display); moreover recall $K_0, p$ depend polynomially in $\delta, 1/\varepsilon$. By adjusting $\eta$ smaller by a $\mathrm{polylog}(K_0/p)$ factor, the same proof as above goes through, up to changing quantities by polylogarithmic factors.

# D  Perturbed GD finding Second Order Stationary Points

## D.1  Proof using the Framework

Here we prove Theorem 3.4. We instantiate Algorithm 1 formally here. The parameters of Algorithm 1 will depend on $L_1(\boldsymbol{w}_0), L_2(\boldsymbol{w}_0)$, which are defined in (4), (21) respectively, and depend only on $\rho_1, \rho_2, F(\boldsymbol{w}_0)$. Given a desired success probability $1 - \delta$ for $\delta > 0$, a tolerance $\varepsilon > 0$, and $F(\boldsymbol{w}_0), L_1(\boldsymbol{w}_0), L_2(\boldsymbol{w}_0)$, the algorithm's other parameters are defined in terms of as follows:

1. $c \le c_{\max}$ is a universal constant, where $c_{\max}$ is a universal constant defined in Lemma D.2.

2. $\tilde{\varepsilon} = \frac{\varepsilon}{L_2(\boldsymbol{w}_0)}$.

3. $\chi \leftarrow 4 \max\left\{\log\left(\frac{2dL_1(\boldsymbol{w}_0)^2 F(\boldsymbol{w}_0)}{c^2 \tilde{\varepsilon}^{2.5} \delta}\right), 5\right\}$.

4. $\eta \leftarrow \frac{c}{L_1(\boldsymbol{w}_0)}$.

5. $r \leftarrow \frac{\sqrt{c}\tilde{\varepsilon}}{\chi^2 L_1(\boldsymbol{w}_0)}$.

6. $g_{\mathrm{thres}} \leftarrow \frac{\sqrt{c}}{\chi^2}\tilde{\varepsilon}$.

7. $f_{\mathrm{thres}} \leftarrow \frac{c}{\chi^3}\sqrt{\frac{\tilde{\varepsilon}^3}{L_2(\boldsymbol{w}_0)}}$.

8. $t_{\mathrm{thres}} \leftarrow \frac{\chi}{c^2}\frac{L_1(\boldsymbol{w}_0)}{\sqrt{L_2(\boldsymbol{w}_0)\tilde{\varepsilon}}}$.

**Proof of Theorem 3.4 given Lemma D.2.** We will first prove the following Lemma, which will define $L_2(\boldsymbol{w}_0)$ and explain its significance.

**Lemma D.1.** *Define $L_1(\boldsymbol{w}_0)$ as in (4), and define*

$$L_2(\boldsymbol{w}_0) = \max\{1, L_1(\boldsymbol{w}_0), \rho_2(F(\boldsymbol{w}_0) + 1)\}. \tag{21}$$

*Then we have the following:*

1. *Suppose $\boldsymbol{u}$ is such that $\|\boldsymbol{u} - \tilde{\boldsymbol{w}}\| \le \frac{1}{\rho_0(F(\boldsymbol{w}_0)+1)}$, where $\tilde{\boldsymbol{w}} \in \mathcal{L}_{F,F(\boldsymbol{w}_0)}$, the $F(\boldsymbol{w}_0)$-sublevel set. Then under Assumption 1.1 (and in particular under Assumption 1.2),*

$$\left\|\nabla^2 F(\boldsymbol{u})\right\|_{\mathrm{op}} \le L_1(\boldsymbol{w}_0).$$

2. *Suppose that $\boldsymbol{u}_1, \boldsymbol{u}_2$ are such that $\|\boldsymbol{u}_1 - \tilde{\boldsymbol{w}}\|, \|\boldsymbol{u}_2 - \tilde{\boldsymbol{w}}\| \le \frac{1}{\rho_0(F(\boldsymbol{w}_0)+1)}$, where $\tilde{\boldsymbol{w}} \in \mathcal{L}_{F,F(\boldsymbol{w}_0)}$. Then*

$$\left\|\nabla^2 F(\boldsymbol{u}_1) - \nabla^2 F(\boldsymbol{u}_2)\right\|_{\mathrm{op}} \le L_2(\boldsymbol{w}_0)\|\boldsymbol{u}_1 - \boldsymbol{u}_2\|.$$

**Remark 8.** Note $L_1(\boldsymbol{w}_0), L_2(\boldsymbol{w}_0) \ge 1$, and $L_2(\boldsymbol{w}_0) \ge L_1(\boldsymbol{w}_0)$.

**Proof of Lemma D.1.** Recall by Corollary 1 that $\|\nabla F(\boldsymbol{w})\| \le \rho_0(F(\boldsymbol{w}))$. Now by Lemma 3.1 and as $\tilde{\boldsymbol{w}} \in \mathcal{L}_{F,F(\boldsymbol{w}_0)}$, for any $\boldsymbol{u}'$ with $\|\boldsymbol{u}' - \tilde{\boldsymbol{w}}\| \le \frac{1}{\rho_0(F(\boldsymbol{w}_0)+1)} \le \frac{1}{\rho_0(F(\tilde{\boldsymbol{w}})+1)}$, we have $F(\boldsymbol{u}') \le F(\tilde{\boldsymbol{w}}) + 1$. The first part now directly follows by Assumption 1.1.

---
**Algorithm 1** Perturbed Gradient Descent, modified from Jin et al. (2017).
---

$\tilde{\varepsilon} = \frac{\varepsilon}{L_2(\boldsymbol{w}_0)}$, $\chi \leftarrow 4\max\left\{\log\left(\frac{2dL_1(\boldsymbol{w}_0)^2 F(\boldsymbol{w}_0)}{c\tilde{\varepsilon}^{2.5}\delta}\right), 5\right\}$, $\eta \leftarrow \frac{c}{L_1(\boldsymbol{w}_0)}$, $r \leftarrow \frac{\sqrt{c}\tilde{\varepsilon}}{\chi^2 L_1(\boldsymbol{w}_0)}$, $g_{\text{thres}} \leftarrow \frac{\sqrt{c}}{\chi^2}\tilde{\varepsilon}$,

$f_{\text{thres}} \leftarrow \frac{c}{\chi^3}\sqrt{\frac{\tilde{\varepsilon}^3}{L_2(\boldsymbol{w}_0)}}$, $t_{\text{thres}} \leftarrow \frac{\chi}{c^2}\frac{L_1(\boldsymbol{w}_0)}{\sqrt{L_2(\boldsymbol{w}_0)\tilde{\varepsilon}}}$. Here $c$ refers to a small enough universal constant upper bounded by $c_{\max}$ in Lemma D.2.

**while** True **do**
    **if** $\|\nabla F(\boldsymbol{w}_t)\| \leq g_{\text{thres}}$ **then**
        $\tilde{\boldsymbol{w}}_t \leftarrow \boldsymbol{w}_t$, $t_{\text{noise}} \leftarrow t$
        $\boldsymbol{w}_t \leftarrow \tilde{\boldsymbol{w}}_t + \boldsymbol{\xi}_t$, $\boldsymbol{\xi}_t$ uniform from $\mathbb{B}(\vec{\boldsymbol{0}}, r)$
        $s \leftarrow 0$
        **while** $s < t_{\text{thres}}$ **do**
            $\boldsymbol{w}_{t+1} = \boldsymbol{w}_t - \eta\nabla F(\boldsymbol{w}_t)$, $s \leftarrow s+1$, $t \leftarrow t+1$
        **end while**
        **if** $F(\boldsymbol{w}_t) - F(\tilde{\boldsymbol{w}}_{t_{\text{noise}}}) > -f_{\text{thres}}$ **then**
            Return $\tilde{\boldsymbol{w}}_{t_{\text{noise}}}$
        **end if**
    **else**
        $\boldsymbol{w}_{t+1} = \boldsymbol{w}_t - \eta\nabla F(\boldsymbol{w}_t)$, $t \leftarrow t+1$
    **end if**
**end while**

---

The second part now follows by noting the line segment $\overline{\boldsymbol{u}_1\boldsymbol{u}_2}$ is contained in $\mathbb{B}\left(\tilde{\boldsymbol{w}}, \frac{1}{\rho_0(F(\boldsymbol{w}_0)+1)}\right)$ via Triangle Inequality, recalling $\tilde{\boldsymbol{w}} \in \mathcal{L}_{F,F(\boldsymbol{w}_0)}$, and then applying Lemma A.6 and Lemma 3.1. $\qquad\square$

We now prove Theorem 3.4 by instantiating our framework.

Define $\tilde{\varepsilon} = \frac{\varepsilon}{L_2(\boldsymbol{w}_0)}$ as we did earlier, and note $L_2(\boldsymbol{w}_0) \geq 1$. It suffices to show for $\tilde{\varepsilon} \leq 1$, that with probability at least $1 - \delta$, we will return $\boldsymbol{w}$ such that $\|\nabla F(\boldsymbol{w})\| \leq \tilde{\varepsilon}$, $\nabla^2 F(\boldsymbol{w}) \succeq -\sqrt{L_2(\boldsymbol{w}_0)\tilde{\varepsilon}}\boldsymbol{I}$ in $T = O\left(\frac{L_1(\boldsymbol{w}_0)\max\{F(\boldsymbol{w}_0),1\}\chi^4}{\tilde{\varepsilon}^2}\right) = O\left(\frac{L_1(\boldsymbol{w}_0)L_2(\boldsymbol{w}_0)^2\max\{F(\boldsymbol{w}_0),1\}\chi^4}{\varepsilon^2}\right)$ oracle calls.[7]

Now let the set of interest

$$\mathcal{S} = \{\boldsymbol{w} : \|\nabla F(\boldsymbol{w})\| \leq g_{\text{thres}}, \nabla^2 F(\boldsymbol{w}) \succeq -\sqrt{L_2(\boldsymbol{w}_0)\tilde{\varepsilon}}\boldsymbol{I}\}.$$

Note $g_{\text{thres}} \leq \tilde{\varepsilon}$, so $\boldsymbol{w} \in \mathcal{S}$ immediately implies $\|\nabla F(\boldsymbol{w})\| \leq \tilde{\varepsilon}$, $\nabla^2 F(\boldsymbol{w}) \succeq -\sqrt{L_2(\boldsymbol{w}_0)\tilde{\varepsilon}}\boldsymbol{I}$. Also note it suffices to show the result for all $\tilde{\varepsilon} \leq \frac{1}{100L_2(\boldsymbol{w}_0)}$; otherwise for larger $\tilde{\varepsilon}$ we can just apply the result for $\tilde{\varepsilon} = \frac{1}{100L_2(\boldsymbol{w}_0)}$. Thus as $L_2(\boldsymbol{w}_0) \geq 1$, we can assume $\tilde{\varepsilon} \leq 1$. Clearly, we also can assume WLOG that $t_{\text{thres}} \geq 1$.

As in Subsection 2.3, we make the following definitions for Algorithm 1. For all $\boldsymbol{u}_0 \in \mathbb{R}^d$, if $\|\nabla F(\boldsymbol{u}_0)\| > g_{\text{thres}}$, we let

$$\mathcal{A}(\boldsymbol{u}_0) = (\boldsymbol{u}_0 - \eta\nabla F(\boldsymbol{u}_0), \boldsymbol{u}_0), \text{ hence } \mathcal{A}_1(\boldsymbol{u}_0) = \boldsymbol{u}_0 - \eta\nabla F(\boldsymbol{u}_0), \mathcal{A}_2(\boldsymbol{u}_0) = \boldsymbol{u}_0.$$

Otherwise if $\|\nabla F(\boldsymbol{u}_0)\| \leq g_{\text{thres}}$, we let $\boldsymbol{p}_0 = \boldsymbol{u}_0 + \boldsymbol{\xi}$ where $\boldsymbol{\xi}$ is uniform from $\mathbb{B}(\vec{\boldsymbol{0}}, r)$, and define a sequence $(\boldsymbol{p}_i)_{0 \leq i \leq t_{\text{thres}}}$ via

$$\boldsymbol{p}_i = \boldsymbol{p}_{i-1} - \eta\nabla F(\boldsymbol{p}_{i-1}).$$

When then take

$$\mathcal{A}(\boldsymbol{u}_0) = (\boldsymbol{p}_{t_{\text{thres}}}, \boldsymbol{u}_0), \text{ hence } \mathcal{A}_1(\boldsymbol{u}_0) = \boldsymbol{p}_{t_{\text{thres}}}, \mathcal{A}_2(\boldsymbol{u}_0) = \boldsymbol{u}_0.$$

We then have

$$t_{\text{oracle}}(\boldsymbol{u}_0) = \begin{cases} t_{\text{thres}} & : \|\nabla F(\boldsymbol{u}_0)\| \leq g_{\text{thres}} \\ 1 & : \|\nabla F(\boldsymbol{u}_0)\| > g_{\text{thres}}. \end{cases}$$

We also define

$$\Delta(\boldsymbol{u}_0) = \begin{cases} f_{\text{thres}} & : \|\nabla F(\boldsymbol{u}_0)\| \leq g_{\text{thres}} \\ \frac{\eta}{2} \cdot g_{\text{thres}}^2 & : \|\nabla F(\boldsymbol{u}_0)\| > g_{\text{thres}}. \end{cases}$$

---

[7]The $\max\{1, F(\boldsymbol{w}_0)\}$ is a proof artifact.

We now establish the crucial Claim 2: for all $\boldsymbol{u}_0 \in \mathcal{L}_{F,F(\boldsymbol{w}_0)}$, $\mathcal{A}$ is a $(\mathcal{S}, t_{\text{oracle}}(\boldsymbol{u}_0), \Delta(\boldsymbol{u}_0), \frac{dL_1(\boldsymbol{w}_0)}{\sqrt{L_2(\boldsymbol{w}_0)\tilde{\varepsilon}}}e^{-\chi}, \boldsymbol{u}_0)$-decrease procedure. (Recall $\tilde{\varepsilon} = \frac{\varepsilon}{L_2(\boldsymbol{w}_0)}$.)

To do this, we use the following crucial Lemma ensuring high-probability decrease around saddle points *in the $F(\boldsymbol{w}_0)$-sublevel set*:

**Lemma D.2** (Equivalent of Lemma 13, Jin et al. (2017)). *There exists a universal constant $c_{max} \leq 1$ such that the following occurs. Suppose we start with a $\tilde{\boldsymbol{w}} \in \mathcal{L}_{F,F(\boldsymbol{w}_0)}$, that is in the $F(\boldsymbol{w}_0)$-sublevel set, satisfying the following conditions:*

$$\|\nabla F(\tilde{\boldsymbol{w}})\| \leq g_{\text{thres}} \quad \text{and} \quad \lambda_{\min}(\nabla^2 F(\tilde{\boldsymbol{w}})) \leq -\sqrt{L_2(\boldsymbol{w}_0)\tilde{\varepsilon}}.$$

*Now let $\boldsymbol{p}_0 = \tilde{\boldsymbol{w}} + \boldsymbol{\zeta}$, where $\boldsymbol{\zeta}$ is sampled uniformly from $\mathbb{B}(\vec{\boldsymbol{0}}, r)$ where $r$ is defined in Lemma D.3, and let $\{\boldsymbol{p}_t\}$ be the iterates of gradient descent starting from $\boldsymbol{p}_0$. Then when the step size $\eta \leq \frac{c_{max}}{L_1(\boldsymbol{w}_0)}$, with probability at least $1 - \frac{dL_1(\boldsymbol{w}_0)}{\sqrt{L_2(\boldsymbol{w}_0)\tilde{\varepsilon}}}e^{-\chi}$, we have:*

$$F(\boldsymbol{p}_{t_{\text{thres}}}) - F(\tilde{\boldsymbol{w}}) < -f_{\text{thres}}.$$

The variables in the above are defined in Algorithm 1. As noted earlier, because we work in the generalized smooth setting, the details require significant care compared to the proof of Lemma 13 in Jin et al. (2017).

With Lemma D.2, we have the ingredients to prove Theorem 3.4. First we establish Claim 2.

**Proof of Claim 2.** We prove this by breaking into the following cases:

- Suppose $\|\nabla F(\boldsymbol{u}_0)\| > g_{\text{thres}}$. Then $\boldsymbol{u}_1 = \mathcal{A}_1(\boldsymbol{u}_0) = \boldsymbol{u}_0 - \eta\nabla F(\boldsymbol{u}_0)$.

  Our condition on $\eta$ implies that

  $$\eta \leq \frac{1}{L_1(\boldsymbol{w}_0)} \leq \frac{1}{\rho_0(F(\boldsymbol{w}_0))\rho_0(F(\boldsymbol{w}_0)+1)}.$$

  As $\boldsymbol{u}_0 \in \mathcal{L}_{F,F(\boldsymbol{w}_0)}$, we have by Corollary 1,

  $$\|\boldsymbol{u}_1 - \boldsymbol{u}_0\| = \eta\|\nabla F(\boldsymbol{u}_0)\| \leq \eta\rho_0(F(\boldsymbol{u}_0)) \leq \eta\rho_0(F(\boldsymbol{w}_0)) \leq \frac{1}{\rho_0(F(\boldsymbol{w}_0)+1)}.$$

  Consequently, by Lemma 3.1,

  $$F(\boldsymbol{p}) \leq F(\boldsymbol{u}_0) + 1 \leq F(\boldsymbol{w}_0) + 1 \text{ for all } \boldsymbol{p} \in \overline{\boldsymbol{u}_0\boldsymbol{u}_1}.$$

  Now by Lemma A.1 and Assumption 1.1,

  $$\begin{aligned}
  F(\boldsymbol{u}_1) &\leq F(\boldsymbol{u}_0) - \eta\|\nabla F(\boldsymbol{u}_0)\|^2 + \frac{L_1(\boldsymbol{w}_0)\eta^2}{2}\|\nabla F(\boldsymbol{u}_0)\|^2 \\
  &\leq F(\boldsymbol{u}_0) - \frac{\eta}{2}\|\nabla F(\boldsymbol{u}_0)\|^2 \\
  &< F(\boldsymbol{u}_0) - \frac{\eta}{2} \cdot g_{\text{thres}}^2 = F(\boldsymbol{u}_0) - \Delta(\boldsymbol{u}_0).
  \end{aligned}$$

- Else suppose $\|\nabla F(\boldsymbol{u}_0)\| \leq g_{\text{thres}}$. Then $\boldsymbol{u}_0$ is perturbed, and we consider the sequence of the next $t_{\text{thres}}$ iterates $\boldsymbol{p}_0 = \boldsymbol{u}_0 + \boldsymbol{\xi}, \boldsymbol{p}_1, \ldots, \boldsymbol{p}_{t_{\text{thres}}}$.

  Consider the event $\mathcal{E}$ from Lemma D.2, which occurs with probability at least $1 - \frac{dL_1(\boldsymbol{w}_0)}{\sqrt{L_2(\boldsymbol{w}_0)\tilde{\varepsilon}}}e^{-\chi}$. Under $\mathcal{E}$, for such $\boldsymbol{u}_0$, we have:

  - Either
    $$F(\boldsymbol{p}_{t_{\text{thres}}}) - F(\boldsymbol{u}_0) < -f_{\text{thres}},$$
    that is
    $$F(\boldsymbol{u}_1) = F(\boldsymbol{p}_{t_{\text{thres}}}) < F(\boldsymbol{u}_0) - f_{\text{thres}}.$$

  - Or
    $$\lambda_{\text{MIN}}(\nabla^2 F(\boldsymbol{u}_0)) \geq -\sqrt{\tilde{\varepsilon}L_2(\boldsymbol{w}_0)}, \text{ hence } \boldsymbol{u}_0 \in \mathcal{S}.$$

In all cases, by definition of $\mathcal{A}$, we conclude that $\mathcal{A}$ is a $(\mathcal{S}, t_{\text{oracle}}(\boldsymbol{u}_0), \Delta(\boldsymbol{u}_0), \frac{dL_1(\boldsymbol{w}_0)}{\sqrt{L_2(\boldsymbol{w}_0)\tilde{\varepsilon}}}e^{-\chi}, \boldsymbol{u}_0)$ decrease procedure for $\boldsymbol{u}_0 \in \mathcal{L}_{F,F(\boldsymbol{w}_0)}$. $\qquad\square$

Consider these two cases, and recall the definition of $\overline{\Delta}$ from Theorem 2.1. Using the definition of $\eta, g_{\text{thres}}, f_{\text{thres}}$, we obtain for $c$ a small enough universal constant,

$$
\begin{aligned}
\overline{\Delta} &\ge \frac{1}{2}\min\left\{\frac{c^2\tilde{\varepsilon}^2}{2L_1(\boldsymbol{w}_0)\chi^4}, \frac{c^3\tilde{\varepsilon}^2}{\chi^4 L_1(\boldsymbol{w}_0)}\right\} \\
&\ge \frac{c^3\tilde{\varepsilon}^2}{\chi^4 L_1(\boldsymbol{w}_0)}.
\end{aligned}
$$

Combining with Theorem 2.1, and note $t_{\text{oracle}}(\boldsymbol{u}_0) \le t_{\text{thres}} \le \frac{\max\{1, F(\boldsymbol{w}_0)\}}{\overline{\Delta}}$ for $\tilde{\varepsilon} \le 1$. We thus obtain the desired oracle complexity of $O\left(\frac{L_1(\boldsymbol{w}_0)\max\{F(\boldsymbol{w}_0),1\}\chi^4}{\tilde{\varepsilon}^2}\right) = O\left(\frac{L_1(\boldsymbol{w}_0)L_2(\boldsymbol{w}_0)^2\max\{F(\boldsymbol{w}_0),1\}\chi^4}{\varepsilon^2}\right)$ to obtain an iterate in $\mathcal{S}$.[8]

We finally show the desired probability of success. Through Theorem 2.1, since $\chi \ge 18$ and by definition of $\chi$, we can verify that the probability of failure is at most

$$
\begin{aligned}
&\frac{dL_1(\boldsymbol{w}_0)}{\sqrt{L_2(\boldsymbol{w}_0)\tilde{\varepsilon}}}e^{-\chi} \cdot \sup_{\boldsymbol{u}\in\mathcal{L}_{F,F(\boldsymbol{w}_0)}}\left\{\frac{F(\boldsymbol{w}_0)}{\Delta(\boldsymbol{w})}\right\} \\
&\le \frac{dL_1(\boldsymbol{w}_0)}{\sqrt{L_2(\boldsymbol{w}_0)\tilde{\varepsilon}}}e^{-\chi} \cdot \frac{F(\boldsymbol{w}_0)}{\frac{c^2\tilde{\varepsilon}^2}{2\chi^4 L_1(\boldsymbol{w}_0)\sqrt{L_2(\boldsymbol{w}_0)}}} \\
&\le \chi^4 e^{-\chi}\frac{2dL_1(\boldsymbol{w}_0)^2 F(\boldsymbol{w}_0)}{c^2\tilde{\varepsilon}^{2.5}} \\
&\le e^{-\chi/4} \cdot \frac{2F(\boldsymbol{w}_0)dL_1^2(\boldsymbol{w}_0)}{c\tilde{\varepsilon}^{2.5}} \\
&\le \delta.
\end{aligned}
$$

This completes the proof, assuming Lemma D.2. $\qquad\square$

## D.2 Proving the key Lemma

We now prove Lemma D.2 to complete the proof. The rest of the proof is similar to that of Jin et al. (2017), but hinges crucially on the fact that the analysis in Jin et al. (2017) is 'local'.

Consider any $\gamma > 0$, and define the 'units' in a similar way as Jin et al. (2017), but now in terms of $L_1(\boldsymbol{w}_0), L_2(\boldsymbol{w}_0) > 0$ defined earlier. First let the new 'condition number' be $\kappa = \kappa(\boldsymbol{w}_0) \coloneqq \frac{L_1(\boldsymbol{w}_0)}{\gamma}$ (note this is *not* the real condition number, but rather is the 'effective condition number' of $\nabla^2 F$ in $\mathcal{L}_{F,F(\boldsymbol{w}_0)}$). Now define the following positive reals:

$$
\begin{aligned}
\mathcal{F}_1 &= \eta L_1(\boldsymbol{w}_0)\frac{\gamma^3}{L_2(\boldsymbol{w}_0)^2}\log^{-3}\left(\frac{d\kappa}{\delta}\right), \\
\mathcal{F}_2 &= \frac{\log\left(\frac{d\kappa}{\delta}\right)}{\eta\gamma}, \\
\mathcal{G} &= \sqrt{\eta L_1(\boldsymbol{w}_0)}\frac{\gamma^2}{L_2(\boldsymbol{w}_0)}\log^{-2}\left(\frac{d\kappa}{\delta}\right), \\
\mathcal{L} &= \sqrt{\eta L_1(\boldsymbol{w}_0)}\frac{\gamma}{L_2(\boldsymbol{w}_0)}\log^{-1}\left(\frac{d\kappa}{\delta}\right).
\end{aligned}
$$

Our goal is to prove the following.

---

[8]Note $t_{\text{thres}}$ generally does not decrease with $F(\boldsymbol{w}_0)$, and this is why the $\max\{1, F(\boldsymbol{w}_0)\}$ comes in.

**Lemma D.3** (equivalent of Lemma 14 in Jin et al. (2017)). *There exists a universal constant $c_{max}$ such that the following holds. For any $F$ satisfying the conditions of Theorem 3.4, for any $\delta \in \left(0, \frac{d\kappa}{e}\right]$, suppose we start with a point $\tilde{\boldsymbol{w}} \in \mathcal{L}_{F,F(\boldsymbol{w}_0)}$ satisfying the following conditions for some $\gamma > 0$, where $\mathcal{G}$ is defined as above:*

$$\|\nabla F(\tilde{\boldsymbol{w}})\| \leq \mathcal{G} \quad and \quad \lambda_{\min}(\nabla^2 F(\tilde{\boldsymbol{w}})) \leq -\gamma.$$

*Let $\boldsymbol{p}_0 = \tilde{\boldsymbol{w}} + \boldsymbol{\zeta}$, where $\boldsymbol{\zeta}$ is sampled from the uniform distribution over a ball with radius $\frac{\mathcal{L}}{\kappa \cdot \log\left(\frac{d\kappa}{\delta}\right)} := r$ and where $\mathcal{L}$ is defined as above. Let $\{\boldsymbol{p}_t\}$ be the iterates of gradient descent starting from $\boldsymbol{p}_0$. Then, when the step size $\eta \leq \frac{c_{max}}{L_1(\boldsymbol{w}_0)}$, with probability at least $1 - \delta$, we have the following for any $T \geq \frac{1}{c_{max}}\mathcal{F}_2$:*

$$F(\boldsymbol{p}_T) - F(\tilde{\boldsymbol{w}}) < -\mathcal{F}_1.$$

Plugging in $\gamma = \sqrt{L_2(\boldsymbol{w}_0)\tilde{\varepsilon}}$, $\eta = \frac{c_{max}}{L_1(\boldsymbol{w}_0)}$, $\delta = \frac{dL_1(\boldsymbol{w}_0)}{\sqrt{L_2(\boldsymbol{w}_0)\tilde{\varepsilon}}}e^{-\chi}$ into the above expressions for $\mathcal{F}_1, \mathcal{F}_2, \mathcal{G}, \mathcal{L}$, using $c \leq c_{max}$, and directly applying Lemma D.3, we immediately obtain Lemma D.2. The rest of Section D is thus devoted to proving Lemma D.3.

**Remark 9.** Note it suffices to prove Lemma D.3 for $\delta$ and $\gamma$ smaller than universal constants, as the result Theorem 3.4 will remain identical under the $O(\cdot)$. Thus we can assume WLOG that $\log(d\kappa/\delta)$ is larger than some universal constant, and that $\gamma \leq \frac{1}{60}$. Also notice by our choice of step size $\eta \leq \frac{c_{max}}{L_1(\boldsymbol{w}_0)}$ and the assumption $\gamma \leq \frac{1}{60}$, for $c \leq c_{max} \leq \frac{1}{12100}$ we obtain

$$\kappa \geq 1, r \leq 1.$$

This in turn implies

$$\mathcal{G} \leq \mathcal{L},$$
$$\mathcal{F}_2 \geq 40,$$
$$\mathcal{L} \leq \sqrt{\eta L_1(\boldsymbol{w}_0)} \cdot \frac{\gamma}{L_2(\boldsymbol{w}_0)} \cdot \log^{-1}\left(\frac{d\kappa}{\delta}\right)$$
$$\leq \frac{1}{6600} \cdot \min\left\{1, \frac{1}{\rho_0(F(\boldsymbol{w}_0) + 1)}, \frac{1}{\rho_0(F(\boldsymbol{w}_0))\rho_0(F(\boldsymbol{w}_0) + 1)}\right\},$$

where the second line uses that

$$L_2(\boldsymbol{w}_0) \geq L_1(\boldsymbol{w}_0) \geq \max\{1, \rho_0(F(\boldsymbol{w}_0) + 1), \rho_0(F(\boldsymbol{w}_0))\rho_0(F(\boldsymbol{w}_0) + 1)\}.$$

As *these assumptions come with no loss of generality*, we make these assumptions for the rest of the proof.

To show Lemma D.3, again as in Jin et al. (2017), we prove that the width of the stuck region is not too large.

**Lemma D.4** (equivalent of Lemma 15 in Jin et al. (2017)). *There exists a universal constant $c_{max}$ such that the following occurs. For any $\delta \in \left(0, \frac{d\kappa}{e}\right]$, let $F$ and $\tilde{\boldsymbol{w}}$ satisfy the conditions in Lemma D.3. Without loss of generality, by rotational symmetry, let $\boldsymbol{e}_1$ be the minimum eigenvector of $\nabla^2 F(\tilde{\boldsymbol{w}})$. Consider two gradient descent sequences $\{\boldsymbol{u}_t\}$ and $\{\boldsymbol{x}_t\}$ with initial points $\boldsymbol{u}_0, \boldsymbol{x}_0$ satisfying (again, denote the radius $r = \frac{\mathcal{L}}{\kappa \cdot \log\left(\frac{d\kappa}{\delta}\right)}$):*

$$\|\boldsymbol{u}_0 - \tilde{\boldsymbol{w}}\| \leq r, \quad \boldsymbol{x}_0 = \boldsymbol{u}_0 \pm \mu \cdot r \cdot \boldsymbol{e}_1, \quad \mu \in \left[\frac{\delta}{2\sqrt{d}}, 1\right].$$

*Then for any step size $\eta \leq \frac{c_{max}}{L_1(\boldsymbol{w}_0)}$, and any $T \geq \frac{1}{c_{max}}\mathcal{F}_2$, we have:*

$$\min\{F(\boldsymbol{u}_T) - F(\boldsymbol{u}_0), F(\boldsymbol{x}_T) - F(\boldsymbol{x}_0)\} \leq -2.5\mathcal{F}_1.$$

Now, we prove Lemma D.3 given Lemma D.4.

**Proof of Lemma D.3 given Lemma D.4.** Recall as per Remark 9 that

$$\|\boldsymbol{p}_0 - \tilde{\boldsymbol{w}}\| \leq r \leq \mathcal{L} \leq \frac{1}{\rho_0(F(\boldsymbol{w}_0) + 1)}.$$

Also recall $\tilde{\boldsymbol{w}} \in \mathcal{L}_{F,F(\boldsymbol{w}_0)}$. Thus by Lemma D.1 we obtain for all $\boldsymbol{u} \in \overline{\boldsymbol{p}_0\tilde{\boldsymbol{w}}}$ that

$$\left\|\nabla^2 F(\boldsymbol{u})\right\|_{\text{op}} \le L_1(\boldsymbol{w}_0).$$

Therefore by Lemma A.1,

$$F(\boldsymbol{p}_0) \le F(\tilde{\boldsymbol{w}}) + \|\nabla F(\tilde{\boldsymbol{w}})\|r + \frac{L_1(\boldsymbol{w}_0)}{2}r^2 \le F(\tilde{\boldsymbol{w}}) + \mathcal{G}r + \frac{L_1(\boldsymbol{w}_0)}{2}r^2 = F(\tilde{\boldsymbol{w}}) + \mathcal{F}_1,$$

where we can readily verify from Remark 9 that $\mathcal{G}r + \frac{L_1(\boldsymbol{w}_0)}{2}r^2 \le \mathcal{F}_1$.

Now let the stuck region be the set of points $\boldsymbol{p}_0$ in $\mathbb{B}(\tilde{\boldsymbol{w}}, r)$ such that

$$F(\boldsymbol{p}_T) - F(\boldsymbol{p}_0) \ge -2.5\mathcal{F}_1.$$

Define the unstuck points by the complement of the stuck points.

We upper bound the volume of the stuck region as done in Jin et al. (2017); this step does not use gradient and Hessian Lispchitzness. Let $1_{\text{Stuck Region}}(\cdot)$ be the indicator function of the stuck region. Write all $\boldsymbol{w} \in \mathbb{R}^d$ as $\boldsymbol{w} = (\boldsymbol{w}^{(1)}, \boldsymbol{w}^{(-1)})$, where $\boldsymbol{w}^{(1)}$ is the component of $\boldsymbol{w}$ along $\boldsymbol{e}_1$ direction and $\boldsymbol{w}^{(-1)}$ is the component of $\boldsymbol{w}$ along the orthogonal complement of $\boldsymbol{e}_1$. By Lemma D.4, for any $\boldsymbol{w} \in \mathbb{B}(\tilde{\boldsymbol{w}}, r)$,

$$1_{\text{Stuck region}}(\boldsymbol{w})\mathrm{d}\boldsymbol{w} = 1_{\text{Stuck region}}(\boldsymbol{w})\mathrm{d}\boldsymbol{w}^{(-1)} \int_{\tilde{\boldsymbol{w}}-\sqrt{r^2-\|\tilde{\boldsymbol{w}}^{(-1)}-\boldsymbol{w}^{(-1)}\|^2}}^{\tilde{\boldsymbol{w}}+\sqrt{r^2-\|\tilde{\boldsymbol{w}}^{(-1)}-\boldsymbol{w}^{(-1)}\|^2}} \mathrm{d}\boldsymbol{w}^{(1)}$$

$$\le \mathrm{d}\boldsymbol{w}^{(-1)} \cdot 2 \cdot \frac{\delta}{2\sqrt{d}}r.$$

Using this, we have:

$$\text{Volume(Stuck region)} = \int_{\mathbb{B}^d(\tilde{\boldsymbol{w}}, r)} 1_{\text{Stuck region}}(\boldsymbol{w})\mathrm{d}\boldsymbol{w}$$

$$= \int_{\mathbb{B}^{d-1}(\tilde{\boldsymbol{w}}, r)} 1_{\text{Stuck region}}(\boldsymbol{w})\mathrm{d}\boldsymbol{w}^{(-1)} \int_{\tilde{\boldsymbol{w}}-\sqrt{r^2-\|\tilde{\boldsymbol{w}}^{(-1)}-\boldsymbol{w}^{(-1)}\|^2}}^{\tilde{\boldsymbol{w}}+\sqrt{r^2-\|\tilde{\boldsymbol{w}}^{(-1)}-\boldsymbol{w}^{(-1)}\|^2}} \mathrm{d}\boldsymbol{w}^{(1)}$$

$$\le \int_{\mathbb{B}^{d-1}(\tilde{\boldsymbol{w}}, r)} \mathrm{d}\boldsymbol{w}^{(-1)} \cdot 2 \cdot \frac{\delta}{2\sqrt{d}}r.$$

$$= \text{Volume}(\mathbb{B}^{d-1}(\vec{\boldsymbol{0}}, r)) \cdot \frac{\delta r}{\sqrt{d}}.$$

Then letting $\Gamma(\cdot)$ denote the Gamma function, we have the following ratio:

$$\frac{\text{Volume(Stuck region)}}{\text{Volume}(\mathbb{B}(\tilde{\boldsymbol{w}}, r))} \le \frac{\delta r}{\sqrt{d}} \cdot \frac{\text{Volume}(\mathbb{B}^{d-1}(\vec{\boldsymbol{0}}, r))}{\text{Volume}(\mathbb{B}^d(\vec{\boldsymbol{0}}, r))}$$

$$= \frac{\delta}{\sqrt{\pi d}} \cdot \frac{\Gamma\left(\frac{d}{2} + 1\right)}{\Gamma\left(\frac{d}{2} + \frac{1}{2}\right)}$$

$$\le \frac{\delta}{\sqrt{\pi d}} \cdot \sqrt{\frac{d}{2} + \frac{1}{2}} \le \delta.$$

Here we use the following property of the Gamma function: for $x \ge 0$, $\frac{\Gamma(x+1)}{\Gamma(\frac{x}{2}+\frac{1}{2})} \le \sqrt{x + \frac{1}{2}}$.

This directly implies that with probability at least $1 - \delta$, $\boldsymbol{p}_0$ is an unstuck point. Consequently with probability at least $1 - \delta$, for any $T \ge \frac{1}{c_{\max}}\mathcal{F}_2$, we have

$$F(\boldsymbol{p}_T) - F(\tilde{\boldsymbol{w}}) = F(\boldsymbol{p}_T) - F(\boldsymbol{p}_0) + F(\boldsymbol{p}_0) - F(\tilde{\boldsymbol{w}}) \le -2.5\mathcal{F}_1 + \mathcal{F}_1 = -1.5\mathcal{F}_1 < -\mathcal{F}_1.$$

This proves Lemma D.3. $\qquad\square$

Now we prove Lemma D.4, which we do with an analogous strategy as Jin et al. (2017) by coupling two gradient descent sequences. We have the following two Lemmas, analogous to

Lemmas 16, 17 in Jin et al. (2017). Again, the reason why they hold in our setting under generalized smoothness is because they all concern 'local' behavior around points in the sublevel set of $F(\boldsymbol{w}_0)$. Consequently Lemma 3.1 and Assumption 1.2 ensure we have the required 'local' smoothness properties.

Again define $\boldsymbol{H}, \tilde{F}_{\boldsymbol{y}}(\boldsymbol{x})$ analogously to page 20, Jin et al. (2017), as follows:

$$\boldsymbol{H} \coloneqq \nabla^2 F(\tilde{\boldsymbol{w}}), \tilde{F}_{\boldsymbol{y}}(\boldsymbol{x}) \coloneqq F(\boldsymbol{y}) + \langle \nabla F(\boldsymbol{y}), \boldsymbol{x} - \boldsymbol{y} \rangle + \frac{1}{2}(\boldsymbol{x} - \boldsymbol{y})^\top \boldsymbol{H}(\boldsymbol{x} - \boldsymbol{y}). \tag{22}$$

That is, $\tilde{F}_{\boldsymbol{y}}$ is a quadratic approximation of $F$, Taylor expanded about $\tilde{\boldsymbol{w}}$.

The aforementioned Lemmas are as follows:

**Lemma D.5** (equivalent of Lemma 16 in Jin et al. (2017)). *Letting $\hat{c} = 11$, there exists a universal constant $c_{max} \le \frac{1}{12100}$ such that following holds. For any $\delta \in (0, \frac{d\kappa}{e}]$, consider $F, \tilde{\boldsymbol{w}}, r$ as in Lemma D.3. For any $\boldsymbol{u}_0$ with $\|\boldsymbol{u}_0 - \tilde{\boldsymbol{w}}\| \le 2r = \frac{2\mathcal{L}}{\kappa \cdot \log\left(\frac{d\kappa}{\delta}\right)}$, define*

$$T = \min\left\{\inf_t\left\{t \mid \tilde{F}_{\boldsymbol{u}_0}(\boldsymbol{u}_t) - F(\boldsymbol{u}_0) \le -3\mathcal{F}_1\right\}, \hat{c}\mathcal{F}_2\right\}.$$

*Then for any $\eta \le \frac{c_{max}}{L(\boldsymbol{w}_0)}$, we have for all $t < T$ that $\|\boldsymbol{u}_t - \tilde{\boldsymbol{w}}\| \le 150\mathcal{L}\hat{c}$.*

**Lemma D.6** (equivalent of Lemma 17 in Jin et al. (2017)). *Letting $\hat{c} = 11$, there exists a universal constant $c_{max} \le \frac{1}{12100}$ such that the following holds. For any $\delta \in \left(0, \frac{d\kappa}{e}\right]$, consider $F, \tilde{\boldsymbol{w}}, r$ as in Lemma D.3, and sequences $\{\boldsymbol{u}_t\}$, $\{\boldsymbol{x}_t\}$ satisfying the conditions in Lemma D.4. Define:*

$$T = \min\left\{\inf_t\left\{t \mid \tilde{F}_{\boldsymbol{x}_0}(\boldsymbol{x}_t) - F(\boldsymbol{x}_0) \le -3\mathcal{F}_1\right\}, \hat{c}\mathcal{F}_2\right\}.$$

*Then, for any $\eta \le \frac{c_{max}}{L_1(\boldsymbol{w}_0)}$, if $\|\boldsymbol{u}_t - \tilde{\boldsymbol{w}}\| \le 150\mathcal{L}\hat{c}$ for all $t < T$, we will have $T < \hat{c}\mathcal{F}_2$. Equivalently, this means that*

$$\inf_t\left\{t : \tilde{F}_{\boldsymbol{x}_0}(\boldsymbol{x}_t) - F(\boldsymbol{x}_0) \le -3\mathcal{F}_1\right\} < \hat{c}\mathcal{F}_2,$$

*i.e. that we escaped the saddle point.*

**Proof of Lemma D.4 given Lemma D.5, Lemma D.6.** Choosing $c_{max}$ to be the minimum of the $c_{max}$ from Lemma D.5, Lemma D.6, we can ensure both Lemmas hold. Clearly this preserves that $c_{max} \le \frac{1}{12100}$.

Define

$$T^\star = \hat{c}\mathcal{F}_2, T' = \inf\left\{t : \tilde{F}_{\boldsymbol{u}_0}(\boldsymbol{u}_t) - F(\boldsymbol{u}_0) \le -3\mathcal{F}_1\right\}.$$

We break into cases on $T'$ versus $T^\star$:

- $T' \le T^\star$: By Lemma D.5, $\|\boldsymbol{u}_{T'-1} - \tilde{\boldsymbol{w}}\| \le 150\mathcal{L}\hat{c}$. Since $\mathcal{L} \le \frac{1}{6600} \cdot \frac{1}{\rho_0(F(\boldsymbol{w}_0)+1)}$ from Remark 9 and $\hat{c} = 11$, this yields

$$\|\boldsymbol{u}_{T'-1} - \tilde{\boldsymbol{w}}\| \le 150\mathcal{L}\hat{c} \le \frac{1}{4} \cdot \frac{1}{\rho_0(F(\boldsymbol{w}_0) + 1)}.$$

Thus because $\tilde{\boldsymbol{w}} \in \mathcal{L}_{F, F(\boldsymbol{w}_0)}$, by Lemma D.1, we have

$$\left\|\nabla^2 F(\boldsymbol{u})\right\| \le L_1(\boldsymbol{w}_0) \text{ for all } \boldsymbol{u} \in \overline{\boldsymbol{u}_{T'-1}\tilde{\boldsymbol{w}}}.$$

Thus, recalling $\mathcal{G} \le \mathcal{L}$ from Remark 9, we obtain

$$\|\nabla F(\boldsymbol{u}_{T'-1})\| \le \|\nabla F(\tilde{\boldsymbol{w}})\| + L_1(\boldsymbol{w}_0)\|\boldsymbol{u}_{T'-1} - \tilde{\boldsymbol{w}}\|$$
$$\le \mathcal{G} + 150\hat{c}L_1(\boldsymbol{w}_0)\mathcal{L} \le \mathcal{L} + 150\hat{c}L_1(\boldsymbol{w}_0)\mathcal{L}.$$

Therefore, as $\eta L_1(\boldsymbol{w}_0) \le c_{max} \le 1$,

$$\|\boldsymbol{u}_{T'} - \tilde{\boldsymbol{w}}\| \le \|\boldsymbol{u}_{T'-1} - \tilde{\boldsymbol{w}}\| + \eta\|\nabla F(\boldsymbol{u}_{T'-1})\|$$
$$\le 150\mathcal{L}\hat{c} + \mathcal{L} + 150\hat{c} \cdot \eta L_1(\boldsymbol{w}_0)\mathcal{L} \le (300\hat{c} + 1)\mathcal{L} \tag{23}$$

Recalling $\kappa, \log\left(\frac{d\kappa}{\delta}\right) \ge 1$, the conditions of Lemma D.4 give

$$\|\boldsymbol{u}_0 - \tilde{\boldsymbol{w}}\| \le r \le \mathcal{L}. \tag{24}$$

Combining (23), (24) and applying Triangle Inequality gives

$$\|\boldsymbol{u}_{T'} - \boldsymbol{u}_0\| \le (300\hat{c} + 2)\mathcal{L}. \tag{25}$$

Also by (24), we have $\|\boldsymbol{u}_0 - \tilde{\boldsymbol{w}}\| \le \mathcal{L} \le \frac{1}{\rho_0(F(\boldsymbol{w}_0)+1)}$. Thus as $\tilde{\boldsymbol{w}} \in \mathcal{L}_{F,F(\boldsymbol{w}_0)}$, by Lemma D.1 we obtain

$$\left\|\nabla^2 F(\boldsymbol{u}_0)\right\| \le L_1(\boldsymbol{w}_0). \tag{26}$$

Moreover, by Triangle Inequality we obtain that for any $\boldsymbol{u} \in \overline{\boldsymbol{u}_0\boldsymbol{u}_{T'}}$, we have

$$\|\boldsymbol{u} - \tilde{\boldsymbol{w}}\| \le (300\hat{c} + 2)\mathcal{L} = 3302\mathcal{L} \le \frac{1}{\rho_0(F(\boldsymbol{w}_0) + 1)}.$$

As $\tilde{\boldsymbol{w}} \in \mathcal{L}_{F,F(\boldsymbol{w}_0)}$, Lemma D.1 implies for all such $\boldsymbol{u}_1, \boldsymbol{u}_2 \in \overline{\boldsymbol{u}_0\boldsymbol{u}_{T'}}$ that

$$\left\|\nabla^2 F(\boldsymbol{u}_1) - \nabla^2 F(\boldsymbol{u}_2)\right\|_{\text{op}} \le \|\boldsymbol{u}_1 - \boldsymbol{u}_2\| L_2(\boldsymbol{w}_0).$$

Now applying Lemma A.2, and by choosing $\eta = \frac{c}{L(\boldsymbol{w}_0)}$ for a small enough universal constant $c$, we obtain:

$$
\begin{aligned}
&F(\boldsymbol{u}_{T'}) - F(\boldsymbol{u}_0) \\
&\le \nabla F(\boldsymbol{u}_0)^\top (\boldsymbol{u}_{T'} - \boldsymbol{u}_0) + \frac{1}{2}(\boldsymbol{u}_{T'} - \boldsymbol{u}_0)^\top \nabla^2 F(\boldsymbol{u}_0)(\boldsymbol{u}_{T'} - \boldsymbol{u}_0) + \frac{L_2(\boldsymbol{w}_0)}{6}\|\boldsymbol{u}_{T'} - \boldsymbol{u}_0\|^3 \\
&\le \tilde{F}_{\boldsymbol{u}_0}(\boldsymbol{u}_{T'}) - F(\boldsymbol{u}_0) + \frac{L_2(\boldsymbol{w}_0)}{2}\|\boldsymbol{u}_{T'} - \boldsymbol{u}_0\|^2\|\boldsymbol{u}_0 - \tilde{\boldsymbol{w}}\| + \frac{L_2(\boldsymbol{w}_0)}{6}\|\boldsymbol{u}_{T'} - \boldsymbol{u}_0\|^3 \\
&\le -3\mathcal{F}_1 + O(L_1(\boldsymbol{w}_0)\mathcal{L}^3) \\
&= -3\mathcal{F}_1 + O(\sqrt{\eta L_1(\boldsymbol{w}_0)}\mathcal{F}_1) \le -2.5\mathcal{F}_1.
\end{aligned}
$$

Here we used (26), (24), (25), and that $\mathcal{L} \le 1$ as per Remark 9. In the above, $O(\cdot)$ only hides universal constants as $\hat{c} = 11$ is a universal constant, and so these final inequalities can be made to hold by choosing $c_{\max}$ a sufficiently small universal constant.

Since $\tilde{\boldsymbol{w}} \in \mathcal{L}_{F,F(\boldsymbol{w}_0)}$ and $\eta \le \frac{2}{L_1(\boldsymbol{w}_0)}$, Lemma A.7 shows that gradient descent will not increase value (this is essentially the same as several steps the proof of Theorem 3.1, combined with induction). Thus for all $T \ge T'$ and hence for all $T \ge \frac{1}{c_{\max}}\mathcal{F}_2 \ge \hat{c}\mathcal{F}_2 \ge T'$ along this gradient descent trajectory, we have

$$F(\boldsymbol{u}_T) - F(\boldsymbol{u}_0) \le F(\boldsymbol{u}_{T'}) - F(\boldsymbol{u}_0) \le -2.5\mathcal{F}_1.$$

- $T' > T^\star$: In this case, by Lemma D.5, we know $\|\boldsymbol{u}_t - \tilde{\boldsymbol{w}}\| \le 150\mathcal{L}\hat{c}$ for all $t < T^\star = \hat{c}\mathcal{F}_2$.

  Define

  $$T'' = \inf_t \left\{ t \mid \tilde{F}_{\boldsymbol{x}_0}(\boldsymbol{x}_t) - F(\boldsymbol{x}_0) \le -3\mathcal{F}_1 \right\}.$$

  Since $\|\boldsymbol{u}_t - \tilde{\boldsymbol{w}}\| \le 150\mathcal{L}\hat{c}$ for all $t < T^\star = \hat{c}\mathcal{F}_2$, it follows that $\|\boldsymbol{u}_t - \tilde{\boldsymbol{w}}\| \le 150\mathcal{L}\hat{c}$ for all $t < \min\{T'', T^\star\}$. Thus by Lemma D.6, we have that $\min\{T'', T^\star\} < T^\star$, and so $T'' < T^\star$. Applying the same argument as in the first case to the $\{\boldsymbol{x}_t\}$, we have that for all $T \ge \frac{1}{c_{\max}}\mathcal{F}_2$ that

  $$F(\boldsymbol{x}_T) - F(\boldsymbol{x}_0) \le -2.5\mathcal{F}_1.$$

This proves Lemma D.4. $\qquad\square$

**Remark 10.** Note that $\tilde{\boldsymbol{w}} \in \mathcal{L}_{F,F(\boldsymbol{w}_0)}$ is central to this argument, unlike the Lipschitz gradient and Hessian case from Jin et al. (2017).

### D.3 Proof of Escaping Saddles Lemmas

Now we prove Lemma D.5, Lemma D.6.

**Proof of Lemma D.5.** We follow the proof of Lemma 16, Jin et al. (2017). Again, we aim to show that if the function value does not decrease, then all the iterates must remain constrained in a small ball. This is done by analyzing the dynamics of the iterates and decomposing the $d$-dimensional space into two subspaces: a subspace $S$, which is the span of the negative enough eigenvectors of the Hessian, and its orthogonal complement.

The main difference now is that now we cannot directly control relevant operator norms with global Lipschitz properties of the gradient and Hessian. However, it turns out that the proof of this Lemma will follow induction on the iterate $\boldsymbol{u}_t$, and consequently we will obtain that all of the prior iterates $\boldsymbol{u}_{t'}$ for $t' < t$ are close enough to $\tilde{\boldsymbol{w}}$. By a similar argument as in Lemma D.3, since $\tilde{\boldsymbol{w}} \in \mathcal{L}_{F,F(\boldsymbol{w}_0)}$, this lets us upper bound the gradient of these points. By the Gradient Descent update rule, this in turn implies the current iterate is also close to $\tilde{\boldsymbol{w}}$, and thus we obtain bounds on the relevant derivatives in terms of $L_1(\boldsymbol{w}_0), L_2(\boldsymbol{w}_0)$ for all points in the convex hull of the relevant iterates.

We begin the argument. Analogously to Jin et al. (2017), since $\delta \in \left(0, \frac{d\kappa}{e}\right]$, we always have $\log\left(\frac{d\kappa}{\delta}\right) \geq 1$. By the gradient descent update function, we have

$$\boldsymbol{u}_{t+1} = \boldsymbol{u}_t - \eta \nabla F(\boldsymbol{u}_t).$$

This can be expanded as:

$$\boldsymbol{u}_{t+1} = \boldsymbol{u}_t - \eta \nabla F(\boldsymbol{u}_0) - \eta \left( \int_0^1 \nabla^2 F(\theta(\boldsymbol{u}_t - \boldsymbol{u}_0) + \boldsymbol{u}_0) \mathrm{d}\theta \right)(\boldsymbol{u}_t - \boldsymbol{u}_0).$$

Recall the definition $\boldsymbol{H} = \nabla^2 F(\tilde{\boldsymbol{w}})$. Let $\Delta_t$ be defined as:

$$\Delta_t := \int_0^1 \nabla^2 F(\theta(\boldsymbol{u}_t - \boldsymbol{u}_0) + \boldsymbol{u}_0) \mathrm{d}\theta - \boldsymbol{H}.$$

Substituting, we obtain:

$$\boldsymbol{u}_{t+1} = (\boldsymbol{I} - \eta \boldsymbol{H} - \eta \Delta_t)(\boldsymbol{u}_t - \boldsymbol{u}_0) - \eta \nabla F(\boldsymbol{u}_0) + \boldsymbol{u}_0.$$

Note we do not immediately have an upper bound on the operator norm of $\Delta_t$. In particular this is because $t$ could diverge (logarithmically) in the dimension, only being upper bounded by $\mathcal{F}_2$.

We now compute the projections of $\boldsymbol{u}_t - \boldsymbol{u}_0$ in different eigenspaces of $\boldsymbol{H}$. Define $\mathcal{S}$ as the subspace spanned by all eigenvectors of $\boldsymbol{H}$ whose eigenvalues are less than $-\frac{\gamma}{\hat{c}\log\left(\frac{d\kappa}{\delta}\right)}$. Let $\mathcal{S}^c$ denote the subspace of the remaining eigenvectors. Let $\boldsymbol{\alpha}_t$ and $\boldsymbol{\beta}_t$ denote the projections of $\boldsymbol{u}_t - \boldsymbol{u}_0$ onto $\mathcal{S}$ and $\mathcal{S}^c$ respectively, i.e., $\boldsymbol{\alpha}_t = \mathcal{P}_{\mathcal{S}}(\boldsymbol{u}_t - \boldsymbol{u}_0)$, and $\boldsymbol{\beta}_t = \mathcal{P}_{\mathcal{S}^c}(\boldsymbol{u}_t - \boldsymbol{u}_0)$.

We can decompose the update equations for $\boldsymbol{u}_{t+1}$ into:

$$\boldsymbol{\alpha}_{t+1} = (\boldsymbol{I} - \eta \boldsymbol{H})\boldsymbol{\alpha}_t - \eta \mathcal{P}_{\mathcal{S}} \Delta_t (\boldsymbol{u}_t - \boldsymbol{u}_0) - \eta \mathcal{P}_{\mathcal{S}} \nabla F(\boldsymbol{u}_0),$$

$$\boldsymbol{\beta}_{t+1} = (\boldsymbol{I} - \eta \boldsymbol{H})\boldsymbol{\beta}_t - \eta \mathcal{P}_{\mathcal{S}^c} \Delta_t (\boldsymbol{u}_t - \boldsymbol{u}_0) - \eta \mathcal{P}_{\mathcal{S}^c} \nabla F(\boldsymbol{u}_0).$$

By the definition of $T$, we know for all $t < T$:

$$-3\mathcal{F}_1 < \tilde{F}_{\boldsymbol{u}_0}(\boldsymbol{u}_t) - F(\boldsymbol{u}_0) = \nabla F(\boldsymbol{u}_0)^\top (\boldsymbol{u}_t - \boldsymbol{u}_0) - \frac{1}{2}(\boldsymbol{u}_t - \boldsymbol{u}_0)^\top \boldsymbol{H}(\boldsymbol{u}_t - \boldsymbol{u}_0)$$

$$\leq \nabla F(\boldsymbol{u}_0)^\top (\boldsymbol{u}_t - \boldsymbol{u}_0) - \frac{\gamma}{2} \frac{\|\boldsymbol{\alpha}_t\|^2}{\hat{c}\log\left(\frac{d\kappa}{\delta}\right)} + \frac{1}{2}\boldsymbol{\beta}_t^\top \boldsymbol{H} \boldsymbol{\beta}_t.$$

Evidently we have $\|\boldsymbol{u}_t - \boldsymbol{u}_0\|^2 = \|\boldsymbol{\alpha}_t\|^2 + \|\boldsymbol{\beta}_t\|^2$, and thus the above rearranges to

$$\|\boldsymbol{u}_t - \boldsymbol{u}_0\|^2 \leq \frac{2\hat{c}\log\left(\frac{d\kappa}{\delta}\right)}{\gamma}\left(3\mathcal{F}_1 + \nabla F(\boldsymbol{u}_0)^\top (\boldsymbol{u}_t - \boldsymbol{u}_0) + \frac{1}{2}\boldsymbol{\beta}_t^\top \boldsymbol{H} \boldsymbol{\beta}_t\right) + \|\boldsymbol{\beta}_t\|^2. \tag{27}$$

Now we control $\|\nabla F(\boldsymbol{u}_0)\|$. We use the fact that $\tilde{\boldsymbol{w}} \in \mathcal{L}_{F,F(\boldsymbol{w}_0)}$ to give us the necessary control over this quantity. Similar ideas were used in the proof of Lemma D.4, and will continue to be used in the rest of the proofs of Lemma D.5, Lemma D.6. In particular, recall as per Remark 9 that

$$\|\boldsymbol{u}_0 - \tilde{\boldsymbol{w}}\| \leq 2r \leq 2\mathcal{L} \leq \frac{1}{\rho_0(F(\boldsymbol{w}_0) + 1)}.$$

Thus by Lemma D.1, as $\tilde{\boldsymbol{w}} \in cL_{F,F(\boldsymbol{w}_0)}$, we obtain

$$\left\|\nabla^2 F(\boldsymbol{u})\right\| \le L_1(\boldsymbol{w}_0) \text{ for all } \boldsymbol{u} \in \overline{\boldsymbol{u}_0 \tilde{\boldsymbol{w}}}.$$

Consequently,

$$\|\nabla F(\boldsymbol{u}_0) - \nabla F(\tilde{\boldsymbol{w}})\| \le L_1(\boldsymbol{w}_0)\|\boldsymbol{u}_0 - \tilde{\boldsymbol{w}}\| \le 2rL_1(\boldsymbol{w}_0) = 2\mathcal{G},$$

which implies

$$\|\nabla F(\boldsymbol{u}_0)\| \le \|\nabla F(\tilde{\boldsymbol{w}})\| + 2\mathcal{G} = 3\mathcal{G}. \tag{28}$$

This gives us an analogous bound on $\|\nabla F(\boldsymbol{u}_0)\|$ as in the proof of Lemma 16, Jin et al. (2017). Substituting this bound on $\|\nabla F(\boldsymbol{u}_0\|$ into (27), we obtain

$$\|\boldsymbol{u}_t - \boldsymbol{u}_0\|^2 \le 14 \max\left\{ \frac{\mathcal{G}\hat{c}\log\left(\frac{d\kappa}{\delta}\right)}{\gamma}\|\boldsymbol{u}_t - \boldsymbol{u}_0\|, \frac{\mathcal{F}_1\hat{c}\log\left(\frac{d\kappa}{\delta}\right)}{\gamma}, \frac{\boldsymbol{\beta}_t^\top \boldsymbol{H}\boldsymbol{\beta}_t\hat{c}\log\left(\frac{d\kappa}{\delta}\right)}{\gamma}, \|\boldsymbol{\beta}_t\|^2 \right\}.$$

In turn this implies

$$\|\boldsymbol{u}_t - \boldsymbol{u}_0\| \le 14 \max\left\{ \frac{\mathcal{G}\hat{c}\log\left(\frac{d\kappa}{\delta}\right)}{\gamma}, \sqrt{\frac{\mathcal{F}_1\hat{c}\log\left(\frac{d\kappa}{\delta}\right)}{\gamma}}, \sqrt{\frac{\boldsymbol{\beta}_t^\top \boldsymbol{H}\boldsymbol{\beta}_t\hat{c}\log\left(\frac{d\kappa}{\delta}\right)}{\gamma}}, \|\boldsymbol{\beta}_t\| \right\}. \tag{29}$$

**The key induction:** Now, we induct on $t$ to prove

$$\|\boldsymbol{u}_t - \boldsymbol{u}_0\| \le 148\mathcal{L}\hat{c} \text{ for all } t < T. \tag{30}$$

Clearly this implies Lemma D.5, upon recalling $\|\boldsymbol{u}_0 - \tilde{\boldsymbol{w}}\| \le 2r = 2\mathcal{L} \le \hat{c}\mathcal{L}$ by our choice $\hat{c} = 11$.

The base case $t = 0$ is evident.

Now for the inductive step, suppose (30) is true for all $\tau \le t$ such that $t + 1 < T$. We show it is true for $t + 1$.

Due to the above bound (29), it suffices to upper bound $\|\boldsymbol{\beta}_{t+1}\|, \boldsymbol{\beta}_{t+1}^\top \boldsymbol{H}\boldsymbol{\beta}_{t+1}$. We note as in the proof of Lemma 16 of Jin et al. (2017) that letting

$$\boldsymbol{\delta}_t := \mathcal{P}_{\mathcal{S}^c}(\Delta_t(\boldsymbol{u}_t - \boldsymbol{u}_0) + \nabla F(\boldsymbol{u}_0)),$$

we have by the Triangle Inequality and properties of projections that

$$\|\boldsymbol{\delta}_t\| \le \|\Delta_t\|_{\text{op}}\|\boldsymbol{u}_t - \boldsymbol{u}_0\| + \|\nabla F(\boldsymbol{u}_0)\|. \tag{31}$$

Furthermore, we have by definition of the update rule for $\boldsymbol{\beta}_{t+1}$ that

$$\boldsymbol{\beta}_{t+1} = (\boldsymbol{I} - \eta\boldsymbol{H})\boldsymbol{\beta}_t + \eta\boldsymbol{\delta}_t. \tag{32}$$

Thus,

$$\|\boldsymbol{\beta}_{t+1}\| \le \|(\boldsymbol{I} - \eta\boldsymbol{H})\boldsymbol{\beta}_t\| + \eta\boldsymbol{\delta}_t \le \|\boldsymbol{\beta}_t\| + \eta\|\boldsymbol{H}\boldsymbol{\beta}_t\| + \eta\boldsymbol{\delta}_t. \tag{33}$$

Now, consider any $\tau, 0 \le \tau \le t$. We upper bound $\|\Delta_\tau\|_{\text{op}}$. Rewrite

$$\Delta_\tau = \int_0^1 \left( \nabla^2 F(\theta(\boldsymbol{u}_\tau - \boldsymbol{u}_0) + \boldsymbol{u}_0) - \nabla^2 F(\boldsymbol{u}_0)\right)\mathrm{d}\theta + \nabla^2 F(\boldsymbol{u}_0) - \nabla^2 F(\tilde{\boldsymbol{w}}).$$

Clearly, as per Remark 9,

$$\|\boldsymbol{u}_0 - \tilde{\boldsymbol{w}}\| \le 2r \le 2\mathcal{L} \le \frac{1}{\rho_0(F(\boldsymbol{w}_0) + 1)}.$$

Recalling $\tilde{\boldsymbol{w}} \in \mathcal{L}_{F,F(\boldsymbol{w}_0)}$ and applying Lemma D.1 gives

$$\left\|\nabla^2 F(\boldsymbol{u}_0) - \nabla^2 F(\tilde{\boldsymbol{w}})\right\|_{\text{op}} \le L_2(\boldsymbol{w}_0)\|\boldsymbol{u}_0 - \tilde{\boldsymbol{w}}\|. \tag{34}$$

Moreover by inductive hypothesis, we know that $\|\boldsymbol{u}_\tau - \boldsymbol{u}_0\| \le 148\mathcal{L}\hat{c}$. Consequently as $\hat{c} = 11 \ge 1$ and following Remark 9, for all $\theta \in [0, 1]$, we have

$$\|(\theta(\boldsymbol{u}_\tau - \boldsymbol{u}_0) + \boldsymbol{u}_0) - \tilde{\boldsymbol{w}}\| \le 2\mathcal{L} + 148\hat{c}\mathcal{L} \le \frac{1}{\rho_0(F(\boldsymbol{w}_0) + 1)}.$$

Since $\tilde{\boldsymbol{w}} \in \mathcal{L}_{F,F(\boldsymbol{w}_0)}$, it follows by Lemma D.1 that

$$\left\|\nabla^2 F(\theta(\boldsymbol{u}_\tau - \boldsymbol{u}_0) + \boldsymbol{u}_0) - \nabla^2 F(\boldsymbol{u}_0)\right\|_{\mathrm{op}} \le L_2(\boldsymbol{w}_0)\|\boldsymbol{u}_\tau - \boldsymbol{u}_0\| \text{ for all } \theta \in [0,1]. \qquad (35)$$

Hence by Triangle Inequality, from (34) and (35), we have

$$\|\Delta_t\|_{\mathrm{op}} \le L_2(\boldsymbol{w}_0)(\|\boldsymbol{u}_\tau - \boldsymbol{u}_0\| + \|\boldsymbol{u}_0 - \tilde{\boldsymbol{w}}\|) \le L_2(\boldsymbol{w}_0)(148\mathcal{L}\hat{c} + \|\boldsymbol{u}_0 - \tilde{\boldsymbol{w}}\|). \qquad (36)$$

Proceeding from here is now exactly the same as in Jin et al. (2017). We detail the argument for completeness.

Combining (31), (36), (28) and applying the inductive hypothesis and the condition of Lemma D.3 that $\|\boldsymbol{u}_0 - \tilde{\boldsymbol{w}}\| \le 2r$, gives

$$\|\boldsymbol{\delta}_\tau\| \le L_2(\boldsymbol{w}_0)(148\mathcal{L}\hat{c} + \|\boldsymbol{u}_0 - \tilde{\boldsymbol{w}}\|)\|\boldsymbol{u}_\tau - \boldsymbol{u}_0\| + \|\nabla F(\boldsymbol{u}_0)\|$$

$$\le L_2(\boldsymbol{w}_0) \cdot 148\hat{c}\left(148\hat{c} + \frac{2}{\kappa \cdot \log\left(\frac{d\kappa}{\delta}\right)}\right)\mathcal{L}^2 + 3\mathcal{G}.$$

Plugging in the choice of $\mathcal{L}$, and choosing a small enough constant $c_{\max} \le \left(\frac{1}{2 \cdot 148\hat{c}(148\hat{c}+2)}\right)^2$ and choosing step size $\eta < \frac{c_{\max}}{L_1(\boldsymbol{w}_0)}$, gives for any $0 \le \tau \le t$:

$$\|\boldsymbol{\delta}_\tau\| \le \left\{148\hat{c}\left(148\hat{c} + \frac{2}{\kappa \cdot \log\left(\frac{d\kappa}{\delta}\right)}\right)\sqrt{\eta L_1(\boldsymbol{w}_0)} + 3\right\}\mathcal{G} \le 3.5\mathcal{G}. \qquad (37)$$

We now bound $\|\boldsymbol{\beta}_{t+1}\|, \boldsymbol{\beta}_{t+1}^\top \boldsymbol{H} \boldsymbol{\beta}_{t+1}$, which combining with (29) finishes the induction and thus the proof.

- In order to bound $\|\boldsymbol{\beta}_{t+1}\|$, combining (33) with (37) and recalling the definition of $\mathcal{S}$ and $\boldsymbol{\beta}_t$ gives:

$$\|\boldsymbol{\beta}_{t+1}\| \le \left(1 + \frac{\eta\gamma}{\hat{c}\log\left(\frac{d\kappa}{\delta}\right)}\right)\|\boldsymbol{\beta}_t\| + 3.5\eta\mathcal{G}.$$

Since $\|\boldsymbol{\beta}_0\| = 0$ and $t + 1 \le T$, by applying the above relation recursively, we have:

$$\|\boldsymbol{\beta}_{t+1}\| \le \sum_{\tau=0}^T 3.5\left(1 + \frac{\eta\gamma}{\hat{c}\log\left(\frac{d\kappa}{\delta}\right)}\right)^\tau \eta\mathcal{G} \le 3.5 \cdot 3 \cdot T\eta\mathcal{G} \le 10.5\mathcal{L}\hat{c}. \qquad (38)$$

In the above we used $T \le \hat{c}\mathcal{F}$, which also implies $\left(1 + \frac{\eta\gamma}{\hat{c}\log\left(\frac{d\kappa}{\delta}\right)}\right)^T \le \left(1 + \frac{\eta\gamma}{\hat{c}\log\left(\frac{d\kappa}{\delta}\right)}\right)^{\hat{c}\mathcal{F}} \le 3$ (one can find an easy upper bound on $\mathcal{F}$ based on its definition and check using $L_2(\boldsymbol{w}_0) \ge L_1(\boldsymbol{w}_0) \ge 1$ that this is the case).

- Now for bounding $\boldsymbol{\beta}_{t+1}^\top \boldsymbol{H} \boldsymbol{\beta}_{t+1}$, notice we can also write the update equation (32) for $\boldsymbol{\beta}_t$ as:

$$\boldsymbol{\beta}_t = \eta \sum_{\tau=0}^{t-1}(\boldsymbol{I} - \eta\boldsymbol{H})^\tau \boldsymbol{\delta}_{t-1-\tau}.$$

As $\boldsymbol{H}$ is symmetric this gives:

$$\boldsymbol{\beta}_{t+1}^\top \boldsymbol{H} \boldsymbol{\beta}_{t+1} = \eta^2 \sum_{\tau_1=0}^t \sum_{\tau_2=0}^t \boldsymbol{\delta}_{t-1-\tau_1}^\top (\boldsymbol{I} - \eta\boldsymbol{H})^{\tau_1} \boldsymbol{H} (\boldsymbol{I} - \eta\boldsymbol{H})^{\tau_2} \boldsymbol{\delta}_{t-1-\tau_2}.$$

Thus we have:

$$\boldsymbol{\beta}_{t+1}^\top \boldsymbol{H} \boldsymbol{\beta}_{t+1} \le \eta^2 \sum_{\tau_1=0}^t \sum_{\tau_2=0}^t \|\boldsymbol{\delta}_{t-1-\tau_1}\|\|(\boldsymbol{I} - \eta\boldsymbol{H})^{\tau_1} \boldsymbol{H} (\boldsymbol{I} - \eta\boldsymbol{H})^{\tau_2}\|\|\boldsymbol{\delta}_{t-1-\tau_2}\|.$$

Since for $0 \le \tau_1, \tau_2 \le t$ we have $\|\boldsymbol{\delta}_{t-1-\tau_1}\|, \|\boldsymbol{\delta}_{t-1-\tau_2}\| \le 3.5\mathcal{G}$ as argued earlier, we have:

$$\boldsymbol{\beta}_{t+1}^\top \boldsymbol{H} \boldsymbol{\beta}_{t+1} \le 3.5^2 \eta^2 \mathcal{G}^2 \sum_{\tau_1=0}^t \sum_{\tau_2=0}^t \|(\boldsymbol{I} - \eta\boldsymbol{H})^{\tau_1} \boldsymbol{H} (\boldsymbol{I} - \eta\boldsymbol{H})^{\tau_2}\|.$$

Let the eigenvalues of $\boldsymbol{H}$ be $\{\lambda_i\}$. Thus for any $\tau_1, \tau_2 \geq 0$, the eigenvalues of $(\boldsymbol{I} - \eta\boldsymbol{H})^{\tau_1}\boldsymbol{H}(\boldsymbol{I} - \eta\boldsymbol{H})^{\tau_2}$ are $\{\lambda_i(1 - \eta\lambda_i)^{\tau_1+\tau_2}\}$. We now detail a calculation from Jin et al. (2017). Letting $g_t(\lambda) := \lambda(1 - \eta\lambda)^t$ and setting its derivative to zero yields

$$\nabla g_t(\lambda) = (1 - \eta\lambda)^t - t\eta\lambda(1 - \eta\lambda)^{t-1} = 0.$$

It is easy to check that $\lambda_t^\star = \frac{1}{(1+t)\eta}$ is the unique maximizer, and $g_t(\lambda)$ is monotonically increasing in $(-\infty, \lambda_t^\star]$.

This gives:

$$\|(\boldsymbol{I} - \eta\boldsymbol{H})^{\tau_1}\boldsymbol{H}(\boldsymbol{I} - \eta\boldsymbol{H})^{\tau_2}\| = \max_i \lambda_i(1 - \eta\lambda_i)^{\tau_1+\tau_2} \leq \hat{\lambda}(1 - \eta\hat{\lambda})^{\tau_1+\tau_2} \leq \frac{1}{(1 + \tau_1 + \tau_2)\eta},$$

where $\hat{\lambda} = \min\{\ell, \lambda_{\tau_1+\tau_2}^\star\}$. Therefore, we have:

$$\boldsymbol{\beta}_{t+1}^\top\boldsymbol{H}\boldsymbol{\beta}_{t+1} \leq 3.5^2\eta\mathcal{G}^2 \sum_{\tau_1=0}^{t}\sum_{\tau_2=0}^{t} \frac{1}{1 + \tau_1 + \tau_2}.$$

To bound the sum note:

$$\sum_{\tau_1=0}^{t}\sum_{\tau_2=0}^{t} \frac{1}{1 + \tau_1 + \tau_2} = \sum_{\tau=0}^{2t} \min\{1 + \tau, 2t + 1 - \tau\} \cdot \frac{1}{1 + \tau} \leq 2t + 1 < 2T.$$

Thus:

$$\boldsymbol{\beta}_{t+1}^\top\boldsymbol{H}\boldsymbol{\beta}_{t+1} \leq 2 \cdot 3.5^2\eta T\mathcal{G}^2 \leq \frac{3.5^2\mathcal{L}^2\gamma\hat{c}}{\log\left(\frac{d\kappa}{\delta}\right)}. \tag{39}$$

Finally, substituting the previous upper bounds (38), (39) for $\|\boldsymbol{\beta}_t\|$, $\boldsymbol{\beta}_{t+1}^\top\boldsymbol{H}\boldsymbol{\beta}_{t+1}$ into our prior display (29) for $\|\boldsymbol{u}_t - \boldsymbol{u}_0\|$, we obtain:

$$\|\boldsymbol{u}_t - \boldsymbol{u}_0\| \leq 14\max\left\{\frac{\mathcal{G}\hat{c}\log\left(\frac{d\kappa}{\delta}\right)}{\gamma}, \sqrt{\frac{\mathcal{F}_1\hat{c}\log\left(\frac{d\kappa}{\delta}\right)}{\gamma}}, \sqrt{\frac{\boldsymbol{\beta}_t^\top\boldsymbol{H}\boldsymbol{\beta}_t\hat{c}\log\left(\frac{d\kappa}{\delta}\right)}{\gamma}}, \|\boldsymbol{\beta}_t\|\right\} \leq 148\mathcal{L}\hat{c}.$$

This finishes the induction, and hence the proof of the Lemma. $\qquad\square$

**Proof of Lemma D.6.** Again, we aim to show that if all iterates from $\boldsymbol{u}_0$ are contained in a small ball, then the iterates from $\boldsymbol{x}_0$ decrease function value. As with the proof of Lemma D.5, the proof combines the proof idea of Lemma 17, Jin et al. (2017) with the self-bounding framework. This time it goes through even easier, because the required new bounds that we need from the relevant iterates being 'local' hold not due to induction, but rather from a direct application of Lemma D.5.

Define $\boldsymbol{v}_t = \boldsymbol{x}_t - \boldsymbol{u}_t$. By the assumptions of this Lemma we have that $\boldsymbol{v}_0 = \pm\mu\left[\frac{\mathcal{L}}{\kappa\cdot\log\left(\frac{d\kappa}{\delta}\right)}\right]\boldsymbol{e}_1$ where $\mu \in \left[\frac{\delta}{2\sqrt{d}}, 1\right]$. Consequently

$$\frac{\delta}{2\sqrt{d}} \cdot r \leq \|\boldsymbol{v}_0\| \leq r. \tag{40}$$

Recall the definition

$$\boldsymbol{H} = \nabla^2 F(\tilde{\boldsymbol{w}})$$

as per (22). Also define

$$\Delta_t' := \int_0^1 \nabla^2 F(\boldsymbol{u}_t + \theta\boldsymbol{v}_t)\mathrm{d}\theta - \boldsymbol{H}.$$

Exactly as in the proof of Lemma 17, Jin et al. (2017), by directly writing the update equations, we have

$$\boldsymbol{u}_{t+1} + \boldsymbol{v}_{t+1} = \boldsymbol{x}_{t+1} = \boldsymbol{x}_t - \eta\nabla F(\boldsymbol{x}_t)$$
$$= \boldsymbol{u}_t + \boldsymbol{v}_t - \eta\nabla F(\boldsymbol{u}_t + \boldsymbol{v}_t)$$

$$= \boldsymbol{u}_t + \boldsymbol{v}_t - \eta \nabla F(\boldsymbol{u}_t) - \eta \left( \int_0^1 \nabla^2 F(\boldsymbol{u}_t + \theta \boldsymbol{v}_t) \mathrm{d}\theta \right) \boldsymbol{v}_t$$

$$= \boldsymbol{u}_t + \boldsymbol{v}_t - \eta \nabla F(\boldsymbol{u}_t) - \eta (\boldsymbol{H} + \Delta_t') \boldsymbol{v}_t$$

$$= \boldsymbol{u}_t - \eta \nabla F(\boldsymbol{u}_t) + (\boldsymbol{I} - \eta \boldsymbol{H} - \eta \Delta_t') \boldsymbol{v}_t.$$

Hence as $\boldsymbol{u}_{t+1} = \boldsymbol{u}_t - \eta \nabla F(\boldsymbol{u}_t)$, we obtain

$$\boldsymbol{v}_{t+1} = (\boldsymbol{I} - \eta \boldsymbol{H} - \eta \Delta_t') \boldsymbol{v}_t. \tag{41}$$

The difference from the proof of Lemma 17, Jin et al. (2017) is now that we do not immediately have an upper bound on $\|\Delta_t'\|_{\mathrm{op}}$ without global Lipschitzness of the gradient and Hessian. However, similarly as in the proof of Lemma D.5, we can obtain such a bound using the self-bounding framework, since the point $\tilde{\boldsymbol{w}}$ in question is in the $F(\boldsymbol{w}_0)$-sublevel set $\mathcal{L}_{F,F(\boldsymbol{w}_0)}$.

Note by hypothesis on $\boldsymbol{u}_0$ from Lemma D.4 and as $\|\boldsymbol{v}_0\| \le r$ by (40),

$$\|\boldsymbol{x}_0 - \tilde{\boldsymbol{w}}\| \le \|\boldsymbol{u}_0 - \tilde{\boldsymbol{w}}\| + \|\boldsymbol{v}_0\| \le r + r = 2r.$$

Applying Lemma D.5 directly to the $\{\boldsymbol{x}_t\}$ implies that

$$\|\boldsymbol{x}_t - \tilde{\boldsymbol{w}}\| \le 150 \mathcal{L}\hat{c} \text{ for all } t < T.$$

By assumption of this Lemma, we have

$$\|\boldsymbol{u}_t - \tilde{\boldsymbol{w}}\| \le 150 \mathcal{L}\hat{c} \text{ for all } t < T.$$

Triangle Inequality thus gives

$$\|\boldsymbol{v}_t\| \le 300 \mathcal{L}\hat{c}, \|\boldsymbol{u}_t - \boldsymbol{u}_0\| \le 300 \mathcal{L}\hat{c} \text{ for all } t < T.$$

Therefore for all $0 \le \theta \le 1$,

$$\boldsymbol{u}_t + \theta \boldsymbol{v}_t \in \mathbb{B}(\tilde{\boldsymbol{w}}, 600 \mathcal{L}\hat{c}).$$

Note as per Remark 9,

$$600 \mathcal{L}\hat{c} = 6600 \mathcal{L} \le \frac{1}{\rho_0(F(\boldsymbol{w}_0) + 1)}.$$

As $\tilde{\boldsymbol{w}} \in \mathcal{L}_{F,F(\boldsymbol{w}_0)}$, it follows from Lemma D.1 that

$$\left\| \nabla^2 F(\boldsymbol{u}_t + \theta \boldsymbol{v}_t) - \nabla^2 F(\boldsymbol{u}_t) \right\|_{\mathrm{op}} \le L_2(\boldsymbol{w}_0) \cdot \theta \boldsymbol{v}_t \text{ for all } \theta \in [0, 1]. \tag{42}$$

Similarly, by the above bound

$$\|\boldsymbol{u}_t - \tilde{\boldsymbol{w}}\| \le 150 \mathcal{L}\hat{c} \le \frac{1}{\rho_0(F(\boldsymbol{w}_0) + 1)}$$

and as $\tilde{\boldsymbol{w}} \in \mathcal{L}_{F,F(\boldsymbol{w}_0)}$, Lemma D.1 proves that

$$\left\| \nabla^2 F(\boldsymbol{u}_t) - \nabla^2 F(\tilde{\boldsymbol{w}}) \right\|_{\mathrm{op}} \le L_2(\boldsymbol{w}_0) \|\boldsymbol{u}_t - \tilde{\boldsymbol{w}}\|. \tag{43}$$

Now, rewrite

$$\Delta_t' = \int_0^1 \left( \nabla^2 F(\boldsymbol{u}_t + \theta \boldsymbol{v}_t) - \nabla^2 F(\boldsymbol{u}_t) \right) \mathrm{d}\theta + \nabla^2 F(\boldsymbol{u}_t) - \nabla^2 F(\tilde{\boldsymbol{w}}).$$

By (42), (43), and the above bounds on $\|\boldsymbol{v}_t\|, \|\boldsymbol{u}_t - \tilde{\boldsymbol{w}}\|$, we obtain for all $\theta \in [0, 1]$ that

$$\|\Delta_t'\|_{\mathrm{op}} \le L_2(\boldsymbol{w}_0)(\theta \|\boldsymbol{v}_t\| + \|\boldsymbol{u}_t - \tilde{\boldsymbol{w}}\|) \le L_2(\boldsymbol{w}_0) \mathcal{L}(450\hat{c} + 1). \tag{44}$$

From here, exactly the same proof as that of Lemma 17, Jin et al. (2017) lets us conclude. We detail it for completeness. Similar to the proof of Lemma 17, Jin et al. (2017), let $S$ be the subspace corresponding to eigenvectors of $\boldsymbol{H}$ with eigenvalues larger or equal in absolute value to $\gamma$, and let $S^\perp$ be its orthogonal complement. Note $\boldsymbol{e}_1 \subseteq S$. Denote the norm of $\boldsymbol{v}_t$ projected onto $S$ by $\psi_t$, and the norm of $\boldsymbol{v}_t$ projected onto $S^\perp$ by $\phi_t$.

Notice therefore from the assumptions of this Lemma that $\phi_0 = 0$ as $\boldsymbol{v}_0$ is a scalar multiple of $\boldsymbol{e}_1$. Similarly, note $\psi_0 = \|\boldsymbol{v}_0\| \ge \frac{\delta}{2\sqrt{d}} \cdot r$ by (40).

Let

$$B := \eta L_2(\boldsymbol{w}_0) \mathcal{L}(450\hat{c} + 1).$$

Observe $B \le 1$, as $\mathcal{L}L_2(\boldsymbol{w}_0) \le 1$ and as $\eta \le c_{\max} \le \frac{1}{12100}, \hat{c} = 11$.

Combining (41) with (44) gives that

$$\psi_{t+1} \ge (1 + \gamma\eta)\psi_t - B\sqrt{\psi_t^2 + \phi_t^2}, \phi_{t+1} \le (1 + \gamma\eta)\phi_t + B\sqrt{\psi_t^2 + \phi_t^2}. \tag{45}$$

**The key induction:**   Now we induct on $t$ to show that for all $t < T$,

$$\phi_t \leq 4Bt \cdot \psi_t.$$

For the base case, recall by hypotheses of the Lemma that $\boldsymbol{v}_0$ is a scalar multiple of $\boldsymbol{e}_1$, thus $\phi_0 = 0$ and the base case holds.

Now, for the inductive step, assume that the inductive hypothesis holds true for all $\tau \leq t$ for some $t$ such that $t + 1 \leq T$. Substituting the inequality (45) for $\phi_{t+1}$ and applying the inductive hypothesis $\phi_t \leq 4Bt \cdot \psi_t$, we obtain

$$\phi_{t+1} \leq 4Bt(1 + \gamma\eta)\psi_t + B\sqrt{\psi_t^2 + \phi_t^2}.$$

Also note (45) gives

$$4B(t + 1)\psi_{t+1} \geq 4B(t + 1)\left((1 + \gamma\eta)\psi_t - B\sqrt{\psi_t^2 + \phi_t^2}\right),$$

which rearranges to

$$4Bt(1 + \gamma\eta)\psi_t \leq 4B(t + 1)\psi_{t+1} + 4B^2(t + 1)\sqrt{\psi_t^2 + \phi_t^2} - 4B(1 + \gamma\eta)\psi_t.$$

Therefore,

$$\phi_{t+1} \leq 4B(t + 1)\psi_{t+1} + \left(4B^2(t + 1)\sqrt{\psi_t^2 + \phi_t^2} + B\sqrt{\psi_t^2 + \phi_t^2} - 4B(1 + \gamma\eta)\psi_t\right).$$

Thus, recalling $B \leq 1$, to complete the induction it suffices to show the following:

$$\left(1 + 4B^2(t + 1)\right)\sqrt{\psi_t^2 + \phi_t^2} \leq 4(1 + \gamma\eta)\psi_t.$$

Choosing $\sqrt{c_{\max}} \leq \frac{1}{450\hat{c}+1}\min\left\{\frac{1}{2\sqrt{2}}, \frac{1}{4\hat{c}}\right\}$ which is a universal constant, and choosing $\eta \leq \frac{c_{\max}}{L_1(\boldsymbol{w}_0)}$, we have:

$$4B(t + 1) \leq 4BT \leq 4\eta L_2(\boldsymbol{w}_0)\mathcal{L}(450\hat{c} + 1)\hat{c}\mathcal{F} = 4\sqrt{\eta L_1(\boldsymbol{w}_0)}(450\hat{c} + 1)\hat{c} \leq 1.$$

By the inductive hypothesis, this gives $\phi_t \leq \psi_t$. In turn this implies that

$$4(1 + \gamma\eta)\psi_t \geq 4\psi_t \geq 2\sqrt{2}\psi_t \geq (1 + 4B(t + 1))\sqrt{\psi_t^2 + \phi_t^2},$$

finishing the induction.

**Finishing the proof from here:**   We thus obtain $\phi_t \leq 4Bt\psi_t \leq \psi_t$ for all $t$, where we use that $4BT \leq 1$ as proven above, which just follows from our choice of parameters. Therefore,

$$\psi_{t+1} \geq (1 + \gamma\eta)\psi_t - B\sqrt{2}\psi_t > \left(1 + \frac{\gamma\eta}{2}\right)\psi_t. \tag{46}$$

The last step follows upon noting $B \leq \eta L_2(\boldsymbol{w}_0)\mathcal{L}(450\hat{c}+1) \leq \sqrt{c_{\max}}(450\hat{c}+1)\gamma\eta\log^{-1}\left(\frac{d\kappa}{\delta}\right) < \frac{\gamma\eta}{2\sqrt{2}}$. The inequality is strict as $\gamma\eta > 0$.

Finally, recalling that $\|\boldsymbol{v}_t\| \leq 300\mathcal{L}\hat{c}$, $\psi_0 \geq \frac{\delta}{2\sqrt{d}} \cdot r$ and using (46), we have for all $t < T$:

$$
\begin{aligned}
300(\mathcal{L} \cdot \hat{c}) &\geq \|\boldsymbol{v}_t\| \\
&\geq \psi_t \\
&> \left(1 + \frac{\gamma\eta}{2}\right)^t \psi_0 \\
&\geq \left(1 + \frac{\gamma\eta}{2}\right)^t \cdot \frac{\delta}{2\sqrt{d}} \cdot \frac{\mathcal{L}}{\kappa \cdot \log\left(\frac{d\kappa}{\delta}\right)}.
\end{aligned}
\tag{47}
$$

Note that $\delta \in \left(0, \frac{d\kappa}{e}\right]$ implies $\log\left(\frac{d\kappa}{\delta}\right) \geq 1$. Applying (47) for $t = T - 1$ we obtain:

$$T < 1 + \log\left(600\kappa\sqrt{d}\delta^{-1} \cdot \hat{c}\log\left(\frac{d\kappa}{\delta}\right)\right) \cdot \log^{-1}\left(1 + \frac{\gamma\eta}{2}\right)$$

---
**Algorithm 2** Restarted SGD, from Fang et al. (2019)
---
Initialize at $\boldsymbol{w}_0$, and consider $K_0 = \tilde{\Theta}(\varepsilon^{-2})$, $\eta = \tilde{\Theta}(\varepsilon^{1.5})$, $B = \tilde{\Theta}(\varepsilon^{0.5})$, $\tilde{\sigma} = 2\sigma_1'(\boldsymbol{w}_0)$, all explicitly defined in Subsection E.1.

Let $t = 0$ (the total number of iterates), $k = 0$ (the restart counter), $\boldsymbol{x}^0 = \boldsymbol{w}_0$ (the point we consider the escape from).

**while** $k < K_0$ **do**

  Let $\boldsymbol{x}^{t+1} = \boldsymbol{x}^t - \eta(\nabla f(\boldsymbol{x}^t; \boldsymbol{\zeta}_{t+1}) + \tilde{\sigma}\boldsymbol{\Lambda}^{t+1})$, where $\boldsymbol{\Lambda}^{t+1}$ is uniform from $\mathbb{B}(\vec{\boldsymbol{0}}, 1)$ and independent of everything else, and $\boldsymbol{\zeta}_{t+1}$ is an i.i.d. minibatch sample

  $t \leftarrow t + 1, k \leftarrow k + 1$

  **if** $\|\boldsymbol{x}^k - \boldsymbol{x}^0\| > B$ **then**

    $\boldsymbol{x}^0 \leftarrow \boldsymbol{x}^k, k \leftarrow 0$

  **end if**

**end while**

Return $\frac{1}{K_0} \sum_{k=0}^{K_0-1} \boldsymbol{x}^k$

---

$$\leq 1 + 2.01 \log\left(600\kappa\sqrt{d}\delta^{-1} \cdot \hat{c}\log\left(\frac{d\kappa}{\delta}\right)\right) \cdot \frac{1}{\gamma\eta}$$

$$\leq 1 + 2.01(\log(600\hat{c}) + 1.01\log(d\kappa/\delta)) \cdot \frac{1}{\gamma\eta}$$

$$\leq \left(\frac{1}{40} + 1 + 2.0301\right)\mathcal{F}_2 \leq \hat{c}\mathcal{F}_2.$$

These last steps follow by:

- Taking $c_{\max}$ a small enough universal constant so that $\gamma\eta \leq \frac{1}{60} \cdot \frac{c_{\max}}{L_1(\boldsymbol{w}_0)} \leq \frac{c_{\max}}{60}$ satisfies $\frac{2.01}{x} > \log^{-1}(1 + x/2)$, which is valid for all $0 < x < 0.02$.

- Remark 9, which states that we can assume WLOG $\log(d\kappa/\delta)$ is larger than a universal constant. In particular we can assume WLOG that $\log(d\kappa/\delta)$ solves $\log x < x^{0.01}$ (hence $\log(\kappa\sqrt{d}\delta^{-1}\log(d\kappa/\delta)) \leq 1.01\log(d\kappa/\delta)$), that $2.01\log(600\hat{c}) = 2.01\log(6600) \leq \log(d\kappa/\delta)$ (recall $\hat{c} = 11$), and that $\mathcal{F}_2 = \frac{\log(d\kappa/\delta)}{\gamma\eta} \geq 40$.

This completes the proof. $\qquad\square$

# E  Restarted SGD finding Second Order Stationary Points

Here, we formally prove Theorem 3.5. We formally instantiate Algorithm 2 here. One may notice a slight difference in Algorithm 2 vs the algorithm of Fang et al. (2019): we artificially inject bounded noise at a particular scale $\tilde{\sigma}$. This ensures we can escape saddle points that are in the $F(\boldsymbol{w}_0)$-sublevel set $\mathcal{L}_{F,F(\boldsymbol{w}_0)}$. Note we may not be able to escape saddle points that are not in $\mathcal{L}_{F,F(\boldsymbol{w}_0)}$, but that does not matter thanks to our framework Theorem 2.1, which effectively lets us consider only behavior within $\mathcal{L}_{F,F(\boldsymbol{w}_0)}$. Also note a practitioner can find such a noise scaling $\tilde{\sigma}$ (depending on suboptimality at initialization $F(\boldsymbol{w}_0)$) via appropriate cross-validation.

The general proof strategy here is similar to the way we adapted the proof of Jin et al. (2017) in Section D. Namely, we use the self-bounding regularity conditions to control the derivatives of $F$ in appropriate neighborhoods of the $F(\boldsymbol{w}_0)$-sublevel set $\mathcal{L}_{F,F(\boldsymbol{w}_0)}$.

## E.1  Notation and Parameters

We set the parameters of the algorithm as follows. We will highlight the significance of these parameters in Subsection E.3.

**Noise Parameters:**  Define

$$\sigma'(\boldsymbol{w}_0) = \sigma(F(\boldsymbol{w}_0) + 1). \tag{48}$$

$$\tilde{\sigma} = 2\sigma'(\boldsymbol{w}_0). \tag{49}$$

$$\sigma_1(\boldsymbol{w}_0) = \max\{\sigma'(\boldsymbol{w}_0) + \tilde{\sigma}, 1\}. \tag{50}$$

Note this only depends on $\rho_0$ (and therefore only on $\rho_1$) and $F(\boldsymbol{w}_0)$. Note $\tilde{\sigma} \in [\sigma'(\boldsymbol{w}_0), 2\sigma'(\boldsymbol{w}_0)]$.[9] Also note $\sigma_1(\boldsymbol{w}_0) \le 3\sigma'(\boldsymbol{w}_0)$.

**Update Rule:** Define
$$\nabla\tilde{f}(\boldsymbol{x}^t; \boldsymbol{\zeta}_{t+1}) := \nabla f(\boldsymbol{x}^t; \boldsymbol{\zeta}_{t+1}) + \tilde{\sigma}\boldsymbol{\Lambda}^{t+1}.$$
Thus the SGD update rule in Algorithm 2 (without considering the restarts) is $\boldsymbol{x}^{t+1} = \boldsymbol{x}^t - \eta\nabla\tilde{f}(\boldsymbol{x}^t; \boldsymbol{\zeta}_{t+1})$. Note the slight abuse of notation; $\nabla\tilde{f}(\boldsymbol{x}^t; \boldsymbol{\zeta}_{t+1})$ is not necessarily an actual gradient.[10] This will not cause issues or ambiguity for the rest of this section.

**Effective Smoothness Parameters in $F(\boldsymbol{w}_0)$-sublevel set:** We define the 'local smoothness parameters' as follows, slightly differently compared to the proof of Theorem 3.4. Define

$$L_1(\boldsymbol{w}_0) := \max\{1, \rho_1(F(\boldsymbol{w}_0) + 1), \rho_3(\rho_0(F(\boldsymbol{w}_0) + 1) + \sigma'(\boldsymbol{w}_0), F(\boldsymbol{w}_0) + 1)\}, \tag{51}$$

$$L_2(\boldsymbol{w}_0) := \max\Big\{1, \rho_2(F(\boldsymbol{w}_0) + 1), \rho_0(F(\boldsymbol{w}_0) + 1)^2 \max\Big\{4, (\sigma_1(\boldsymbol{w}_0) + \rho_0(F(\boldsymbol{w}_0) + 1))^2\Big\}\Big\}. \tag{52}$$

Note all of these parameters only depend on $F(\boldsymbol{w}_0)$, through $\rho_1(\cdot), \rho_2(\cdot), \rho_3(\cdot, \cdot)$ (recall $\rho_0(\cdot)$ can be defined in terms of $\rho_1(\cdot)$).

**Parameters of Algorithm 2:** We define the remaining parameters of Algorithm 2 as follows. Consider any $\varepsilon > 0$ and $p \in (0, 1)$. We choose:

$$\tilde{C}_1 = 2\Big\lfloor \frac{\log(3/p)}{\log(0.8^{-1})} + 1\Big\rfloor \log\Big(\frac{24\sqrt{d}}{\eta}\Big),$$

$$\delta = \sqrt{L_2(\boldsymbol{w}_0)\varepsilon},$$

$$\delta_2 = 16\delta,$$

$$B = \frac{\delta}{L_2(\boldsymbol{w}_0)\tilde{C}_1},$$

$$K_0 = \tilde{C}_1\eta^{-1}\delta_2^{-1},$$

$$\eta \le \frac{B^2\delta}{512\max(\sigma_1(\boldsymbol{w}_0)^2, 1)\tilde{C}_1\log(48K_0/p)} \cdot \frac{1}{3(1 + \log(K_0))}. \tag{53}$$

Also define

$$K_o = 2\log\Big(\frac{24\sqrt{d}}{\eta}\Big)\eta^{-1}\delta_2^{-1}, \quad \text{thus } K_0 = \Big\lfloor \frac{\log(3/p)}{\log(0.8^{-1})} + 1\Big\rfloor K_o.$$

**Remark 11.** To choose $\eta$ satisfying the above inequality, one can perform the same analysis as on footnote 4, page 7 of Fang et al. (2019). We first choose $\tilde{\eta}$ appropriately by setting

$$\tilde{\eta} = \frac{B^2\delta}{4096\max(\sigma_1(\boldsymbol{w}_0)^2, 1)\log(48/p)\log(p)\big\lfloor \frac{\log(3/p)}{\log(0.8^{-1})} + 1\big\rfloor},$$

and then set $\eta = \tilde{\eta}\log^{-3}(1/\tilde{\eta})$.

**Remark 12.** Analogously to the proof of Theorem 3.4, note it suffices to show the result for $\varepsilon \le \frac{1}{L_2(\boldsymbol{w}_0)}$; for $\varepsilon > \frac{1}{L_2(\boldsymbol{w}_0)}$, we can just apply the result for $\varepsilon = \frac{1}{L_2(\boldsymbol{w}_0)}$, and the result remains the same up to $F(\boldsymbol{w}_0)$-dependent parameters in the $O(\cdot)$. Thus we can suppose that $\delta_2$ (and $\delta$) are at most some universal constant. We also can take $L_1(\boldsymbol{w}_0), L_2(\boldsymbol{w}_0), \sigma_1(\boldsymbol{w}_0)$ to be the max between their currently definition and an appropriate universal constant. Thus due to the choice of parameters above, we may assume that

$$\tilde{C}_1, K_0 \ge 1,$$

---

[9]In fact, this is the only condition we need on $\tilde{\sigma}$. In practice, such a $\tilde{\sigma}$ by fine-enough cross validation in terms of only $F(\boldsymbol{w}_0)$.

[10]This choice of notation is made to demonstrate the artificial noise injections $\tilde{\sigma}\boldsymbol{\Lambda}^{t+1}$ are not fundamentally needed. They are not necessary if the stochastic gradient $\nabla f(\cdot; \cdot)$ enjoys suitable anticoncentration properties.

$$\log(K_0), \sigma_1(\boldsymbol{w}_0) \geq 1,$$

$$B \leq \min\left(1, \frac{\sigma_1(\boldsymbol{w}_0)}{L_1(\boldsymbol{w}_0)}, \frac{1}{L_1(\boldsymbol{w}_0)}, \frac{1}{L_2(\boldsymbol{w}_0)}\right),$$

$$\eta \leq \min\left\{1, \frac{1}{\sigma_1(\boldsymbol{w}_0)^2}\right\}.$$

From here note we have $\eta L_1(\boldsymbol{w}_0) \leq 1$. As *these assumptions come with no loss of generality*, we make these assumptions for the rest of the proof.

**Notation:** Consider a sequence of iterates $\boldsymbol{x}^0, \boldsymbol{x}^1, \ldots$ beginning at $\boldsymbol{x}^0$ comprising an instance of the while loop in Algorithm 2. For such a sequence, let $\mathfrak{F}^k$ be the $\sigma$-algebra defined by all the prior iterates and the noise up through $\boldsymbol{x}^k$, namely $\sigma\{\boldsymbol{x}^0, \boldsymbol{\zeta}_1, \boldsymbol{\Lambda}^1, \boldsymbol{x}^1, \ldots, \boldsymbol{x}^{k-1}, \boldsymbol{\zeta}_k, \boldsymbol{\Lambda}^k\}$. Let $\mathcal{K}_0$ be a stopping time given by

$$\mathcal{K}_0 = \inf_k\{k \geq 0 : \|\boldsymbol{x}^k - \boldsymbol{x}^0\| \geq B\}.$$

Note $\boldsymbol{x}^k$ and $1_{\mathcal{K}_0 \geq k}, 1_{\mathcal{K}_0 > k}$ are $\mathfrak{F}^k$-measurable. Thus, $1_{\mathcal{K}_0 > k-1} \equiv 1_{\mathcal{K}_0 \geq k}$ is $\mathfrak{F}^{k-1}$-measurable.

## E.2   Result

We now formally prove Theorem 3.5. The following Theorem E.1 can readily be seen to imply Theorem 3.5.

**Theorem E.1.** *Suppose $F$ satisfies Assumption 1.2 and the stochastic gradient oracle satisfies Assumption 3.1 and Assumption 3.2. Run Algorithm 2 initialized at $\boldsymbol{w}_0$, run with parameters chosen as per Subsection E.1.*

*Consider any $p \in (0, 1)$. With probability at least $1 - \frac{7}{4}p \cdot \frac{(F(\boldsymbol{w}_0)+1)7\eta K_0}{B^2}$, upon making*

$$K_0 + \frac{7\eta K_0^2(F(\boldsymbol{w}_0) + 1)}{B^2} \text{ oracle calls to } \nabla f(\cdot; \cdot),$$

*Algorithm 2 will output $\tilde{O}\left(\frac{7\eta K_0^2(F(\boldsymbol{w}_0)+1)}{B^2}\right)$ candidate vectors $\boldsymbol{w}$, one of which satisfies*

$$\|\nabla F(\boldsymbol{w})\| \leq 18L_2(\boldsymbol{w}_0)B^2, \lambda_{\text{MIN}}(\nabla^2 F(\boldsymbol{w})) \geq -17\delta.$$

**Remark 13.** Before proceeding, we justify why Theorem E.1 implies Theorem 3.5. Simply take $\varepsilon \leftarrow \frac{\varepsilon}{289L_2(\boldsymbol{w}_0)}$ in Theorem E.1. Plugging this in, we obtain a result on finding a SOSP as per the definition in (2).[11] The oracle complexity has the desired dependence on $\varepsilon$ and polylog dependence on $d, p$. The probability is at least $1 - p \cdot \tilde{\Theta}(\varepsilon^{-1.5})$, where the $\tilde{\Theta}$ are hiding polylog terms in $d, 1/\varepsilon, 1/p$ and dependence on $F(\boldsymbol{w}_0)$ (through $\rho_1(\cdot), \rho_2(\cdot), \rho_3(\cdot), \sigma(\cdot)$). This holds for any $p \in (0, 1)$.

Now consider the final desired success probability $1 - \tilde{\delta}$ governed in terms of $\tilde{\delta} \in (0, 1)$ in Theorem 3.5. Let $p = \tilde{\delta}\varepsilon^{1.5} \cdot \text{polylog}(d, 1/\varepsilon)$ in the guarantee from the above paragraph. This gives Theorem 3.5, with the requested probability and oracle complexity.

We now prove Theorem E.1 via our framework, Theorem 2.1.

**Proof of Theorem E.1 and thus Theorem 3.5.** We again use our framework Theorem 2.1. Consider any $p \in (0, 1)$, and choose parameters as per Subsection E.1.

Let

$$\mathcal{S} = \{\boldsymbol{w} : \|\nabla F(\boldsymbol{w})\| \leq 18L_2(\boldsymbol{w}_0)B^2, \lambda_{\text{MIN}}(\nabla^2 F(\boldsymbol{w})) \geq -17\delta\}.$$

Define $\mathcal{A}$ as follows, identically to how we defined them for Restarted SGD in Subsection 2.3. Consider any given $\boldsymbol{u}_0 \in \mathbb{R}^d$. Let $\boldsymbol{p}_0 = \boldsymbol{u}_0$. We define a sequence $(\boldsymbol{p}_i)_{0 \leq i \leq K_0}$ via $\boldsymbol{p}_i = \boldsymbol{p}_{i-1} - \eta(\nabla f(\boldsymbol{p}_{i-1}; \boldsymbol{\zeta}_i) + \tilde{\sigma}\boldsymbol{\Lambda}^i)$. Note this sequence can be equivalently defined by repeatedly composing the function $\boldsymbol{u} \to \boldsymbol{u} - \eta(\nabla f(\boldsymbol{u}; \boldsymbol{\zeta}) + \tilde{\sigma}\boldsymbol{\Lambda})$.

---

[11]Recall this definition refers to $\boldsymbol{w}$ such that $\|\nabla F(\boldsymbol{w})\| \leq \varepsilon, \nabla^2 F(\boldsymbol{w}) \geq -\sqrt{\varepsilon}\boldsymbol{I}$.

If it exists, let $i, 1 \leq i \leq K_0$ be the minimal index such that $\|\boldsymbol{p}_i - \boldsymbol{p}_0\| > B$. Otherwise let $i = K_0$. In either case, we define

$$\mathcal{A}(\boldsymbol{u}_0) = \left(\boldsymbol{p}_i, \frac{1}{i}\sum_{t=0}^{i-1}\boldsymbol{p}_t\right), \text{ hence } \mathcal{A}_1(\boldsymbol{u}_0) = \boldsymbol{p}_i, \mathcal{A}_2(\boldsymbol{u}_0) = \frac{1}{i}\sum_{t=0}^{i-1}\boldsymbol{p}_t.$$

We now let

$$t_{\text{oracle}}(\boldsymbol{u}_0) = K_0, \text{ and } \Delta = \frac{B^2}{7\eta K_0}.$$

Following the notation from Algorithm 2, notice that $\mathcal{A}(\boldsymbol{u}_0)$ corresponds to next vector set to $\boldsymbol{x}^0$ in the while loop of Algorithm 2, when the while loop begins at $\boldsymbol{x}^0 = \boldsymbol{u}_0$.

Crucial to this proof are the following two Lemmas. While inspired from Fang et al. (2019), a crucial difference is that *they hold only in the $F(\boldsymbol{w}_0)$-sublevel set $\mathcal{L}_{F,F(\boldsymbol{w}_0)}$*.

**Lemma E.1** (Equivalent of Proposition 10, Fang et al. (2019)). *Consider $\boldsymbol{x}^0$ in the while loop of Algorithm 2. Suppose $\boldsymbol{x}^0 \in \mathcal{L}_{F,F(\boldsymbol{w}_0)}$. With probability at least $1 - p$, if $\boldsymbol{x}^k$ does not move out of the ball $\mathbb{B}(\boldsymbol{x}^0, B)$ within the first $K_0$ iterations in the while loop of Algorithm 2, letting $\overline{\boldsymbol{x}} = \frac{1}{K_0}\sum_{k=0}^{K_0-1}\boldsymbol{x}^k$, we have*

$$\|\nabla F(\overline{\boldsymbol{x}})\| \leq 18L_2(\boldsymbol{w}_0)B^2, \lambda_{\text{MIN}}(\nabla^2 F(\overline{\boldsymbol{x}})) \geq -17\delta.$$

**Lemma E.2** (Equivalent of Proposition 9, Fang et al. (2019)). *Consider $\boldsymbol{x}^0$ in the while loop of Algorithm 2. Suppose $\boldsymbol{x}^0 \in \mathcal{L}_{F,F(\boldsymbol{w}_0)}$. With probability at least $1 - \frac{3}{4}p$, if $\boldsymbol{x}^k$ moves out of $\mathbb{B}(\boldsymbol{x}^0, B)$ in $K_0$ iterations or fewer in the while loop of Algorithm 2, we have*

$$F(\boldsymbol{x}^{\mathcal{K}_0}) < F(\boldsymbol{x}^0) - \frac{B^2}{7\eta K_0}.$$

**Finishing the proof:** The main point is to prove the following Claim.

**Claim 7.** *For any $\boldsymbol{u}_0 \in \mathcal{L}_{F,F(\boldsymbol{w}_0)}$, $\mathcal{A}$ is a $(\mathcal{S}, K_0, \Delta, \frac{7}{4}p, \boldsymbol{u}_0)$-decrease procedure.*

**Proof of Claim 7.** Apply Lemma E.1 and Lemma E.2 to the sequence $(\boldsymbol{p}_i)_{0\leq i\leq K_0}$, recalling that $\mathcal{A}(\boldsymbol{u}_0)$ corresponds to next vector set to $\boldsymbol{x}^0$ in the while loop of Algorithm 2 when the while loop begins at $\boldsymbol{x}^0 = \boldsymbol{p}_0 = \boldsymbol{u}_0$. By a Union Bound over the events of Lemma E.1 and Lemma E.2, with probability at least $1 - \frac{7}{4}p$, we have the following:

- Suppose there exists $t < K_0$ such that $\boldsymbol{p}_t \notin \mathbb{B}(\boldsymbol{p}_0, B) = \mathbb{B}(\boldsymbol{u}_0, B)$. Let $t'$ be the minimal such $t$. By Lemma E.2, we have

$$F(\mathcal{A}_1(\boldsymbol{u}_0)) = F(\boldsymbol{p}_{t'}) \leq F(\boldsymbol{p}_0) - \frac{B^2}{7\eta K_0} = F(\boldsymbol{u}_0) - \Delta.$$

- Otherwise, we have $\mathcal{A}_2(\boldsymbol{u}_0) = \overline{\boldsymbol{p}}$ where $\overline{\boldsymbol{p}} = \frac{1}{K_0}\sum_{k=0}^{K_0-1}\boldsymbol{p}_k$. In this case, by Lemma E.1, we have

$$\mathcal{A}_2(\boldsymbol{u}_0) = \overline{\boldsymbol{p}} \in \mathcal{S}.$$

Consequently, $\mathcal{A}$ is a $(\mathcal{S}, K_0, \Delta, \frac{7}{4}p, \boldsymbol{u})$-decrease procedure. $\square$

Now with Claim 7, directly applying Theorem 2.1 and plugging in the relevant parameters, we obtain Theorem E.1. $\square$

**Remark 14.** To sanity check these results, note the rate from Lemma E.2 will get worse as $\eta$ gets smaller because $K_0\eta = 2\lfloor\frac{\log(3/p)}{\log(0.8^{-1})} + 1\rfloor\log\left(\frac{24\sqrt{d}}{\eta}\right)\delta_2^{-1}$ will increase as $\eta$ gets smaller.

The rest of Section E will now be devoted to the proofs of Lemma E.1 and Lemma E.2. For the rest of Section E, we suppose $F$ satisfies Assumption 1.2 and the stochastic gradient oracle satisfies Assumption 3.1 and Assumption 3.2. These proofs are similar to that of Fang et al. (2019), but hinges crucially on the fact that the analysis in Fang et al. (2019) is 'local'.

### E.3 Preliminaries

We now establish useful properties of the parameters of the algorithm defined in Subsection E.1, analogously to Lemma D.1.

**Locality of balls $\mathbb{B}(\boldsymbol{x}^0, B)$:**

**Lemma E.3.** *We have $B \leq \frac{1}{2\rho_0(F(\boldsymbol{w}_0)+1)}$. In particular, for any $\boldsymbol{u} \in \mathbb{B}(\boldsymbol{w}, B)$ for $\boldsymbol{w} \in \mathcal{L}_{F,F(\boldsymbol{w}_0)}$, we have $\|\boldsymbol{u} - \boldsymbol{w}\| \leq \frac{1}{2\rho_0(F(\boldsymbol{w}_0)+1)} \leq \frac{1}{2\rho_0(F(\boldsymbol{w})+1)}$.*

**Proof.** As per Remark 12, we have $\varepsilon \leq 1$. Thus by the choice of parameters in (52),

$$B \leq \frac{\delta}{L_2(\boldsymbol{w}_0)} \leq \frac{1}{\sqrt{L_2(\boldsymbol{w}_0)}} \leq \frac{1}{2\rho_0(F(\boldsymbol{w}_0)+1)}.$$

This completes the proof. $\qquad\square$

**Control over the stochastic gradient oracle:**

**Lemma E.4.** *For all $\boldsymbol{u}$ such that $\boldsymbol{u} \in \mathbb{B}\left(\boldsymbol{w}, \frac{1}{\rho_0(F(\boldsymbol{w}_0)+1)}\right)$ for $\boldsymbol{w} \in \mathcal{L}_{F,F(\boldsymbol{w}_0)}$, we have $\|\nabla f(\boldsymbol{u};\boldsymbol{\zeta}) - \nabla F(\boldsymbol{u})\| \leq \sigma'(\boldsymbol{w}_0)$ for all $\boldsymbol{\zeta}$.*

**Proof.** By Assumption 3.1, we have

$$\|\nabla f(\boldsymbol{u};\boldsymbol{\zeta}) - \nabla F(\boldsymbol{u})\| \leq \sigma(F(\boldsymbol{u})).$$

Now as $\boldsymbol{w} \in \mathcal{L}_{F,F(\boldsymbol{w}_0)}$, we have

$$\frac{1}{\rho_0(F(\boldsymbol{w}_0)+1)} \leq \frac{1}{\rho_0(F(\boldsymbol{w})+1)}.$$

Thus by Lemma 3.1 and again as $\boldsymbol{w} \in \mathcal{L}_{F,F(\boldsymbol{w}_0)}$, we have

$$F(\boldsymbol{u}) \leq F(\boldsymbol{w}) + 1 \leq F(\boldsymbol{w}_0) + 1.$$

Combining these gives Lemma E.4. $\qquad\square$

**Lemma E.5.** *For all $\boldsymbol{u}$ such that $\boldsymbol{u} \in \mathbb{B}\left(\boldsymbol{w}, \frac{1}{\rho_0(F(\boldsymbol{w}_0)+1)}\right)$ for $\boldsymbol{w} \in \mathcal{L}_{F,F(\boldsymbol{w}_0)}$, $\left\|\nabla \tilde{f}(\boldsymbol{u};\boldsymbol{\zeta}) - \nabla F(\boldsymbol{u})\right\| \leq \sigma_1(\boldsymbol{w}_0)$ for all $\boldsymbol{\zeta}$.*

**Proof.** This immediately follows from Lemma E.4 and the definition of $\nabla \tilde{f}(\boldsymbol{u};\boldsymbol{\zeta})$, as $\left\|\tilde{\sigma}\boldsymbol{\Lambda}^t\right\| \leq \tilde{\sigma}$. $\quad\square$

**Locality after one step of SGD:**

**Lemma E.6.** *Consider any $\boldsymbol{u} \in \mathbb{B}(\boldsymbol{w}, B)$ for $\boldsymbol{w} \in \mathcal{L}_{F,F(\boldsymbol{w}_0)}$. Then for all points $\boldsymbol{p}$ in the line segment between $\boldsymbol{u}$ and $\boldsymbol{u} - \eta\nabla \tilde{f}(\boldsymbol{u};\boldsymbol{\zeta})$ for any $\boldsymbol{\zeta}$, we have $\boldsymbol{p} \in \mathbb{B}\left(\boldsymbol{w}, \frac{1}{\rho_0(F(\boldsymbol{w}_0)+1)}\right)$.*

**Proof.** It suffices to show $\boldsymbol{u} - \eta\nabla \tilde{f}(\boldsymbol{u};\boldsymbol{\zeta}) \in \mathbb{B}\left(\boldsymbol{w}, \frac{1}{2(\rho_0(F(\boldsymbol{w}_0)+1))}\right)$; after establishing this, the result then follows by Triangle Inequality and Lemma E.3. To this end, by Triangle Inequality, it suffices to show that

$$\eta\left\|\nabla \tilde{f}(\boldsymbol{u};\boldsymbol{\zeta})\right\| \leq \frac{1}{2\rho_0(F(\boldsymbol{w}_0)+1)}.$$

Indeed, the same reasoning as in the proof of Lemma E.3 gives

$$F(\boldsymbol{u}) \leq F(\boldsymbol{w}_0) + 1.$$

Thus, Assumption 3.2 gives

$$\|\nabla F(\boldsymbol{u})\| \leq \rho_0(F(\boldsymbol{w}_0)+1),$$

and so Lemma E.5 gives

$$\left\|\nabla \tilde{f}(\boldsymbol{u};\boldsymbol{\zeta})\right\| \leq \sigma_1(\boldsymbol{w}_0) + \rho_0(F(\boldsymbol{w}_0) + 1).$$

As per Remark 12, we have

$$\eta \leq \frac{1}{2}B^2\delta \leq \frac{1}{2} \cdot \frac{\delta^3}{L_2(\boldsymbol{w}_0)^2} \leq \frac{1}{2L_2(\boldsymbol{w}_0)^{0.5}}.$$

Combining all the above gives

$$\eta\left\|\nabla \tilde{f}(\boldsymbol{u};\boldsymbol{\zeta})\right\| \leq \frac{1}{2L_2(\boldsymbol{w}_0)^{0.5}} \cdot (\sigma_1(\boldsymbol{w}_0) + \rho_0(F(\boldsymbol{w}_0) + 1))$$

$$\leq \frac{1}{2\rho_0(F(\boldsymbol{w}_0) + 1)(\sigma_1(\boldsymbol{w}_0) + \rho_0(F(\boldsymbol{w}_0) + 1))} \cdot (\sigma_1(\boldsymbol{w}_0) + \rho_0(F(\boldsymbol{w}_0) + 1))$$

$$\leq \frac{1}{2\rho_0(F(\boldsymbol{w}_0) + 1)},$$

which by our earlier remarks completes the proof. $\qquad\square$

**Properties of the effective smoothness parameters:**

**Lemma E.7.** *Consider any $\boldsymbol{x}^0 \in \mathcal{L}_{F,F(\boldsymbol{w}_0)}$. Then we have $\left\|\nabla^2 F(\boldsymbol{u})\right\|_{\mathrm{op}} \leq L_1(\boldsymbol{w}_0)$ for all $\boldsymbol{u}$ such that either:*

- $\boldsymbol{u} \in \mathbb{B}(\boldsymbol{x}^0, B)$,

- *Or $\boldsymbol{u}$ lies in the line segment between some $\boldsymbol{u}' \in \mathbb{B}(\boldsymbol{x}^0, B)$ and $\boldsymbol{u}' - \eta\nabla \tilde{f}(\boldsymbol{u}';\boldsymbol{\zeta})$, for any $\boldsymbol{\zeta}$.*

**Proof.** By Lemma E.3 and Lemma E.6, irrespective of which case for $\boldsymbol{u}$ in the conditions of Lemma E.7 holds, we have

$$\boldsymbol{u} \in \mathbb{B}\left(\boldsymbol{x}^0, \frac{1}{\rho_0(F(\boldsymbol{w}_0) + 1)}\right).$$

As $\boldsymbol{x}^0 \in \mathcal{L}_{F,F(\boldsymbol{w}_0)}$, this implies

$$\left\|\boldsymbol{u} - \boldsymbol{x}^0\right\| \leq \frac{1}{\rho_0(F(\boldsymbol{w}_0) + 1)} \leq \frac{1}{\rho_0(F(\boldsymbol{x}^0) + 1)}.$$

By Lemma 3.1 and as $\boldsymbol{x}^0 \in \mathcal{L}_{F,F(\boldsymbol{w}_0)}$, it follows that

$$F(\boldsymbol{u}) \leq F(\boldsymbol{x}^0) + 1 \leq F(\boldsymbol{w}_0) + 1.$$

The conclusion now follows by Assumption 1.1. $\qquad\square$

**Lemma E.8.** *Consider any $\boldsymbol{x}^0 \in \mathcal{L}_{F,F(\boldsymbol{w}_0)}$. Consider any $\boldsymbol{u}_1, \boldsymbol{u}_2$ such that each $\boldsymbol{u}_i$, $i = 1, 2$ is such that either:*

- $\boldsymbol{u}_i \in \mathbb{B}(\boldsymbol{x}^0, B)$,

- *Or $\boldsymbol{u}_i$ lies in the line segment between some $\boldsymbol{u}' \in \mathbb{B}(\boldsymbol{x}^0, B)$ and $\boldsymbol{u}' - \eta\nabla \tilde{f}(\boldsymbol{u}';\boldsymbol{\zeta})$, for any $\boldsymbol{\zeta}$.*

*Then*

$$\left\|\nabla^2 F(\boldsymbol{u}_1) - \nabla^2 F(\boldsymbol{u}_2)\right\|_{\mathrm{op}} \leq L_2(\boldsymbol{w}_0)\|\boldsymbol{u}_1 - \boldsymbol{u}_2\|.$$

**Proof.** Irrespective of which condition applies to $\boldsymbol{u}_i$, By Lemma E.3 and Lemma E.6, we have

$$\boldsymbol{u}_i \in \mathbb{B}\left(\boldsymbol{x}^0, \frac{1}{\rho_0(F(\boldsymbol{w}_0) + 1)}\right)$$

for $i = 1, 2$. Thus the line segment $\overline{\boldsymbol{u}_1\boldsymbol{u}_2}$ is contained in $\mathbb{B}\left(\tilde{\boldsymbol{w}}, \frac{1}{\rho_0(F(\boldsymbol{w}_0)+1)}\right)$. As $\boldsymbol{x}^0 \in \mathcal{L}_{F,F(\boldsymbol{w}_0)}$, the result now follows from applying Lemma A.6 and Lemma 3.1. $\qquad\square$

**Remark 15.** The reason for the second case in the condition on $\boldsymbol{u}$ or $\boldsymbol{u}_i$ from Lemma E.7, Lemma E.8 will become clear in the proof of Lemma E.2. In particular, to prove Lemma E.2, we will consider $\boldsymbol{u} - \eta \nabla \tilde{f}(\boldsymbol{u}; \boldsymbol{\zeta})$ for $\boldsymbol{u} \in \mathbb{B}(\boldsymbol{x}^0, B)$ where $\boldsymbol{x}^0 \in \mathcal{L}_{F, F(\boldsymbol{w}_0)}$.

**Lemma E.9.** *Consider any $\boldsymbol{x}^0 \in \mathcal{L}_{F, F(\boldsymbol{w}_0)}$. Then for any $\boldsymbol{u} \in \mathbb{B}(\boldsymbol{x}^0, B)$ and any $\boldsymbol{\zeta}$,*

$$\left\| \nabla^2 f(\boldsymbol{u}; \boldsymbol{\zeta}) \right\|_{\mathrm{op}} \le L_1(\boldsymbol{w}_0).$$

**Proof.** By Lemma E.3, we have

$$\boldsymbol{u} \in \mathbb{B}\left( \boldsymbol{x}^0, \frac{1}{\rho_0(F(\boldsymbol{w}_0) + 1)} \right).$$

By Lemma 3.1, because $\boldsymbol{x}^0 \in \mathcal{L}_{F, F(\boldsymbol{w}_0)}$, we have

$$F(\boldsymbol{u}) \le F(\boldsymbol{w}_0) + 1.$$

Moreover, as $\boldsymbol{x}^0 \in \mathcal{L}_{F, F(\boldsymbol{w}_0)}$ and by Lemma E.4 and Corollary 1,

$$\| \nabla f(\boldsymbol{u}; \boldsymbol{\zeta}) \| \le \| \nabla F(\boldsymbol{u}) \| + \sigma'(\boldsymbol{w}_0) \le \rho_0(F(\boldsymbol{w}_0) + 1) + \sigma'(\boldsymbol{w}_0).$$

Thus the result follows from Assumption 3.2. $\qquad\qquad\square$

**Remark 16.** While Lemma E.9 is phrased as an upper bound on the operator norm of $\nabla^2 f(\cdot; \boldsymbol{\zeta})$, it can be easily phrased in terms of the local Lipschitz constant of $\nabla f(\cdot; \boldsymbol{\zeta})$, similar to one of the possibilities in Assumption 1.2.

**Enough noise to escape saddles:** Now we verify that the noise scheme here gives us enough noise to escape saddle points in the $F(\boldsymbol{w}_0)$-sublevel set $\mathcal{L}_{F, F(\boldsymbol{w}_0)}$.

**Definition E.1** $((q^*, \boldsymbol{v})$-narrow property; Definition 2 in Fang et al. (2019)). *A Borel set $\mathcal{A} \subset \mathbb{R}^d$ satisfies the $(q^*, \boldsymbol{v})$-narrow property if for any $\boldsymbol{u} \in \mathcal{A}$, $q \ge q^*$, $\boldsymbol{u} + q\boldsymbol{v} \in \mathcal{A}^c$.*

Immediately, we obtain the following properties of this definition, as also noted in Fang et al. (2019).

**Lemma E.10.** *If $\mathcal{A}$ satisfies the $(q^*, \boldsymbol{v})$-narrow property, then for all $c_1 \in \mathbb{R}^d$, $c_2 \in \mathbb{R}$, $c_1 + c_2\mathcal{A}$ satisfies the $(|c_2|q^*, \boldsymbol{v})$-narrow property.*

We now introduce the following definition:

**Definition E.2** ($\boldsymbol{v}$-dispersive Property; Equivalent of Definition 3 in Fang et al. (2019)). *We say that a random vector $\tilde{\boldsymbol{\xi}}$ has the $\boldsymbol{v}$-dispersive property if for any $\mathcal{A}$ satisfying the $\left( \frac{\sigma_1(\boldsymbol{w}_0)}{4\sqrt{d}}, \boldsymbol{v} \right)$-narrow property, we have*

$$\mathbb{P}(\tilde{\boldsymbol{\xi}} \in \mathcal{A}) \le \frac{1}{2}.$$

Note the slight change of the constant $\frac{1}{2}$ rather than $\frac{1}{4}$ in the above definition compared to that of Fang et al. (2019); this subtle difference will appear in the following proofs, although this will not change too much conceptually.

Now we prove the following Lemma, which shows that our update rule contains enough noise to escape saddle points:

**Lemma E.11** (Dispersive Noise; see also Algorithm 3, Fang et al. (2019)). *The update $\nabla \tilde{f}(\boldsymbol{x}^t; \boldsymbol{\zeta}_{t+1})$ admits the $\boldsymbol{v}$-dispersive property for all unit vectors $\boldsymbol{v}$, for any $\boldsymbol{x}^t$.*

Note this does not necessarily hold for the stochastic gradient oracle itself under our assumptions, hence the artificial noise injection of $\tilde{\sigma} \boldsymbol{\Lambda}^t$.

**Proof of Lemma E.11.** First, we prove that the random vector $\tilde{\sigma} \boldsymbol{\Lambda}^{t+1}$ satisfies the Dispersive Noise property for all unit vectors $\boldsymbol{v}$. Consider any $\mathcal{A}$ satisfying the $\left( \frac{\sigma_1(\boldsymbol{w}_0)}{4\sqrt{d}}, \boldsymbol{v} \right)$-narrow property. Note we have

$$\mathbb{P}(\tilde{\sigma} \boldsymbol{\Lambda}^{t+1} \in \mathcal{A}) = \mathbb{P}(\boldsymbol{\Lambda}^{t+1} \in \tilde{\sigma}^{-1} \mathcal{A})$$

$$\leq \frac{\sigma_1(\boldsymbol{w}_0)/4\sqrt{d}}{\tilde{\sigma}} \cdot \frac{\mathrm{Vol}^{d-1}\mathbb{B}(\vec{\boldsymbol{0}},1)}{\mathrm{Vol}^d\mathbb{B}(\vec{\boldsymbol{0}},1)}$$

$$\leq \frac{\sigma_1(\boldsymbol{w}_0)/4\sqrt{d}}{\tilde{\sigma}} \cdot \sqrt{d} = \frac{\sigma_1(\boldsymbol{w}_0)}{4\tilde{\sigma}}.$$

Here, the inequality follows from an elementary calculation with multivariate calculus, analogous to the calculation in the proof of Lemma D.3, which we detailed in full in this article. An analogous calculation can also be found in Jin et al. (2017), proof of Lemma 14, and in Appendix F, Fang et al. (2019).

Now, note as $\tilde{\sigma} \geq \sigma'(\boldsymbol{w}_0)$, we have

$$\frac{\sigma_1(\boldsymbol{w}_0)}{4\tilde{\sigma}} \leq \frac{\sigma'(\boldsymbol{w}_0) + \tilde{\sigma}}{4\tilde{\sigma}} \leq \frac{1}{2},$$

and so

$$\mathbb{P}(\tilde{\sigma}\boldsymbol{\Lambda}^{t+1} \in \mathcal{A}) \leq \frac{1}{2}.$$

Consequently the random vector $\tilde{\sigma}\boldsymbol{\Lambda}^t$ satisfies the Dispersive Noise property for all unit vectors $\boldsymbol{v}$.

Now, we show that $\nabla \tilde{f}(\boldsymbol{x}^t; \boldsymbol{\zeta}_{t+1})$ satisfies the $\boldsymbol{v}$-dispersive property as wanted. The proof is analogous to part iii, Proposition 4 of Fang et al. (2019). Consider any unit vector $\boldsymbol{v}$. Recall that $\Lambda^t$ and $\nabla f(\boldsymbol{x}^t; \boldsymbol{\zeta}_{t+1})$ are independent. Since the $(q^\star, \boldsymbol{v})$-narrow property is evidently preserved with the same parameters by adding a fixed vector to $\mathcal{A}$, we obtain the following bound on the following conditional probability:

$$\mathbb{P}\big(\nabla \tilde{f}(\boldsymbol{x}^t; \boldsymbol{\zeta}_{t+1}) \in \mathcal{A} | \nabla f(\boldsymbol{x}^t; \boldsymbol{\zeta}_{t+1})\big) = \mathbb{P}\big(\nabla f(\boldsymbol{x}^t; \boldsymbol{\zeta}_{t+1}) + \tilde{\sigma}\boldsymbol{\Lambda}^{t+1} \in \mathcal{A} | \nabla f(\boldsymbol{x}^t; \boldsymbol{\zeta}_{t+1})\big)$$

$$= \mathbb{P}\big(\tilde{\sigma}\boldsymbol{\Lambda}^{t+1} \in -\nabla f(\boldsymbol{x}^t; \boldsymbol{\zeta}_{t+1}) + \mathcal{A} | \nabla f(\boldsymbol{x}^t; \boldsymbol{\zeta}_{t+1})\big) \leq \frac{1}{2}.$$

This holds irrespective of conditioning, which implies that $\nabla \tilde{f}(\boldsymbol{x}^t; \boldsymbol{\zeta}_{t+1})$ satisfies the $\boldsymbol{v}$-dispersive property. $\qquad\square$

### E.4 Escaping Saddles

We first aim to prove that we can efficiently escape strict saddle points in the $F(\boldsymbol{w}_0)$-sublevel set, similarly to Fang et al. (2019). In particular, we aim to prove the following Lemma E.12. The contrapositive of Lemma E.12 will in turn be used to prove Lemma E.1, which establishes that Algorithm 2 can find SOSPs.

**Lemma E.12** (Equivalent of Proposition 7 in Fang et al. (2019)). *Consider a sequence of iterates* $\boldsymbol{x}^0, \boldsymbol{x}^1, \ldots$ *beginning at* $\boldsymbol{x}^0$ *comprising an instance of the while loop in Algorithm 2. Suppose* $\boldsymbol{x}^0 \in \mathcal{L}_{F,F(\boldsymbol{w}_0)}$ *and that* $\lambda_{\mathrm{MIN}}(\nabla^2 F(\boldsymbol{x}^0)) \leq -\delta_2$ *for* $\delta_2 > 0$. *Then when the while loop of Algorithm 2 is initialized at* $\boldsymbol{x}^0$, *with probability at least* $1 - \frac{p}{3}$, *we have*

$$\mathcal{K}_0 \leq K_0 = \lfloor \frac{\log(3/p)}{\log(0.8^{-1})} + 1 \rfloor K_o.$$

**Remark 17.** For $\delta_2$ very small, note the guarantee from Lemma E.12 will deteriorate because $K_0$ scales with $\delta_2^{-1}$.

To prove Lemma E.12, we use the same strategy as in Fang et al. (2019). However, as we do not have global Lipschitzness of the gradient and Hessian, we must be careful. We use that the strategy only requires control over points that are 'local', i.e. near $\boldsymbol{x}^0$, since the proof strategy studies escape from the ball $\mathbb{B}(\boldsymbol{x}^0, B)$. We then appeal to control over $F$ in $\mathbb{B}(\boldsymbol{x}^0, B)$ that we have by Subsection E.3.

**Remark 18.** In this section Subsection E.4, probability is over the samples $\boldsymbol{\zeta}_k$ and the artificial noise injections $\Lambda^k$.

Now we go into the details. As in Fang et al. (2019), let $\boldsymbol{w}^k(\boldsymbol{u})$ be the iterates of SGD starting from a given $\boldsymbol{u}$ using the *same* stochastic samples as $\boldsymbol{x}^k$ and the same noise additions $\tilde{\sigma}\boldsymbol{\Lambda}^k$. In particular

$$\boldsymbol{w}^k(\boldsymbol{u}) = \boldsymbol{w}^{k-1}(\boldsymbol{u}) - \eta\nabla \tilde{f}(\boldsymbol{w}^{k-1}(\boldsymbol{u}); \boldsymbol{\zeta}_k).$$

Thus $\boldsymbol{x}^k = \boldsymbol{w}^k(\boldsymbol{x}^0)$.

Also for all $\boldsymbol{u}$, let $\mathcal{K}_{\text{exit}}(\boldsymbol{u})$ be the stopping time defined by

$$\mathcal{K}_{\text{exit}}(\boldsymbol{u}) := \inf\{k \geq 0 : \|\boldsymbol{w}^k(\boldsymbol{u}) - \boldsymbol{x}^0\| > B\}.$$

Thus $\mathcal{K}_0 = \mathcal{K}_{\text{exit}}(\boldsymbol{x}^0)$.

The high-level idea from Fang et al. (2019), similar to as in Jin et al. (2017), is to consider the 'bad initialization region' around $\mathbb{B}(\boldsymbol{x}^0, B)$ where iterates initialized in this bad region escape with low probability. We then prove that this bad initialization region is 'narrow', and consequently we can escape the saddle point efficiently.

In particular, define

$$\mathcal{S}_{K_o}^B(\boldsymbol{x}^0) = \{\boldsymbol{u} \in \mathbb{R}^d : \mathbb{P}(\mathcal{K}_{\text{exit}}(\boldsymbol{u}) < K_o) \leq 0.4\}.$$

Note by definition that $\mathcal{S}_{K_0}^B(\boldsymbol{x}^0) \subseteq \mathbb{B}(\boldsymbol{x}^0, B)$.

First let $q_0 = \frac{\sigma_1(\boldsymbol{w}_0)\eta}{4\sqrt{d}}$. We establish the following Lemma, which verifies that $\mathcal{S}_{K_o}^B(\boldsymbol{x}^0)$ is 'narrow' in a suitable sense.

**Lemma E.13** (Equivalent of Lemma 8 in Fang et al. (2019); also similar to Lemma 15, Jin et al. (2017)). *Suppose the assumptions of Lemma E.12 hold. Let $\boldsymbol{e}_1$ be an arbitrary unit eigenvector of $\nabla^2 F(\boldsymbol{x}^0)$ corresponding to its smallest eigenvalue $-\delta_m \leq -\delta_2$. Then for any $q \geq q_0 = \frac{\sigma_1(\boldsymbol{w}_0)\eta}{4\sqrt{d}}$ and any $\boldsymbol{u}, \boldsymbol{u} + q\boldsymbol{e}_1 \in \mathbb{B}(\boldsymbol{x}^0, B)$, we have that*

$$\mathbb{P}((\mathcal{K}_{\text{exit}}(\boldsymbol{u}) \geq K_o) \text{ and } (\mathcal{K}_{\text{exit}}(\boldsymbol{u} + q\boldsymbol{e}_1) \geq K_o)) \leq 0.1.$$

*Here probability is over the single sequence of samples used to compute stochastic gradients and the artificial noise injection.*

**Remark 19.** The proof of Lemma E.13 crucially uses that $\nabla^2 F(\boldsymbol{x}^0)$ has a negative eigenvector, as one would expect.

Note we have, as in Fang et al. (2019), that

$$K_o = 2\log\left(\frac{24\sqrt{d}}{\eta}\right)\eta^{-1}\delta_2^{-1} \geq \frac{\log(6/q_0)}{\log(1 + \eta\delta_2)} \geq \frac{\log(6B/q_0)}{\log(1 + \eta\delta_2)}. \tag{54}$$

This follows evidently from the choice of parameters and definition of $q_0$, and Remark 12 which states that it is enough to show the result for $\eta\delta_2$ at most a universal constant, namely one satisfying $\log(1 + x) \geq \frac{x}{2}$. Now using Lemma E.13, we prove Lemma E.12:

**Proof of Lemma E.12 given Lemma E.13.** Given Lemma E.13, we first prove that the bad initialization region $\mathcal{S}_{K_o}^B(\boldsymbol{x}^0)$ satisfies the $(q_0, \boldsymbol{e}_1)$-narrow property, i.e. that there are no points $\boldsymbol{u}, \boldsymbol{u} + q\boldsymbol{e}_1 \in \mathcal{S}_{K_o}^B(\boldsymbol{x}^0)$ where $q \geq q_0 = \frac{\sigma_1(\boldsymbol{w}_0)\eta}{4\sqrt{d}}$. This part of the proof is identical to Proposition 7, Fang et al. (2019). If such points existed we would have

$$\mathbb{P}(\mathcal{K}_{\text{exit}}(\boldsymbol{u}) \geq K_o) \geq 0.6, \mathbb{P}(\mathcal{K}_{\text{exit}}(\boldsymbol{u} + q\boldsymbol{e}_1) \geq K_o) \geq 0.6.$$

This implies

$$\mathbb{P}((\mathcal{K}_{\text{exit}}(\boldsymbol{u}) \geq K_o) \text{ and } (\mathcal{K}_{\text{exit}}(\boldsymbol{u} + q\boldsymbol{e}_1) \geq K_o)) \geq \mathbb{P}(\mathcal{K}_{\text{exit}}(\boldsymbol{u}) \geq K_o) + \mathbb{P}(\mathcal{K}_{\text{exit}}(\boldsymbol{u} + q\boldsymbol{e}_1)) - 1$$
$$\geq 0.2,$$

which contradicts Lemma E.13.

From here, we prove Lemma E.12. For this rest of the proof of Lemma E.12, we only consider $\boldsymbol{u}$ and do not consider the iterates from $\boldsymbol{u} + q\boldsymbol{e}_1$. Recall $\mathcal{S}_{K_o}^B$ satisfies the $(q_0, \boldsymbol{e}_1)$-narrow property with $q_0 = \frac{\eta\sigma_1(\boldsymbol{w}_0)}{4\sqrt{d}}$ as shown above. Thus we have for any $\boldsymbol{u} \in \mathbb{B}(\boldsymbol{x}^0, B)$,

$$\mathbb{P}(\boldsymbol{w}^1(\boldsymbol{u}) \in \mathcal{S}_{K_o}^B(\boldsymbol{x}^0)) = \mathbb{P}(\boldsymbol{u} - \eta\nabla\tilde{f}(\boldsymbol{u}; \boldsymbol{\zeta}_1) \in \mathcal{S}_{K_o}^B(\boldsymbol{x}^0))$$
$$= \mathbb{P}(\nabla\tilde{f}(\boldsymbol{u}; \boldsymbol{\zeta}_1) \in \eta^{-1}(-\mathcal{S}_{K_o}^B(\boldsymbol{x}^0) + \boldsymbol{u})) \leq \frac{1}{2}. \tag{55}$$

The last step follows from the definition of the $\boldsymbol{w}^k(\boldsymbol{u})$, the scale and translation properties of the $(q_0, \boldsymbol{e}_1)$-narrow property which implies that $\eta^{-1}\left(-S_{K_o}^B(\boldsymbol{x}^0) + \boldsymbol{u}\right)$ satisfies the $\left(\frac{\sigma_1(\boldsymbol{w}_0)}{4\sqrt{d}}, \boldsymbol{e}_1\right)$-narrow property, and that $\nabla \tilde{f}(\boldsymbol{u}; \boldsymbol{\zeta}_1)$ satisfies the $\boldsymbol{e}_1$-dispersive property by Lemma E.11.

Note as events we have $\{\mathcal{K}_{\text{exit}}(\boldsymbol{w}^1(\boldsymbol{u})) < K_o\} \subseteq \{\mathcal{K}_{\text{exit}}(\boldsymbol{u}) \le K_o\}$. Thus by Law of Total Expectation, for all $\boldsymbol{u} \in \mathbb{B}(\boldsymbol{x}^0, B)$,

$$\mathbb{P}(\mathcal{K}_{\text{exit}}(\boldsymbol{u}) \le K_o) \ge \mathbb{P}\left(\mathcal{K}_{\text{exit}}(\boldsymbol{w}^1(\boldsymbol{u})) < K_o\right)$$
$$\ge \mathbb{E}\left[\mathbb{P}\left(\mathcal{K}_{\text{exit}}(\boldsymbol{w}^1(\boldsymbol{u})) < K_o|\mathfrak{F}^1\right)|\{\boldsymbol{w}^1(\boldsymbol{u}) \in (\mathcal{S}_{K_o}^B(\boldsymbol{x}^0))^c\}\right]. \quad (56)$$

Conditioned on $\boldsymbol{w}^1(\boldsymbol{u}) \le (\mathcal{S}_{K_o}^B(\boldsymbol{x}^0))^c$, we have by definition of $\mathcal{S}_{K_o}^B(\boldsymbol{x}^0)$ that $\mathbb{P}\left(\mathcal{K}_{\text{exit}}(\boldsymbol{w}^1(\boldsymbol{u})) < K_o|\mathfrak{F}^1\right) \ge 0.4$. By (55), for all $\boldsymbol{u} \in \mathbb{B}(\boldsymbol{x}^0, B)$, we have

$$\mathbb{P}\left(\boldsymbol{w}^1(\boldsymbol{u}) \in \mathcal{S}_{K_o}^B(\boldsymbol{x}^0)^c\right) \ge \frac{1}{2}.$$

Thus combining with (56) implies for all $\boldsymbol{u} \in \mathbb{B}(\boldsymbol{x}^0, B)$,

$$\mathbb{P}(\mathcal{K}_{\text{exit}}(\boldsymbol{u}) \le K_o) \ge 0.4 \cdot \frac{1}{2} = 0.2. \quad (57)$$

Now consider any $N' \ge 1$. Notice as events,

$$\{\mathcal{K}_{\text{exit}}(\boldsymbol{u}) > N'K_o\} = \left\{\mathcal{K}_{\text{exit}}(\boldsymbol{w}^{(N'-1)K_o}(\boldsymbol{u})) > K_o\right\}$$
$$= \left\{\mathcal{K}_{\text{exit}}(\boldsymbol{w}^{(N'-1)K_o}(\boldsymbol{u})) > K_o\right\} \cap \{\mathcal{K}_{\text{exit}}(\boldsymbol{u}) > (N'-1)K_o\}.$$

Therefore,

$$\mathbb{P}(\mathcal{K}_{\text{exit}}(\boldsymbol{u}) > N'K_o) = \mathbb{E}\left[\mathbb{P}\left(\mathcal{K}_{\text{exit}}(\boldsymbol{w}^{(N'-1)K_o}(\boldsymbol{u})) > K_o|\mathfrak{F}^{K_o}\right)|\{\mathcal{K}_{\text{exit}}(\boldsymbol{u}) > (N'-1)K_o\}\right].$$

Note that conditioned on $\mathcal{K}_{\text{exit}}(\boldsymbol{u}) > (N'-1)K_o$, it follows that $\mathcal{K}_{\text{exit}}(\boldsymbol{w}^{(N'-1)K_o}(\boldsymbol{u})) \in \mathbb{B}(\boldsymbol{x}^0, B)$. Therefore $\mathbb{P}\left(\mathcal{K}_{\text{exit}}(\boldsymbol{w}^{(N'-1)K_o}(\boldsymbol{u})) > K_o|\mathfrak{F}^{K_o}\right) \le \sup_{\boldsymbol{u}' \in \mathbb{B}(\boldsymbol{x}^0, B)} \mathbb{P}(\mathcal{K}_{\text{exit}}(\boldsymbol{u}') > K_o)$. Using (57), we can upper bound

$$\mathbb{P}(\mathcal{K}_{\text{exit}}(\boldsymbol{u}) > N'K_o) \le \mathbb{P}(\mathcal{K}_{\text{exit}}(\boldsymbol{u}) > (N'-1)K_o) \cdot \sup_{\boldsymbol{u}' \in \mathbb{B}(\boldsymbol{x}^0, B)} \mathbb{P}(\mathcal{K}_{\text{exit}}(\boldsymbol{u}') > K_o)$$
$$\le 0.8\mathbb{P}(\mathcal{K}_{\text{exit}}(\boldsymbol{u}) > (N'-1)K_o). \quad (58)$$

Recall that $K_0 = \lfloor \frac{\log(3/p)}{\log(0.8^{-1})} + 1 \rfloor K_o$. Let $N = \lfloor \frac{\log(3/p)}{\log(0.8^{-1})} + 1 \rfloor$. We obtain by repeatedly applying (58) for $N' = N, N-1, \ldots$ that

$$\mathbb{P}(\mathcal{K}_{\text{exit}}(\boldsymbol{u}) > NK_o) \le 0.8^N \le p/3.$$

This gives the desired result. $\qquad \square$

Now we prove Lemma E.13.

**Proof of Lemma E.13.** Again, we proceed similarly as the proof of Lemma 8, Fang et al. (2019). The main difference is we only have control over the relevant derivatives prior to the escape from $\mathbb{B}(\boldsymbol{x}^0, B)$ (recall $\boldsymbol{x}^0 \in \mathcal{L}_{F, F(\boldsymbol{w}_0)}$). However, it turns out that this is sufficient for the proof to go through.

**Setup.** Recall that we have $\boldsymbol{w}^0(\boldsymbol{u}) = \boldsymbol{u}$, and

$$\boldsymbol{w}^k(\boldsymbol{u}) = \boldsymbol{w}^{k-1}(\boldsymbol{u}) - \eta\nabla\tilde{f}(\boldsymbol{w}^{k-1}(\boldsymbol{u}); \boldsymbol{\zeta}_k),$$
$$\boldsymbol{w}^k(\boldsymbol{u} + q\boldsymbol{e}_1) = \boldsymbol{w}^{k-1}(\boldsymbol{u} + q\boldsymbol{e}_1) - \eta\nabla\tilde{f}(\boldsymbol{w}^{k-1}(\boldsymbol{u} + q\boldsymbol{e}_1); \boldsymbol{\zeta}_k).$$

Now define the following stopping time:

$$\mathcal{K}_1 = \mathcal{K}_{\text{exit}}(\boldsymbol{u}) \wedge \mathcal{K}_{\text{exit}}(\boldsymbol{u} + q\boldsymbol{e}_1).$$

For solely the purpose of analysis, consider the following sequence:

$$\boldsymbol{z}^k = \begin{cases} \boldsymbol{w}^k(\boldsymbol{u} + q\boldsymbol{e}_1) - \boldsymbol{w}^k(\boldsymbol{u}) & : k \le \mathcal{K}_1 \\ \left(\boldsymbol{I} - \eta\nabla^2 F(\boldsymbol{x}^0)\right)\boldsymbol{z}^{k-1} & : k > \mathcal{K}_1 \end{cases}. \quad (59)$$

Clearly the $\boldsymbol{z}^k$ are $\mathfrak{F}^k$-measurable, because the event $\{k \le \mathcal{K}_1\}$ is $\mathfrak{F}^k$-measurable.

**Remark 20.** Note unlike Fang et al. (2019), the first case holds when $k \leq \mathcal{K}_1$ rather than $k < \mathcal{K}_1$. That being said we expect that if one uses the exact same definition as in Fang et al. (2019) for the $\boldsymbol{z}^k$, the proof this generalized smooth setting will still work, with a slightly modified argument compared to the proof we present.

Notice by definition of $\boldsymbol{w}^0(\boldsymbol{u}), \boldsymbol{w}^0(\boldsymbol{u}+q\boldsymbol{e}_1)$ and assumption of Lemma E.13 that $\boldsymbol{u}, \boldsymbol{u}+q\boldsymbol{e}_1 \in \mathbb{B}(\boldsymbol{x}^0, B)$, we have $\mathcal{K}_1 > 0$. Thus,

$$\boldsymbol{z}^0 = q\boldsymbol{e}_1.$$

**Controlling the $\boldsymbol{z}^k$.** Let $\boldsymbol{H} = \nabla^2 F(\boldsymbol{x}^0)$. We have the following lemma to control the $\boldsymbol{z}^k$ from (59). For all $k$, define

$$\boldsymbol{D}^k := \nabla^2 F(\boldsymbol{x}^0) - \int_0^1 \nabla^2 F\big(\boldsymbol{w}^k(\boldsymbol{u}) + \theta(\boldsymbol{w}^k(\boldsymbol{u} + q\boldsymbol{e}_1) - \boldsymbol{w}^k(\boldsymbol{u}))\big)\mathrm{d}\theta, \tag{60}$$

$$\boldsymbol{\xi}_d^k := \big(\nabla F(\boldsymbol{w}^{k-1}(\boldsymbol{u} + q\boldsymbol{e}_1)) - \nabla F(\boldsymbol{w}^{k-1}(\boldsymbol{u}))\big) - \big(\nabla \tilde{f}(\boldsymbol{w}^{k-1}(\boldsymbol{u} + q\boldsymbol{e}_1); \boldsymbol{\zeta}_k) - \nabla \tilde{f}(\boldsymbol{w}^{k-1}(\boldsymbol{u}); \boldsymbol{\zeta}_k)\big). \tag{61}$$

Recall by definition of $\boldsymbol{w}^k(\boldsymbol{u})$, we have

$$\nabla \tilde{f}(\boldsymbol{w}^{k-1}(\boldsymbol{u} + q\boldsymbol{e}_1); \boldsymbol{\zeta}_k) = \nabla f(\boldsymbol{w}^{k-1}(\boldsymbol{u} + q\boldsymbol{e}_1); \boldsymbol{\zeta}_k) + \tilde{\sigma}\boldsymbol{\Lambda}^k,$$

$$\nabla \tilde{f}(\boldsymbol{w}^{k-1}(\boldsymbol{u}); \boldsymbol{\zeta}_k) = \nabla f(\boldsymbol{w}^{k-1}(\boldsymbol{u}); \boldsymbol{\zeta}_k) + \tilde{\sigma}\boldsymbol{\Lambda}^k,$$

for the *same* noise sequence $\boldsymbol{\Lambda}^k$. Thus we also have

$$\boldsymbol{\xi}_d^k = \big(\nabla F(\boldsymbol{w}^{k-1}(\boldsymbol{u} + q\boldsymbol{e}_1)) - \nabla F(\boldsymbol{w}^{k-1}(\boldsymbol{u}))\big) - \big(\nabla f(\boldsymbol{w}^{k-1}(\boldsymbol{u} + q\boldsymbol{e}_1); \boldsymbol{\zeta}_k) - \nabla f(\boldsymbol{w}^{k-1}(\boldsymbol{u}); \boldsymbol{\zeta}_k)\big). \tag{62}$$

**Lemma E.14** (Equivalent of Lemma 13, Fang et al. (2019))**.** *We have that for all $k \leq \mathcal{K}_1$,*

$$\boldsymbol{z}^k = (\boldsymbol{I} - \eta\boldsymbol{H})\boldsymbol{z}^{k-1} + \eta\boldsymbol{D}^{k-1}\boldsymbol{z}^{k-1} + \eta\boldsymbol{\xi}_d^k.$$

*Furthermore, we have the following properties of the $\boldsymbol{D}^k$ and $\boldsymbol{\xi}_d^k$ defined in (60), (61):*

1. *For all such $k \leq \mathcal{K}_1$, we have*

$$\big\|\boldsymbol{D}^{k-1}\big\| \leq L_2(\boldsymbol{w}_0) \max\big(\big\|\boldsymbol{w}^{k-1}(\boldsymbol{u} + q\boldsymbol{e}_1) - \boldsymbol{x}^0\big\|, \big\|\boldsymbol{w}^{k-1}(\boldsymbol{u}) - \boldsymbol{x}^0\big\|\big) \leq L_2(\boldsymbol{w}_0)B.$$

2. *For all $k$, we have*

$$\mathbb{E}\big[\boldsymbol{\xi}_d^k|\mathfrak{F}^{k-1}\big] = 0.$$

3. *For all $k \leq \mathcal{K}_1$, we have*

$$\big\|\boldsymbol{\xi}_d^k\big\| \leq 2L_1(\boldsymbol{w}_0)\big\|\boldsymbol{z}^{k-1}\big\|.$$

**Proof.** We prove each part one at a time:

1. For $k \leq \mathcal{K}_1$, using the definition of $\boldsymbol{z}^k$, it follows that

$$\begin{aligned}
\boldsymbol{z}^k &= \boldsymbol{w}^k(\boldsymbol{u} + q\boldsymbol{e}_1) - \boldsymbol{w}^k(\boldsymbol{u}) \\
&= \boldsymbol{w}^{k-1}(\boldsymbol{u} + q\boldsymbol{e}_1) - \boldsymbol{w}^{k-1}(\boldsymbol{u}) - \eta\big(\nabla \tilde{f}(\boldsymbol{w}^{k-1}(\boldsymbol{u} + q\boldsymbol{e}_1); \boldsymbol{\zeta}_k) - \nabla \tilde{f}(\boldsymbol{w}^{k-1}(\boldsymbol{u}); \boldsymbol{\zeta}_k)\big) \\
&= \boldsymbol{z}^{k-1} - \eta\big(\nabla F(\boldsymbol{w}^{k-1}(\boldsymbol{u} + q\boldsymbol{e}_1)) - \nabla F(\boldsymbol{w}^{k-1}(\boldsymbol{u}))\big) \\
&\quad + \eta\big[\big(\nabla F(\boldsymbol{w}^{k-1}(\boldsymbol{u} + q\boldsymbol{e}_1)) - \nabla F(\boldsymbol{w}^{k-1}(\boldsymbol{u}))\big) - \big(\nabla \tilde{f}(\boldsymbol{w}^{k-1}(\boldsymbol{u} + q\boldsymbol{e}_1); \boldsymbol{\zeta}_k) - \nabla \tilde{f}(\boldsymbol{w}^{k-1}(\boldsymbol{u}); \boldsymbol{\zeta}_k)\big)\big] \\
&= \boldsymbol{z}^{k-1} - \eta\bigg[\int_0^1 \nabla^2 F\big(\boldsymbol{w}^{k-1}(\boldsymbol{u}) + \theta(\boldsymbol{w}^{k-1}(\boldsymbol{u} + q\boldsymbol{e}_1) - \boldsymbol{w}^{k-1}(\boldsymbol{u}))\big)\mathrm{d}\theta\bigg]\boldsymbol{z}^{k-1} + \eta\boldsymbol{\xi}_d^k \\
&= \boldsymbol{z}^{k-1} - \eta\big(\boldsymbol{H} - \boldsymbol{D}^{k-1}\big)\boldsymbol{z}^{k-1} + \eta\boldsymbol{\xi}_d^k.
\end{aligned}$$

This proves the desired property of the $\boldsymbol{z}^k$.

2. For the required properties of the $\boldsymbol{D}^{k-1}$, consider any $k \leq \mathcal{K}_1$. First, notice $\boldsymbol{w}^{k-1}(\boldsymbol{u}) + \theta(\boldsymbol{w}^{k-1}(\boldsymbol{u} + q\boldsymbol{e}_1) - \boldsymbol{w}^{k-1}(\boldsymbol{u})) = \theta \boldsymbol{w}^{k-1}(\boldsymbol{u} + q\boldsymbol{e}_1) + (1 - \theta)\boldsymbol{w}^{k-1}(\boldsymbol{u})$ for any $\theta \in [0, 1]$. For $k \leq \mathcal{K}_1$, both $\boldsymbol{w}^{k-1}(\boldsymbol{u} + q\boldsymbol{e}_1), \boldsymbol{w}^{k-1}(\boldsymbol{u}) \in \mathbb{B}(\boldsymbol{x}^0, B)$. Note this still remains true for $k = \mathcal{K}_1$ because for $k - 1 = \mathcal{K}_1 - 1 < \mathcal{K}_1$, the definition of $\mathcal{K}_1$ implies that the iterates $\boldsymbol{w}^{k-1}(\boldsymbol{u} + q\boldsymbol{e}_1), \boldsymbol{w}^{k-1}(\boldsymbol{u}) \in \mathbb{B}(\boldsymbol{x}^0, B)$.

Thus for any $\theta \in [0, 1]$, $\boldsymbol{w}^{k-1}(\boldsymbol{u}) + \theta(\boldsymbol{w}^{k-1}(\boldsymbol{u} + q\boldsymbol{e}_1) - \boldsymbol{w}^{k-1}(\boldsymbol{u})) \in \mathbb{B}(\boldsymbol{x}^0, B)$, and so all points $\boldsymbol{p}$ on the line segment between $\boldsymbol{x}^0$ and $\boldsymbol{w}^{k-1}(\boldsymbol{u}) + \theta(\boldsymbol{w}^{k-1}(\boldsymbol{u} + q\boldsymbol{e}_1) - \boldsymbol{w}^{k-1}(\boldsymbol{u}))$ lie in $\mathbb{B}(\boldsymbol{x}^0, B)$. Thus by Lemma E.8,

$$
\begin{aligned}
\left\|\boldsymbol{D}^{k-1}\right\| &= \left\|\nabla^2 F(\boldsymbol{x}^0) - \int_0^1 \nabla^2 F(\boldsymbol{w}^{k-1}(\boldsymbol{u}) + \theta(\boldsymbol{w}^{k-1}(\boldsymbol{u} + q\boldsymbol{e}_1) - \boldsymbol{w}^{k-1}(\boldsymbol{u})))\mathrm{d}\theta\right\| \\
&\leq \int_0^1 \left\|\nabla^2 F(\boldsymbol{x}^0) - \nabla^2 F\big(\boldsymbol{w}^{k-1}(\boldsymbol{u}) + \theta(\boldsymbol{w}^{k-1}(\boldsymbol{u} + q\boldsymbol{e}_1) - \boldsymbol{w}^{k-1}(\boldsymbol{u}))\big)\right\|\mathrm{d}\theta \\
&\leq L_2(\boldsymbol{w}_0) \int_0^1 \left\|\theta\big(\boldsymbol{w}^{k-1}(\boldsymbol{u} + q\boldsymbol{e}_1) - \boldsymbol{x}^0\big) + (1 - \theta)\big(\boldsymbol{w}^{k-1}(\boldsymbol{u}) - \boldsymbol{x}^0\big)\right\|\mathrm{d}\theta \\
&\leq L_2(\boldsymbol{w}_0) \max\left\{\left\|\boldsymbol{w}^{k-1}(\boldsymbol{u} + q\boldsymbol{e}_1) - \boldsymbol{x}^0\right\|, \left\|\boldsymbol{w}^{k-1}(\boldsymbol{u}) - \boldsymbol{x}^0\right\|\right\} \\
&\leq L_2(\boldsymbol{w}_0) B.
\end{aligned}
$$

The last line follows since $k \leq \mathcal{K}_1$, hence $k - 1 < \mathcal{K}_1$, thus $\boldsymbol{w}^{k-1}(\boldsymbol{u} + q\boldsymbol{e}_1), \boldsymbol{w}^{k-1}(\boldsymbol{u}) \in \mathbb{B}(\boldsymbol{x}^0, B)$.

3. Next as the stochastic gradient oracle $\nabla f(\cdot; \boldsymbol{\zeta})$ is unbiased, applying Linearity of Expectation on (62), it follows that $\mathbb{E}\big[\boldsymbol{\xi}_d^k | \mathfrak{F}^{k-1}\big] = 0$ for all $k$.

For the bound on the magnitude of $\boldsymbol{\xi}_d^k$, again recall by the above that for $k \leq \mathcal{K}_1$, we have

$$
\boldsymbol{w}^{k-1}(\boldsymbol{u} + q\boldsymbol{e}_1), \boldsymbol{w}^{k-1}(\boldsymbol{u}) \in \mathbb{B}(\boldsymbol{x}^0, B).
$$

Thus for all $\boldsymbol{p}$ on the line segment between $\boldsymbol{w}^{k-1}(\boldsymbol{u} + q\boldsymbol{e}_1), \boldsymbol{w}^{k-1}(\boldsymbol{u})$, we have $\boldsymbol{p} \in \mathbb{B}(\boldsymbol{x}^0, B)$. Thus by Lemma E.7, $\left\|\nabla^2 F(\boldsymbol{p})\right\| \leq L_1(\boldsymbol{w}_0)$. By Lemma E.9, for any $\boldsymbol{\zeta}$, $\left\|\nabla^2 f(\boldsymbol{p}; \boldsymbol{\zeta})\right\| \leq L_1(\boldsymbol{w}_0)$. Recalling (62) gives

$$
\begin{aligned}
&\left\|\boldsymbol{\xi}_d^k\right\| \\
&\leq \left\|\nabla F(\boldsymbol{w}^{k-1}(\boldsymbol{u} + q\boldsymbol{e}_1)) - \nabla F(\boldsymbol{w}^{k-1}(\boldsymbol{u}))\right\| + \left\|\nabla f(\boldsymbol{w}^{k-1}(\boldsymbol{u} + q\boldsymbol{e}_1); \boldsymbol{\zeta}_k) - \nabla f(\boldsymbol{w}^{k-1}(\boldsymbol{u}); \boldsymbol{\zeta}_k)\right\| \\
&\leq 2L_1(\boldsymbol{w}_0)\left\|\boldsymbol{w}^{k-1}(\boldsymbol{u} + q\boldsymbol{e}_1) - \boldsymbol{w}^{k-1}(\boldsymbol{u})\right\| \\
&= 2L_1(\boldsymbol{w}_0)\left\|\boldsymbol{z}^{k-1}\right\|.
\end{aligned}
$$

In the last step, we used the definition of $\boldsymbol{z}^k$ for $k \leq \mathcal{K}_1$.

This proves all the desired parts of Lemma E.14. $\qquad\square$

**Controlling iterates under a high probability event.** We now consider a rescaled iteration as considered in Fang et al. (2019). Recall the definition of $\delta_m \geq \delta_2$ in the statement of Lemma E.13. For each $k = 0, 1, \ldots$, we define:

$$
\boldsymbol{\psi}_k := q^{-1}(1 + \eta\delta_m)^{-k}\boldsymbol{z}_k.
$$

**Lemma E.15** (Equivalent of the first part of Lemma 14, Fang et al. (2019)). *Define* $\hat{\boldsymbol{D}}_k := (1 + \eta\delta_m)^{-1}\boldsymbol{D}_k$, *and slightly overloading notation, define*

$$
\boldsymbol{\zeta}_d^k := q^{-1}(1 + \eta\delta_m)^{-k}\boldsymbol{\xi}_d^k.
$$

*Then for* $k \leq \mathcal{K}_1$, *we have* $\boldsymbol{\psi}^0 = \boldsymbol{e}_1$ *and*

$$
\boldsymbol{\psi}^k = \frac{\boldsymbol{I} - \eta\boldsymbol{H}}{1 + \eta\delta_m}\boldsymbol{\psi}^{k-1} + \eta\hat{\boldsymbol{D}}^{k-1}\boldsymbol{\psi}^{k-1} + \eta\boldsymbol{\zeta}_d^k,
$$

*as well as the properties*

$$\left\|\hat{\boldsymbol{D}}^k\right\| \le L_2(\boldsymbol{w}_0)B \text{ for all } 0 \le k < \mathcal{K}_1,$$

$$\left\|\boldsymbol{\zeta}_d^k\right\| \le 2L_1(\boldsymbol{w}_0)\left\|\boldsymbol{\psi}^{k-1}\right\| \text{ for all } 1 \le k \le \mathcal{K}_1.$$

**Proof.** We prove all the desired parts of Lemma E.15.

- The fact that $\boldsymbol{\psi}^0 = \boldsymbol{e}_1$ follows immediately, because $\boldsymbol{z}^0 = q\boldsymbol{e}_1$. For the general recursion for $\boldsymbol{\psi}^k$, consider any $k \le \mathcal{K}_1$. First note that by the recursion for the $\boldsymbol{z}^k$ for $k \le \mathcal{K}_1$ in Lemma E.14, we have

$$\begin{aligned}
\boldsymbol{\psi}^k &= q^{-1}(1 + \eta\delta_m)^{-k}\boldsymbol{z}^k \\
&= \frac{\boldsymbol{I} - \eta\boldsymbol{H}}{1 + \eta\delta_m}q^{-1}(1 + \eta\delta_m)^{-(k-1)}\boldsymbol{z}^{k-1} \\
&\quad + \eta\frac{\boldsymbol{D}^{k-1}}{1 + \eta\delta_m}q^{-1}(1 + \eta\delta_m)^{-(k-1)}\boldsymbol{z}^{k-1} + \eta q^{-1}(1 + \eta\delta_m)^{-k}\boldsymbol{\xi}_d^k \\
&= \frac{\boldsymbol{I} - \eta\boldsymbol{H}}{1 + \eta\delta_m}\boldsymbol{\psi}^{k-1} + \eta\hat{\boldsymbol{D}}^{k-1}\boldsymbol{\psi}^{k-1} + \eta\boldsymbol{\zeta}_d^k.
\end{aligned}$$

- Consider any $k \le \mathcal{K}_1$. For the requisite properties of $\hat{\boldsymbol{D}}^k$ for $k < \mathcal{K}_1$, the upper bound on the norm of $\hat{\boldsymbol{D}}^k$ follows immediately from Lemma E.14.

  Next from the definition of $\boldsymbol{\zeta}_d^k$ and Lemma E.14, for $k \le \mathcal{K}_1$ we have that

$$\begin{aligned}
\left\|\boldsymbol{\zeta}_d^k\right\| &\le q^{-1}(1 + \eta\delta_m)^{-k}\left\|\boldsymbol{\xi}_d^k\right\| \\
&\le 2L_1(\boldsymbol{w}_0)q^{-1}\frac{(1 + \eta\delta_m)^{-(k-1)}}{1 + \eta\delta_m}\left\|\boldsymbol{z}^{k-1}\right\| \\
&\le 2L_1(\boldsymbol{w}_0)\left\|\boldsymbol{\psi}^{k-1}\right\|.
\end{aligned}$$

This proves Lemma E.15. $\qquad\square$

**Lemma E.16** (Equivalent of the rest of Lemma 14, Fang et al. (2019))**.** *With the step size $\eta$ from (53), there exists an event $\mathcal{H}_o$ (namely, from (66)) with probability at least 0.9, such that for all $k \le \min(\mathcal{K}_1 - 1, K_0)$ we have*

$$\left\|\boldsymbol{\psi}^k\right\|^2 \le 4, \tag{63}$$

*and*

$$\boldsymbol{e}_1^\top\boldsymbol{\psi}^k > \frac{1}{2}. \tag{64}$$

**Proof.** Define

$$\hat{\boldsymbol{\psi}}^{k-1} = \frac{\boldsymbol{I} - \eta\boldsymbol{H}}{1 + \eta\delta_m}\boldsymbol{\psi}^{k-1}.$$

Recall that $\boldsymbol{H} = \nabla^2 F(\boldsymbol{x}^0)$ and $\boldsymbol{x}^0$ is in the $F(\boldsymbol{w}_0)$-sublevel set $\mathcal{L}_{F,F(\boldsymbol{w}_0)}$. Therefore, from Assumption 1.1, $\|\boldsymbol{H}\| \le L_1(\boldsymbol{w}_0)$. By definition of $\delta_m$, it follows that

$$-\delta_m\boldsymbol{I} \preceq \boldsymbol{H} \preceq L_1(\boldsymbol{w}_0)\boldsymbol{I}.$$

Since $\eta L_1(\boldsymbol{w}_0) \le 1$, it follows that the matrix $\boldsymbol{I} - \eta\boldsymbol{H}$ is symmetric and has all eigenvalues in $[0, 1 + \eta\delta_m]$. This implies

$$\left\|\hat{\boldsymbol{\psi}}^{k-1}\right\| \le \left\|\boldsymbol{\psi}^{k-1}\right\|. \tag{65}$$

Note that $\hat{\boldsymbol{\psi}}^{k-1}$ and $\boldsymbol{\psi}^{k-1}$ are measurable on $\mathfrak{F}^{k-1}$. This combined with Lemma E.14 and Lemma E.15 implies that for all $1 \le k \le \mathcal{K}_1$,

$$\mathbb{E}\left[(\hat{\boldsymbol{\psi}}^{k-1})^\top\boldsymbol{\zeta}_d^k \cdot 1_{\|\boldsymbol{\psi}^{k-1}\|\le2}|\mathfrak{F}^{k-1}\right] = 1_{\|\boldsymbol{\psi}^{k-1}\|\le2} \cdot \mathbb{E}\left[(\hat{\boldsymbol{\psi}}^{k-1})^\top\boldsymbol{\zeta}_d^k|\mathfrak{F}^{k-1}\right] = 0,$$

and
$$|(\hat{\boldsymbol{\psi}}^{k-1})^\top \boldsymbol{\zeta}_d^k \cdot 1_{\|\boldsymbol{\psi}^{k-1}\|\leq 2}|^2 \leq 1_{\|\boldsymbol{\psi}^{k-1}\|\leq 2} \cdot 4L_1^2(\boldsymbol{w}_0)\|\boldsymbol{\psi}^{k-1}\|^4 \leq (8L_1(\boldsymbol{w}_0))^2.$$
Now define the following real-valued stochastic process:
$$Y_k = (\hat{\boldsymbol{\psi}}^{k-1})^\top \boldsymbol{\zeta}_d^k 1_{\|\boldsymbol{\psi}^{k-1}\|\leq 2} 1_{k-1<\mathcal{K}_1} = \begin{cases} (\hat{\boldsymbol{\psi}}^{k-1})^\top \boldsymbol{\zeta}_d^k \cdot 1_{\|\boldsymbol{\psi}^{k-1}\|\leq 2} & : k \leq \mathcal{K}_1 \\ 0 & : k > \mathcal{K}_1. \end{cases}$$

Note $Y_k$ is $\mathfrak{F}_k$-measurable, and that $(\hat{\boldsymbol{\psi}}^{k-1})^\top, 1_{\|\boldsymbol{\psi}^{k-1}\|\leq 2}, 1_{k-1<\mathcal{K}_1} \equiv 1_{k\leq\mathcal{K}_1}$ are all $\mathfrak{F}_{k-1}$-measurable. Thus, by Lemma E.14 and the definition of $\boldsymbol{\zeta}_d^k$ from Lemma E.15,
$$\mathbb{E}[Y_k|\mathfrak{F}_{k-1}] = 0.$$
Furthermore combining the above justification with the trivial case $k > \mathcal{K}_1$, we obtain
$$|Y_k| \leq 8L_1(\boldsymbol{w}_0).$$
By the (standard) Azuma's Inequality, with probability $1 - 0.1/(2K_0)$, for any given $l, 1 \leq l \leq K_0$:
$$\left|\sum_{k=1}^l Y_k\right| \leq 8L_1(\boldsymbol{w}_0)\sqrt{2l\log(40K_0)} \leq 8L_1(\boldsymbol{w}_0)\sqrt{2K_0\log(40K_0)} \leq \frac{1}{\eta},$$
where the last inequality follows from the given choice of parameters.

Analogously, by Lemma E.14 and Lemma E.15, we also have for $1 \leq k \leq \mathcal{K}_1$:
$$\mathbb{E}[\boldsymbol{e}_1^\top \boldsymbol{\zeta}_d^k \cdot 1_{\|\boldsymbol{\psi}^{k-1}\|\leq 2}|\mathfrak{F}^{k-1}] = 0, |\boldsymbol{e}_1^\top \boldsymbol{\zeta}_d^k \cdot 1_{\|\boldsymbol{\psi}^{k-1}\|\leq 2}| \leq 4L_1(\boldsymbol{w}_0).$$
Define
$$Y_k' := \boldsymbol{e}_1^\top \boldsymbol{\zeta}_d^k \cdot 1_{\|\boldsymbol{\psi}^{k-1}\|\leq 2} 1_{k\leq\mathcal{K}_1}.$$
The (standard) Azuma's Inequality now implies that with probability at least $1 - 0.1/(2K_0)$, for any given $l, 1 \leq l \leq K_0$:
$$\left|\sum_{k=1}^l Y_k'\right| \leq 4L_1(\boldsymbol{w}_0)\sqrt{2l\log(40K_0)} \leq \frac{1}{4\eta}.$$
By the Union Bound, there exists an event $\mathcal{H}_o$ happening with probability at least $0.9$ such that the following inequalities hold for each $l = 1, 2, \ldots, K_0$:
$$\left|\sum_{k=1}^l Y_k\right| \leq \frac{1}{\eta}, \left|\sum_{k=1}^l Y_k'\right| \leq \frac{1}{4\eta}. \tag{66}$$
In particular under the event $\mathcal{H}_o$, for any $l \leq \min(\mathcal{K}_1 - 1, K_0)$, using the definitions of $Y_k, Y_k'$ we obtain
$$\left|\sum_{k=1}^l \hat{\boldsymbol{\psi}}_{k-1}^\top \boldsymbol{\zeta}_d^k \cdot 1_{\|\boldsymbol{\psi}^{k-1}\|\leq 2}\right| \leq \frac{1}{\eta}, \left|\sum_{k=1}^l \boldsymbol{e}_1^\top \boldsymbol{\zeta}_d^k \cdot 1_{\|\boldsymbol{\psi}_{k-1}\|\leq 2}\right| \leq \frac{1}{4\eta}. \tag{67}$$
Now from Lemma E.15, it follows for all $k \leq \mathcal{K}_1$ that
$$\begin{aligned}\|\boldsymbol{\psi}^k\|^2 &= \left\|\frac{\boldsymbol{I} - \eta\boldsymbol{H}}{1 + \eta\delta_m}\boldsymbol{\psi}^{k-1} + \eta\hat{\boldsymbol{D}}^{k-1}\boldsymbol{\psi}^{k-1} + \eta\boldsymbol{\zeta}_d^k\right\|^2 \\ &= \left\|\hat{\boldsymbol{\psi}}^{k-1}\right\|^2 + 2\eta(\hat{\boldsymbol{\psi}}_{k-1})^\top \hat{\boldsymbol{D}}_{k-1}\boldsymbol{\psi}_{k-1} + \eta^2\left\|\hat{\boldsymbol{D}}_{k-1}\boldsymbol{\psi}^{k-1} + \boldsymbol{\zeta}_d^k\right\|^2 + 2\eta(\hat{\boldsymbol{\psi}}^{k-1})^\top \boldsymbol{\zeta}_d^k \\ &= \|\boldsymbol{\psi}^{k-1}\|^2 + Q_{1,k} + Q_{2,k} + Q_{3,k}\end{aligned} \tag{68}$$
where we define
$$Q_{1,k} := 2\eta(\hat{\boldsymbol{\psi}}^{k-1})^\top \hat{\boldsymbol{D}}^{k-1}\boldsymbol{\psi}^{k-1}, Q_{2,k} := \eta^2\left\|\hat{\boldsymbol{D}}_{k-1}\boldsymbol{\psi}_{k-1} + \boldsymbol{\zeta}_d^k\right\|^2, Q_{3,k} := 2\eta(\hat{\boldsymbol{\psi}}^{k-1})^\top \boldsymbol{\zeta}_d^k.$$
For $k \leq \mathcal{K}_1$, we have $k - 1 < \mathcal{K}_1$. Thus by Lemma E.15 and (65), we have
$$Q_{1,k} \leq 2\eta L_2(\boldsymbol{w}_0)B\|\boldsymbol{\psi}^{k-1}\|^2, \tag{69}$$
and
$$Q_{2,k} \leq 2\eta^2\left\|\hat{\boldsymbol{D}}^{k-1}\boldsymbol{\psi}^{k-1}\right\|^2 + 2\eta^2\|\boldsymbol{\zeta}_d^k\|^2$$

$$\leq 2\eta^2 \cdot L_2(\boldsymbol{w}_0)^2 B^2 \|\boldsymbol{\psi}^{k-1}\|^2 + 8\eta^2 L_1(\boldsymbol{w}_0)^2 \|\boldsymbol{\psi}^{k-1}\|^2$$

$$\leq 16\eta^2 L_1(\boldsymbol{w}_0)^2 \|\boldsymbol{\psi}^{k-1}\|^2. \tag{70}$$

The last inequality above follows as per Remark 12.

Now we complete the proof. Under the event $\mathcal{H}_o$ from (66), we prove (63) by induction on $k$ (recall our condition for $k$ for Lemma E.16 is that $0 \leq k \leq \min(\mathcal{K}_1 - 1, K_0)$).

When $k = 0$, by Lemma E.15, $\boldsymbol{\psi}^0 = \boldsymbol{e}_1$, so $\|\boldsymbol{\psi}^0\| = \|\boldsymbol{e}_1\| = 1 \leq 2$ and $\boldsymbol{e}_1^\top \boldsymbol{\psi}^0 = \|\boldsymbol{e}_1\|^2 = 1$ (recall $\boldsymbol{e}_1$ is a *unit* eigenvector), proving the base case.

Now for the inductive step, consider some $k \leq \min(\mathcal{K}_1 - 1, K_0)$. Suppose $\|\boldsymbol{\psi}^l\| \leq 2$ holds for all $l, 0 \leq l \leq k - 1$. Then because $k < \mathcal{K}_1$, upon applying the above bounds (68), (69), (70) we have:

$$\|\boldsymbol{\psi}^k\|^2 \leq \|\boldsymbol{\psi}^0\|^2 + \sum_{s=1}^{k} Q_{1,s} + \sum_{s=1}^{k} Q_{2,s} + \sum_{s=1}^{k} Q_{3,s}$$

$$\leq 1 + 2\eta \sum_{s=1}^{k} L_2(\boldsymbol{w}_0) B \|\boldsymbol{\psi}^{s-1}\|^2 + 16\eta^2 L_1(\boldsymbol{w}_0)^2 \sum_{s=1}^{k} \|\boldsymbol{\psi}^s\|^2 + 2\eta \sum_{s=1}^{k} (\hat{\boldsymbol{\psi}}^{s-1})^\top \boldsymbol{\zeta}_d^s$$

$$\leq 1 + 2 L_2(\boldsymbol{w}_0) B \cdot 4 \cdot \eta k + 16\eta^2 \cdot L_1(\boldsymbol{w}_0)^2 \cdot 4 \cdot k + 2\eta \sum_{s=1}^{k} (\hat{\boldsymbol{\psi}}^{s-1})^\top \boldsymbol{\zeta}_d^s \cdot 1_{\|\boldsymbol{\psi}^{s-1}\| \leq 2}$$

$$\leq 1 + 16 L_2(\boldsymbol{w}_0) B \cdot \eta K_0 + 2\eta \sum_{s=1}^{k} \hat{\boldsymbol{\psi}}_{s-1}^\top \boldsymbol{\zeta}_d^s \cdot 1_{\|\boldsymbol{\psi}_{s-1}\| \leq 2} \leq 1 + 1 + 2\eta \cdot \frac{1}{\eta} = 4.$$

To upper bound the above, we used our choice of step size $\eta \leq \frac{L_2(\boldsymbol{w}_0) B}{8 L_1(\boldsymbol{w}_0)^2}$ and $B \leq \frac{1}{L_1(\boldsymbol{w}_0)}$ as per Remark 12, our above upper bounds on $Q_{1,s}, Q_{2,s}$, and that the event $\mathcal{H}_o$ implies (67).

This completes the induction and proves (63).

With (63), we prove (64). Namely note for $k \leq \min(\mathcal{K}_1 - 1, K_0)$, summing and telescoping the recursion for $\boldsymbol{\psi}^k$ from Lemma E.15, we have:

$$\boldsymbol{e}_1^\top \boldsymbol{\psi}_k = \boldsymbol{e}_1^\top \boldsymbol{\psi}_0 + \sum_{s=0}^{k-1} \eta \boldsymbol{e}_1^\top \hat{\boldsymbol{D}}_s \boldsymbol{\psi}^s + \sum_{s=0}^{k-1} \eta \boldsymbol{e}_1^\top \boldsymbol{\zeta}_d^s$$

$$\geq 1 - \eta \sum_{s=0}^{k-1} 2 L_2(\boldsymbol{w}_0) B \|\boldsymbol{\psi}^s\| + \eta \sum_{s=0}^{k-1} \boldsymbol{e}_1^\top \boldsymbol{\zeta}_d^s \cdot 1_{\|\boldsymbol{\psi}^{s-1}\| \leq 2}$$

$$\geq 1 - \eta \cdot K_0 \cdot 2 L_2(\boldsymbol{w}_0) B \cdot 2 + \eta \sum_{s=0}^{k-1} \boldsymbol{e}_1^\top \boldsymbol{\zeta}_d^s \cdot 1_{\|\boldsymbol{\psi}^{s-1}\| \leq 2} \geq 1 - \frac{1}{8} - \frac{2}{8} \geq \frac{1}{2}.$$

Here to lower bound the final sum, we used that $\boldsymbol{\psi}_0 = \boldsymbol{e}_1$ and the upper bound on $\|\hat{\boldsymbol{D}}_s\|$ from Lemma E.15, the fact that we have already established $\|\boldsymbol{\psi}^s\| \leq 2$ for all $s < k$ as we showed (63), and that the event $\mathcal{H}_o$ implies (67).

This proves all parts of Lemma E.16. $\qquad\square$

**Finish.** Now we prove Lemma E.13 via the same high-level strategy as the proof of Lemma 8, Fang et al. (2019). Note on the event $\{\mathcal{K}_1 > K_o\}$, we have

$$\boldsymbol{z}^{K_o} = \boldsymbol{w}^{K_o}(\boldsymbol{u} + q\boldsymbol{e}_1) - \boldsymbol{w}^{K_o}(\boldsymbol{u}) = (\boldsymbol{w}^{K_o}(\boldsymbol{u} + q\boldsymbol{e}_1) - \boldsymbol{x}^0)) - (\boldsymbol{w}^{K_o}(\boldsymbol{u}) - \boldsymbol{x}^0).$$

Thus by definition of $\mathcal{K}_1$, the event $\{\mathcal{K}_1 > K_o\}$ implies that

$$\|\boldsymbol{z}^{K_o}\| \leq \|\boldsymbol{w}^{K_o}(\boldsymbol{u} + q\boldsymbol{e}_1) - \boldsymbol{x}^0\| + \|\boldsymbol{w}^{K_o}(\boldsymbol{u}) - \boldsymbol{x}^0\| \leq 2B.$$

That is,

$$\{\mathcal{K}_1 > K_o\} \subseteq \{\|\boldsymbol{z}^{K_o}\| \leq 2B\}.$$

However, consider the event $\mathcal{H}_o$, (66) from Lemma E.16. On the event $\{\mathcal{K}_1 > K_o\} \cap \mathcal{H}_o$, we have $K_o \leq \min(\mathcal{K}_1 - 1, K_0)$, and so by Lemma E.16, we have

$$\boldsymbol{e}_1^\top \boldsymbol{\psi}^{K_o} > \frac{1}{2}.$$

Thus by definition of $\boldsymbol{\psi}^k$ and recalling $\delta_m \geq \delta_2 > 0$, on the event $\{\mathcal{K}_1 > K_o\} \cap \mathcal{H}_o$ we have

$$\left\|\boldsymbol{z}^{K_o}\right\| = q(1 + \eta\delta_m)^{K_o}\left\|\boldsymbol{\psi}^{K_o}\right\| \geq q_0(1 + \eta\delta_2)^{K_o}\left|\boldsymbol{e}_1^\top \boldsymbol{\psi}^{K_o}\right| > q_0 \cdot \frac{6B}{q_0} \cdot \frac{1}{2} = 3B,$$

where the last inequality uses (54). This means that

$$\{\mathcal{K}_1 > K_o\} \cap \mathcal{H}_o \subseteq \{\left\|\boldsymbol{z}^{K_o}\right\| \geq 3B\}.$$

Putting our work together, we see that

$$\{\mathcal{K}_1 > K_o\} \cap \mathcal{H}_o \subseteq \{\left\|\boldsymbol{z}^{K_o}\right\| \geq 3B\} \cap \{\left\|\boldsymbol{z}^{K_o}\right\| \leq 2B\} = \varnothing.$$

Therefore

$$\{\mathcal{K}_1 > K_o\} \subseteq \mathcal{H}_o^c \implies \mathbb{P}(\mathcal{K}_1 > K_o) \leq \mathbb{P}(\mathcal{H}_0^c) \leq 0.1.$$

Recalling the definition of $\mathcal{K}_1$, we conclude Lemma E.13. $\qquad\square$

**Remark 21.** Note we only have $\boldsymbol{e}_1^T \boldsymbol{\psi}^k > \frac{1}{2}$ for $k < \mathcal{K}_1$ due to the lack of global Lipschitz bounds on the graedient and Hessian of $F$, unlike in the proof of Lemma 8, Fang et al. (2019).

### E.5 Faster Descent

**Setup:** As in Subsection E.4, let $\mathcal{K}_0$ denote the escape time of $\mathbb{B}(\boldsymbol{x}^0, B)$ for while loop of Algorithm 2 when the while loop begins at $\boldsymbol{x}^0$. In this section, we aim to prove Lemma E.2.

As in Subsection E.4, the difference between Lemma E.2 and Proposition 9 of Fang et al. (2019) is that *this result only holds at points in the $F(\boldsymbol{w}_0)$-sublevel set $\mathcal{L}_{F,F(\boldsymbol{w}_0)}$*. For the rest of this section, we work under the assumptions of Lemma E.2; thus for the rest of this section, $\boldsymbol{x}^0$ is in the $F(\boldsymbol{w}_0)$-sublevel set $\mathcal{L}_{F,F(\boldsymbol{w}_0)}$.

The idea here is similar to that of Subsection E.4. At a high level, we have the requisite control over the gradient and Hessian since the iterates we consider are in a neighborhood of a point $\boldsymbol{x}^0 \in \mathcal{L}_{F,F(\boldsymbol{w}_0)}$. As in the previous part and as in Fang et al. (2019), we let

$$\boldsymbol{H} := \nabla^2 F(\boldsymbol{x}^0),$$

and let

$$\boldsymbol{\xi}^{k+1} := \nabla\tilde{f}(\boldsymbol{x}^k; \boldsymbol{\zeta}_{k+1}) - \nabla F(\boldsymbol{x}^k), \quad k \geq 0. \tag{71}$$

Note as $\boldsymbol{\Lambda}^{k+1}$ has mean 0 and as the stochastic gradient oracle is unbiased, we have that for all $k \geq 0$,

$$\mathbb{E}\left[\boldsymbol{\xi}^{k+1} | \mathfrak{F}^k\right] = 0.$$

Let $\mathcal{S}$ be the subspace spanned by all eigenvectors of $\nabla^2 F(\boldsymbol{x}^0)$ whose eigenvalue is greater than $0$, and $\mathcal{S}^\perp$ denotes the complement space. Also, let $\boldsymbol{\mathcal{P}}_{\mathcal{S}} \in \mathbb{R}^{d \times d}$ and $\boldsymbol{\mathcal{P}}_{\mathcal{S}^\perp} \in \mathbb{R}^{d \times d}$ denote the projection matrices onto the spaces $\mathcal{S}$ and $\mathcal{S}^\perp$, respectively. Let $\boldsymbol{u}^k = \boldsymbol{\mathcal{P}}_{\mathcal{S}}(\boldsymbol{x}^k - \boldsymbol{x}^0)$, and $\boldsymbol{v}^k = \boldsymbol{\mathcal{P}}_{\mathcal{S}^\perp}(\boldsymbol{x}^k - \boldsymbol{x}^0)$. We can decompose the update equation of SGD as:

$$\boldsymbol{u}^{k+1} = \boldsymbol{u}^k - \eta\boldsymbol{\mathcal{P}}_{\mathcal{S}}\nabla F(\boldsymbol{x}^k) - \eta\boldsymbol{\mathcal{P}}_{\mathcal{S}}\boldsymbol{\xi}^{k+1},$$

$$\boldsymbol{v}^{k+1} = \boldsymbol{v}^k - \eta\boldsymbol{\mathcal{P}}_{\mathcal{S}^\perp}\nabla F(\boldsymbol{x}^k) - \eta\boldsymbol{\mathcal{P}}_{\mathcal{S}^\perp}\boldsymbol{\xi}^{k+1},$$

for $k \geq 0$. Clearly $\boldsymbol{u}^0 = \boldsymbol{0}$, $\boldsymbol{v}^0 = \boldsymbol{0}$.

Now decompose $\boldsymbol{H} = \boldsymbol{U}\boldsymbol{\Lambda}\boldsymbol{U}^T$ by the Spectral Theorem where $\boldsymbol{U} \in \mathbb{R}^{d \times d}$ is unitary and $\boldsymbol{\Lambda} \in \mathbb{R}^{d \times d}$ is diagonal. Let $\boldsymbol{\Lambda}_{>0}$ denote the diagonal matrix with diagonal entries equal to the positive (diagonal) entries of $\boldsymbol{\Lambda}$. Let $\boldsymbol{\Lambda}_{\leq 0}$ denote the diagonal matrix with diagonal entries equal to the zero or negative (diagonal) entries of $\boldsymbol{\Lambda}$. Now define

$$\boldsymbol{H}_{\mathcal{S}} := \boldsymbol{U}\boldsymbol{\Lambda}_{>0}\boldsymbol{U}^T, \boldsymbol{H}_{\mathcal{S}^\perp} := \boldsymbol{U}\boldsymbol{\Lambda}_{\leq 0}\boldsymbol{U}^T.$$

Thus $\boldsymbol{H}_{\mathcal{S}}$ has range in $\mathcal{S}$, and $\boldsymbol{H}_{\mathcal{S}^\perp}$ has range in $\mathcal{S}^\perp$. Note $\boldsymbol{H}_{\mathcal{S}}, \boldsymbol{H}_{\mathcal{S}^\perp}$ are both symmetric.

From here, define the following quadratic approximations:

$$G_{\mathcal{S}}(\boldsymbol{u}) := \left[\boldsymbol{P}_{\mathcal{S}}\nabla F(\boldsymbol{x}^0)\right]^\top \boldsymbol{u} + \frac{1}{2}\boldsymbol{u}^\top \boldsymbol{H}_{\mathcal{S}}\boldsymbol{u}, \ G_{\mathcal{S}^\perp}(\boldsymbol{v}) := \left[\boldsymbol{P}_{\mathcal{S}^\perp}\nabla F(\boldsymbol{x}^0)\right]^\top \boldsymbol{v} + \frac{1}{2}\boldsymbol{v}^\top \boldsymbol{H}_{\mathcal{S}^\perp}\boldsymbol{v}.$$

Now define the quadratic approximation

$$G(\boldsymbol{x}) = G_{\mathcal{S}}(\boldsymbol{u}) + G_{\mathcal{S}^\perp}(\boldsymbol{v}) \text{ where } \boldsymbol{u} = \boldsymbol{P}_{\mathcal{S}}(\boldsymbol{x} - \boldsymbol{x}^0), \boldsymbol{v} = \boldsymbol{P}_{\mathcal{S}^\perp}(\boldsymbol{x} - \boldsymbol{x}^0).$$

It is easy to see that

$$G(\boldsymbol{x}) = \left[\nabla F(\boldsymbol{x}^0)\right]^\top (\boldsymbol{x} - \boldsymbol{x}^0) + \frac{1}{2}(\boldsymbol{x} - \boldsymbol{x}^0)^\top \boldsymbol{H}(\boldsymbol{x} - \boldsymbol{x}^0).$$

For convenience, let

$$\nabla_{\boldsymbol{u}} F(\boldsymbol{x}^k) = \boldsymbol{P}_{\mathcal{S}}\nabla F(\boldsymbol{x}^k), \nabla_{\boldsymbol{v}} F(\boldsymbol{x}^k) = \boldsymbol{P}_{\mathcal{S}^\perp}\nabla F(\boldsymbol{x}^k).$$

Similarly, let

$$\boldsymbol{\xi}_{\boldsymbol{u}}^k = \boldsymbol{P}_{\mathcal{S}}\boldsymbol{\xi}^k, \boldsymbol{\xi}_{\boldsymbol{v}}^k = \boldsymbol{P}_{\mathcal{S}^\perp}\boldsymbol{\xi}^k.$$

Also denote the stopping time

$$\mathcal{K} = \mathcal{K}_0 \wedge K_0.$$

Due to its 'local' nature around the $\boldsymbol{x}^0$ in the $F(\boldsymbol{w}_0)$-sublevel set, we still have the following result from Fang et al. (2019):

**Lemma E.17** (Equivalent of Lemma 15, Fang et al. (2019)). *Consider any* $\boldsymbol{u} \in \mathcal{L}_{F,F(\boldsymbol{w}_0)}$, *and consider any* $\boldsymbol{x} \in \mathbb{B}(\boldsymbol{u}, B)$. *Then we have*

$$\|\nabla F(\boldsymbol{x}) - \nabla G(\boldsymbol{x})\| \le \frac{L_2(\boldsymbol{w}_0)B^2}{2}.$$

*Furthermore, for any symmetric matrix* $\boldsymbol{A}$, *with* $0 < a \le \frac{1}{\|\boldsymbol{A}\|_2}$, *for any* $i = 0, 1, \dots$, *and* $j = 0, 1, \dots$, *we have*

$$\left\|(\boldsymbol{I} - a\boldsymbol{A})^i \boldsymbol{A}(\boldsymbol{I} - a\boldsymbol{A})^j\right\|_2 \le \frac{1}{a(i + j + 1)}.$$

**Proof.** Notice that for all $0 \le \theta \le 1$, $\theta\boldsymbol{x} + (1 - \theta)\boldsymbol{u} \in \mathbb{B}(\boldsymbol{u}, B)$. Thus as $\boldsymbol{u} \in \mathcal{L}_{F,F(\boldsymbol{w}_0)}$, by Lemma E.8, we have

$$\left\|\nabla^2 F(\theta\boldsymbol{x} + (1 - \theta)\boldsymbol{u}) - \nabla^2 F(\boldsymbol{u})\right\| \le L_2(\boldsymbol{w}_0) \cdot \theta\|\boldsymbol{x} - \boldsymbol{u}\| \text{ for all } 0 \le \theta \le 1.$$

Thus we have

$$\|\nabla F(\boldsymbol{x}) - \nabla G(\boldsymbol{x})\| = \left\|\nabla F(\boldsymbol{x}) - \nabla F(\boldsymbol{x}^0) - \nabla^2 F(\boldsymbol{u})(\boldsymbol{x} - \boldsymbol{u})\right\|$$

$$= \left\|\left\{\int_0^1 \left(\nabla^2 F(\boldsymbol{x}^0 + \theta(\boldsymbol{x} - \boldsymbol{u})) - \nabla^2 F(\boldsymbol{u})\right)\mathrm{d}\theta\right\}(\boldsymbol{x} - \boldsymbol{u})\right\|$$

$$\le \left\|\int_0^1 \{L_2(\boldsymbol{w}_0) \cdot \theta\|\boldsymbol{x} - \boldsymbol{u}\|\}\mathrm{d}\theta\right\| \cdot \|\boldsymbol{x} - \boldsymbol{u}\|$$

$$\le \frac{L_2(\boldsymbol{w}_0)B^2}{2}.$$

The second part of the Lemma follows from the exact same proof of Lemma D.5 in Section D. It is also proved in the proofs of Lemma 15, Fang et al. (2019), and in the proof of Lemma 16 of Jin et al. (2017). For more detail, let the eigenvalues of $\boldsymbol{A}$ be $\{\lambda_k\}$. Thus for any $i, j \ge 0$, the eigenvalues of $(\boldsymbol{I} - a\boldsymbol{A})^i \boldsymbol{A}(\boldsymbol{I} - a\boldsymbol{A})^j$ are $\{\lambda_k(1 - a\lambda_k)^{i+j}\}$. We now detail a calculation from Jin et al. (2017). Letting $g_t(\lambda) := \lambda(1 - a\lambda)^t$ and setting its derivative to zero yields

$$\nabla g_t(\lambda) = (1 - a\lambda)^t - ta\lambda(1 - a\lambda)^{t-1} = 0.$$

It is easy to check that $\lambda_t^\star = \frac{1}{(1+t)a}$ is the unique maximizer, and $g_t(\lambda)$ is monotonically increasing in $(-\infty, \lambda_t^\star]$.

This gives:

$$\left\|(\boldsymbol{I} - a\boldsymbol{A})^i \boldsymbol{A}(\boldsymbol{I} - a\boldsymbol{A})^j\right\| = \max_k \lambda_i(1 - a\lambda_k)^{i+j} \le \hat{\lambda}(1 - a\hat{\lambda})^{i+j} \le \frac{1}{(1 + i + j)a},$$

where $\hat{\lambda} = \min\{\ell, \lambda_{i+j}^\star\}$. $\qquad\square$

**Lemma E.18.** *For any $k \leq \mathcal{K}_0$, we have*

$$\left\|\boldsymbol{\xi}^k\right\| \leq \sigma_1(\boldsymbol{w}_0).$$

**Proof.** Note for $k \leq \mathcal{K}_0$, we have $k - 1 < \mathcal{K}_0$ and so $\boldsymbol{x}^{k-1} \in \mathbb{B}(\boldsymbol{x}^0, B)$. Recall furthermore that $\boldsymbol{x}^0 \in \mathcal{L}_{F, F(\boldsymbol{w}_0)}$. Thus, by Lemma E.5 and Lemma E.3,

$$\left\|\boldsymbol{\xi}^k\right\| = \left\|\nabla \tilde{f}(\boldsymbol{x}^{k-1}; \boldsymbol{\zeta}_k) - \nabla F(\boldsymbol{x}^{k-1})\right\| \leq \sigma_1(\boldsymbol{w}_0),$$

as desired. $\qquad \square$

**Analyzing the Quadratic Approximation:** We now analyze the quadratic approximation $G(\boldsymbol{x})$ as done in Fang et al. (2019). First we analyze the part in $\mathcal{S}$:

**Lemma E.19** (Equivalent of Lemma 16, Fang et al. (2019)). *Set hyperparameters from (8). With probability at least $1 - p/4$, we have*

$$G_{\mathcal{S}}(\boldsymbol{u}^{\mathcal{K}}) - G_{\mathcal{S}}(\boldsymbol{u}^0)$$

$$\leq -\frac{25\eta}{32} \sum_{k=0}^{\mathcal{K}-1} \left\|\nabla G_{\mathcal{S}}(\boldsymbol{y}^k)\right\|^2 + 4\eta \sigma_1(\boldsymbol{w}_0)^2 (\log(K_0) + 3) \log\left(\frac{48K_0}{p}\right) + \eta L_2(\boldsymbol{w}_0)^2 B^4 K_0$$

$$= -\frac{25\eta}{32} \sum_{k=0}^{\mathcal{K}-1} \left\|\nabla G_{\mathcal{S}}(\boldsymbol{y}^k)\right\|^2 + \tilde{O}(\varepsilon^{1.5}).$$

**Proof.** We follow a similar strategy as before of combining the proof of Fang et al. (2019) with our self-bounding framework. To analyze $G_{\mathcal{S}}(\cdot)$ we first consider an auxiliary Gradient Descent trajectory, which performs the update:

$$\boldsymbol{y}^{k+1} = \boldsymbol{y}^k - \eta \nabla G_{\mathcal{S}}(\boldsymbol{y}^k), \quad k \geq 0,$$

and $\boldsymbol{y}^0 = \boldsymbol{u}^0$. $\boldsymbol{y}^k$ performs Gradient Descent on $G_{\mathcal{S}}(\cdot)$, which is deterministic given $\boldsymbol{x}^0$.

Noting $G_{\mathcal{S}}$ has Hessian $\boldsymbol{H}_{\mathcal{S}}$, and that $\boldsymbol{H}$ is the Hessian of $F$ at the point $\boldsymbol{x}^0 \in \mathcal{L}_{F, F(\boldsymbol{w}_0)}$, we obtain from Assumption 1.1 that

$$\|\boldsymbol{H}_{\mathcal{S}}\| \leq \|\boldsymbol{H}\| \leq L_1(\boldsymbol{w}_0).$$

Since the following only concern $G_{\mathcal{S}}$, then identically to the proof of Lemma 16, Fang et al. (2019), we obtain the following:

- By $L_1(\boldsymbol{w}_0)$-smoothness of $G_{\mathcal{S}}$ (recall $G_{\mathcal{S}}$ has Hessian $\boldsymbol{H}_S$), we obtain the so-called 'Descent Lemma':

$$G_{\mathcal{S}}(\boldsymbol{y}^{k+1}) \leq G_{\mathcal{S}}(\boldsymbol{y}^k) + \langle \nabla G_{\mathcal{S}}(\boldsymbol{y}^k), \boldsymbol{y}^{k+1} - \boldsymbol{y}^k \rangle + \frac{L_1(\boldsymbol{w}_0)}{2} \left\|\boldsymbol{y}^{k+1} - \boldsymbol{y}^k\right\|^2.$$

$$= G_{\mathcal{S}}(\boldsymbol{y}^k) - \eta\left(1 - \frac{L_1(\boldsymbol{w}_0)\eta}{2}\right)\left\|\nabla G_{\mathcal{S}}(\boldsymbol{y}^k)\right\|^2.$$

- Telescoping the above for $0 \leq k \leq \mathcal{K}-1$, and by our choice of $\eta$ which satisfies $\eta L_1(\boldsymbol{w}_0) \leq \frac{1}{16}$ as per Remark 12, we obtain

$$G_{\mathcal{S}}(\boldsymbol{y}^{\mathcal{K}}) \leq G_{\mathcal{S}}(\boldsymbol{y}^0) - \frac{31\eta}{32} \sum_{k=0}^{\mathcal{K}-1} \left\|\nabla G_{\mathcal{S}}(\boldsymbol{y}^k)\right\|^2. \tag{72}$$

To obtain Lemma E.19, we upper bound the difference between $\boldsymbol{u}^{\mathcal{K}}$ and $\boldsymbol{y}^{\mathcal{K}}$. For all $k \geq 0$, define

$$\boldsymbol{z}^k := \boldsymbol{u}^k - \boldsymbol{y}^k.$$

We aim to upper bound $\boldsymbol{z}^{\mathcal{K}}$ (in an appropriate sense) using the concentration argument of Fang et al. (2019):

**Lemma E.20** (Equivalent of Lemma 17, Fang et al. (2019)). *With probability at least $1 - p/6$, we have*

$$\|\boldsymbol{z}^k\| \le \frac{3B}{32} \approx \tilde{\Theta}(\varepsilon^{0.5}), \tag{73}$$

*and*

$$\boldsymbol{z}^{k\top}\boldsymbol{H}_\mathcal{S}\boldsymbol{z}^k \le 8\sigma_1(\boldsymbol{w}_0)^2\eta(\log(K_0)+1)\log\left(\frac{48K_0}{p}\right) + \eta L_2(\boldsymbol{w}_0)^2 B^4 K_0 \approx \tilde{\Theta}(\varepsilon^{0.5}). \tag{74}$$

*Here $\tilde{\Theta}(\cdot)$ hides $F(\boldsymbol{w}_0)$-dependence.*

**Proof of Lemma E.20.** Clearly $\boldsymbol{z}^0 = \boldsymbol{0}$. From the definitions of $\boldsymbol{u}^k, \boldsymbol{y}^k$, we have

$$\begin{aligned}
\boldsymbol{z}^{k+1} &= \boldsymbol{z}^k - \eta(\nabla G_\mathcal{S}(\boldsymbol{u}^k) - \nabla G_\mathcal{S}(\boldsymbol{y}^k)) - \eta(\nabla_{\boldsymbol{u}}F(\boldsymbol{x}^k) - \nabla G_\mathcal{S}(\boldsymbol{u}^k)) - \eta\boldsymbol{\xi}_{\boldsymbol{u}}^{k+1} \\
&= (\boldsymbol{I} - \eta\boldsymbol{H}_\mathcal{S})\boldsymbol{z}^k - \eta(\nabla_{\boldsymbol{u}}F(\boldsymbol{x}^k) - \nabla G_\mathcal{S}(\boldsymbol{u}^k)) - \eta\boldsymbol{\xi}_{\boldsymbol{u}}^{k+1}, \quad k \ge 0.
\end{aligned} \tag{75}$$

Unraveling the above recursion gives:

$$\boldsymbol{z}^k = -\sum_{j=1}^{k}\eta(\boldsymbol{I}-\eta\boldsymbol{H}_\mathcal{S})^{k-j}\boldsymbol{\xi}_{\boldsymbol{u}}^j - \eta\sum_{j=0}^{k-1}(\boldsymbol{I}-\eta\boldsymbol{H}_\mathcal{S})^{k-1-j}(\nabla_{\boldsymbol{u}}F(\boldsymbol{x}^j) - \nabla G_\mathcal{S}(\boldsymbol{u}^j)), \quad k \ge 0. \tag{76}$$

Setting $k = \mathcal{K}$, Triangle Inequality gives

$$\|\boldsymbol{z}^\mathcal{K}\| \le \left\|\sum_{j=1}^{\mathcal{K}}\eta(\boldsymbol{I}-\eta\boldsymbol{H}_\mathcal{S})^{\mathcal{K}-j}\boldsymbol{\xi}_{\boldsymbol{u}}^j\right\| + \left\|\eta\sum_{j=0}^{\mathcal{K}-1}(\boldsymbol{I}-\eta\boldsymbol{H}_\mathcal{S})^{\mathcal{K}-1-j}(\nabla_{\boldsymbol{u}}F(\boldsymbol{x}^j) - \nabla G_\mathcal{S}(\boldsymbol{u}^j))\right\|.$$

We separately bound these two terms:

- For the first term, for any fixed $l$ from 1 to $K_0$, and any $j$ from 1 to $\min(l, \mathcal{K}_0)$, we have

$$\mathbb{E}\big[\eta(\boldsymbol{I}-\eta\boldsymbol{H}_\mathcal{S})^{l-j}\boldsymbol{\xi}_{\boldsymbol{u}}^j|\mathfrak{F}^{j-1}\big] = 0, \left\|\eta(\boldsymbol{I}-\eta\boldsymbol{H}_\mathcal{S})^{l-j}\boldsymbol{\xi}_{\boldsymbol{u}}^j\right\| \le \eta\sigma_1(\boldsymbol{w}_0).$$

  The first equality uses $\left\|\boldsymbol{\xi}_{\boldsymbol{u}}^j\right\| = \left\|\boldsymbol{\mathcal{P}}_\mathcal{S}\boldsymbol{\xi}^j\right\|$ and that the stochastic gradient oracle is unbiased. The inequality uses that $\boldsymbol{\mathcal{P}}$ is a projection matrix, $\left\|\boldsymbol{\xi}_{\boldsymbol{u}}^j\right\| = \left\|\boldsymbol{\mathcal{P}}_\mathcal{S}\boldsymbol{\xi}^j\right\| \le \sigma_1(\boldsymbol{w}_0)$ which follows as $j \le \mathcal{K}_0$ and Lemma E.18, and $\left\|(\boldsymbol{I}-\eta\boldsymbol{H}_\mathcal{S})^{l-j}\right\| \le 1$ which follows as $l \ge j$ and $\boldsymbol{H}_\mathcal{S} \succeq 0$. (Note the importance that $j \le \mathcal{K}_0$, which gives us enough control over the noise term $\boldsymbol{\xi}_{\boldsymbol{u}}^j$.)

  Now to deal with the fact that the above control only applies for certain $j$, we define a stochastic process as follows, analogously to our proof of Lemma E.13. For all fixed $1 \le l \le K_0$, define a stochastic process $Y_{l,j}$ over all $1 \le j \le l$ by:

$$Y_{l,j} = \eta(\boldsymbol{I}-\eta\boldsymbol{H}_\mathcal{S})^{l-j}\boldsymbol{\xi}_{\boldsymbol{u}}^j 1_{j-1<\mathcal{K}} = \begin{cases} \eta(\boldsymbol{I}-\eta\boldsymbol{H}_\mathcal{S})^{l-j}\boldsymbol{\xi}_{\boldsymbol{u}}^j & : j \le \mathcal{K} \\ 0 & : j > \mathcal{K}. \end{cases}$$

  Recalling $\mathcal{K} = \mathcal{K}_0 \wedge K_0$, it's easy to check that for any *fixed* $l$, $Y_{l,j}$ is $\mathfrak{F}^j$-measurable. Furthermore, $\eta(\boldsymbol{I}-\eta\boldsymbol{H}_\mathcal{S})^{l-j}, 1_{j-1<\mathcal{K}}$ are both $\mathfrak{F}^{j-1}$-measurable. Thus combining with the earlier observations, we obtain that

$$\mathbb{E}\big[Y_{l,j}|\mathfrak{F}^{j-1}\big] = 0, \|Y_{l,j}\| \le \eta\sigma_1(\boldsymbol{w}_0).$$

  Thus, by the Vector-Martingale Concentration Inequality Theorem C.1, we have with probability $1 - p/(12K_0)$,

$$\left\|\sum_{j=1}^{l}Y_{l,j}\right\| \le 2\eta\sigma_1(\boldsymbol{w}_0)\sqrt{l\log\left(\frac{48K_0}{p}\right)} \le 2\eta\sigma_1(\boldsymbol{w}_0)\sqrt{K_0\log\left(\frac{48K_0}{p}\right)} \le \frac{B}{16}. \tag{77}$$

  The last inequality uses our choice of parameters.

  By a Union Bound, with probability at least $1 - p/12$, (77) holds for all $l$ from 1 to $K_0$. In particular, with probability at least $1 - p/12$ we have for $\mathcal{K}$ (recall $\mathcal{K} \le K_0$) that

$$\left\|\sum_{j=1}^{\mathcal{K}}\eta(\boldsymbol{I}-\eta\boldsymbol{H}_\mathcal{S})^{\mathcal{K}-j}\boldsymbol{\xi}_{\boldsymbol{u}}^j\right\| = \left\|\sum_{j=1}^{\mathcal{K}}Y_{\mathcal{K},j}\right\| \le \frac{B}{16},$$

  where we define $Y_{\mathcal{K},j}$ the obvious way. This holds because with probability at least $1 - p/12$, we have the bound (77) on $\left\|\sum_{j=1}^{l}Y_{l,j}\right\|$ irrespective of which value of $1 \le l \le K_0$ that $\mathcal{K}$ takes on. The first equality holds by our definition of $Y_{l,j}$ for $j \le l = \mathcal{K}$.

- For the second term, we have

$$\left\| \eta \sum_{j=0}^{\mathcal{K}-1} (\boldsymbol{I} - \eta \boldsymbol{H}_{\mathcal{S}})^{\mathcal{K}-1-j} (\nabla_{\boldsymbol{u}} F(\boldsymbol{x}^j) - \nabla G_{\mathcal{S}}(\boldsymbol{u}^j)) \right\| \le \eta \sum_{j=0}^{\mathcal{K}-1} \left\| \nabla_{\boldsymbol{u}} F(\boldsymbol{x}^j) - \nabla G_{\mathcal{S}}(\boldsymbol{u}^j) \right\|$$

$$\le \eta \sum_{j=0}^{\mathcal{K}-1} \left\| \nabla F(\boldsymbol{x}^j) - \nabla G(\boldsymbol{x}^j) \right\|$$

$$\le \frac{\eta L_2(\boldsymbol{w}_0) B^2 K_0}{2} \le \frac{B}{32}.$$

The first inequality uses the Triangle Inequality and that $\left\| (\boldsymbol{I} - \eta \boldsymbol{H}_{\mathcal{S}})^{\mathcal{K}-1-j} \right\|_2 \le 1$ for $j$ from 0 to $\mathcal{K} - 1$; this follows because $\|\boldsymbol{H}_{\mathcal{S}}\| \le L_1(\boldsymbol{w}_0)$ and as $\eta \le \frac{1}{L_1(\boldsymbol{w}_0)}$. The second inequality uses $\|\boldsymbol{\mathcal{P}}_{\mathcal{S}}(\nabla F(\boldsymbol{x}) - \nabla G(\boldsymbol{x}))\| \le \|\nabla F(\boldsymbol{x}) - \nabla G(\boldsymbol{x})\|$ because $\boldsymbol{\mathcal{P}}_{\mathcal{S}}$ is a projection matrix. The third inequality follows from Lemma E.17, and the fact that for all $j \le \mathcal{K} - 1$, $\boldsymbol{x}^j \in \mathbb{B}(\boldsymbol{x}^0, B)$. The last inequality uses the choice of parameters.

Combining the above gives (73), the first part of Lemma E.20.

Now prove the second part of Lemma E.20, namely (74). Using the fact that $(\boldsymbol{a} + \boldsymbol{b})^\top \boldsymbol{A}(\boldsymbol{a} + \boldsymbol{b}) \le 2\boldsymbol{a}^\top \boldsymbol{A}\boldsymbol{a} + 2\boldsymbol{b}^\top \boldsymbol{A}\boldsymbol{b}$ for any symmetric positive definite matrix $\boldsymbol{A}$ and the recursion (76) for $\boldsymbol{z}^k$, we have

$$\left( \boldsymbol{z}^{\mathcal{K}} \right)^\top \boldsymbol{H}_{\mathcal{S}} \boldsymbol{z}^{\mathcal{K}}$$

$$\le 2\eta^2 \left( \sum_{j=1}^{\mathcal{K}} (\boldsymbol{I} - \eta \boldsymbol{H}_{\mathcal{S}})^{\mathcal{K}-j-1} \right)^\top \boldsymbol{H}_{\mathcal{S}} \left( \sum_{j=1}^{\mathcal{K}} (\boldsymbol{I} - \eta \boldsymbol{H}_{\mathcal{S}})^{K-j} \boldsymbol{\xi}_u^j \right)$$

$$+ 2\eta^2 \left( \sum_{j=0}^{\mathcal{K}-1} (\boldsymbol{I} - \eta \boldsymbol{H}_{\mathcal{S}})^{\mathcal{K}-1-j} \left( \nabla_{\boldsymbol{u}} F(\boldsymbol{x}^j) - \nabla G_{\mathcal{S}}(\boldsymbol{u}^j) \right) \right)^\top \boldsymbol{H}_{\mathcal{S}} \left( \sum_{j=0}^{\mathcal{K}-1} (\boldsymbol{I} - \eta \boldsymbol{H}_{\mathcal{S}})^{\mathcal{K}-1-j} \left( \nabla_{\boldsymbol{u}} F(\boldsymbol{x}^j) - \nabla G_{\mathcal{S}}(\boldsymbol{u}^j) \right) \right)$$

$$= 2 \left\| \eta \sum_{j=1}^{\mathcal{K}} \boldsymbol{H}_{\mathcal{S}}^{1/2} (\boldsymbol{I} - \eta \boldsymbol{H}_{\mathcal{S}})^{\mathcal{K}-j} \boldsymbol{\xi}_{\boldsymbol{u}}^j \right\|^2$$

$$+ 2\eta^2 \sum_{j=0}^{\mathcal{K}-1} \sum_{l=0}^{\mathcal{K}-1} \left( \nabla_{\boldsymbol{u}} F(\boldsymbol{x}^j) - \nabla G_{\mathcal{S}}(\boldsymbol{u}^j) \right)^\top (\boldsymbol{I} - \eta \boldsymbol{H}_{\mathcal{S}})^{\mathcal{K}-1-j} \boldsymbol{H}_{\mathcal{S}} (\boldsymbol{I} - \eta \boldsymbol{H}_{\mathcal{S}})^{\mathcal{K}-1-l} \left( \nabla_{\boldsymbol{u}} F(\boldsymbol{x}^l) - \nabla G_{\mathcal{S}}(\boldsymbol{u}^l) \right)$$

$$\le 2 \left\| \eta \sum_{j=1}^{\mathcal{K}} \boldsymbol{H}_{\mathcal{S}}^{1/2} (\boldsymbol{I} - \eta \boldsymbol{H}_{\mathcal{S}})^{\mathcal{K}-j} \boldsymbol{\xi}_{\boldsymbol{u}}^j \right\|^2 + 2\eta^2 \frac{L_2(\boldsymbol{w}_0)^2 B^4}{4} \sum_{j=0}^{\mathcal{K}-1} \sum_{l=0}^{\mathcal{K}-1} \left\| (\boldsymbol{I} - \eta \boldsymbol{H}_{\mathcal{S}})^{\mathcal{K}-1-j} \boldsymbol{H}_{\mathcal{S}} (\boldsymbol{I} - \eta \boldsymbol{H}_{\mathcal{S}})^{\mathcal{K}-1-l} \right\|.$$

The last inequality follows by properties of projection matrices and by Lemma E.17, recalling that for $j \le \mathcal{K} - 1$, $\boldsymbol{x}^j \in \mathbb{B}(\boldsymbol{x}^0, B)$.

Now we bound each of these two terms separately:

- For the first term, for any fixed $l, 1 \le l \le K_0$, again we define a stochastic process for any $j, 1 \le j \le l$ by:

$$Y_{l,j} = \eta \left( \boldsymbol{H}_{\mathcal{S}}^{1/2} (\boldsymbol{I} - \eta \boldsymbol{H}_{\mathcal{S}})^{l-j} \boldsymbol{\xi}_{\boldsymbol{u}}^j \right) 1_{j-1<\mathcal{K}} = \begin{cases} \eta \left( \boldsymbol{H}_{\mathcal{S}}^{1/2} (\boldsymbol{I} - \eta \boldsymbol{H}_{\mathcal{S}})^{l-j} \boldsymbol{\xi}_{\boldsymbol{u}}^j \right) & : j \le \mathcal{K} \\ 0 & : j > \mathcal{K}. \end{cases}$$

Analogously to earlier, recalling $\mathcal{K} \le \mathcal{K}_0$, for *fixed* $l$, it is evident that $Y_{l,j}$ is $\mathfrak{F}^j$-measurable, $\eta \boldsymbol{H}_{\mathcal{S}}^{1/2} (\boldsymbol{I} - \eta \boldsymbol{H}_{\mathcal{S}})^{l-j} 1_{j-1<\mathcal{K}}$ is $\mathfrak{F}^{j-1}$-measurable, and thus

$$\mathbb{E}\left[ Y_{l,j} | \mathfrak{F}^{j-1} \right] = 0.$$

We furthermore have

$$\|Y_{l,j}\|^2 \le \frac{\eta \sigma_1(\boldsymbol{w}_0)^2}{1 + 2(l-j)},$$

which follows by noting for any $1 \le l \le K_0$ and $j \le \mathcal{K} \le \mathcal{K}_0$,

$$\left\| \eta (\boldsymbol{H}_{\mathcal{S}}^{1/2} (\boldsymbol{I} - \eta \boldsymbol{H}_{\mathcal{S}})^{l-j} \boldsymbol{\xi}_{\boldsymbol{u}}^j) \right\|^2 \le \eta^2 \|\boldsymbol{\xi}_{\boldsymbol{u}}^j\|^2 \left\| \boldsymbol{H}_{\mathcal{S}}^{1/2} (\boldsymbol{I} - \eta \boldsymbol{H}_{\mathcal{S}})^{l-j} \boldsymbol{H}_{\mathcal{S}} (\boldsymbol{I} - \eta \boldsymbol{H}_{\mathcal{S}})^{l-j} \right\| \|\boldsymbol{\xi}_{\boldsymbol{u}}^j\|^2$$

$$\leq \frac{\eta \sigma_1(\boldsymbol{w}_0)^2}{1 + 2(l - j)}.$$

This uses the second part of Lemma E.17, that $\|\boldsymbol{H}_{\mathcal{S}}\| \leq L_1(\boldsymbol{w}_0)$, that $j \leq \mathcal{K}_0$ which gives $\|\boldsymbol{\xi}_{\boldsymbol{u}}^j\| \leq \sigma_1(\boldsymbol{w}_0)$ by Lemma E.18, and our choice of $\eta$ (which cancels one of the $\sigma_1(\boldsymbol{w}_0)^2$ factors).

For a given $l$, by the Vector-Martingale Concentration Inequality Theorem C.1, we have with probability $1 - p/(12K_0)$ that

$$\left\| \sum_{j=1}^{l} Y_{l,j} \right\|^2 \leq 4\eta \sigma_1(\boldsymbol{w}_0)^2 \log\left(\frac{48K_0}{p}\right) \sum_{j=1}^{l} \frac{1}{1 + 2(l - j)}$$

$$\leq 4\eta \sigma_1(\boldsymbol{w}_0)^2 (\log(K_0) + 1) \log\left(\frac{48K_0}{p}\right). \tag{78}$$

The last step above uses $l \leq K_0$, $\sum_{j=1}^{l} \frac{1}{1+j} \leq \log(K_0) + 1$.

By the Union Bound, with probability at least $1 - \frac{p}{12}$, (78) holds for all $l$ from 1 to $K_0$. Because $1 \leq \mathcal{K} \leq K_0$, using the definition of $Y_{l,j}$ for $l \leq \mathcal{K}$, we obtain with probability at least $1 - \frac{p}{12}$ that

$$\eta \left\| \sum_{j=1}^{\mathcal{K}} \boldsymbol{H}_{\mathcal{S}}^{1/2} (\boldsymbol{I} - \eta \boldsymbol{H}_{\mathcal{S}})^{K-j} \boldsymbol{\xi}_{\boldsymbol{u}}^j \right\|^2 = \left\| \sum_{j=1}^{\mathcal{K}} Y_{\mathcal{K},j} \right\|^2 \leq 4\eta \sigma_1(\boldsymbol{w}_0)^2 (\log(K_0) + 1) \log\left(\frac{48K_0}{p}\right).$$

- For the second term, using the second part of Lemma E.17 and that $\mathcal{K} \leq K_0$, and then rearranging order of the sum and performing explicit calculation yields

$$\eta^2 \frac{L_2(\boldsymbol{w}_0)^2 B^4}{4} \sum_{j=0}^{\mathcal{K}-1} \sum_{l=0}^{\mathcal{K}-1} \left\| (\boldsymbol{I} - \eta \boldsymbol{H}_{\mathcal{S}})^{\mathcal{K}-1-j} \boldsymbol{H}_{\mathcal{S}} (\boldsymbol{I} - \eta \boldsymbol{H}_{\mathcal{S}})^{\mathcal{K}-1-l} \right\|$$

$$\leq \eta \frac{L_2(\boldsymbol{w}_0)^2 B^4}{4} \sum_{j=0}^{K_0-1} \sum_{l=0}^{K_0-1} \frac{1}{1 + j + l}$$

$$\leq \eta \frac{L_2(\boldsymbol{w}_0)^2 B^4}{4} \sum_{l=0}^{2(K_0-1)} \frac{\min(1 + j, 2K_0 - 1 - j)}{1 + j}$$

$$\leq \frac{\eta L_2(\boldsymbol{w}_0)^2 B^4 K_0}{2}.$$

Combining the above two bounds proves (74), the second part of Lemma E.20. $\qquad \square$

We introduce one more Lemma, an intermediate step in the proof of Fang et al. (2019).

**Lemma E.21.** *We have with probability at least $1 - p/12$ that*

$$\left\langle \nabla G_{\mathcal{S}}(\boldsymbol{y}^{\mathcal{K}}), \boldsymbol{u}^{\mathcal{K}} - \boldsymbol{y}^{\mathcal{K}} \right\rangle \leq \frac{3\eta \sum_{k=0}^{\mathcal{K}} \left\| \nabla G_{\mathcal{S}}(\boldsymbol{y}^k) \right\|^2}{16} + 8\eta \sigma_1(\boldsymbol{w}_0)^2 \log(48K_0/p) + \eta L_2(\boldsymbol{w}_0)^2 B^4 K_0/2.$$

**Proof of Lemma E.21.** Let $\boldsymbol{y}^* = \arg\min_{\boldsymbol{y}} G_{\mathcal{S}}(\boldsymbol{y})$; this exists as $G$ is convex in the subspace $\mathcal{S}$, by the definition of $\mathcal{S}$. By the optimality condition of $\boldsymbol{y}^*$, we have:

$$\nabla_{\boldsymbol{u}} F(\boldsymbol{x}^0) = -\boldsymbol{H}_{\mathcal{S}} \boldsymbol{y}^*. \tag{79}$$

Let $\tilde{\boldsymbol{y}}^k = \boldsymbol{y}^k - \boldsymbol{y}^*$. From the update rule of $\boldsymbol{y}^k$ and the optimality condition (79), we obtain:

$$\boldsymbol{H}_{\mathcal{S}} \tilde{\boldsymbol{y}}^k = \nabla G_{\mathcal{S}}(\boldsymbol{y}^k), \tilde{\boldsymbol{y}}^{k+1} = \tilde{\boldsymbol{y}}^k - \eta \boldsymbol{H}_{\mathcal{S}} \tilde{\boldsymbol{y}}^k. \tag{80}$$

Consequently, using (80) and (76), we have:

$$\left\langle \nabla G_{\mathcal{S}}(\boldsymbol{y}^{\mathcal{K}}), \boldsymbol{u}^{\mathcal{K}} - \boldsymbol{y}^{\mathcal{K}} \right\rangle$$

$$= \langle \tilde{\boldsymbol{y}}^{\mathcal{K}}, \boldsymbol{z}^{\mathcal{K}} \rangle_{\boldsymbol{H}_{\mathcal{S}}}$$

$$= \eta \sum_{k=1}^{\mathcal{K}} \langle \tilde{\boldsymbol{y}}^{k-1}, \boldsymbol{\xi}_{\boldsymbol{u}}^{k} \rangle_{\boldsymbol{H}_{\mathcal{S}}(\boldsymbol{I}-\eta \boldsymbol{H}_{\mathcal{S}})^{\mathcal{K}-k+1}} - \eta \sum_{k=0}^{\mathcal{K}-1} \langle \tilde{\boldsymbol{y}}^{k}, \nabla_{\boldsymbol{u}} F(\boldsymbol{x}^{k}) - \nabla G_{\mathcal{S}}(\boldsymbol{u}^{k}) \rangle_{\boldsymbol{H}_{\mathcal{S}}(\boldsymbol{I}-\eta \boldsymbol{H}_{\mathcal{S}})^{\mathcal{K}-k}}.$$

Now we bound both of these sums in a manner similar to the proof of Lemma E.20:

- For the first term: For any fixed $l$, $1 \le l \le K_0$, define a real-valued stochastic process for any $k$, $1 \le k \le \min(l, \mathcal{K}_0)$ by:

$$Y_{l,k} = \langle \tilde{\boldsymbol{y}}^{k-1}, \boldsymbol{\xi}_{\boldsymbol{u}}^{k} \rangle_{\boldsymbol{H}_{\mathcal{S}}(\boldsymbol{I}-\eta \boldsymbol{H}_{\mathcal{S}})^{l-k+1}} 1_{k-1<\mathcal{K}} = \begin{cases} \langle \tilde{\boldsymbol{y}}^{k-1}, \boldsymbol{\xi}_{\boldsymbol{u}}^{k} \rangle_{\boldsymbol{H}_{\mathcal{S}}(\boldsymbol{I}-\eta \boldsymbol{H}_{\mathcal{S}})^{l-k+1}} & : k \le \mathcal{K} \\ 0 & : k > \mathcal{K}. \end{cases}$$

Analogously to earlier, recalling $\mathcal{K} \le \mathcal{K}_0$, it's easy to check that for any *fixed* $l$, $Y_{l,k}$ is $\mathfrak{F}^{k}$ measurable, and that all terms defining $Y_{l,k}$ are $\mathfrak{F}^{k-1}$ measurable except $\boldsymbol{\xi}_{\boldsymbol{u}}^{k}$. Thus,

$$\mathbb{E}\big[Y_{l,k} | \mathfrak{F}^{k-1}\big] = 0.$$

We furthermore have for any fixed $l$, $1 \le l \le K_0$ and $k$, $1 \le k \le l$,

$$\|Y_{l,k}\|^{2} \le \sigma_1(\boldsymbol{w}_0)^2 \|\nabla G_{\mathcal{S}}(\boldsymbol{y}^{k-1})\|^{2}.$$

To justify why the above holds, clearly this is evident for $k > \mathcal{K}$. For $k \le \mathcal{K} \le \mathcal{K}_0$, note that

$$|Y_{l,k}|^{2} = \left\| \langle \tilde{\boldsymbol{y}}^{k-1}, \boldsymbol{\xi}_{\boldsymbol{u}}^{k} \rangle_{\boldsymbol{H}_{\mathcal{S}}(\boldsymbol{I}-\eta \boldsymbol{H}_{\mathcal{S}})^{l-k+1}} \right\|^{2} = \left| \langle \boldsymbol{H}_{\mathcal{S}} \tilde{\boldsymbol{y}}^{k-1}, \boldsymbol{\xi}_{\boldsymbol{u}}^{k} \rangle \right|^{2}_{(\boldsymbol{I}-\eta \boldsymbol{H}_{\mathcal{S}})^{l-k+1}}$$

$$= \left| \langle \nabla G_{\mathcal{S}}(\boldsymbol{y}^{k-1}), \boldsymbol{\xi}_{\boldsymbol{u}}^{k} \rangle \right|^{2}_{(\boldsymbol{I}-\eta \boldsymbol{H}_{\mathcal{S}})^{l-k+1}}$$

$$\le \sigma_1(\boldsymbol{w}_0)^2 \|\nabla G_{\mathcal{S}}(\boldsymbol{y}^{k-1})\|^{2} \|(\boldsymbol{I} - \eta \boldsymbol{H}_{\mathcal{S}})^{l-k+1}\|^{2}$$

$$\le \sigma_1(\boldsymbol{w}_0)^2 \|\nabla G_{\mathcal{S}}(\boldsymbol{y}^{k-1})\|^{2}.$$

Here we used that $\boldsymbol{H}_{\mathcal{S}}$ is symmetric, that (80), that $\|\boldsymbol{I} - \eta \boldsymbol{H}_{\mathcal{S}}^{l-k+1}\| \le 1$ which we have argued earlier in the proof of Lemma E.20, and that $\|\boldsymbol{\xi}_{\boldsymbol{u}}^{k}\| \le \sigma_1(\boldsymbol{w}_0)$ as $k \le l \le \mathcal{K}_0$ by Lemma E.18 and properties of projection matrices.

Now for any $l$, $1 \le l \le K_0$, by the Azuma–Hoeffding inequality, we have with probability at least $1 - p/(12 K_0)$ that

$$\left| \eta \sum_{k=1}^{l} Y_{l,k} \right| \le \sqrt{2\eta^2 \sigma_1(\boldsymbol{w}_0)^2 \log(24 K_0/p) \sum_{k=0}^{l-1} \|\nabla G_{\mathcal{S}}(\boldsymbol{y}^{k})\|^{2}}.$$

Taking a Union Bound, it follows that with probability at least $1 - p/12$, the above holds for all $l$ with $1 \le l \le K_0$.

Because $1 \le \mathcal{K} \le K_0$ always holds, using the definition of $Y_{l,k}$ for $k \le \mathcal{K}$, we obtain with probability at least $1 - \frac{p}{12}$ that

$$\left| \eta \sum_{k=1}^{\mathcal{K}} \langle \tilde{\boldsymbol{y}}_{k-1}, \boldsymbol{\xi}_{\boldsymbol{u}}^{k} \rangle_{\boldsymbol{H}_{\mathcal{S}}(\boldsymbol{I}-\eta \boldsymbol{H}_{\mathcal{S}})^{\mathcal{K}-k+1}} \right| = \left| \eta \sum_{k=1}^{\mathcal{K}} Y_{\mathcal{K},k} \right|$$

$$\le \sqrt{2\eta^2 \sigma_1(\boldsymbol{w}_0)^2 \log(24 K_0/p) \sum_{k=0}^{\mathcal{K}-1} \|\nabla G_{\mathcal{S}}(\boldsymbol{y}^{k})\|^{2}}$$

$$\le \frac{\eta}{16} + 8\eta \sigma_1(\boldsymbol{w}_0)^2 \log(48 K_0/p)$$

where we used AM-GM in the last step. This holds because we have this upper bound on $\left| \sum_{k=1}^{l} Y_{l,k} \right|$ irrespective of which value of $l$, $1 \le l \le K_0$ that $\mathcal{K}$ takes on. The first equality holds by our definition of $Y_{l,k}$ for $k \le \mathcal{K}$.

- For the second term: note

$$\eta \sum_{k=0}^{\mathcal{K}-1} \langle \tilde{\boldsymbol{y}}^{k}, \nabla_{\boldsymbol{u}} F(\boldsymbol{x}^{k}) - \nabla G_{\mathcal{S}}(\boldsymbol{u}^{k}) \rangle_{\boldsymbol{H}_{\mathcal{S}}(\boldsymbol{I}-\eta \boldsymbol{H}_{\mathcal{S}})^{\mathcal{K}-k}}$$

$$= \eta \sum_{k=0}^{\mathcal{K}-1} \langle \nabla G_{\mathcal{S}}(\boldsymbol{y}^{\mathcal{K}}), \nabla_{\boldsymbol{u}} F(\boldsymbol{x}^k) - \nabla G_{\mathcal{S}}(\boldsymbol{u}^k) \rangle_{(\boldsymbol{I} - \eta \boldsymbol{H}_{\mathcal{S}})^{\mathcal{K}-k}}$$

$$\leq \eta \sum_{k=0}^{\mathcal{K}-1} \left\| \nabla G_{\mathcal{S}}(\boldsymbol{y}^{\mathcal{K}}) \right\| \left\| \nabla_{\boldsymbol{u}} F(\boldsymbol{x}^k) - \nabla G_{\mathcal{S}}(\boldsymbol{u}^k) \right\|$$

$$\leq \frac{\eta \sum_{k=0}^{\mathcal{K}-1} \left\| \nabla G_{\mathcal{S}}(\boldsymbol{y}^{\mathcal{K}}) \right\|^2}{8} + 2\eta \sum_{k=0}^{\mathcal{K}-1} \left\| \nabla_{\boldsymbol{u}} F(\boldsymbol{x}^k) - \nabla G_{\mathcal{S}}(\boldsymbol{u}^k) \right\|^2$$

$$\leq \frac{\eta \sum_{k=0}^{\mathcal{K}-1} \left\| \nabla G_{\mathcal{S}}(\boldsymbol{y}^{\mathcal{K}}) \right\|^2}{8} + \frac{1}{2} \eta L_2(\boldsymbol{w}_0)^2 B^4 K_0.$$

The first step above uses that $\boldsymbol{H}_{\mathcal{S}}$ is symmetric and (80). The second step uses that $k \leq \mathcal{K}$ and that $\|\boldsymbol{I} - \eta \boldsymbol{H}_{\mathcal{S}}\| \leq 1$, as argued in the proof of Lemma E.20. The third step uses AM-GM. The last step uses that $\mathcal{K} \leq K_0$ and Lemma E.17; for $k < \mathcal{K}$, we have $\boldsymbol{x}^k \in \mathbb{B}(\boldsymbol{x}^0, B)$.

Combining these above two bounds proves Lemma E.21. $\qquad \square$

Now we finish the proof of Lemma E.19. As done in Fang et al. (2019), we combine Lemma E.20, Lemma E.21 with (72) to prove Lemma E.19 as follows. In particular, taking a Union Bound over the events from Lemma E.20 and Lemma E.21, we obtain with probability at least $1 - p/4$ that

$$G_{\mathcal{S}}(\boldsymbol{u}^{\mathcal{K}}) = G_{\mathcal{S}}(\boldsymbol{y}^{\mathcal{K}}) + \langle \nabla G_{\mathcal{S}}(\boldsymbol{y}^{\mathcal{K}}), \boldsymbol{u}^{\mathcal{K}} - \boldsymbol{y}^{\mathcal{K}} \rangle + \frac{1}{2} (\boldsymbol{u}^{\mathcal{K}} - \boldsymbol{y}^{\mathcal{K}})^\top \boldsymbol{H} (\boldsymbol{u}^{\mathcal{K}} - \boldsymbol{y}^{\mathcal{K}})$$

$$\leq G_{\mathcal{S}}(\boldsymbol{y}^{\mathcal{K}}) + \langle \nabla G_{\mathcal{S}}(\boldsymbol{y}^{\mathcal{K}}), \boldsymbol{u}^{\mathcal{K}} - \boldsymbol{y}^{\mathcal{K}} \rangle + \frac{1}{2} (\boldsymbol{u}^{\mathcal{K}} - \boldsymbol{y}^{\mathcal{K}})^\top \boldsymbol{H}_{\mathcal{S}} (\boldsymbol{u}^{\mathcal{K}} - \boldsymbol{y}^{\mathcal{K}})$$

$$\leq G_{\mathcal{S}}(\boldsymbol{y}^{\mathcal{K}}) + \frac{3\eta}{16} \sum_{k=0}^{\mathcal{K}-1} \left\| \nabla G_{\mathcal{S}}(\boldsymbol{y}^k) \right\|^2$$
$$+ 4\eta \sigma_1(\boldsymbol{w}_0)^2 (\log(K_0) + 3) \log(48K_0/p) + L_2(\boldsymbol{w}_0)^2 \eta B^4 K_0.$$

Here the first two lines used the definition of $G_{\mathcal{S}}$ and $\mathcal{S}$. The last line above applied Lemma E.21 together with the second part of Lemma E.20.

Now combining the above with (72), we obtain

$$G_{\mathcal{S}}(\boldsymbol{u}^{\mathcal{K}}) \leq G_{\mathcal{S}}(\boldsymbol{y}^{\mathcal{K}}) + \frac{3\eta}{16} \sum_{k=0}^{\mathcal{K}-1} \left\| \nabla G_{\mathcal{S}}(\boldsymbol{y}^k) \right\|^2$$
$$+ 4\eta \sigma_1(\boldsymbol{w}_0)^2 (\log(K_0) + 3) \log(48K_0/p) + L_2(\boldsymbol{w}_0)^2 \eta B^4 K_0$$

$$\leq G_{\mathcal{S}}(\boldsymbol{u}^0) - \frac{25}{32} \sum_{k=0}^{\mathcal{K}-1} \left\| \nabla G_{\mathcal{S}}(\boldsymbol{y}^k) \right\|^2$$
$$+ 4\eta \sigma_1(\boldsymbol{w}_0)^2 (\log(K_0) + 3) \log(48K_0/p) + \eta L_2(\boldsymbol{w}_0)^2 B^4 K_0,$$

where we also used $\boldsymbol{y}^0 = \boldsymbol{u}^0$. This proves Lemma E.19. $\qquad \square$

We now analyze the orthogonal complement of $\mathcal{S}$, $\mathcal{S}^\perp$ as in Fang et al. (2019), where the analysis again goes through since the iterates are 'local', being prior to the escape time $\mathcal{K}$:

**Lemma E.22** (Equivalent of Lemma 18, Fang et al. (2019)). *Deterministically, we have:*

$$G_{\mathcal{S}^\perp}(\boldsymbol{v}^{\mathcal{K}}) \leq G_{\mathcal{S}^\perp}(\boldsymbol{v}^0) - \sum_{k=1}^{\mathcal{K}} \eta \langle \nabla G_{\mathcal{S}^\perp}(\boldsymbol{v}_{\mathcal{K}-1}), \boldsymbol{\xi}_{\boldsymbol{v}}^k \rangle - \frac{7\eta}{8} \sum_{k=0}^{\mathcal{K}-1} \left\| \nabla G_{\mathcal{S}^\perp}(\boldsymbol{x}^k) \right\|^2 + L_2(\boldsymbol{w}_0)^2 B^4 \eta K_0^2.$$

*Note by choice of parameters that $L_2(\boldsymbol{w}_0)^2 B^4 \eta K_0^2 = \tilde{O}(\varepsilon^{1.5})$, where again the $\tilde{O}(\cdot)$ hides $F(\boldsymbol{w}_0)$-dependence.*

**Proof.** By definition of $G_{\mathcal{S}^\perp}$, and using definition of $\mathcal{S}^\perp$ which implies $\boldsymbol{H}_{\mathcal{S}^\perp} \leq 0$, we obtain

$$G_{\mathcal{S}^\perp}(\boldsymbol{v}^{k+1}) = G_{\mathcal{S}^\perp}(\boldsymbol{v}^k) + \langle \nabla G_{\mathcal{S}^\perp}(\boldsymbol{v}^k), \boldsymbol{v}^{k+1} - \boldsymbol{v}^k \rangle + \frac{1}{2} (\boldsymbol{v}^{k+1} - \boldsymbol{v}^k)^\top \boldsymbol{H}_{\mathcal{S}^\perp} (\boldsymbol{v}^{k+1} - \boldsymbol{v}^k)$$

$$\leq G_{\mathcal{S}^\perp}\left(\boldsymbol{v}^k\right) + \left\langle \nabla G_{\mathcal{S}^\perp}\left(\boldsymbol{v}^k\right), \boldsymbol{v}^{k+1} - \boldsymbol{v}^k \right\rangle$$

$$= G_{\mathcal{S}^\perp}\left(\boldsymbol{v}^k\right) - \eta\left\langle \nabla G_{\mathcal{S}^\perp}\left(\boldsymbol{v}^k\right), \nabla_{\boldsymbol{v}} F(\boldsymbol{x}^k) + \boldsymbol{\xi}_{\boldsymbol{v}}^{k+1} \right\rangle$$

$$= G_{\mathcal{S}^\perp}\left(\boldsymbol{v}^k\right) - \eta\left\|\nabla G_{\mathcal{S}^\perp}(\boldsymbol{v}^k)\right\|^2 - \left\langle \eta\nabla G_{\mathcal{S}^\perp}\left(\boldsymbol{v}^k\right), \nabla_{\boldsymbol{v}} F(\boldsymbol{x}^k) - \nabla G_{\mathcal{S}^\perp}\left(\boldsymbol{v}^k\right) \right\rangle$$
$$\qquad - \eta\left\langle \nabla G_{\mathcal{S}^\perp}\left(\boldsymbol{v}^k\right), \boldsymbol{\xi}_{\boldsymbol{v}}^{k+1} \right\rangle$$

$$\leq G_{\mathcal{S}^\perp}\left(\boldsymbol{v}^k\right) - \eta\left\langle \nabla G_{\mathcal{S}^\perp}\left(\boldsymbol{v}^k\right), \boldsymbol{\xi}_{\boldsymbol{v}}^{k+1} \right\rangle - \frac{7\eta}{8}\left\|\nabla G_{\mathcal{S}^\perp}(\boldsymbol{v}^k)\right\|^2 + 2\eta\left\|\nabla_{\boldsymbol{v}} F(\boldsymbol{x}^k) - \nabla G_{\mathcal{S}^\perp}\left(\boldsymbol{v}^k\right)\right\|^2.$$

The last step uses AM-GM.

Substituting and telescoping the above for $k$ from 0 to $\mathcal{K}-1$, we have:

$$G_{\mathcal{S}^\perp}(\boldsymbol{v}^{\mathcal{K}})$$

$$\leq G_{\mathcal{S}^\perp}(\boldsymbol{v}^0) - \sum_{k=1}^{\mathcal{K}} \eta\langle \nabla G_{\mathcal{S}^\perp}(\boldsymbol{v}^{k-1}), \boldsymbol{\xi}_{\boldsymbol{v}}^k \rangle - \frac{7\eta}{8}\sum_{k=0}^{\mathcal{K}-1}\left\|\nabla G_{\mathcal{S}^\perp}(\boldsymbol{x}^k)\right\|^2 + 2\eta\sum_{k=0}^{\mathcal{K}-1}\left\|\nabla_{\boldsymbol{v}} F(\boldsymbol{x}^k) - \nabla G_{\mathcal{S}^\perp}(\boldsymbol{v}^k)\right\|^2$$

$$\leq G_{\mathcal{S}^\perp}(\boldsymbol{v}^0) - \sum_{k=1}^{K} \eta\langle \nabla G_{\mathcal{S}^\perp}(\boldsymbol{v}^{k-1}), \boldsymbol{\xi}_{\boldsymbol{v}}^k \rangle - \frac{7\eta}{8}\sum_{k=0}^{K-1}\left\|\nabla G_{\mathcal{S}^\perp}(\boldsymbol{x}^k)\right\|^2 + \frac{L_2(\boldsymbol{w}_0)^2 B^4 \eta K_0}{2}.$$

Here, the second inequality uses that by Lemma E.17, for all $k \leq \mathcal{K}-1$, we have $\boldsymbol{x}^k \in \mathbb{B}(\boldsymbol{x}^0, B)$ and so

$$\left\|\nabla_{\boldsymbol{v}} F(\boldsymbol{x}^k) - G_{\mathcal{S}^\perp}(\boldsymbol{v}^k)\right\| = \left\|\mathcal{P}_{\mathcal{S}_\perp}(\nabla F(\boldsymbol{x}^k) - \nabla G(\boldsymbol{x}^k))\right\| \leq \left\|\nabla F(\boldsymbol{x}^k) - \nabla G(\boldsymbol{x}^k)\right\| \leq \frac{L_2(\boldsymbol{w}_0)B^2}{2}.$$

This completes the proof. $\qquad\square$

**Completing the Proof:** Now we have all the ingredients in hand to prove Lemma E.2.

**Proof of Lemma E.2.** Again, we follow the strategy of Fang et al. (2019) and adapt it to our setting here where we do not have global bounds on the Lipschitz constants of the gradient and Hessian. With Lemma E.19 and Lemma E.22 in hand, the idea will be to show

$$\sum_{k=0}^{\mathcal{K}_0-1}\left\|\nabla G_{\mathcal{S}_\perp}(\boldsymbol{v}^k)\right\|^2 + \sum_{k=0}^{\mathcal{K}_0-1}\left\|\nabla G_{\mathcal{S}}(\boldsymbol{y}^k)\right\|^2 = \tilde{\Omega}(1),$$

and to bound the noise term

$$-\sum_{k=1}^{K} \eta\langle \nabla G_{\mathcal{S}^\perp}(\boldsymbol{v}^{k-1}), \boldsymbol{\xi}_{\boldsymbol{v}}^k \rangle.$$

We break the proof of Lemma E.2 into two cases:

1. $\left\|\nabla F(\boldsymbol{x}^0)\right\| > 5\sigma_1(\boldsymbol{w}_0)$.

2. $\left\|\nabla F(\boldsymbol{x}^0)\right\| \leq 5\sigma_1(\boldsymbol{w}_0)$.

**Case 1:** This case is more straightforward as the gradient is large, and will not use the quadratic approximation we developed earlier.

Consider any $k, 0 \leq k \leq \mathcal{K}-1$. Thus $\boldsymbol{x}^k \in \mathbb{B}(\boldsymbol{x}^0, B)$, and so $\boldsymbol{u} \in \mathbb{B}(\boldsymbol{x}^0, B)$ for all $\boldsymbol{u} \in \overline{\boldsymbol{x}^0 \boldsymbol{x}^k}$. By Lemma E.7, as $\boldsymbol{x}^0 \in \mathcal{L}_{F, F(\boldsymbol{w}_0)}$, we have $\left\|\nabla^2 F(\boldsymbol{u})\right\| \leq L_1(\boldsymbol{w}_0)$ for all such $\boldsymbol{u}$. Thus as $\left\|\nabla F(\boldsymbol{x}^0)\right\| > 5\sigma_1(\boldsymbol{w}_0)$ and by our choice of parameters,

$$\left\|\nabla F(\boldsymbol{x}^k)\right\| \geq \left\|\nabla F(\boldsymbol{x}^0)\right\| - \left\|\nabla F(\boldsymbol{x}^k) - \nabla F(\boldsymbol{x}^0)\right\| \geq 5\sigma_1(\boldsymbol{w}_0) - L_1(\boldsymbol{w}_0)B \geq \frac{9}{2}\sigma_1(\boldsymbol{w}_0). \tag{81}$$

Similarly, as $\boldsymbol{x}^{k+1} = \boldsymbol{x}^k - \eta\nabla \tilde{f}(\boldsymbol{x}^k; \boldsymbol{\zeta}_{k+1})$ and again as $\boldsymbol{x}^0 \in \mathcal{L}_{F, F(\boldsymbol{w}_0)}$, we have $\left\|\nabla^2 F(\boldsymbol{u})\right\| \leq L_1(\boldsymbol{w}_0)$ for all $\boldsymbol{u} \in \overline{\boldsymbol{x}^k \boldsymbol{x}^{k+1}}$ by Lemma E.7. Applying Lemma A.1, for all $0 \leq k \leq \mathcal{K}-1$, we obtain:

$$F(\boldsymbol{x}^{k+1}) - F(\boldsymbol{x}^k) \leq \left\langle \nabla F(\boldsymbol{x}^k), \boldsymbol{x}^{k+1} - \boldsymbol{x}^k \right\rangle + \frac{L_1(\boldsymbol{w}_0)}{2}\left\|\boldsymbol{x}^{k+1} - \boldsymbol{x}^k\right\|^2$$

$$= -\eta\|\nabla F(\boldsymbol{x}^k)\|^2 - \eta\langle\nabla F(\boldsymbol{x}^k),\boldsymbol{\xi}^{k+1}\rangle + \frac{L_1(\boldsymbol{w}_0)\eta^2}{2}\|\nabla F(\boldsymbol{x}^k)+\boldsymbol{\xi}^{k+1}\|^2.$$

$$\leq -\eta\|\nabla F(\boldsymbol{x}^k)\|^2 - \eta\langle\nabla F(\boldsymbol{x}^k),\boldsymbol{\xi}^{k+1}\rangle + L_1(\boldsymbol{w}_0)\eta^2\|\nabla F(\boldsymbol{x}^k)\|^2 + L_1(\boldsymbol{w}_0)\eta^2\|\boldsymbol{\xi}^{k+1}\|^2.$$

$$\leq \eta\left(-\frac{15}{16}+\frac{5}{32}\right)\|\nabla F(\boldsymbol{x}^k)\|^2 + \frac{8}{5}\eta\sigma_1(\boldsymbol{w}_0)^2 + L_1(\boldsymbol{w}_0)\eta^2\sigma_1(\boldsymbol{w}_0)^2$$

$$\leq -\frac{25\eta}{32}\|\nabla F(\boldsymbol{x}^k)\|^2 + 2\eta\sigma^2.$$

$$\leq -\eta\left(\frac{25}{32}-\frac{8}{81}\right)\|\nabla F(\boldsymbol{x}^k)\|^2.$$

Note here that we need to consider a bound on the Lipschitz constant of the gradient between $\boldsymbol{x}^{\mathcal{K}-1}$ and $\boldsymbol{x}^{\mathcal{K}}$; see Remark 15. Here, we used the update rule of SGD, AM-GM and Young's Inequality, that $L_1(\boldsymbol{w}_0)\eta \leq \frac{1}{16}$ by our choice of hyperparameters, Lemma E.18, and finally (81) in the last step.

Telescoping the above inequality from $k = 0$ to $\mathcal{K} - 1$, we get:

$$F(\boldsymbol{x}^{\mathcal{K}}) - F(\boldsymbol{x}^0) \leq -\eta\left(\frac{25}{32}-\frac{8}{81}\right)\sum_{k=0}^{\mathcal{K}-1}\|\nabla F(\boldsymbol{x}^k)\|^2. \tag{82}$$

To upper bound the right hand side above, note by Triangle Inequality that

$$\left\|\eta\sum_{k=0}^{\mathcal{K}-1}\nabla F(\boldsymbol{x}^k)\right\| = \left\|-\eta\sum_{k=0}^{\mathcal{K}-1}\nabla F(\boldsymbol{x}^k)\right\|$$

$$= \left\|\boldsymbol{x}^{\mathcal{K}}-\boldsymbol{x}^0 + \eta\sum_{k=1}^{\mathcal{K}}\boldsymbol{\xi}^k\right\|$$

$$\geq \|\boldsymbol{x}^{\mathcal{K}}-\boldsymbol{x}^0\| - \left\|\eta\sum_{k=1}^{\mathcal{K}}\boldsymbol{\xi}^k\right\|. \tag{83}$$

By the Vector-Martingale Concentration Inequality Theorem C.1 and the bound $\|\boldsymbol{\xi}^k\| \leq \sigma_1(\boldsymbol{w}_0)$ for all $k \leq \mathcal{K}$ by Lemma E.5, we obtain with probability at least $1 - p/12$:

$$\left\|\eta\sum_{k=1}^{\mathcal{K}}\boldsymbol{\xi}^k\right\| = \left\|\eta\sum_{k=1}^{K_0}\boldsymbol{\xi}^k 1_{k\leq\mathcal{K}}\right\| \leq 2\eta\sigma_1(\boldsymbol{w}_0)\sqrt{K_0\log(48/p)} \leq \frac{B}{16}. \tag{84}$$

Here, we used the fact that $1_{k\leq\mathcal{K}} \equiv 1_{k-1<\mathcal{K}}$ and consequently $1_{k\leq\mathcal{K}}$ is $\mathfrak{F}^{k-1}$-measurable, and that $\mathbb{E}[\boldsymbol{\xi}^k|\mathfrak{F}^{k-1}] = 0$, $\|\boldsymbol{\xi}^k\| \leq \sigma_1(\boldsymbol{w}_0)$ for all $k \leq \mathcal{K}$.

Suppose the above event implying (84) occurs, which has probability at least $1 - \frac{p}{12}$. Under this event, suppose that $\boldsymbol{x}^k$ is able to leave the ball $\mathbb{B}(\boldsymbol{x}^0, B)$ in $K_0$ iterations or less. If this is the case, then we have $\mathcal{K} = \mathcal{K}_0 \leq K_0$, and so $\|\boldsymbol{x}^{\mathcal{K}}-\boldsymbol{x}^0\| \geq B$. Thus conditioned on the aforementioned event implying (84), if $\boldsymbol{x}^k$ is able to leave the ball $\mathbb{B}(\boldsymbol{x}^0, B)$ in $K_0$ iterations or less, we obtain

$$\eta\sum_{k=0}^{\mathcal{K}-1}\|\nabla F(\boldsymbol{x}^k)\|^2 \geq \frac{1}{\eta\mathcal{K}}\left\|\sum_{k=0}^{\mathcal{K}-1}\eta\nabla F(\boldsymbol{x}^k)\right\|^2 \geq \frac{1}{\eta\mathcal{K}}\left(B - \frac{1}{16}B\right)^2 \geq \frac{15^2 B^2}{16^2\eta\mathcal{K}} \geq \frac{15^2 B^2}{16^2\eta K_0},$$

where we combined (83), (84) to lower bound $\left\|\sum_{k=0}^{\mathcal{K}-1}\eta\nabla F(\boldsymbol{x}^k)\right\|$. Here the first step holds by the elementary inequality $\left\|\sum_{i=0}^{l}\boldsymbol{a}_i\right\|^2 \leq l\sum_{i=0}^{l}\|\boldsymbol{a}_i\|^2$, and the last step uses $K_0 \geq \mathcal{K}$.

Consequently by combining with (82), with probability at least $1 - \frac{p}{12}$, if $\boldsymbol{x}^k$ is able to leave the ball $\mathbb{B}(\boldsymbol{x}^0, B)$ in $K_0$ iterations or less, we have

$$F(\boldsymbol{x}^{\mathcal{K}}) \leq F(\boldsymbol{x}^0) - \left(\frac{25}{32}-\frac{8}{81}\right)\cdot\frac{15^2 B^2}{16^2\eta K_0} < F(\boldsymbol{x}^0) - \frac{B^2}{7\eta K_0}.$$

**Case 2:** Suppose $\|\nabla F(\boldsymbol{x}^0)\| \leq 5\sigma_1(\boldsymbol{w}_0)$. To obtain the desired result, we first define and prove the following Lemmas. Proving these Lemmas in turn utilizes the Lemmas on quadratic approximation we have established earlier.

**Lemma E.23.** *For all $0 \le k \le \mathcal{K} - 1$, we have*

$$\left\| \nabla G_{\mathcal{S}^{\perp}}(\boldsymbol{v}^k) \right\| \le \frac{11}{2} \sigma_1(\boldsymbol{w}_0).$$

**Proof.** By the condition in this case, properties of projection matrices, and as $\boldsymbol{v}^0 = 0$,

$$\left\| \nabla G_{\mathcal{S}^{\perp}}(\boldsymbol{v}^0) \right\| = \left\| \nabla_{\boldsymbol{v}} F(\boldsymbol{x}^0) \right\| \le \left\| \nabla F(\boldsymbol{x}^0) \right\| \le 5 \sigma_1(\boldsymbol{w}_0).$$

Note for $k \le \mathcal{K} - 1$, we have

$$\left\| \boldsymbol{v}^k - \boldsymbol{v}^0 \right\| = \left\| \boldsymbol{\mathcal{P}}_{\mathcal{S}^{\perp}}(\boldsymbol{x}^k - \boldsymbol{x}^0) \right\| \le B.$$

Thus

$$\begin{aligned}
\left\| \nabla G_{\mathcal{S}^{\perp}}(\boldsymbol{v}^k) \right\| &\le \left\| \nabla G_{\mathcal{S}^{\perp}}(\boldsymbol{v}^0) \right\| + \left\| \nabla G_{\mathcal{S}^{\perp}}(\boldsymbol{v}^k) - \nabla G_{\mathcal{S}^{\perp}}(\boldsymbol{v}^0) \right\| \\
&\le 5\sigma_1(\boldsymbol{w}_0) + L_1(\boldsymbol{w}_0) B \\
&\le \frac{11}{2} \sigma.
\end{aligned}$$

The above uses our choice of hyperparameters, and that

$$\left\| \nabla G_{\mathcal{S}^{\perp}}(\boldsymbol{v}^k) - \nabla G_{\mathcal{S}^{\perp}}(\boldsymbol{v}^0) \right\| = \left\| \boldsymbol{H}_{\mathcal{S}^{\perp}}(\boldsymbol{v}^k - \boldsymbol{v}^0) \right\| \le \|\boldsymbol{H}\| \left\| \boldsymbol{v}^k - \boldsymbol{v}^0 \right\| \le L_1(\boldsymbol{w}_0) \left\| \boldsymbol{v}^k - \boldsymbol{v}^0 \right\|,$$

which in turn follows because $\boldsymbol{x}^0 \in \mathcal{L}_{F, F(\boldsymbol{w}_0)}$ and by Assumption 1.1. $\qquad \square$

The next Lemma is obtained by combining Lemma E.19 and Lemma E.22, and it gives us a way to upper bound $F(\boldsymbol{x}^k) - F(\boldsymbol{x}^0)$.

**Lemma E.24** (Equivalent of Lemma 19 in Fang et al. (2019)). *If $\left\| \nabla F(\boldsymbol{x}^0) \right\| \le 5\sigma_1(\boldsymbol{w}_0)$, with probability $1 - \frac{p}{4}$, we have*

$$\begin{aligned}
F(\boldsymbol{x}^{\mathcal{K}}) \le F(\boldsymbol{x}^0) &- \eta \sum_{k=1}^{\mathcal{K}} \left\langle \nabla G_{\mathcal{S}^{\perp}}(\boldsymbol{v}^{k-1}), \boldsymbol{\xi}_{\boldsymbol{v}}^k \right\rangle + \left( \frac{3}{256} + \frac{1}{80} \right) \frac{B^2}{\eta K_0} \\
&- \frac{7\eta}{8} \sum_{k=0}^{\mathcal{K}-1} \left\| \nabla G_{\mathcal{S}^{\perp}}(\boldsymbol{v}^k) \right\|^2 - \frac{25\eta}{32} \sum_{k=0}^{\mathcal{K}-1} \left\| \nabla G_{\mathcal{S}}(\boldsymbol{y}^k) \right\|^2.
\end{aligned}$$

**Proof.** For $k \le \mathcal{K} - 1$, we have $\boldsymbol{x}^k \in \mathbb{B}(\boldsymbol{x}^0, B)$. Consequently the entire line segment $\overline{\boldsymbol{x}^0 \boldsymbol{x}^k}$ lies in $\mathbb{B}(\boldsymbol{x}^0, B)$. As $\boldsymbol{x}^0 \in \mathcal{L}_{F, F(\boldsymbol{w}_0)}$, by Lemma E.7, we have

$$\left\| \nabla F(\boldsymbol{x}^k) - \nabla F(\boldsymbol{x}^0) \right\| \le L_1(\boldsymbol{w}_0) \left\| \boldsymbol{x}^k - \boldsymbol{x}^0 \right\| \le L_1(\boldsymbol{w}_0) B.$$

Thus by our choice of parameters, as per Remark 12,

$$\left\| \nabla F(\boldsymbol{x}^k) \right\| \le \left\| \nabla F(\boldsymbol{x}^0) \right\| + \left\| \nabla F(\boldsymbol{x}^k) - \nabla F(\boldsymbol{x}^0) \right\| \le 5\sigma_1(\boldsymbol{w}_0) + L_1(\boldsymbol{w}_0) B \le \frac{11}{2} \sigma_1(\boldsymbol{w}_0).$$

Recalling $\left\| \boldsymbol{\xi}^{\mathcal{K}} \right\| \le \sigma_1(\boldsymbol{w}_0)$ by Lemma E.18, we obtain from our choice of parameters as per Remark 12 that

$$\left\| \boldsymbol{x}^{\mathcal{K}} - \boldsymbol{x}^0 \right\| \le \left\| \boldsymbol{x}^0 - \boldsymbol{x}^{\mathcal{K}-1} \right\| + \eta \left\| \nabla F(\boldsymbol{x}^{\mathcal{K}-1}) + \boldsymbol{\xi}^{\mathcal{K}} \right\| \le B + \frac{13}{2} \eta \sigma_1(\boldsymbol{w}_0) \le B + \frac{B}{100}. \tag{85}$$

Using this, we then bound the difference between $F(\boldsymbol{x}^{\mathcal{K}})$ and $G(\boldsymbol{x}^{\mathcal{K}})$. As $\boldsymbol{x}^{\mathcal{K}} = \boldsymbol{x}^{\mathcal{K}-1} - \eta \nabla \tilde{f}(\boldsymbol{x}^{\mathcal{K}-1}; \boldsymbol{\zeta}_{\mathcal{K}})$, as $\boldsymbol{x}^{\mathcal{K}-1} \in \mathbb{B}(\boldsymbol{x}^0, B)$, and as $\boldsymbol{x}^0 \in \mathcal{L}_{F, F(\boldsymbol{w}_0)}$, we have $\left\| \nabla^2 F(\boldsymbol{u}) - \nabla^2 F(\boldsymbol{x}^0) \right\| \le L_2(\boldsymbol{w}_0) \left\| \boldsymbol{u} - \boldsymbol{x}^0 \right\|$ for all $\boldsymbol{u} \in \overline{\boldsymbol{x}^{\mathcal{K}-1} \boldsymbol{x}^{\mathcal{K}}}$ by Lemma E.8. Applying Lemma A.2 and recalling that $G_{\mathcal{S}}(\boldsymbol{u}^{\mathcal{K}}) + G_{\mathcal{S}^{\perp}}(\boldsymbol{v}^{\mathcal{K}}) = G(\boldsymbol{x}^{\mathcal{K}} - \boldsymbol{x}^0)$, we obtain

$$F(\boldsymbol{x}^{\mathcal{K}}) - F(\boldsymbol{x}^0) - G_{\mathcal{S}}(\boldsymbol{u}^{\mathcal{K}}) - G_{\mathcal{S}^{\perp}}(\boldsymbol{v}^{\mathcal{K}}) \le \frac{L_2(\boldsymbol{w}_0)}{6} \left\| \boldsymbol{x}^{\mathcal{K}} - \boldsymbol{x}^0 \right\|^3 \le \frac{L_2(\boldsymbol{w}_0) B^3}{5}. \tag{86}$$

Here, we used (85) in the last step. Note here that we need to consider a bound on the Lipschitz constant of the Hessian between $\boldsymbol{x}^{\mathcal{K}-1}$ and $\boldsymbol{x}^{\mathcal{K}}$; see Remark 15.

Now, take a Union Bound over Lemma E.19 and Lemma E.22. We now add the bounds from Lemma E.19 and Lemma E.22 to upper bound $G_{\mathcal{S}}(\boldsymbol{u}^{\mathcal{K}}) + G_{\mathcal{S}^\perp}(\boldsymbol{v}^{\mathcal{K}})$ and use that $G_{\mathcal{S}}(\boldsymbol{u}^0) + G_{\mathcal{S}^\perp}(\boldsymbol{v}^0) = 0$. Combining with (86), we obtain with probability at least $1 - p/4$ that

$$F(\boldsymbol{x}^{\mathcal{K}}) \le F(\boldsymbol{x}^0) - \eta \sum_{k=1}^{\mathcal{K}} \langle \nabla G_{\mathcal{S}^\perp}(v_{k-1}), \boldsymbol{\xi}_v^k \rangle + 4\eta \sigma_1(\boldsymbol{w}_0)^2 (1 + 3\log(K_0)) \log\left(\frac{48}{p}\right)$$

$$- \frac{7\eta}{8} \sum_{k=0}^{\mathcal{K}-1} \left\| \nabla G_{\mathcal{S}^\perp}(\boldsymbol{v}^k) \right\|^2 - \frac{25\eta}{32} \sum_{k=0}^{\mathcal{K}-1} \left\| \nabla G_{\mathcal{S}}(\boldsymbol{y}^k) \right\|^2 + \frac{3L_2(\boldsymbol{w}_0)B^4 \eta K_0}{2} + \frac{L_2(\boldsymbol{w}_0)B^3}{5}. \quad (87)$$

Note by our choice of hyperparameters (analogous to the choice of hyperparameters from Fang et al. (2019)), we have the following bounds: $4\eta\sigma_1(\boldsymbol{w}_0)^2(1 + 3\log(K_0))\log\left(\frac{48}{p}\right) \le \frac{B^2}{256\eta K_0}$, $\frac{3L_2(\boldsymbol{w}_0)B^4 \eta K_0}{2} \le \frac{B^2}{128\eta K_0}$, $\frac{L_2(\boldsymbol{w}_0)B^3}{5} \le \frac{B^2}{80\eta K_0}$.

Combining these above inequalities with (87), with probability at least $1 - p/4$, we obtain

$$F(\boldsymbol{x}^{\mathcal{K}}) \le F(\boldsymbol{x}^0) - \eta \sum_{k=1}^{\mathcal{K}} \langle \nabla G_{\mathcal{S}^\perp}(\boldsymbol{v}_{k-1}), \boldsymbol{\xi}_v^k \rangle + \left(\frac{3}{256} + \frac{1}{80}\right)\frac{B^2}{\eta K_0}$$

$$- \frac{7\eta}{8} \sum_{k=0}^{\mathcal{K}-1} \left\| \nabla G_{\mathcal{S}^\perp}(\boldsymbol{v}^k) \right\|^2 - \frac{25\eta}{32} \sum_{k=0}^{\mathcal{K}-1} \left\| \nabla G_{\mathcal{S}}(\boldsymbol{y}^k) \right\|^2.$$

This implies Lemma E.24. $\qquad\square$

By Lemma E.24, we want to lower bound the gradient norm of $G_{\mathcal{S}^\perp}, G_{\mathcal{S}}$. We do this in the following Lemma, assuming $\boldsymbol{x}^k$ leaves the ball $\mathbb{B}(\boldsymbol{x}^0, B)$ in $K_0$ iterations.

**Lemma E.25** (Equivalent of Lemma 20 in Fang et al. (2019)). *With probability $1 - \frac{p}{6}$, if $\boldsymbol{x}^k$ exits $\mathbb{B}(\boldsymbol{x}^0, B)$ in $K_0$ iterations (i.e. $\mathcal{K} = \mathcal{K}_0 \le K_0$), we have*

$$\eta \sum_{k=0}^{\mathcal{K}-1} \left\| \nabla G_{\mathcal{S}^\perp}(\boldsymbol{v}^k) \right\|^2 + \eta \sum_{k=0}^{\mathcal{K}-1} \left\| \nabla G_{\mathcal{S}}(\boldsymbol{y}^k) \right\|^2 \ge \frac{169B^2}{512\eta K_0}.$$

**Proof.** At a high level, the proof idea is similar to the proof of Case 1 earlier. Telescoping the recursions $\boldsymbol{v}^k = \boldsymbol{v}^{k-1} - \eta \boldsymbol{\xi}_v^k - \eta \nabla_{\boldsymbol{v}} F(\boldsymbol{x}^k)$ and $\boldsymbol{y}^k = \boldsymbol{y}^{k-1} - \eta \nabla G_{\mathcal{S}}(\boldsymbol{y}^k)$, we obtain

$$\left\| \eta \sum_{k=0}^{\mathcal{K}-1} \left( \nabla G_{\mathcal{S}^\perp}(\boldsymbol{v}^k) + \nabla G_{\mathcal{S}}(\boldsymbol{y}^k) \right) \right\| = \left\| -\eta \sum_{k=0}^{\mathcal{K}-1} \left( \nabla G_{\mathcal{S}^\perp}(\boldsymbol{v}^k) + \nabla G_{\mathcal{S}}(\boldsymbol{y}^k) \right) \right\|$$

$$= \left\| \boldsymbol{v}^{\mathcal{K}} - \boldsymbol{v}^0 + \eta \sum_{k=0}^{\mathcal{K}-1} \left( \boldsymbol{\xi}_v^{k+1} - \nabla G_{\mathcal{S}^\perp}(\boldsymbol{v}^k) + \nabla_{\boldsymbol{v}} F(\boldsymbol{x}^k) \right) + \boldsymbol{y}^{\mathcal{K}} - \boldsymbol{y}^0 \right\|$$

$$\ge \left\| \boldsymbol{v}^{\mathcal{K}} - \boldsymbol{v}^0 + \eta \sum_{k=0}^{\mathcal{K}-1} \boldsymbol{\xi}_v^{k+1} + \left( \boldsymbol{u}^{\mathcal{K}} - \boldsymbol{u}^0 \right) - \left( \boldsymbol{z}^{\mathcal{K}} - \boldsymbol{z}^0 \right) \right\|$$

$$- \left\| \eta \sum_{k=0}^{\mathcal{K}-1} \left( \nabla G_{\mathcal{S}^\perp}(\boldsymbol{v}^k) - \nabla_{\boldsymbol{v}} F(\boldsymbol{x}^k) \right) \right\|.$$

Here, we used that $\boldsymbol{z}^k = \boldsymbol{u}^k - \boldsymbol{y}^k$ and the Triangle Inequality.

Next, recall $\boldsymbol{x}^k - \boldsymbol{x}^0 = \boldsymbol{u}^k + \boldsymbol{v}^k$ for all $k \ge 0$, and $\boldsymbol{u}^0 = \boldsymbol{v}^0 = 0$. Thus $\boldsymbol{x}^k - \boldsymbol{x}^0 = \boldsymbol{v}^k - \boldsymbol{v}^0 + \boldsymbol{u}^k - \boldsymbol{u}^0$. Furthermore notice

$$\nabla G_{\mathcal{S}^\perp}(\boldsymbol{v}^k) - \nabla_{\boldsymbol{v}} F(\boldsymbol{x}^k) = \boldsymbol{H}_{\mathcal{S}^\perp} \left( \nabla G(\boldsymbol{x}^k) - \nabla F(\boldsymbol{x}^k) \right).$$

For all $k \le \mathcal{K} - 1$ we have $\boldsymbol{x}^k \in \mathbb{B}(\boldsymbol{x}^0, B)$, so as $\boldsymbol{x}^0 \in \mathcal{L}_{F,F(\boldsymbol{w}_0)}$, Lemma E.17 gives

$$\left\| \eta \sum_{k=0}^{\mathcal{K}-1} \left( \nabla G_{\mathcal{S}^\perp}(\boldsymbol{v}^k) - \nabla_{\boldsymbol{v}} F(\boldsymbol{x}^k) \right) \right\| \le \eta K_0 \cdot \frac{L_2(\boldsymbol{w}_0)B^2}{2}.$$

Applying these observations and Triangle Inequality again, we obtain

$$\left\| \eta \sum_{k=0}^{\mathcal{K}-1} \left( \nabla G_{\mathcal{S}^\perp}(\boldsymbol{v}^k) + \nabla G_{\mathcal{S}}(\boldsymbol{y}^k) \right) \right\| \ge \left\| \boldsymbol{x}^{\mathcal{K}} - \boldsymbol{x}^0 \right\| - \left\| \boldsymbol{z}^{\mathcal{K}} - \boldsymbol{z}^0 \right\| - \eta \left\| \sum_{k=1}^{\mathcal{K}} \boldsymbol{\xi}_v^k \right\| - \frac{\eta K_0 L_2(\boldsymbol{w}_0)B^2}{2}$$

$$\geq \|\boldsymbol{x}^{\mathcal{K}} - \boldsymbol{x}^0\| - \|\boldsymbol{z}^{\mathcal{K}} - \boldsymbol{z}^0\| - \frac{B}{32} - \eta \left\| \sum_{k=1}^{\mathcal{K}} \boldsymbol{\xi}_{\boldsymbol{v}}^k \right\|. \tag{88}$$

and Lemma E.17 combined with the fact that projection matrices do not increase norm and that $\boldsymbol{x}^k \in \mathbb{B}(\boldsymbol{x}^0, B)$ for $k < \mathcal{K}$, and the final statement is by the choice of hyperparameters.

Using Lemma E.20 and that $\boldsymbol{z}^0 = 0$, we obtain with probability at least $1 - \frac{p}{12}$ that

$$\|\boldsymbol{z}^{\mathcal{K}} - \boldsymbol{z}^0\| \leq \frac{3B}{32}. \tag{89}$$

Now recall that $1_{k \leq \mathcal{K}} \equiv 1_{k-1 < \mathcal{K}}$ is $\mathfrak{F}^{k-1}$-measurable, which implies

$$\mathbb{E}\big[\boldsymbol{\xi}_{\boldsymbol{v}}^k 1_{\{k \leq \mathcal{K}\}} | \mathfrak{F}^{k-1}\big] = \boldsymbol{0},$$

as the stochastic gradient oracle is unbiased. Furthermore, recall $\|\boldsymbol{\xi}^k\| \leq \sigma_1(\boldsymbol{w}_0)$ for $k \leq \mathcal{K}$, and projection matrices do not increase norm. Thus by the Vector-Martingale Concentration Inequality Theorem C.1, with probability at least $1 - \frac{p}{12}$, we have

$$\left\| \eta \sum_{k=1}^{\mathcal{K}} \boldsymbol{\xi}_{\boldsymbol{v}}^k \right\| = \left\| \eta \sum_{k=1}^{K_0} \boldsymbol{\xi}_{\boldsymbol{v}}^k 1_{\{k \leq \mathcal{K}\}} \right\| \leq 2\eta \sigma_1(\boldsymbol{w}_0) \sqrt{K_0 \log\left(\frac{48}{p}\right)} \leq \frac{B}{16}. \tag{90}$$

Thus taking a Union Bound over the events implying (89), (89) and combining with the earlier display (88), with probability at least $1 - \frac{p}{6}$, we have

$$\left\| \eta \sum_{k=0}^{\mathcal{K}-1} \nabla G_{\mathcal{S}^\perp}(\boldsymbol{v}^k) + \nabla G_{\mathcal{S}}(\boldsymbol{y}^k) \right\| \geq \|\boldsymbol{x}^{\mathcal{K}} - \boldsymbol{x}^0\| - \frac{3B}{16}.$$

Thus with probability at least $1 - \frac{p}{6}$, if $\boldsymbol{x}^k$ exits $\mathbb{B}(\boldsymbol{x}^0, B)$ in $K_0$ iterations (that is, if we have $K_0 \geq \mathcal{K}$), we have

$$\left\| \eta \sum_{k=0}^{\mathcal{K}-1} \nabla G_{\mathcal{S}^\perp}(\boldsymbol{v}^k) + \nabla G_{\mathcal{S}}(\boldsymbol{y}^k) \right\| \geq \|\boldsymbol{x}^{\mathcal{K}} - \boldsymbol{x}^0\| - \frac{3B}{16} \geq B - \frac{3B}{16},$$

and so

$$\eta \sum_{k=0}^{\mathcal{K}-1} \left\| \nabla G_{\mathcal{S}^\perp}(\boldsymbol{v}^k) \right\|^2 + \eta \sum_{k=0}^{\mathcal{K}-1} \left\| \nabla G_{\mathcal{S}}(\boldsymbol{y}^k) \right\|^2 \geq \frac{1}{2\eta\mathcal{K}} \left\| \eta \sum_{k=0}^{\mathcal{K}-1} \big( \nabla G_{\mathcal{S}^\perp}(\boldsymbol{v}^k) + \nabla G_{\mathcal{S}}(\boldsymbol{y}^k) \big) \right\|^2$$

$$\geq \frac{1}{2\eta\mathcal{K}} \left( B - \frac{3B}{16} \right)^2 = \frac{169B^2}{512\eta\mathcal{K}} \geq \frac{169B^2}{512\eta K_0}.$$

In the first step above we used the elementary inequality $\left\| \sum_{i=1}^l \boldsymbol{a}_i \right\|^2 \leq l \sum_{i=1}^l \|\boldsymbol{a}_i\|^2$ and Young's Inequality. This proves Lemma E.25. $\qquad\square$

We now combine Lemma E.24, Lemma E.25 to prove Lemma E.2. First recall by Lemma E.24, with probability $1 - p/4$, we have

$$F(\boldsymbol{x}^{\mathcal{K}}) \leq F(\boldsymbol{x}^0) - \eta \sum_{k=1}^{\mathcal{K}} \big\langle \nabla G_{\mathcal{S}^\perp}(\boldsymbol{v}^{k-1}), \boldsymbol{\xi}_{\boldsymbol{v}}^k \big\rangle + \left( \frac{3}{256} + \frac{1}{80} \right) \frac{B^2}{\eta K_0}$$

$$- \frac{7\eta}{8} \sum_{k=0}^{\mathcal{K}-1} \left\| \nabla G_{\mathcal{S}^\perp}(\boldsymbol{v}^k) \right\|^2 - \frac{25\eta}{32} \sum_{k=0}^{\mathcal{K}-1} \left\| \nabla G_{\mathcal{S}}(\boldsymbol{y}^k) \right\|^2. \tag{91}$$

We first control $\sum_{k=1}^{\mathcal{K}} \big\langle \nabla G_{\mathcal{S}^\perp}(\boldsymbol{v}^{k-1}), \boldsymbol{\xi}_{\boldsymbol{v}}^k \big\rangle$ by concentration. For all $k$ from 1 to $K_0$, note

$$\mathbb{E}\big[\eta \big\langle \nabla G_{\mathcal{S}^\perp}(\boldsymbol{v}^{k-1}), \boldsymbol{\xi}_{\boldsymbol{v}}^k \big\rangle 1_{k \leq \mathcal{K}} | \mathfrak{F}_{k-1}\big] = 0,$$

because $1_{k \leq \mathcal{K}} \equiv 1_{k-1 \leq \mathcal{K}}$, so all terms in $\eta \big\langle \nabla G_{\mathcal{S}^\perp}(\boldsymbol{v}^{k-1}), \boldsymbol{\xi}_{\boldsymbol{v}}^k \big\rangle 1_{k \leq \mathcal{K}}$ except $\boldsymbol{\xi}_{\boldsymbol{v}}^k$ are $\mathfrak{F}^{k-1}$-measurable.

Furthermore, by Lemma E.23 and Lemma E.18, for all $k \leq \mathcal{K}$, we have

$$\left\| \eta \big\langle \nabla G_{\mathcal{S}^\perp}(\boldsymbol{v}^{k-1}), \boldsymbol{\xi}_{\boldsymbol{v}}^k \big\rangle 1_{k \leq \mathcal{K}} \right\| \leq \frac{11\eta \sigma_1(\boldsymbol{w}_0)^2}{2},$$

and

$$\mathbb{E}\left[\left\{\eta\langle\nabla G_{\mathcal{S}^\perp}(\boldsymbol{v}^{k-1}),\boldsymbol{\xi}_{\boldsymbol{v}}^k\rangle 1_{k\le\mathcal{K}}\right\}^2\Big|\mathfrak{F}^{k-1}\right]\le\eta^2\sigma_1(\boldsymbol{w}_0)^2 1_{k\le K}\left\|\nabla G_{\mathcal{S}^\perp}(\boldsymbol{v}^k)\right\|^2.$$

Taking $\delta=\frac{p}{3\log(K_0)}$ in the Data-Dependent Bernstein Inequality Theorem C.2, we obtain with probability at least $1-\frac{p}{3}$,

$$\sum_{k=1}^{\mathcal{K}}-\eta\langle\nabla G_{\mathcal{S}^\perp}(\boldsymbol{v}^{k-1}),\boldsymbol{\xi}_{\boldsymbol{v}}^k\rangle$$

$$=\sum_{k=1}^{K_0}-\eta\langle\nabla G_{\mathcal{S}^\perp}(\boldsymbol{v}^{k-1}),\boldsymbol{\xi}_{\boldsymbol{v}}^k\rangle 1_{k\le\mathcal{K}}$$

$$\le\max\left\{11\eta\sigma_1(\boldsymbol{w}_0)^2\log\left(\frac{3\log(K_0)}{p}\right),4\sqrt{\eta^2\sigma_1(\boldsymbol{w}_0)^2\sum_{k=0}^{\mathcal{K}-1}\left\|\nabla G_{\mathcal{S}^\perp}(\boldsymbol{v}^k)\right\|^2\log\left(\frac{3\log(K_0)}{p}\right)}\right\}.$$
(92)

We upper bound each of these terms in the maximum. With our choice of parameters and one application of AM-GM, we have

$$11\eta\sigma_1(\boldsymbol{w}_0)^2\log\left(\frac{3\log(K_0)}{p}\right)\le\frac{B^2}{100\eta K_0},$$

and

$$4\sqrt{\eta^2\sigma^2\sum_{k=0}^{K-1}\left\|\nabla G_{\mathcal{S}^\perp}(\boldsymbol{v}^k)\right\|^2\log\left(\frac{3\log(K_0)}{p}\right)}\le32\log\left(\frac{3\log(K_0)}{p}\right)\eta\sigma_1(\boldsymbol{w}_0)^2+\frac{\eta}{8}\sum_{k=0}^{\mathcal{K}-1}\left\|\nabla G_{\mathcal{S}^\perp}(\boldsymbol{v}^k)\right\|^2$$

$$\le\frac{B^2}{32\eta K_0}+\frac{\eta}{8}\sum_{k=0}^{\mathcal{K}-1}\left\|\nabla G_{\mathcal{S}^\perp}(\boldsymbol{v}^k)\right\|^2.$$

Consequently the second upper bound dominates the maximum from (92). Substituting the above into (92), with probability at least $1-\frac{p}{3}$, we obtain

$$\sum_{k=1}^{\mathcal{K}}-\eta\langle\nabla G_{\mathcal{S}^\perp}(\boldsymbol{v}^{k-1}),\boldsymbol{\xi}_{\boldsymbol{v}}^k\rangle\le\frac{B^2}{32\eta K_0}+\frac{\eta}{8}\sum_{k=0}^{\mathcal{K}-1}\left\|\nabla G_{\mathcal{S}^\perp}(\boldsymbol{v}^k)\right\|^2.$$

Combining with (91), we obtain with probability at least $1-\frac{7p}{12}$ that

$$F(\boldsymbol{x}^{\mathcal{K}})-F(\boldsymbol{x}^0)\le\left(\frac{3}{256}+\frac{1}{80}+\frac{1}{32}\right)\frac{B^2}{\eta K_0}-\frac{3\eta}{4}\sum_{k=0}^{\mathcal{K}-1}\left\|\nabla G_{\mathcal{S}^\perp}(\boldsymbol{v}^k)\right\|^2-\frac{3\eta}{4}\sum_{k=0}^{\mathcal{K}-1}\left\|\nabla G_{\mathcal{S}}(\boldsymbol{y}^k)\right\|^2$$

Taking a Union Bound with the event from Lemma E.25, we obtain with probability at least $1-\frac{3}{4}p$, if $\boldsymbol{x}^k$ moves out of the ball $\boldsymbol{B}(\boldsymbol{x}^0,B)$ within $K_0$ iterations (i.e. $\mathcal{K}=\mathcal{K}_0\le K_0$), then

$$F(\boldsymbol{x}^{\mathcal{K}_0})-F(\boldsymbol{x}^0)=F(\boldsymbol{x}^{\mathcal{K}})-F(\boldsymbol{x}^0)\le-\left(\frac{3}{4}\cdot\frac{169}{512}-\frac{3}{256}-\frac{1}{80}-\frac{1}{32}\right)\frac{B^2}{\eta K_0}<-\frac{B^2}{7\eta K_0}.$$

This proves Lemma E.2 in Case 2.

Combining Case 1 and Case 2, we obtain Lemma E.2. $\qquad\square$

### E.6 Finding Second Order Stationary Points

Here, we finish the proof by showing with high probability, if the algorithm does not escape $\mathbb{B}(\boldsymbol{x}^0,B)$ in $K_0$ iterates, then the average of the $K_0$ iterates is a SOSP. In particular, we aim to prove Lemma E.1. Here is where Lemma E.12 is used. In the following, we define $\boldsymbol{\xi}^k$ as in (71). Furthermore, note the proofs of Lemma E.17 and Lemma E.18 still go through under the conditions of Lemma E.1, so we may apply those Lemmas in our proof here.

**Proof.** We adopt the proof strategy of Fang et al. (2019) in a similar way as we have thus far.

- By Lemma E.12, with probability $1 - \frac{p}{3}$ (namely if the event (66) from Lemma E.12 occurs), then if $\lambda_{\mathrm{MIN}}(\nabla^2 F(\overline{\boldsymbol{x}})) \le -\delta_2$, $\boldsymbol{x}^k$ will move out of the ball $\mathbb{B}(\boldsymbol{x}^0, B)$ within $K_0$ iterations. By taking the contrapositive, we see that with probability $1 - \frac{p}{3}$, if $\boldsymbol{x}^k$ does not move out of the ball $\mathbb{B}(\boldsymbol{x}^0, B)$ in $K_0$ iterations, then $\lambda_{\mathrm{MIN}}(\nabla^2 F(\boldsymbol{x}^0)) \ge -\delta_2$. In this case, we have $\boldsymbol{x}^k \in \mathbb{B}(\boldsymbol{x}^0, B)$ for all $1 \le k \le K_0$, so $\overline{\boldsymbol{x}} \in \mathbb{B}(\boldsymbol{x}^0, B)$. Thus by Lemma E.8 and as $\boldsymbol{x}^0 \in \mathcal{L}_{F, F(\boldsymbol{w}_0)}$,

$$\lambda_{\mathrm{MIN}}(\nabla^2 F(\overline{\boldsymbol{x}})) \ge \lambda_{\mathrm{MIN}}(\nabla^2 F(\boldsymbol{x}^0)) - L_2(\boldsymbol{w}_0)\|\overline{\boldsymbol{x}} - \boldsymbol{x}^0\| \ge -\delta_2 - L_2(\boldsymbol{w}_0)B \ge -17\delta,$$

where the final inequality follows from our choice of parameters. That is, with probability $1 - \frac{p}{3}$, if $\boldsymbol{x}^k$ does not move out of the ball $\mathbb{B}(\boldsymbol{x}^0, B)$ in $K_0$ iterations, then $\lambda_{\mathrm{MIN}}(\nabla^2 F(\overline{\boldsymbol{x}})) \ge -17\delta$.

- To complete the proof and show $\overline{\boldsymbol{x}}$ is a SOSP, we will show that $\|\nabla F(\overline{\boldsymbol{x}})\|$ is small. To this end, we upper bound $\frac{1}{K_0}\|\sum_{k=1}^{K_0} \boldsymbol{\xi}^k\|$ using concentration. In deriving this bound we do *not* yet suppose that $\boldsymbol{x}^k$ does not move out of $\mathbb{B}(\boldsymbol{x}^0, B)$ in its first $K_0$ iterations. Consider

$$\left\|\sum_{k=1}^{K_0} \boldsymbol{\xi}^k 1_{k \le \mathcal{K}_0}\right\| = \left\|\sum_{k=1}^{K_0} \boldsymbol{\xi}^k 1_{k-1 < \mathcal{K}_0}\right\|.$$

As $1_{k-1 < \mathcal{K}_0}$ is $\mathfrak{F}^{k-1}$-measurable,

$$\mathbb{E}\left[\boldsymbol{\xi}^k 1_{k \le \mathcal{K}_0} | \mathfrak{F}^{k-1}\right] = \boldsymbol{0}.$$

Furthermore by Lemma E.18, for $k \le \mathcal{K}_0$ we have

$$\left\|\boldsymbol{\xi}^k 1_{k \le \mathcal{K}_0}\right\| \le \sigma_1(\boldsymbol{w}_0).$$

Thus the Vector-Martingale Concentration Inequality Theorem C.1 gives with probability at least $1 - 2p/3$ that

$$\frac{1}{K_0}\left\|\sum_{k=1}^{K_0} \boldsymbol{\xi}^k 1_{k \le \mathcal{K}_0}\right\| \le \frac{2\sigma_1(\boldsymbol{w}_0)\sqrt{K_0 \log(6/p)}}{K_0} \le L_2(\boldsymbol{w}_0)B^2. \tag{93}$$

The last inequality follows from our choice of parameters.

Now conditioning on the above event implying (93) which occurs with probability at least $1 - 2p/3$, suppose $\boldsymbol{x}^k$ does not move out of the ball $\mathbb{B}(\boldsymbol{x}^0, B)$ in $K_0$ iterations. Then we have $\mathcal{K}_0 > K_0$, and so from (93), we have

$$\frac{1}{K_0}\left\|\sum_{k=1}^{K_0} \boldsymbol{\xi}^k\right\| = \frac{1}{K_0}\left\|\sum_{k=1}^{K_0} \boldsymbol{\xi}^k 1_{k \le \mathcal{K}_0}\right\| \le L_2(\boldsymbol{w}_0)B^2.$$

Furthermore, if $\boldsymbol{x}^k$ does not move out of the ball $\mathbb{B}(\boldsymbol{x}^0, B)$ in $K_0$ iterations, then we have $\overline{\boldsymbol{x}} \in \mathbb{B}(\boldsymbol{x}^0, B)$. We find an upper bound $\|\nabla F(\overline{\boldsymbol{x}})\|^2$. We again consider the quadratic approximation $G(\boldsymbol{x})$ at $\boldsymbol{x}^0$ defined in Subsection E.5, and follow the notation from there. Noting $G(\cdot)$ is a quadratic and so its gradient is a linear map, we obtain

$$
\begin{aligned}
\|G(\overline{\boldsymbol{x}})\| &= \left\|\frac{1}{K_0}\sum_{k=0}^{K_0-1} \nabla G(\boldsymbol{x}^k)\right\| \\
&\le \left\|\frac{1}{K_0}\sum_{k=0}^{K_0-1} \nabla F(\boldsymbol{x}^k)\right\| + \left\|\frac{1}{K_0}\sum_{k=0}^{K_0-1} \nabla G(\boldsymbol{x}^k) - \nabla F(\boldsymbol{x}^k)\right\| \\
&= \frac{1}{K_0\eta}\left\|\boldsymbol{x}^{K_0-1} - \boldsymbol{x}^0 - \eta \sum_{k=1}^{K_0} \boldsymbol{\xi}^k\right\| + \left\|\frac{1}{K_0}\sum_{k=0}^{K_0-1} \nabla G(\boldsymbol{x}^k) - \nabla F(\boldsymbol{x}^k)\right\| \\
&\le \frac{B}{K_0\eta} + \frac{1}{K_0}\left\|\sum_{k=1}^{K_0} \boldsymbol{\xi}^k\right\| + \frac{1}{K_0} \cdot K_0 \cdot \frac{L_2(\boldsymbol{w}_0)B^2}{2} \\
&\le \left(\frac{16}{\widetilde{C}_1} + \frac{1}{2}\right)L_2(\boldsymbol{w}_0)B^2 + \frac{1}{K_0}\left\|\sum_{k=1}^{K_0} \boldsymbol{\xi}^k\right\|.
\end{aligned}
$$

Here we used the choice of parameters, that $\boldsymbol{x}^k \in \mathbb{B}(\boldsymbol{x}^0, B)$ for all $0 \le k \le K_0$ combined with Lemma E.17 and that $\boldsymbol{x}^0 \in \mathcal{L}_{F,F(\boldsymbol{w}_0)}$, and Triangle Inequality repeatedly.

Note because $\boldsymbol{x}^0 \in \mathcal{L}_{F,F(\boldsymbol{w}_0)}$ and as $\overline{\boldsymbol{x}} \in \mathbb{B}(\boldsymbol{x}^0, B)$, by Lemma E.17, the above implies

$$\|\nabla F(\overline{\boldsymbol{x}})\| \le \|\nabla G(\overline{\boldsymbol{x}})\| + \frac{L_2(\boldsymbol{w}_0)B^2}{2} \le 17L_2(\boldsymbol{w}_0)B^2 + \frac{1}{K_0}\left\|\sum_{k=1}^{K_0} \boldsymbol{\xi}^k\right\| \le 18L_2(\boldsymbol{w}_0)B^2.$$

Consequently, with probability at least $1 - 2p/3$, if $\boldsymbol{x}^k$ does not move out of the ball $\mathbb{B}(\boldsymbol{x}^0, B)$ within $K_0$ iterations, then
$$\|\nabla F(\overline{\boldsymbol{x}})\| \le 18L_2(\boldsymbol{w}_0)B^2.$$

Taking a Union Bound, it follows that with probability at least $1 - p$, if $\boldsymbol{x}^k$ does not escape $\mathbb{B}(\boldsymbol{x}^0, B)$ within the first $K_0$ iterations, we have both

$$\|\nabla F(\overline{\boldsymbol{x}})\| \le 18L_2(\boldsymbol{w}_0)B^2, \lambda_{\mathrm{MIN}}(\nabla^2 F(\overline{\boldsymbol{x}})) \ge -17\delta.$$

This proves Lemma E.1. $\qquad\qquad\qquad\qquad\qquad\qquad\qquad\qquad\qquad\qquad\qquad\square$

# F  Examples

## F.1  Phase Retrieval

By Theorem 3.4 and Theorem 3.5, it suffices to show that 1) $F_{\mathrm{pr}}$ satisfies Assumption 1.2 and 2) $F_{\mathrm{pr}}$ is a strict saddle problem (that is, all SOSPs are near-optima in a suitable sense). In the rest of this subsection, denote $F_{\mathrm{pr}}$ by $F$ for short. As shown in Candes et al. (2015); De Sa et al. (2022), Section 2.3 and Lemma 16 part a respectively, direct calculation shows $F(\boldsymbol{w})$ takes the form

$$F(\boldsymbol{w}) = \boldsymbol{w}^\top(\boldsymbol{I} - (\boldsymbol{w}^\star)(\boldsymbol{w}^\star)^\top)\boldsymbol{w} + \frac{3}{4}(\|\boldsymbol{w}\|^2 - 1)^2. \tag{94}$$

As $\|\boldsymbol{w}^\star\| = 1$, we have $F(\boldsymbol{w}) \ge 0$. Furthermore, we have $\inf_{\boldsymbol{w} \in \mathbb{R}^d} F(\boldsymbol{w}) = 0$, attained for example at $\boldsymbol{w} = \pm\boldsymbol{w}^\star$. Also note for any fixed $\boldsymbol{w}$, $F$ is absolutely continuous on a compact neighborhood of $\boldsymbol{w}$.

$F$ **satisfies Assumption 1.2:**  By De Sa et al. (2022), Lemma 20, we have that

$$\left\|\nabla^2 F(\boldsymbol{w})\right\| \le \rho_1(F(\boldsymbol{w}))$$

for $\rho_1(x) = 9\sqrt{x} + 10$. It remains to show that

$$\left\|\nabla^3 F(\boldsymbol{w})\right\| \le \rho_2(F(\boldsymbol{w}))$$

for some increasing, non-negative $\rho_2$, where $\left\|\nabla^3 F(\boldsymbol{w})\right\|$ refers to operator norm of the third order tensor. Equivalently, we will show that for any $\boldsymbol{w}$ and any unit vector $\boldsymbol{u}$, we have

$$\lim_{\delta \to 0} \frac{\left\|\nabla^2 F(\boldsymbol{w} + \delta\boldsymbol{u}) - \nabla^2 F(\boldsymbol{w})\right\|_{\mathrm{op}}}{\delta\|\boldsymbol{u}\|} \le \rho_2(F(\boldsymbol{w})).$$

As shown in the proof of Lemma 20, De Sa et al. (2022), we obtain from direct calculation that

$$\nabla^2 F(\boldsymbol{w}) = 2\boldsymbol{I} - 2(\boldsymbol{w}^\star)(\boldsymbol{w}^\star)^\top + 3(\|\boldsymbol{w}\|^2 - 1)\boldsymbol{I} + 6\boldsymbol{w}\boldsymbol{w}^\top. \tag{95}$$

Thus, by repeatedly applying Triangle Inequality and Lemma A.3 and as $\|\boldsymbol{u}\| = 1$,

$$\left\|\nabla^2 F(\boldsymbol{w} + \delta\boldsymbol{u}) - \nabla^2 F(\boldsymbol{w})\right\|_{\mathrm{op}}$$
$$= \left\|3(\|\boldsymbol{w} + \delta\boldsymbol{u}\|^2 - \|\boldsymbol{w}\|^2)\boldsymbol{I} + 6(\boldsymbol{w} + \delta\boldsymbol{u})(\boldsymbol{w} + \delta\boldsymbol{u})^\top - 6\boldsymbol{w}\boldsymbol{w}^\top\right\|_{\mathrm{op}}$$
$$\le 3|\|\boldsymbol{w} + \delta\boldsymbol{u}\| - \|\boldsymbol{w}\|| \cdot (\|\boldsymbol{w} + \delta\boldsymbol{u}\| + \|\boldsymbol{w}\|)$$
$$\qquad\qquad + 6\left\|(\boldsymbol{w} + \delta\boldsymbol{u})(\boldsymbol{w} + \delta\boldsymbol{u})^\top - \boldsymbol{w}(\boldsymbol{w} + \delta\boldsymbol{u})^\top + \boldsymbol{w}(\boldsymbol{w} + \delta\boldsymbol{u})^\top - \boldsymbol{w}\boldsymbol{w}^\top\right\|_{\mathrm{op}}$$
$$\le 3\delta\|\boldsymbol{u}\|(2\|\boldsymbol{w}\| + \delta) + 6\left(\left\|\delta\boldsymbol{u}(\boldsymbol{w} + \delta\boldsymbol{u})^\top\right\|_{\mathrm{op}} + \left\|\boldsymbol{w}(\delta\boldsymbol{u})^\top\right\|_{\mathrm{op}}\right)$$

$$\leq \delta\|\boldsymbol{u}\|(3(2\|\boldsymbol{w}\| + \delta) + 6\|\boldsymbol{w} + \delta\boldsymbol{u}\| + 6\|\boldsymbol{w}\|)$$
$$\leq \delta\|\boldsymbol{u}\|(18\|\boldsymbol{w}\| + 9\delta).$$

Here, we used the inequality $|\|\boldsymbol{x} + \boldsymbol{y}\| - \|\boldsymbol{x}\|| \leq \|\boldsymbol{y}\|$.

Consequently,

$$\lim_{\delta \to 0} \frac{\|\nabla^2 F(\boldsymbol{w} + \delta\boldsymbol{u}) - \nabla^2 F(\boldsymbol{w})\|_{\text{op}}}{\delta\|\boldsymbol{u}\|} \leq \lim_{\delta \to 0} 18\|\boldsymbol{w}\| + 9\delta \leq 18\|\boldsymbol{w}\| + 1.$$

By Lemma 16 part d, De Sa et al. (2022), using Jensen's Inequality we have

$$F(\boldsymbol{w}) \geq (\|\boldsymbol{w}\|^2 - 1)^2.$$

Note for $\|\boldsymbol{w}\| \geq 2$, this implies

$$18\|\boldsymbol{w}\| + 1 \leq 18(\|\boldsymbol{w}\| + 1)^2(\|\boldsymbol{w}\| - 1)^2 \leq 18F(\boldsymbol{w}).$$

Combining with the case $\|\boldsymbol{w}\| < 2$, we obtain

$$\lim_{\delta \to 0} \frac{\|\nabla^2 F(\boldsymbol{w} + \delta\boldsymbol{u}) - \nabla^2 F(\boldsymbol{w})\|_{\text{op}}}{\delta\|\boldsymbol{u}\|} \leq 18\|\boldsymbol{w}\| + 1 \leq 18F(\boldsymbol{w}) + 37,$$

so we can just take $\rho_2(x) = 18x + 37$.

**Next, we check that $F$ is a strict saddle problem:** We check this here. Similar results, in slightly different of a setting where we solve phase retrieval from samples from data, are shown in Sun et al. (2018).

Suppose $\|\nabla F(\boldsymbol{w})\| \leq \delta$ for $\delta \leq (\frac{1}{20})^4$. Note by Lemma 16 part b, De Sa et al. (2022), $\langle \boldsymbol{w}^\star, \nabla F(\boldsymbol{w}) \rangle = 3(\|\boldsymbol{w}\|^2 - 1)\langle \boldsymbol{w}, \boldsymbol{w}^\star \rangle$. By Cauchy-Schwartz and recalling $\boldsymbol{w}^\star$ is a unit vector, this gives

$$\delta \geq \|\boldsymbol{w}^\star\|\|\nabla F(\boldsymbol{w})\| \geq |\langle \boldsymbol{w}^\star, \nabla F(\boldsymbol{w}) \rangle| = 3|\|\boldsymbol{w}\|^2 - 1| \cdot |\langle \boldsymbol{w}, \boldsymbol{w}^\star \rangle|. \tag{96}$$

- Suppose $|\langle \boldsymbol{w}, \boldsymbol{w}^\star \rangle| \geq \sqrt{\delta}$. Combining this with (96) gives

$$\left|\|\boldsymbol{w}\|^2 - 1\right| \leq \frac{\sqrt{\delta}}{3}.$$

By Lemma 16 part c, De Sa et al. (2022),

$$\|\nabla F(\boldsymbol{w})\|^2 = 12\|\boldsymbol{w}\|^2 F(\boldsymbol{w}) - 8(\|\boldsymbol{w}\|^2 - \langle \boldsymbol{w}, \boldsymbol{w}^\star \rangle^2)$$
$$= (12\|\boldsymbol{w}\|^2 - 8)F(\boldsymbol{w}) + 6(\|\boldsymbol{w}\|^2 - 1)^2,$$

where the last equality follows from the explicit form $F(\boldsymbol{w})$ from (94). Thus using $\left|\|\boldsymbol{w}\|^2 - 1\right| \leq \frac{\sqrt{\delta}}{3}$, we obtain

$$\delta^2 \geq \|\nabla F(\boldsymbol{w})\|^2 = (12\|\boldsymbol{w}\|^2 - 8)F(\boldsymbol{w}) + 6(\|\boldsymbol{w}\|^2 - 1)^2 \geq (4 - 4\sqrt{\delta})F(\boldsymbol{w}).$$

For $\delta \leq \frac{1}{4}$, this gives

$$F(\boldsymbol{w}) \leq \frac{\delta^2}{4 - 4\sqrt{\delta}} \leq \frac{\delta^2}{2}.$$

- Otherwise, suppose $|\langle \boldsymbol{w}, \boldsymbol{w}^\star \rangle| \leq \sqrt{\delta}$. Note by differentiating (94), as shown in the proof of Lemma 16 part b, De Sa et al. (2022),

$$\nabla F(\boldsymbol{w}) = 2\boldsymbol{w} - 2\langle \boldsymbol{w}, \boldsymbol{w}^\star \rangle\boldsymbol{w}^\star + 3(\|\boldsymbol{w}\|^2 - 1)\boldsymbol{w} = -2\langle \boldsymbol{w}, \boldsymbol{w}^\star \rangle\boldsymbol{w}^\star + (3\|\boldsymbol{w}\|^2 - 1)\boldsymbol{w}.$$

Thus by Triangle Inequality,

$$\left|3\|\boldsymbol{w}\|^2 - 1\right| \cdot \|\boldsymbol{w}\| \leq \|\nabla F(\boldsymbol{w})\| + 2|\langle \boldsymbol{w}, \boldsymbol{w}^\star \rangle|\|\boldsymbol{w}^\star\| \leq \delta + 2\sqrt{\delta} \leq 4\sqrt{\delta}.$$

Consequently either $\|\boldsymbol{w}\| \leq 2\delta^{1/4}$ or $\left|3\|\boldsymbol{w}\|^2 - 1\right| \leq 2\delta^{1/4}$.

In the first case, by Cauchy Schwartz and (95), notice for any unit vector $\boldsymbol{u}$ that

$$\boldsymbol{u}^\top \nabla^2 F(\boldsymbol{w})\boldsymbol{u} = \boldsymbol{u}^\top \Big(2\boldsymbol{I} - 2(\boldsymbol{w}^\star)(\boldsymbol{w}^\star)^\top + 3(\|\boldsymbol{w}\|^2 - 1)\boldsymbol{I} + 6\boldsymbol{w}\boldsymbol{w}^T\Big)]\boldsymbol{u}$$

$$\leq -\|\boldsymbol{u}\|^2 + 3\|\boldsymbol{u}\|^2 \cdot (2\delta^{1/4})^2 + 6\|\boldsymbol{u}\|^2 \cdot (2\delta^{1/4})^2$$

$$\leq -1 + 36\delta^{1/2} \leq -\frac{9}{10},$$

since $\delta \leq \left(\frac{1}{20}\right)^4$.

In the second case, using (95), notice as $\|\boldsymbol{w}^\star\| = 1$, we have

$$\boldsymbol{w}^{\star\top} \nabla^2 F(\boldsymbol{w})\boldsymbol{w}^\star = \boldsymbol{w}^{\star\top}(3\|\boldsymbol{w}\|^2 - 1)\boldsymbol{w}^\star - 2\|\boldsymbol{w}^\star\|^2 + 6|\langle \boldsymbol{w}, \boldsymbol{w}^\star \rangle|^2$$

$$\leq 2\delta^{1/4} - 2 + 6\delta \leq -\frac{9}{5}.$$

Consequently in either case, $\nabla^2 F(\boldsymbol{w})$ has at least one negative eigenvalue with value at most $-\frac{9}{10}$.

Consider $\varepsilon$ smaller than a universal constant, and take $\delta = \sqrt{\varepsilon}$ in the above result. It follows from the analysis here that if we find an SOSP to tolerance $\varepsilon$ as per the definition (2), we obtain $\boldsymbol{w}$ with $F(\boldsymbol{w}) \leq \frac{\varepsilon}{2}$.

Thus, it follows that running Perturbed GD or Restarted SGD as described in Theorem 3.4 or Theorem 3.5 respectively, we will obtain $\boldsymbol{w}$ with suboptimality $F(\boldsymbol{w}) \leq \varepsilon$, where the number of oracle calls depends on $1/\varepsilon, d, F(\boldsymbol{w}_0)$ in the same way as in Theorem 3.4 or Theorem 3.5 respectively.

## F.2 Matrix PCA

Again by Theorem 3.4, Theorem 3.5, it suffices to show that 1) $F_{\text{pca}}$ satisfies Assumption 1.2 and 2) is a strict saddle problem (that is, all SOSPs are near-optima in a suitable sense). We will show this, with the parameters governing the strict saddle property depending on the spectral gap $\lambda_1(\boldsymbol{M}) - \lambda_2(\boldsymbol{M})$.[12] In the rest of this subsection, denote $F_{\text{pca}}$ by $F$ for short. Recall the loss function for PCA takes the form

$$F(\boldsymbol{w}) = \frac{1}{2}\big\|\boldsymbol{w}\boldsymbol{w}^\top - \boldsymbol{M}\big\|_F^2,$$

where $\boldsymbol{M}$ is a symmetric PD matrix. Note for any fixed $\boldsymbol{w}$, $F$ is absolutely continuous on a compact neighborhood of $\boldsymbol{w}$. Note $F(\boldsymbol{w}) \geq 0$ always holds. While it is not true that $\inf_{\boldsymbol{w} \in \mathbb{R}^d} F(\boldsymbol{w}) = 0$, to enforce this, we can consider the shifted function $G := F - \inf_{\boldsymbol{w} \in \mathbb{R}^d} F(\boldsymbol{w})$. The derivatives of $G$ are identical to those of $F$, and furthermore $G(\boldsymbol{x}) - G(\boldsymbol{y}) = F(\boldsymbol{x}) - F(\boldsymbol{y})$ for all $\boldsymbol{x}, \boldsymbol{y}$. Thus to apply Theorem 3.4, Theorem 3.5 and show that Perturbed GD or Restarted SGD can globally optimize $G$ and therefore $F$ by finding SOSPs, it remains to show $F$ satisfies Assumption 1.2 and is strict saddle.

$F$ **satisfies Assumption 1.2:** Direct calculation, also in Jin et al. (2021a), yields

$$\nabla F(\boldsymbol{w}) = (\boldsymbol{w}\boldsymbol{w}^\top - \boldsymbol{M})\boldsymbol{w}, \nabla^2 F(\boldsymbol{w}) = \|\boldsymbol{w}\|^2 \boldsymbol{I} + 2\boldsymbol{w}\boldsymbol{w}^\top - \boldsymbol{M}. \tag{97}$$

We now check self-bounding regularity for the Hessian and third order derivative tensor. First observe

$$\boldsymbol{w}^\top(\boldsymbol{w}\boldsymbol{w}^\top)\boldsymbol{w} = \|\boldsymbol{w}\|^4.$$

Combining with Lemma A.3, we obtain

$$\|\boldsymbol{w}\| = \big\|\boldsymbol{w}\boldsymbol{w}^\top\big\|_{\text{op}}^{1/2}$$

$$\leq \Big(\big\|\boldsymbol{w}\boldsymbol{w}^\top - \boldsymbol{M}\big\|_{\text{op}} + \|\boldsymbol{M}\|_{\text{op}}\Big)^{1/2}$$

$$\leq \big\|\boldsymbol{w}\boldsymbol{w}^\top - \boldsymbol{M}\big\|_F^{1/2} + \|\boldsymbol{M}\|_{\text{op}}^{1/2}$$

$$\leq 2F(\boldsymbol{w})^{1/4} + \|\boldsymbol{M}\|_{\text{op}}^{1/2}. \tag{98}$$

---

[12]Thus our result will be vacuous when the spectral gap is 0.

Now we check the self bounding conditions. For the Hessian, note from (97) and (98) and using Lemma A.3,

$$\left\|\nabla^2 F(\boldsymbol{w})\right\|_{\mathrm{op}} \le 3\|\boldsymbol{w}\|^2 + \|\boldsymbol{M}\|_{\mathrm{op}} \le 3(2F(\boldsymbol{w})^{1/4} + \|\boldsymbol{M}\|_{\mathrm{op}}^{1/2})^2 + \|\boldsymbol{M}\|_{\mathrm{op}}.$$

Thus we can take $\rho_1(x) = 3(2x^{1/4} + \|\boldsymbol{M}\|_{\mathrm{op}}^{1/2})^2 + \|\boldsymbol{M}\|_{\mathrm{op}}$.

For the third order derivative tensor, following the strategy in Subsection F.1, we will show that for any $\boldsymbol{w}$ and any unit vector $\boldsymbol{u}$, we have

$$\lim_{\delta \to 0} \frac{\left\|\nabla^2 F(\boldsymbol{w} + \delta \boldsymbol{u}) - \nabla^2 F(\boldsymbol{w})\right\|_{\mathrm{op}}}{\delta \|\boldsymbol{u}\|} \le \rho_3(F(\boldsymbol{w})).$$

Applying (97) and Lemma A.3 and note

$$(\boldsymbol{w} + \delta \boldsymbol{u})(\boldsymbol{w} + \delta \boldsymbol{u})^\top - \boldsymbol{w}\boldsymbol{w}^\top = (\boldsymbol{w} + \delta \boldsymbol{u})(\boldsymbol{w} + \delta \boldsymbol{u})^\top - (\boldsymbol{w} + \delta \boldsymbol{u})\boldsymbol{w}^\top + (\boldsymbol{w} + \delta \boldsymbol{u})\boldsymbol{w}^\top - \boldsymbol{w}\boldsymbol{w}^\top$$
$$= (\boldsymbol{w} + \delta \boldsymbol{u})(\delta \boldsymbol{u})^\top + \delta \boldsymbol{u}\boldsymbol{w}^\top.$$

This gives

$$\lim_{\delta \to 0} \frac{\left\|\nabla^2 F(\boldsymbol{w} + \delta \boldsymbol{u}) - \nabla^2 F(\boldsymbol{w})\right\|_{\mathrm{op}}}{\delta \|\boldsymbol{u}\|}$$
$$= \lim_{\delta \to 0} \frac{(\|\boldsymbol{w} + \delta \boldsymbol{u}\|^2 - \|\boldsymbol{w}\|^2) + 2\|(\boldsymbol{w} + \delta \boldsymbol{u})(\boldsymbol{w} + \delta \boldsymbol{u})^\top - \boldsymbol{w}\boldsymbol{w}^\top\|_{\mathrm{op}}}{\delta \|\boldsymbol{u}\|}$$
$$\le \lim_{\delta \to 0} \frac{\|\|\boldsymbol{w} + \delta \boldsymbol{u}\| - \|\boldsymbol{w}\|\| \cdot (2\|\boldsymbol{w}\| + \delta \|\boldsymbol{u}\|) + \delta \|\boldsymbol{u}\|(2\|\boldsymbol{w}\| + \delta \|\boldsymbol{u}\|)}{\delta \|\boldsymbol{u}\|}$$
$$\le \lim_{\delta \to 0} \frac{\delta \|\boldsymbol{u}\|(2\|\boldsymbol{w}\| + \delta \|\boldsymbol{u}\|) + \delta \|\boldsymbol{u}\|(2\|\boldsymbol{w}\| + \delta \|\boldsymbol{u}\|)}{\delta \|\boldsymbol{u}\|}$$
$$= \lim_{\delta \to 0} 4\|\boldsymbol{w}\| + 2\delta \|\boldsymbol{u}\|$$
$$= 4\|\boldsymbol{w}\|$$
$$\le 8F(\boldsymbol{w})^{1/4} + 4\|\boldsymbol{M}\|_{\mathrm{op}}^{1/2}.$$

Here we used the inequality $\|\|\boldsymbol{x} + \boldsymbol{y}\| - \|\boldsymbol{x}\|\| \le \|\boldsymbol{y}\|$. The last step used (98). Thus we can take $\rho_2(x) = 8x^{1/4} + 4\|\boldsymbol{M}\|_{\mathrm{op}}^{1/2}$.

**Next, we check $F$ is a strict saddle problem:** We check this here. A similar verification is done in Ge et al. (2017).

Let $\boldsymbol{v}_1, \ldots, \boldsymbol{v}_d$ be the (unit) eigenvectors of $\boldsymbol{M}$ corresponding to $\lambda_1(\boldsymbol{M}) \ge \lambda_2(\boldsymbol{M}) \ge \cdots \ge \lambda_d(\boldsymbol{M}) > 0$ respectively (recall $\boldsymbol{M}$ is assumed to be PD). Thus the $\boldsymbol{v}_i$ form an orthonormal basis of $\mathbb{R}^d$. Furthermore for convenience let $\lambda_i := \lambda_i(\boldsymbol{M})$ for all $1 \le i \le d$. As $\boldsymbol{M}$ is symmetric and PD, by the Spectral Theorem, we can write

$$\boldsymbol{M} = \sum_{i=1}^d \lambda_i \boldsymbol{v}_i \boldsymbol{v}_i^\top.$$

Suppose $\boldsymbol{w}$ is a SOSP to tolerance $\varepsilon$ for $\varepsilon < \min\left\{1, \frac{(\lambda_1 - \lambda_2)^2}{16}, \frac{3}{8}(\lambda_1 - \lambda_2)^{5/2}\right\}$. Note the minimizers of $F$ are $\boldsymbol{w} = \pm\sqrt{\lambda_1}\boldsymbol{v}_1$. We will show that $\boldsymbol{w}$ is close to these minimizers: in particular, that $\min\left\{\|\boldsymbol{w} - \sqrt{\lambda_1}\boldsymbol{v}_1\|^2, \|\boldsymbol{w} + \sqrt{\lambda_1}\boldsymbol{v}_1\|^2\right\} \le \varepsilon$.

Write $\boldsymbol{w} = c_1 \boldsymbol{v}_1 + \cdots + c_d \boldsymbol{v}_d$. Thus, our goal is to show that $|(c_1^2 + \cdots + c_d^2) - \lambda_1| < \sqrt{\varepsilon}$. By (97), we have

$$\varepsilon \ge \|\nabla F(\boldsymbol{w})\| = \left\|\boldsymbol{M}\boldsymbol{w} - \|\boldsymbol{w}\|^2 \boldsymbol{w}\right\| = \left\|\sum_{i=1}^d \left((c_1^2 + \cdots + c_d^2) - \lambda_i\right)c_i \boldsymbol{v}_i\right\|.$$

That is, we have

$$\sum_{i=1}^d c_i^2 \left((c_1^2 + \cdots + c_d^2) - \lambda_i\right)^2 \le \varepsilon^2. \tag{99}$$

Furthermore by (97), we have

$$\nabla^2 F(\boldsymbol{w}) = (c_1^2 + \cdots + c_d^2)\boldsymbol{I} + 2\sum_{i,j} c_i c_j \boldsymbol{v}_i \boldsymbol{v}_j^\top - \sum_{i=1}^d \lambda_i \boldsymbol{v}_i \boldsymbol{v}_i^\top.$$

Since $\boldsymbol{w}$ is a SOSP, for all $\boldsymbol{v}_k, 1 \le k \le d$, we have

$$-\sqrt{\varepsilon} \le \boldsymbol{v}_k^\top \nabla^2 F(\boldsymbol{w})\boldsymbol{v}_k = (c_1^2 + \cdots + c_d^2) + 2c_k^2 - \lambda_k. \tag{100}$$

We now break into cases:

- Suppose for all $i$, we have $\left|(c_1^2 + \cdots + c_d^2) - \lambda_i\right| \ge \sqrt{\varepsilon}$. From (99), this gives $\sum_{i=1}^d c_i^2 \le \varepsilon$. Taking $k = 1$ in (100), we obtain

$$-\sqrt{\varepsilon} \le 3\sum_{i=1}^d c_i^2 - \lambda_1 \le 3\varepsilon - \lambda_1 \implies \lambda_1 \le \sqrt{\varepsilon} + 3\varepsilon,$$

  contradicting that $\varepsilon < \min\left\{1, \frac{(\lambda_1 - \lambda_2)^2}{16}\right\}$.

- Else, suppose there exists $i$ such that $\left|(c_1^2 + \cdots + c_d^2) - \lambda_i\right| < \sqrt{\varepsilon}$. Suppose that $i \ge 2$. Then taking $k = 1$ in (100), we obtain

$$-\sqrt{\varepsilon} \le \lambda_i + \sqrt{\varepsilon} + 2c_1^2 - \lambda_1 \implies c_1^2 \ge \frac{\lambda_1 - \lambda_i}{2} - \sqrt{\varepsilon} \ge \frac{\lambda_1 - \lambda_2}{4},$$

  where the last inequality uses $\lambda_i \le \lambda_2$ and $\varepsilon < \left(\frac{\lambda_1 - \lambda_2}{4}\right)^2$.

  Note furthermore that as $\varepsilon \le \left(\frac{\lambda_1 - \lambda_2}{4}\right)^2$, as $\left|(c_1^2 + \cdots + c_d^2) - \lambda_i\right| < \sqrt{\varepsilon}$, and as $\lambda_i \le \lambda_2 < \lambda_1$, we have $\left|(c_1^2 + \cdots + c_d^2) - \lambda_1\right| > \frac{3(\lambda_1 - \lambda_2)}{4}$. Thus (99) implies

$$\varepsilon^2 > 0 + \frac{\lambda_1 - \lambda_2}{4} \cdot \frac{9}{16}(\lambda_1 - \lambda_2)^2,$$

  contradicting that $\varepsilon < \frac{3}{8}(\lambda_1 - \lambda_2)^{5/2}$.

Therefore, we must have $i = 1$ in the second case above. That is, $\left|(c_1^2 + \cdots + c_d^2) - \lambda_1\right| < \sqrt{\varepsilon}$, as desired.

Thus, it follows that running Perturbed GD or Restarted SGD as described in Theorem 3.4 or Theorem 3.5 respectively, we will obtain $\boldsymbol{w}$ that is distance at most $\sqrt{\varepsilon}$ from a global minimizer of $F$ for $\varepsilon < \min\left\{1, \frac{(\lambda_1 - \lambda_2)^2}{16}, \frac{3}{8}(\lambda_1 - \lambda_2)^{5/2}\right\}$. Here the number of oracle calls depends on $1/\varepsilon, d, F(\boldsymbol{w}_0)$ the same way as in Theorem 3.4 or Theorem 3.5 respectively. For $\varepsilon \ge \min\left\{1, \frac{(\lambda_1 - \lambda_2)^2}{16}, \frac{3}{8}(\lambda_1 - \lambda_2)^{5/2}\right\}$, we can replace $\varepsilon$ by any real strictly smaller than $\min\left\{1, \frac{(\lambda_1 - \lambda_2)^2}{16}, \frac{3}{8}(\lambda_1 - \lambda_2)^{5/2}\right\}$ in the guarantees from Theorem 3.4 or Theorem 3.5.

## G Simulations

Our algorithmic results Theorem 3.1, Theorem 3.2, Theorem 3.3, Theorem 3.4, and Theorem 3.5 have strong practical implications. They directly suggest that under generalized smoothness, the step sizes $\eta$ that lead to convergence/successful optimization become smaller for larger initialization $F(\boldsymbol{w}_0)$ and larger self-bounding functions $\rho_1(\cdot), \rho_2(\cdot)$. For example in Theorem 3.1, we set $\eta = \frac{1}{L_1(\boldsymbol{w}_0)}$ where $L_1(\boldsymbol{w}_0) = \max\{1, \rho_0(F(\boldsymbol{w}_0) + 1), \rho_0(F(\boldsymbol{w}_0))\rho_0(F(\boldsymbol{w}_0) + 1), \rho_1(F(\boldsymbol{w}_0) + 1)\}$ was defined in (4).

That is, our work suggests that larger suboptimality at initialization and larger self-bounding functions shrink the 'window' for choosing a working $\eta$ in practice, when the loss function satisfies generalized smoothness. This has strong practical implications: it implies that for losses with non-Lipschitz gradient/Hessian, one should tune $\eta$ based on suboptimality at initialization. This contrasts sharply with the Lipschitz gradient/Hessian case, see e.g. (Bubeck et al., 2015; Jin et al., 2017; Fang et al., 2019), where the range of working $\eta$ is fixed in terms of the Lipschitz constant of the gradient and/or Hessian, and does not depend on the initialization.

In this section, we empirically validate this implication of our work.

### G.1 Synthetic Simulations with GD

**Simulation Details:** We consider $F(\boldsymbol{w}) = \|\boldsymbol{Aw}\|^p$ for $p = 2, 3, 4, 5, 6$, where $\boldsymbol{A} = \operatorname{diag}\left(\frac{1}{20}, \frac{1}{19}, \ldots, \frac{1}{2}, 1\right)$. When $p = 2$, $F(\boldsymbol{w})$ is smooth. When $p \geq 3$, $F(\boldsymbol{w})$ is not smooth, but it is straightforward to verify that it satisfies Assumption 1.1, similar to our verifications in Subsection A.2. One can furthermore verify that as $p$ increases, the corresponding self-bounding function $\rho_1(\cdot)$ from Assumption 1.1 increase. This choice of generalized smooth function was motivated by Gaash et al. (2025), who used $\|\boldsymbol{Aw}\|^4$ with the exact same $\boldsymbol{A}$ in their experiments to study optimization with first-order methods under generalized smoothness.

For each $p = 2, 3, 4, 5, 6$, we consider the following settings for GD:

- Step sizes: We consider 30 step sizes $\{\eta_i\}_{i=1}^{30}, \eta_1 < \cdots < \eta_{30}$ evenly spaced on a log scale between $10^{-8}$ and $10^1$, inclusive.

- Initialization: For each step size $\eta_i$, we initialize GD at 4 distributions $\pi_j = \mathcal{N}(\vec{\boldsymbol{0}}, c_j \boldsymbol{I}_{20})$ for $c_j \in \{2.5, 5, 7.5, 10\}$. For each of these 4 distributions $\pi_j$, we draw 100 points $\boldsymbol{w}_0 \sim \pi_j$ to use as our initialization.

- Number of steps: For each $\eta_i$ and each $\boldsymbol{w}_0 \sim \pi_j$, we run GD initialized at $\boldsymbol{w}_0$ with step size $\eta_i$ for $T = 1000$ iterations. Here as $F$ is known, we analytically compute the gradient.

For each $p$ and initialization $\pi_j$, we consider all 30 possible $\eta_i$, which we plot on the $x$-axis. For each $\eta_i$, we consider all 100 initializations $\boldsymbol{w}_0 \sim \pi_j$. For each initialization $\boldsymbol{w}_0$, letting $\{\boldsymbol{w}_t\}$ be the resulting sequence of iterates of GD, we compute $\frac{\|\nabla F(\boldsymbol{w}_T)\|}{F(\boldsymbol{w}_0)}$ for $T = 1000$. For $\eta_i$ that led to faithful convergence of GD, on the $y$-axis, we then plot the mean of $\frac{\|\nabla F(\boldsymbol{w}_T)\|}{F(\boldsymbol{w}_0)}$ over those 100 initializations as a blue dot, with blue vertical error bars indicating $\pm 2$ standard deviations. We considered the ratio $\frac{\|\nabla F(\boldsymbol{w}_T)\|}{F(\boldsymbol{w}_0)}$ because for $L$-smooth functions, established optimization theory predicts that this converges at a rate independent of $F(\boldsymbol{w}_0)$ and only depending on $T$ and $L$ (Bubeck et al., 2015).

The simulations for Subsection G.1 were run on a Jupyter notebook in Python in Google Colab Pro, connected to a single NVIDIA T4 GPU. Our code can be found in the attached files.

**Divergence of GD and working step sizes:** We observe that for some $\eta_i$ larger than some threshold depending on $p$ and $\pi_j$, the iterates of GD diverge. In particular, the resulting ratio $\frac{\|\nabla F(\boldsymbol{w}_T)\|}{F(\boldsymbol{w}_0)}$ becomes massive, often on the order of $10^5$ or more, indicating that $\eta_i$ was too large for GD to converge. To identify the smallest $\eta_i$ where this first occurs, or equivalently find the largest working step size among $\{\eta_i\}_{i=1}^{30}$, for a given $\pi_j$ and $\eta_i$, we computed the average $\frac{\|\nabla F(\boldsymbol{w}_T)\|}{F(\boldsymbol{w}_0)}$ over the 100 initializations. If this average was 100 or more times larger than this average for $\eta_{i-1}$, we took this as an indication that the iterates of GD with this step size $\eta_i$ or larger step sizes diverge, and for this $p$ and $\pi_j$, we stopped considering any larger $\eta_{i'}$, $i' > i$. We then save this $\eta_i$ to indicate the smallest $\eta_i$ for which divergence occurred. This $\eta_i$ is indicated with a red line in the following plots.

This smallest $\eta_i$ for which divergence occurred plays a crucial role in validating our theoretical claims. Established optimization theory predicts that for smooth functions (here, when $p = 2$), this $\eta_i$ is identical across different initializations (Bubeck et al., 2015). Meanwhile for generalized smooth functions, as per our remarks earlier and from Subsection 3.6, we predict that as $F(\boldsymbol{w}_0)$ increases, the range of working step sizes, and consequently also the smallest $\eta_i$ for which divergence occurs, will *decrease*. Note as $c_j$ increases (recall $\pi_j \sim \mathcal{N}(\vec{\boldsymbol{0}}, c_j \boldsymbol{I}_{20})$ and $c_j \in \{2.5, 5, 7.5, 10\}$), we expect $F(\boldsymbol{w}_0)$ to increase, at least on average or with high probability over the 100 initializations $\boldsymbol{w}_0 \sim \pi_j$.

**Results:** Our simulations validate this theory very accurately. Note in the following figures that the $y$-axis is normalized, as we plot $\frac{\|\nabla F(\boldsymbol{w}_T)\|}{F(\boldsymbol{w}_0)}$ where $T = 1000$. Thus larger $c_j$ lead to comparable values on the $y$-axis.

- When $p = 2$: In Figure 1, we plot the results in the manner described above for all 4 initializations $\pi_j$. As is predicted by established optimization theory for smooth functions (Bubeck et al., 2015), the first step size leading to divergence $\eta_i$ is identical across all the $\pi_j$.

- When $p = 3, 4, 5, 6$: We plot the results in the manner described above for all 4 initializations $\pi_j$ in Figure 2, Figure 3, Figure 4, Figure 5 respectively. Unlike the $p = 2$ case, in all of

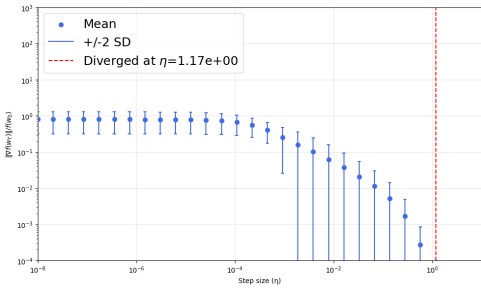
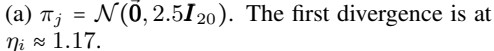

(a) $\pi_j = \mathcal{N}(\vec{0}, 2.5\boldsymbol{I}_{20})$. The first divergence is at $\eta_i \approx 1.17$.

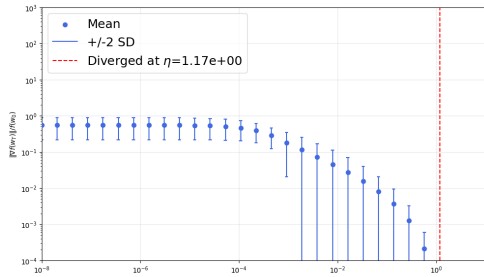

(b) $\pi_j = \mathcal{N}(\vec{0}, 5.0\boldsymbol{I}_{20})$. The first divergence is at $\eta_i \approx 1.17$.

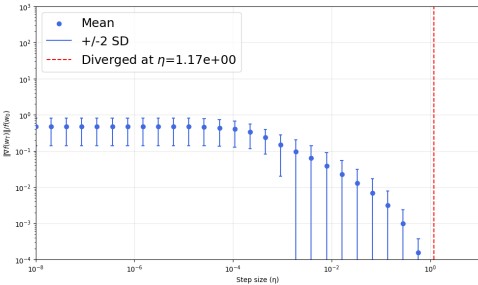

(c) $\pi_j = \mathcal{N}(\vec{0}, 7.5\boldsymbol{I}_{20})$. The first divergence is at $\eta_i \approx 1.17$.

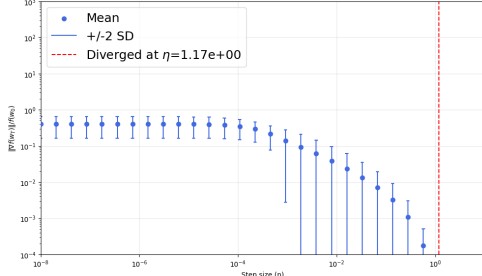

(d) $\pi_j = \mathcal{N}(\vec{0}, 10\boldsymbol{I}_{20})$. The first divergence is at $\eta_i \approx 1.17$.

Figure 1: GD simulation results for $p = 2$. For all $\pi_j$, the smallest $\eta_i$ leading to divergence is $\approx 1.17$.

these cases, the first step size leading to divergence $\eta_i$ generally decreases as the covariance $c_j\boldsymbol{I}_{20}$ of $\pi_j$ increases from 2.5 to 10.

We also notice the following, both in line with our theoretical claims:

- For a given $p$, consider how this first step size $\eta_i$ leading to divergence decreases as the covariance $c_j\boldsymbol{I}_{20}$ of $\pi_j$ increases from 2.5 to 10. We find that the rate of this decrease increases as $p$ increases. The ratio of the first $\eta_i$ leading to divergence for $\pi_1$ vs $\pi_4$ is approximately $4.18, 4.18, 8.53, 17.43$ for $p = 3, 4, 5, 6$ respectively.

  As remarked earlier, for larger $p$, the corresponding self-bounding function $\rho_1(\cdot)$ is larger for $F(\boldsymbol{w}) = \|\boldsymbol{A}\boldsymbol{w}\|^p$ (see Subsection A.2 for a similar verification). Thus this behavior is consistent with our results, as the step size from all of our results depends on $F(\boldsymbol{w}_0)$ through $\rho_1(\cdot)$.

- Fixing $\pi_j$ and comparing across $p$, we see that the first step size leading to divergence $\eta_i$ decreases as $p$ increases. Again this is not a surprise considering our theoretical results, as for larger $p$, both $F(\boldsymbol{w}_0)$ for $\boldsymbol{w}_0 \sim \pi_j$ and the self-bounding function $\rho_1(\cdot)$ become larger.

For each $p \in \{2, 3, 4, 5, 6\}$ and $\pi_j$, we also record the smallest $\eta_i$ for which divergence occurred in Table 1 on page 92, which highlights the aforementioned trends.

|  | $\pi_j = \mathcal{N}(\vec{0}, 2.5\boldsymbol{I}_{20})$ | $\pi_j = \mathcal{N}(\vec{0}, 5.0\boldsymbol{I}_{20})$ | $\pi_j = \mathcal{N}(\vec{0}, 7.5\boldsymbol{I}_{20})$ | $\pi_j = \mathcal{N}(\vec{0}, 10\boldsymbol{I}_{20})$ |
|---|---|---|---|---|
| $p = 2$ | $1.17 \cdot 10^0$ | $1.17 \cdot 10^0$ | $1.17 \cdot 10^0$ | $1.17 \cdot 10^0$ |
| $p = 3$ | $2.81 \cdot 10^{-1}$ | $1.37 \cdot 10^{-1}$ | $1.37 \cdot 10^{-1}$ | $6.72 \cdot 10^{-2}$ |
| $p = 4$ | $3.29 \cdot 10^{-2}$ | $3.29 \cdot 10^{-2}$ | $1.61 \cdot 10^{-2}$ | $7.88 \cdot 10^{-3}$ |
| $p = 5$ | $7.88 \cdot 10^{-3}$ | $3.86 \cdot 10^{-3}$ | $9.24 \cdot 10^{-4}$ | $9.24 \cdot 10^{-4}$ |
| $p = 6$ | $9.24 \cdot 10^{-4}$ | $4.52 \cdot 10^{-4}$ | $5.30 \cdot 10^{-5}$ | $5.30 \cdot 10^{-5}$ |

Table 1: The smallest $\eta_i$ leading to divergence for a given $p$ and initialization $\pi_j$.

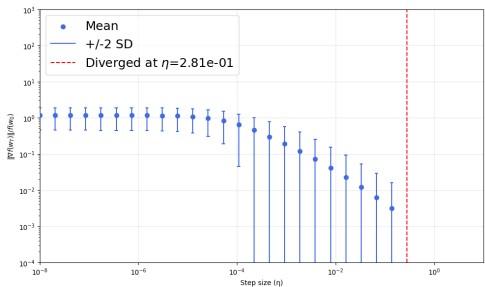

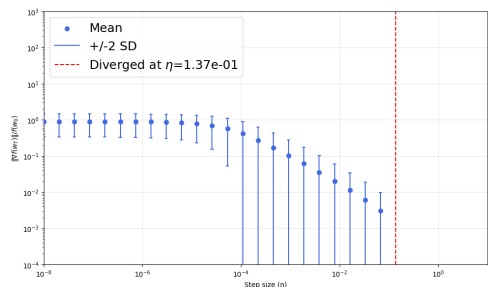

(a) $\pi_j = \mathcal{N}(\vec{0}, 2.5\boldsymbol{I}_{20})$. The first divergence is at $\eta_i \approx 0.281$.

(b) $\pi_j = \mathcal{N}(\vec{0}, 5.0\boldsymbol{I}_{20})$. The first divergence is at $\eta_i \approx 0.137$.

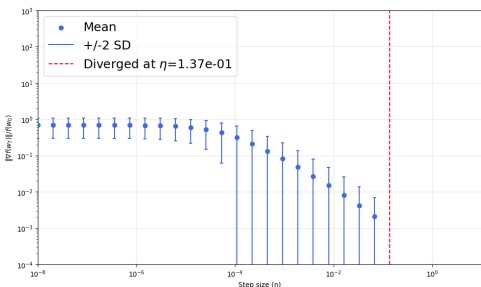

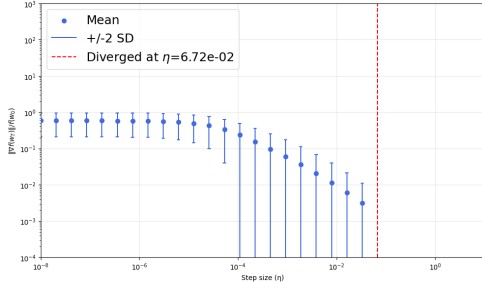

(c) $\pi_j = \mathcal{N}(\vec{0}, 7.5\boldsymbol{I}_{20})$. The first divergence is at $\eta_i \approx 0.137$.

(d) $\pi_j = \mathcal{N}(\vec{0}, 10\boldsymbol{I}_{20})$. The first divergence is at $\eta_i \approx 0.0672$.

Figure 2: GD simulation results for $p = 3$. For $\pi_j = \mathcal{N}(\vec{0}, 2.5\boldsymbol{I}_{20})$, the first divergence is at $\eta_i \approx 0.281$. For $\pi_j = \mathcal{N}(\vec{0}, 5\boldsymbol{I}_{20}), \mathcal{N}(\vec{0}, 7.5\boldsymbol{I}_{20})$, the first divergence is at $\eta_i \approx 0.137$. For $\pi_j = \mathcal{N}(\vec{0}, 10\boldsymbol{I}_{20})$, the first divergence is at $\eta_i \approx 0.0672$.

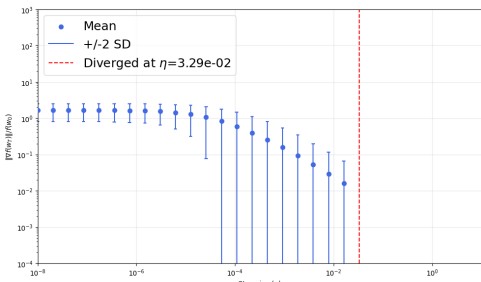

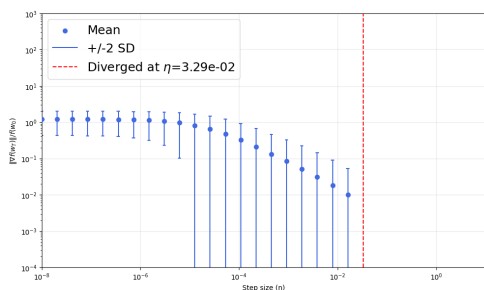

(a) $\pi_j = \mathcal{N}(\vec{0}, 2.5\boldsymbol{I}_{20})$. The first divergence is at $\eta_i \approx 3.29 \cdot 10^{-2}$.

(b) $\pi_j = \mathcal{N}(\vec{0}, 5.0\boldsymbol{I}_{20})$. The first divergence is at $\eta_i \approx 3.29 \cdot 10^{-2}$.

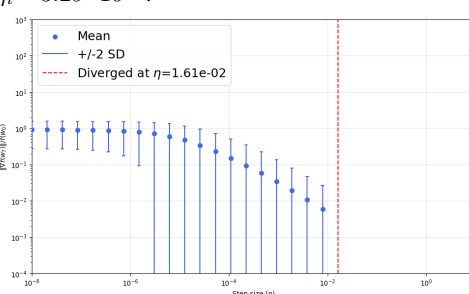

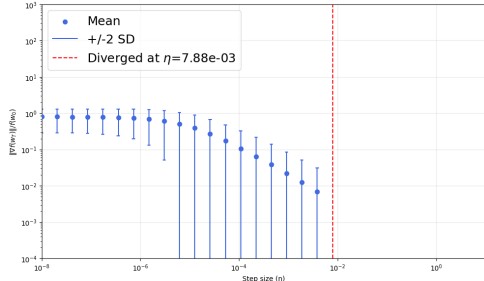

(c) $\pi_j = \mathcal{N}(\vec{0}, 7.5\boldsymbol{I}_{20})$. The first divergence is at $\eta_i \approx 1.61 \cdot 10^{-2}$.

(d) $\pi_j = \mathcal{N}(\vec{0}, 10\boldsymbol{I}_{20})$. The first divergence is at $\eta_i \approx 7.88 \cdot 10^{-3}$.

Figure 3: GD simulation results for $p = 4$. For $\pi_j = \mathcal{N}(\vec{0}, 2.5\boldsymbol{I}_{20}), \mathcal{N}(\vec{0}, 5\boldsymbol{I}_{20})$, the first divergence is at $\eta_i \approx 3.29 \cdot 10^{-2}$. For $\pi_j = \mathcal{N}(\vec{0}, 7.5\boldsymbol{I}_{20})$, the first divergence is at $\eta_i \approx 1.61 \cdot 10^{-2}$. For $\pi_j = \mathcal{N}(\vec{0}, 10\boldsymbol{I}_{20})$, the first divergence is at $\eta_i \approx 7.88 \cdot 10^{-3}$.

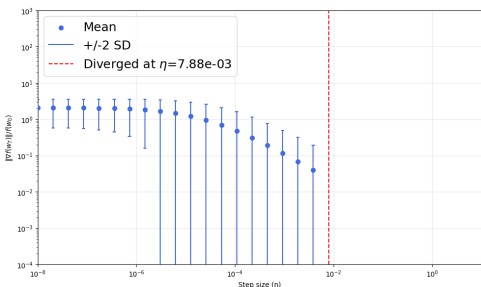

(a) $\pi_j = \mathcal{N}(\vec{0}, 2.5\boldsymbol{I}_{20})$. The first divergence is at $\eta_i \approx 7.88 \cdot 10^{-3}$.

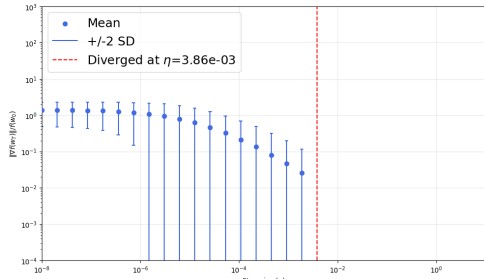

(b) $\pi_j = \mathcal{N}(\vec{0}, 5.0\boldsymbol{I}_{20})$. The first divergence is at $\eta_i \approx 3.86 \cdot 10^{-3}$.

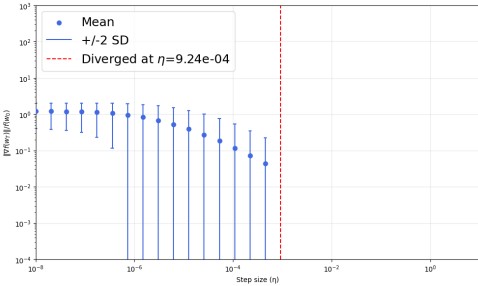

(c) $\pi_j = \mathcal{N}(\vec{0}, 7.5\boldsymbol{I}_{20})$. The first divergence is at $\eta_i \approx 9.24 \cdot 10^{-4}$.

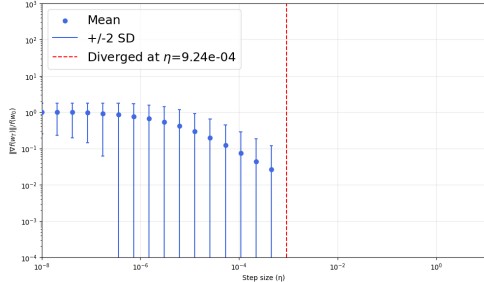

(d) $\pi_j = \mathcal{N}(\vec{0}, 10\boldsymbol{I}_{20})$. The first divergence is at $\eta_i \approx 9.24 \cdot 10^{-4}$.

Figure 4: GD simulation results for $p = 5$. For $\pi_j = \mathcal{N}(\vec{0}, 2.5\boldsymbol{I}_{20})$, the first divergence is at $\eta_i \approx 7.88 \cdot 10^{-3}$. For $\pi_j = \mathcal{N}(\vec{0}, 5\boldsymbol{I}_{20})$, the first divergence is at $\eta_i \approx 3.86 \cdot 10^{-3}$. For $\pi_j = \mathcal{N}(\vec{0}, 7.5\boldsymbol{I}_{20}), \mathcal{N}(\vec{0}, 10\boldsymbol{I}_{20})$, the first divergence is at $\eta_i \approx 9.24 \cdot 10^{-4}$.

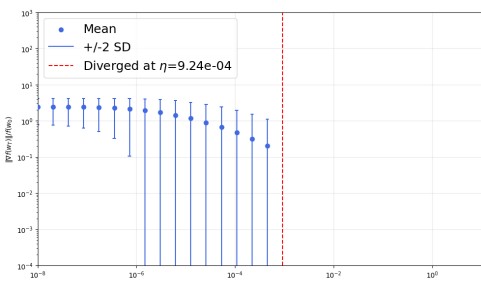

(a) $\pi_j = \mathcal{N}(\vec{0}, 2.5\boldsymbol{I}_{20})$. The first divergence is at $\eta_i \approx 9.24 \cdot 10^{-4}$.

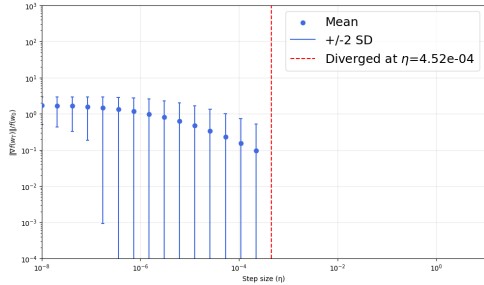

(b) $\pi_j = \mathcal{N}(\vec{0}, 5.0\boldsymbol{I}_{20})$. The first divergence is at $\eta_i \approx 4.52 \cdot 10^{-4}$.

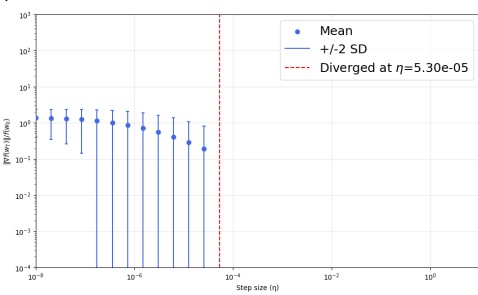

(c) $\pi_j = \mathcal{N}(\vec{0}, 7.5\boldsymbol{I}_{20})$. The first divergence is at $\eta_i \approx 5.30 \cdot 10^{-5}$.

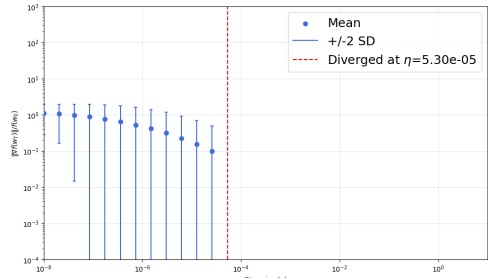

(d) $\pi_j = \mathcal{N}(\vec{0}, 10\boldsymbol{I}_{20})$. The first divergence is at $\eta_i \approx 5.30 \cdot 10^{-5}$.

Figure 5: GD simulation results for $p = 6$. For $\pi_j = \mathcal{N}(\vec{0}, 2.5\boldsymbol{I}_{20})$, the first divergence is at $\eta_i \approx 9.24 \cdot 10^{-4}$. For $\pi_j = \mathcal{N}(\vec{0}, 5\boldsymbol{I}_{20})$, the first divergence is at $\eta_i \approx 4.52 \cdot 10^{-4}$. For $\pi_j = \mathcal{N}(\vec{0}, 7.5\boldsymbol{I}_{20}), \mathcal{N}(\vec{0}, 10\boldsymbol{I}_{20})$, the first divergence is at $\eta_i \approx 5.30 \cdot 10^{-5}$.

## G.2 Synthetic Simulations with SGD

**Simulation Details:** We adopt the exact same settings as in Subsection G.1. The only difference is that we study SGD rather than GD, and hence we simulate stochastic gradients. We do so similarly to Gaash et al. (2025): we artificially add $\mathcal{N}\left(\vec{\mathbf{0}}, 0.01\boldsymbol{I}_{20}\right)$ to $\nabla F$ at each iteration of SGD.[13] The simulations for Subsection G.2 were again run on a Jupyter notebook in Python in Google Colab Pro, connected to a single NVIDIA T4 GPU. Our code is in the attached files.

**Results:** Our conclusions are similar to those from Subsection G.1. When $p = 2$, as predicted by established optimization theory for smooth functions, the first step size leading to divergence $\eta_i$ is identical across the $\pi_j$ (see Figure 6). In contrast for $p = 3, 4, 5, 6$, this $\eta_i$ generally decreases as the covariance $c_j\boldsymbol{I}_{20}$ of $\pi_j$ increases from 2.5 to 10 (see Figure 7, Figure 8, Figure 9, Figure 10). We note that while the general trends are similar to those from Subsection G.1, we can clearly see the presence of the stochastic gradients in these plots. In many of these plots, $\frac{\|\nabla F(\boldsymbol{w}_T)\|}{F(\boldsymbol{w}_0)}$ becomes roughly constant for $\eta$ large enough such that $T = 1000$ yields reasonable convergence; for such $\eta$, by $T = 1000$, the true gradients are small enough and the noise from the stochastic gradients takes over.

Once more, consider how the first step size leading to divergence $\eta_i$ decreases as the covariance $c_j\boldsymbol{I}_{20}$ of $\pi_j$ increases from 2.5 to 10. We find that the rate of this decrease generally increases as $p$ increases. We also again see that fixing $\pi_j$ and comparing across $p$, the first step size leading to divergence $\eta_i$ decreases as $p$ increases. As discussed in Subsection G.1, both of these phenomena are consistent with our theoretical results. For each $p \in \{2, 3, 4, 5, 6\}$ and $\pi_j$, we again record the smallest $\eta_i$ for which divergence occurred in Table 2 on page 95.

|  | $\pi_j = \mathcal{N}(\vec{\mathbf{0}}, 2.5\boldsymbol{I}_{20})$ | $\pi_j = \mathcal{N}(\vec{\mathbf{0}}, 5.0\boldsymbol{I}_{20})$ | $\pi_j = \mathcal{N}(\vec{\mathbf{0}}, 7.5\boldsymbol{I}_{20})$ | $\pi_j = \mathcal{N}(\vec{\mathbf{0}}, 10\boldsymbol{I}_{20})$ |
|---|---|---|---|---|
| $p = 2$ | $1.17 \cdot 10^0$ | $1.17 \cdot 10^0$ | $1.17 \cdot 10^0$ | $1.17 \cdot 10^0$ |
| $p = 3$ | $2.81 \cdot 10^{-1}$ | $1.37 \cdot 10^{-1}$ | $6.72 \cdot 10^{-2}$ | $1.37 \cdot 10^{-1}$ |
| $p = 4$ | $3.29 \cdot 10^{-2}$ | $3.29 \cdot 10^{-2}$ | $1.61 \cdot 10^{-2}$ | $7.88 \cdot 10^{-3}$ |
| $p = 5$ | $7.88 \cdot 10^{-3}$ | $1.89 \cdot 10^{-3}$ | $9.24 \cdot 10^{-4}$ | $4.52 \cdot 10^{-4}$ |
| $p = 6$ | $4.52 \cdot 10^{-4}$ | $4.52 \cdot 10^{-4}$ | $1.08 \cdot 10^{-4}$ | $5.30 \cdot 10^{-5}$ |

Table 2: Smallest $\eta_i$ leading to divergence for a given $p$ and initialization $\pi_j$.

---

[13]Note our result for convergence of SGD to FOSPs, Theorem 3.3, applies for Gaussian noise as per Remark 7.

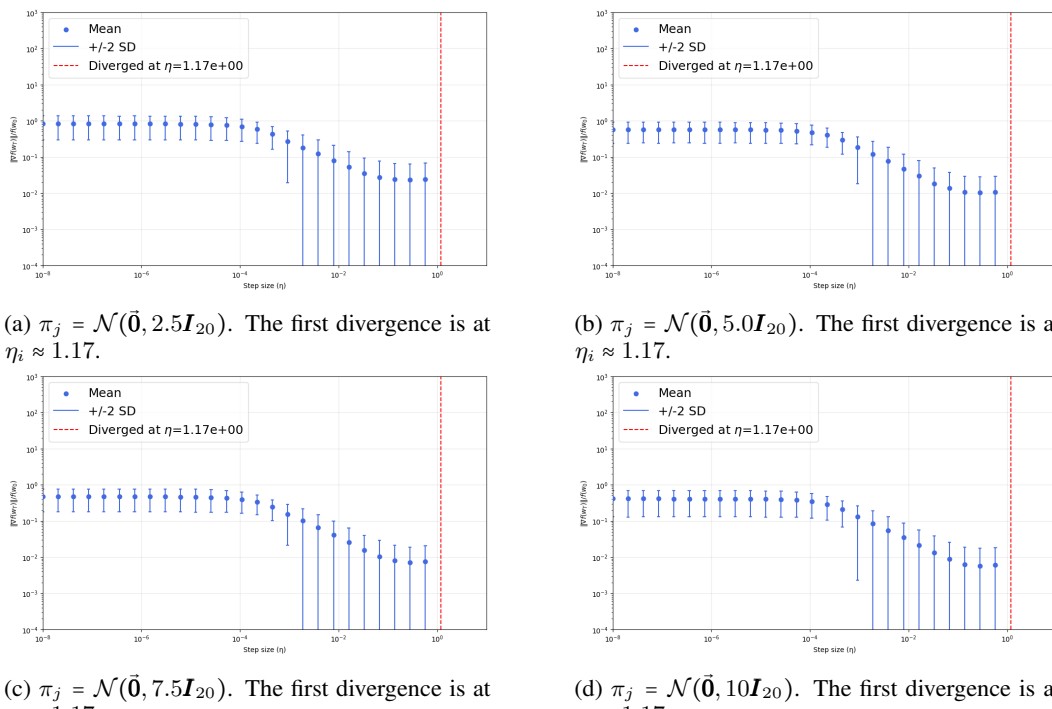

(a) $\pi_j = \mathcal{N}(\vec{\mathbf{0}}, 2.5\boldsymbol{I}_{20})$. The first divergence is at $\eta_i \approx 1.17$.

(b) $\pi_j = \mathcal{N}(\vec{\mathbf{0}}, 5.0\boldsymbol{I}_{20})$. The first divergence is at $\eta_i \approx 1.17$.

(c) $\pi_j = \mathcal{N}(\vec{\mathbf{0}}, 7.5\boldsymbol{I}_{20})$. The first divergence is at $\eta_i \approx 1.17$.

(d) $\pi_j = \mathcal{N}(\vec{\mathbf{0}}, 10\boldsymbol{I}_{20})$. The first divergence is at $\eta_i \approx 1.17$.

Figure 6: SGD simulation results for $p = 2$. For all $\pi_j$, the smallest $\eta_i$ leading to divergence is $\approx 1.17$.

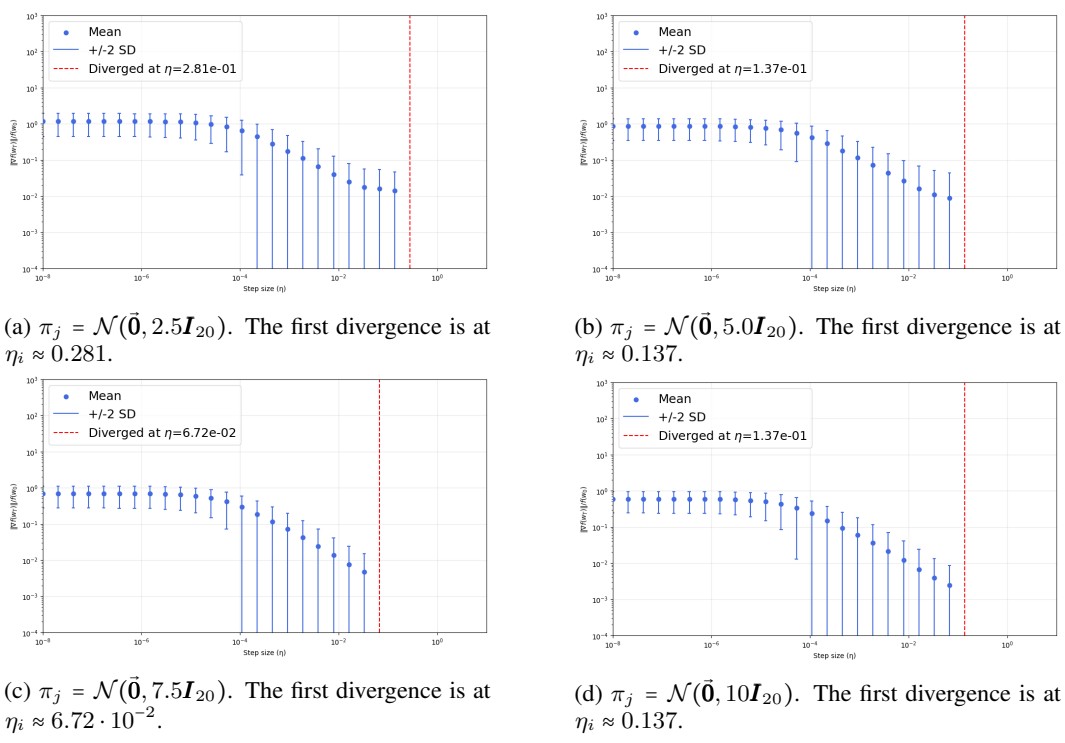

(a) $\pi_j = \mathcal{N}(\vec{\mathbf{0}}, 2.5\boldsymbol{I}_{20})$. The first divergence is at $\eta_i \approx 0.281$.

(b) $\pi_j = \mathcal{N}(\vec{\mathbf{0}}, 5.0\boldsymbol{I}_{20})$. The first divergence is at $\eta_i \approx 0.137$.

(c) $\pi_j = \mathcal{N}(\vec{\mathbf{0}}, 7.5\boldsymbol{I}_{20})$. The first divergence is at $\eta_i \approx 6.72 \cdot 10^{-2}$.

(d) $\pi_j = \mathcal{N}(\vec{\mathbf{0}}, 10\boldsymbol{I}_{20})$. The first divergence is at $\eta_i \approx 0.137$.

Figure 7: SGD simulation results for $p = 3$. For $\pi_j = \mathcal{N}(\vec{\mathbf{0}}, 2.5\boldsymbol{I}_{20})$, the first divergence is at $\eta_i \approx 0.281$. For $\pi_j = \mathcal{N}(\vec{\mathbf{0}}, 7.5\boldsymbol{I}_{20})$, the first divergence is at $\eta_i \approx 6.72 \cdot 10^{-2}$. For the other $\pi_j$, the first divergence is at $\eta_i \approx 0.137$.

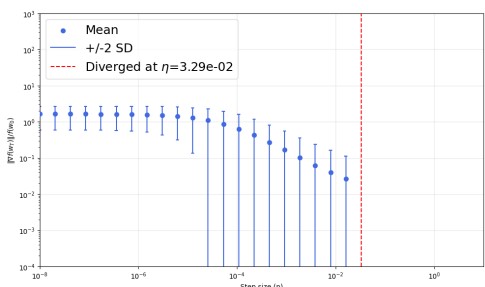

(a) $\pi_j = \mathcal{N}(\vec{\mathbf{0}}, 2.5\boldsymbol{I}_{20})$. The first divergence is at $\eta_i \approx 3.29 \cdot 10^{-2}$.

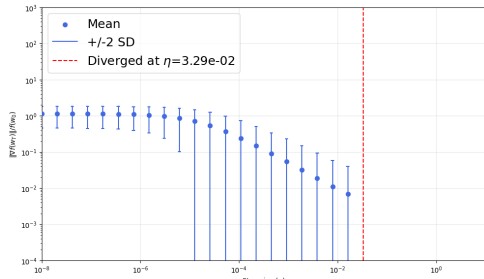

(b) $\pi_j = \mathcal{N}(\vec{\mathbf{0}}, 5.0\boldsymbol{I}_{20})$. The first divergence is at $\eta_i \approx 3.29 \cdot 10^{-2}$.

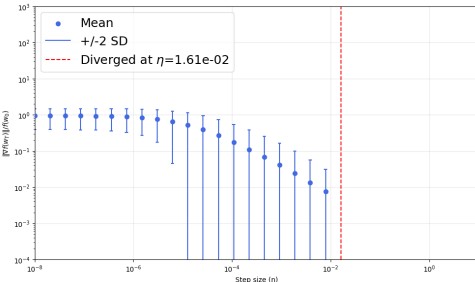

(c) $\pi_j = \mathcal{N}(\vec{\mathbf{0}}, 7.5\boldsymbol{I}_{20})$. The first divergence is at $\eta_i \approx 1.61 \cdot 10^{-2}$.

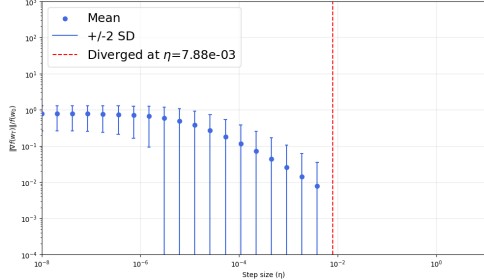

(d) $\pi_j = \mathcal{N}(\vec{\mathbf{0}}, 10\boldsymbol{I}_{20})$. The first divergence is at $\eta_i \approx 7.88 \cdot 10^{-3}$.

Figure 8: SGD simulation results for $p = 4$. For $\pi_j = \mathcal{N}(\vec{\mathbf{0}}, 2.5\boldsymbol{I}_{20}), \mathcal{N}(\vec{\mathbf{0}}, 5.0\boldsymbol{I}_{20})$, the first divergence is at $\eta_i \approx 3.29 \cdot 10^{-2}$. For $\pi_j = \mathcal{N}(\vec{\mathbf{0}}, 7.5\boldsymbol{I}_{20})$, the first divergence is at $\eta_i \approx 1.61 \cdot 10^{-2}$. For $\pi_j \sim \mathcal{N}(\vec{\mathbf{0}}, 10\boldsymbol{I}_{20})$, the first divergence is at $\eta_i \approx 7.88 \cdot 10^{-3}$.

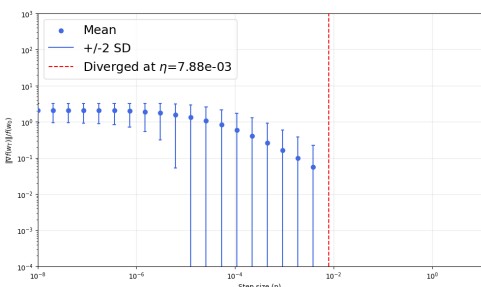

(a) $\pi_j = \mathcal{N}(\vec{\mathbf{0}}, 2.5\boldsymbol{I}_{20})$. The first divergence is at $\eta_i \approx 7.88 \cdot 10^{-3}$.

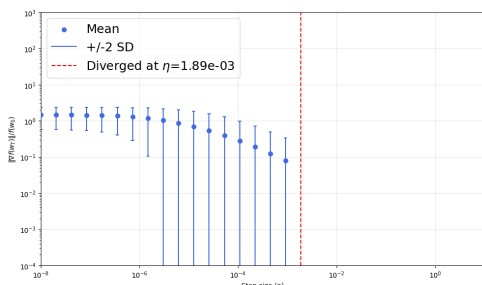

(b) $\pi_j = \mathcal{N}(\vec{\mathbf{0}}, 5.0\boldsymbol{I}_{20})$. The first divergence is at $\eta_i \approx 1.89 \cdot 10^{-3}$.

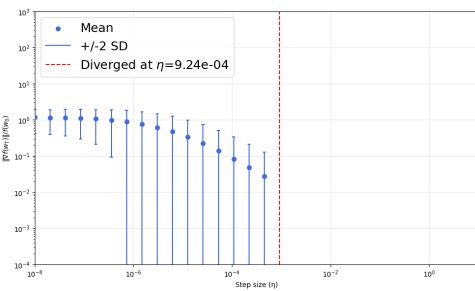

(c) $\pi_j = \mathcal{N}(\vec{\mathbf{0}}, 7.5\boldsymbol{I}_{20})$. The first divergence is at $\eta_i \approx 9.24 \cdot 10^{-4}$.

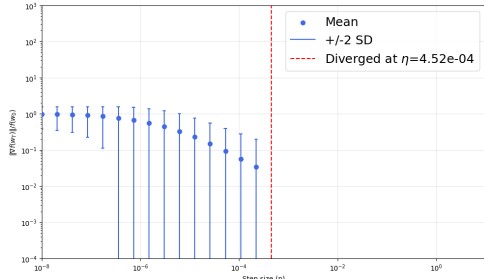

(d) $\pi_j = \mathcal{N}(\vec{\mathbf{0}}, 10\boldsymbol{I}_{20})$. The first divergence is at $\eta_i \approx 4.52 \cdot 10^{-4}$.

Figure 9: SGD simulation results for $p = 5$. For $\pi_j = \mathcal{N}(\vec{\mathbf{0}}, 2.5\boldsymbol{I}_{20}), \mathcal{N}(\vec{\mathbf{0}}, 5.0\boldsymbol{I}_{20}), \mathcal{N}(\vec{\mathbf{0}}, 7.5\boldsymbol{I}_{20}), \mathcal{N}(\vec{\mathbf{0}}, 10\boldsymbol{I}_{20})$, the first divergence is at $\eta_i \approx 7.88 \cdot 10^{-3}, 1.89 \cdot 10^{-3}, 9.24 \cdot 10^{-4}, 4.52 \cdot 10^{-4}$ respectively.

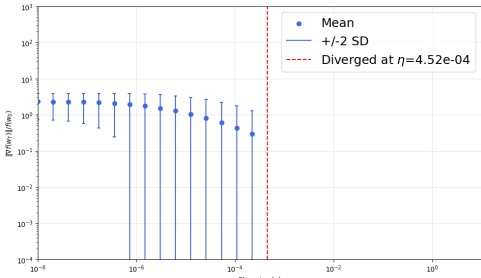

(a) $\pi_j = \mathcal{N}(\vec{\mathbf{0}}, 2.5\mathbf{I}_{20})$. The first divergence is at $\eta_i \approx 4.52 \cdot 10^{-4}$.

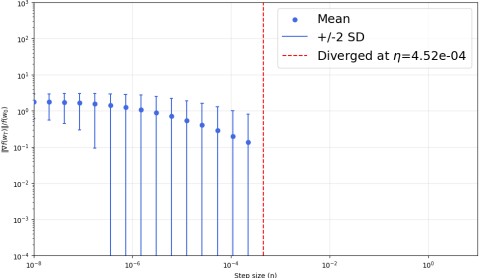

(b) $\pi_j = \mathcal{N}(\vec{\mathbf{0}}, 5.0\mathbf{I}_{20})$. The first divergence is at $\eta_i \approx 4.52 \cdot 10^{-4}$.

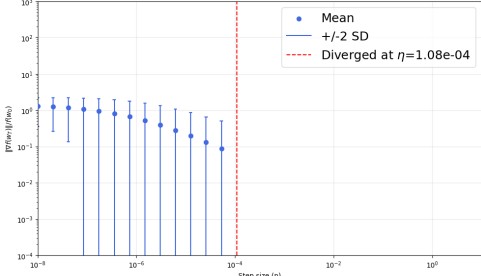

(c) $\pi_j = \mathcal{N}(\vec{\mathbf{0}}, 7.5\mathbf{I}_{20})$. The first divergence is at $\eta_i \approx 1.08 \cdot 10^{-4}$.

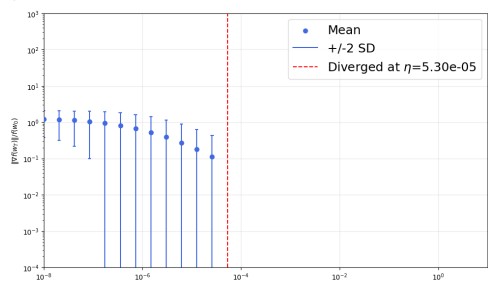

(d) $\pi_j = \mathcal{N}(\vec{\mathbf{0}}, 10\mathbf{I}_{20})$. The first divergence is at $\eta_i \approx 5.30 \cdot 10^{-5}$.

Figure 10: SGD simulation results for $p = 6$. For $\pi_j = \mathcal{N}(\vec{\mathbf{0}}, 2.5\mathbf{I}_{20}), \mathcal{N}(\vec{\mathbf{0}}, 5.0\mathbf{I}_{20}), \mathcal{N}(\vec{\mathbf{0}}, 7.5\mathbf{I}_{20}), \mathcal{N}(\vec{\mathbf{0}}, 10\mathbf{I}_{20})$, the first divergence are at $\eta_i \approx 4.52 \cdot 10^{-4}, 4.52 \cdot 10^{-4}, 1.08 \cdot 10^{-4}, 5.30 \cdot 10^{-5}$ respectively.

