# OpenReview forum: "Efficiently Escaping Saddle Points under Generalized Smoothness via Self-Bounding Regularity"
_NeurIPS.cc/2025/Conference — NeurIPS 2025 poster_

### Official Review · Reviewer_SgYJ · 2025-07-03

**Clarity:** 3
**Significance:** 3
**Originality:** 3
**Rating:** 5
**Confidence:** 4

**Summary:**

This paper proposes a general framework for establishing high-probability convergence of an algorithm to a target solution set $\mathcal{S}$ and for analyzing its oracle complexity.
Using this framework, the authors prove convergence of several algorithms to first-order or second-order stationary points, assuming the objective function $F$ satisfies self-bounding regularity properties, a generalization of $(L\_0, L\_1)$-smoothness.

**Questions:**

- How meaningful is the third-order self-bounding regularity? While I understand that the second-order version is related to $(L\_0, L\_1)$-smoothness (which is already well understood and whose relevance to applications is widely accepted in the literature) it is unclear whether the same holds for the third-order condition. Although the authors demonstrate that it holds for phase retrieval and matrix PCA, it would be helpful to provide a more general discussion regarding its practical significance.

- How general is Assumption 3.1? It is clearly more general than the assumption of uniformly bounded variance, but it may be beneficial if the authors could further clarify its scope by drawing more explicit connections to prior work such as [1], devoted to relaxing assumptions on the noise.

- Are there no experiments on phase retrieval or matrix PCA? If so, the last paragraph of Subsection 3.6 (lines 370–374) may be better separated from the rest of the subsection, as it gave the misleading impression that it was pointing to experimental results on those problems.

[1] Gower et al., SGD: General Analysis and Improved Rates. ICML, 2019.

**Ethical Concerns:**

["NO or VERY MINOR ethics concerns only"]

**Final Justification:**

The authors have committed to clean up the notations to improve the presentation of the paper, which was my main concern. While I previously stated that I will maintain my previous rating as I have not seen authors' concrete plans on how they will address the presentation issues, based on their last comment and the final remarks, I decided to trust the authors that they will take care of it.

**Limitations:**

The paper does not explicitly discuss its limitations, but this seems acceptable given the theoretical nature of the work.

**Quality:**

2

**Strengths And Weaknesses:**

**Strengths**

The proposed perspective of unifying convergence proofs offers a systematic and fundamental view of how such arguments are structured, making it a valuable contribution to the literature.
The central question of finding first- and second-order stationary points without assuming Lipschitz continuity of the gradient or Hessian is well-motivated and clearly explained.
Given the fact that self-bounding regularity conditions generalize the recently popular $(L_0, L_1)$-smoothness, the main results showing convergence to SOSPs under second- and third-order self-bounding conditions seem compelling.


**Weaknesses**

The unification comes at a cost. Section 2.2 is difficult to follow and contains formalisms that appear unnecessarily complex.
The precise roles and meaning of $\mathcal{A}\_2$ and $\mathcal{R}$ are not clearly explained and lead to confusion (lines 156–158).
For instance, why not simply define $\mathcal{A}$ as a procedure that outputs $\mathcal{A}\_1$ (the output with high probability decrease of $F$) and $\mathcal{R}(\mathcal{A}\_2)$ (the candidate solution), given that $\mathcal{R}$ and $\mathcal{A}\_2$ always appear together?
This would eliminate the need to repeatedly use cumbersome notation involving the set of sequences $\cup\_{n=0}^\infty (\mathbb{R}^d)^n$ in defining $\mathcal{A}$.
Overall, the writing and notation in this section can, and should, be significantly improved for clarity and simplicity.

There are also numerous issues with notation, typos, and minor errors in the proofs. A few examples include:
- The notation $\mathcal{A}(u\_0) = (u\_0 - \eta \nabla F(u\_0)) \times u\_0$ is not appropriate for denoting an element of a product set and should instead be written as a tuple; the current form is confusing.
- The definition of a decrease procedure does not clearly distinguish between randomness due to the algorithm itself and noise from sampling (e.g., for SGD and restarted SGD).
- In line 229, is it necessary to define $\mathcal{R}$ for sequences of any length?
- In the proof of Theorem 2.1, using $\mathcal{A}$ to denote an event overlaps confusingly with the notation for the algorithm.
- Also in that proof, the authors write $\mathcal{E}_n \cap \mathcal{B} \cap \mathcal{C}$ even though $\mathcal{C} \subseteq \mathcal{B} \subseteq \mathcal{E}_n$, which is needlessly confusing.
- In lines 1181 and 1182, $p\_{n-1}$ should likely be $p\_{n+1}$.
- In line 1205, the first inequality appears to have an incorrect sign.
- In line 1206, the authors presumably meant $k'=1$ is not possible so $k' \ge 2$, as with $k'=0$ the summation up to $k'-1$ is vacuous?
- In line 1209, $p_i$ is in place of $w_i$.
- In line 1211, the left side of the implication should be $N \le \sum_{i=0}^{k'} t_{oracle}(w_i)$?

Even though the paper is lengthy and involves many intricate parts, I believe it is the authors' responsibility to ensure that the paper is presented in a clear and polished manner to merit publication.

---

> ### Author Rebuttal · Authors · 2025-07-31
>
> Dear reviewer, we thank you for your review and comments. We sincerely appreciate your supportive review, careful reading and feedback, and belief that our main results seem compelling to a well-motivated problem and that our work “offers a systematic and fundamental view of how such arguments are structured, making it a valuable contribution to the literature”. We will be sure to improve the notation and presentation by taking your suggestions fully into account.
>
> We would like to respond to the questions and weaknesses you raised as follows. All the sources here are the same as in our manuscript or your review.
>
> **Third-order self-bounding regularity:** This is a good question, and we thank the reviewer for raising it. The third-order self-bounding regularity condition does carry further significance. Namely, Xie et al 2024 very interestingly demonstrates that under mild assumptions, the objective of Distributionally Robust Optimization (DRO) satisfies their Assumption 3 (see Theorem 3 in their paper), which is subsumed by our third-order self-bounding regularity condition as per our Example 2. Thus our results apply to DRO. DRO is a general optimization problem that has significant applications in fairness in machine learning and in learning under distribution shifts; see Xie et al 2024 for more discussion. We will be sure to add this discussion to our manuscript. We also demonstrate in Example 2 that this condition covers several natural growth rates of interest.
>
> **Notation for the framework:** We thank you very much for the detailed comments on the presentation. We will be sure to clean it up as best we can. Regarding $\mathcal{R}(\mathcal{A}_2(\cdot))$, this was a presentation choice we made. We believe defining $\mathcal{R}$ separately is more natural when presenting high probability results for the algorithm returning exactly one candidate solution in $\mathcal{S}$ (which we did not present in our submission for simplicity). If the reviewer feels strongly that this notation is cumbersome, we are happy to change this in our manuscript.
>
> That said, we believe some of the framework’s notation is both helpful and necessary. In particular, we define $\mathcal{R}$ for sequences of any length because for Restarted SGD, $\mathcal{R}$ takes in the average of $K_0$ points, which can be of arbitrary length. Likewise for SGD, $\mathcal{R}$ outputs a sequence of $K_0$ consecutive iterates, which also can be of arbitrary length.
>
> **Generality of assumption 3.1:** We believe Assumption 3.1 is quite general; it allows for the variance of gradient noise to increase with function value, considered a more realistic setting for gradient noise studied in Wojtowytsch 2023, 2024 and De Sa et al 2022. We can also extend Theorem 3.3 to sub-Gaussian noise with sub-Gaussian parameter depending on function value as we discuss in Remark 4, which was also considered in De Sa et al 2022 as a more general assumption for gradient noise in SGD. As per your suggestion, we will be sure to add more comparisons of Assumption 3.1 to other works studying relaxed gradient noise assumptions, such as Gower et al 2019.
>
>
> **Re experiments/simulations:** Indeed, they are not on phase retrieval or matrix PCA. Instead our experiments/simulations empirically verify that when optimizing natural generalized smooth functions with GD and SGD, the larger the loss $F(w_0)$ is at initialization, the smaller the largest working step size is (in contrast to optimizing smooth functions). These results show that the dependence on initialization via $\rho_1(F(w_0))$ in the range of working step sizes from our theorems matches behavior in practice. Thus our experiments/simulations validate the practical implications of our theoretical results.
> We thank you for the feedback and we will separate the last paragraph of Subsection 3.6 in our manuscript as per your suggestion.
>
> Please let us know if you have further questions; we are very open to discussion. Thank you again for your time and feedback!

---

> > ### Comment · Reviewer_SgYJ · 2025-08-07
> >
> > I thank the authors for their detailed response.
> >
> > I continue to believe that the paper offers contributions and perspectives that would be valuable to the literature, and I understand the authors’ intent to maintain the generality of the notations introduced in Section 2.2.
> >
> > That said, I still feel there is a trade-off between generality and clarity, and the current level of abstraction may be unnecessarily heavy for the algorithmic settings considered in Section 3. One potential improvement to enhance the readability of Section 2.2 would be to present a simplified version of the framework—e.g., by merging $\mathcal{R}$ and $\mathcal{A}_2$ into a single object, which appears sufficient for capturing the current results (though I may be mistaken)—and reserve the more general formulation for when it becomes essential, with a clear explanation of why the added complexity is needed.
> >
> > Ultimately, this is up to the authors’ discretion, but as it stands, I do not find the current presentation particularly reader-friendly, and the writing does not fully convey the rationale behind the authors’ choice of notation. I believe there is room for improvement in this regard.
> >
> > Aside from this point, I have no further concerns and remain overall positive about the paper. However, I will keep my original rating, as I did not see in the response a concrete plan or vision for addressing this writing issue.

---

> > > ### Author Response · Authors · 2025-08-09
> > > **Thank you**
> > >
> > > Dear reviewer, thank you for the reply and engagement. We appreciate both your valuable feedback and overall positive impression of our manuscript. We will make an earnest effort to make the presentation more reader-friendly. For example, we will merge $\mathcal{R}$ and $\mathcal{A}_2$ into a single object throughout at the reviewer’s suggestion (and explain elsewhere, e.g. in the appendix, about further results that can be obtained by separating them). We will also start Subsection 2.2 by presenting a simplified version of the framework that covers GD and Adaptive GD where $\mathcal{A}_2$ and $\mathcal{R}$ are not needed, to provide more intuition to readers. We hope these steps can make the presentation more friendly.

---

### Official Review · Reviewer_er1q · 2025-07-03

**Clarity:** 3
**Significance:** 3
**Originality:** 3
**Rating:** 5
**Confidence:** 3

**Summary:**

An open problem in ML is the development of optimisers that can provably avoid strong saddle points. The authors define a condition SOSP of the estimated optimal point, saying that the norm of the gradient at the should be less than epsilon, and the minimal eigenvalue of the Hessian should be more than -sqrt(epsilon). The paper proposes some new variants of gradient descent, adaptive gradient descent, and stochastic gradient descent, and shows that these variants, under suitable assumptions on the target function, are able to find solutions of the SOSP problem, and give complexity bounds for these solutions.

**Questions:**

Would it be possible to develop a truly adaptive version of the algorithm where the algorithm's steps do not depend on the parameters in Assumptions 1.1-1.2, but these only affect the convergence rates?

**Ethical Concerns:**

["NO or VERY MINOR ethics concerns only"]

**Final Justification:**

I have increased my score based on carefully considering the rebuttal, that made me better appreciate the contributions of the paper.
The authors have done a significant amount of work on this paper, and obtained many interesting mathematical proofs that prove sharp convergence rates in the order of the precision parameter $\epsilon$  for most algorithms. These results are sufficiently interesting to merit publication in NeurIPS.

**Limitations:**

Limitations have been adequately addressed.

**Paper Formatting Concerns:**

No concerns with paper formatting.

**Quality:**

3

**Strengths And Weaknesses:**

Strengths: The paper has some novel theoretical results for finding FOSP and SOSP about several deterministic and stochastic optimisation algorithms. Many of these bounds are sharp.

Weaknesses: Assumption 1.1 of the paper on the Hessian being upper bounded by a function of function value (||Hess f(x)||<= rho_1(f(x))$ are not unreasonable, but it is unfortunate that the algorithms explicitly depends on this function rho_1 (see equation (4)).
Since in practical ML problems, Assumption 1.1 may be difficult to verify.
There are no convincing numerical experiments showing the usefulness of the algorithms developed in this paper.

---

> ### Author Rebuttal · Authors · 2025-07-31
>
> Dear reviewer, we thank you for your review and comments. We sincerely appreciate your reading and feedback, your engagement with our manuscript, and the recognition of our “quite interesting theoretical results about the developed algorithms” in the work. We will be sure to take your feedback into account in the presentation of our manuscript.
>
> We would like to respond to the weakness and question you posed as follows. We believe this response will help clarify the results of our manuscript and we hope for a productive discussion. All the sources here are the same as in our manuscript.
>
> **We first would like to clarify Assumption 1.1.** Our Assumption 1.1 is not that $||\nabla^2 F(w)|| \le \rho_1(||\nabla F(w)||)$, but rather $||\nabla^2 F(w)|| \le \rho_1(F(w))$ (the difference being an upper bound in terms of function value rather than gradient), and in fact our Assumption 1.1 is **more general**. Namely, the assumption $||\nabla^2 F(w)|| \le \rho_1(||\nabla F(w)||)$ for sub-quadratic $\rho_1(\cdot)$ (from Li et al 2023a, generalizing Zhang et al 2020) is subsumed by our assumption, see Proposition A.1. In particular when $\rho_1(x)=L_1 x + L_0$, the $(L_0, L_1)$-smoothness assumption pioneered in Zhang et al 2020 and followed up in many subsequent works, this implies Assumption 1.1 with $\rho_1(x) = \frac32 L_0 + 4 L_1^2 x$ (Proposition A.1). Moreover, there are natural univariate functions where our Assumption 1.1 holds but the $||\nabla^2 F(w)|| \le \rho_1(||\nabla F(w)||)$ condition does not hold for sub-quadratic $\rho_1(\cdot)$, see Example 1.
>
> **We respectfully disagree with the assessment that our algorithms are not practical in their current form.** In our results, $\eta$ is a fixed step size which only depends on $\rho_1(F(w_0))$, a fixed value **depending only on initialization**. Moreover, the expressions depending on $\rho_1(F(w_0))$ are only an **upper bound** for working step sizes. We do **not** need to know these exact values. Therefore to implement our algorithms, all that is needed is an upper bound on fixed quantities such as $\rho_1(F(w_0))$. Thus a working step size for our algorithms can be found using cross validation or binary search. We will be sure to add this discussion to our manuscript and we thank the reviewer for bringing this point up.
>
> **Furthermore as mentioned above, $(L_0, L_1)$-smoothness implies our Assumption 1.1 with $\rho_1(x) = \frac32 L_0 + 4 L_1^2 x$.** Thus under $(L_0, L_1)$ smoothness, it is simple to back out a range of working step sizes in terms of $L_0, L_1, F(w_0)$ for all of our Theorems. This dependence on $L_0, L_1, F(w_0)$ is similar to the dependence in the step sizes provided in Zhang et al 2020, Jin et al 2021b, Crawshaw et al 2022, Li et al 2023a, Li et al 2023b, and many other works.
>
> **On adaptivity:** Indeed as the reviewer pointed out, it would be interesting to get a completely adaptive algorithm. That said, one can interpret cross validation or binary search over $\eta$ as adaptive algorithms in their own right. This leads to results for cross validation or binary search over $\eta$ that are analogous to our current results, and it is relatively straightforward to make these results formal. In the learning from data setting, one can make the cross validation result formal using classic techniques. Again, we will be sure to add this point to our manuscript and we thank the reviewer for bringing this point up.
>
> **Finally, we wish to clarify the goal of our manuscript. Our goal is not to propose new algorithms. Rather, we perform a theoretical analysis of existing, practical, and widely used first-order optimization algorithms.** We systematically establish stronger theoretical results for these widely used algorithms – in particular, novel results on SOSPs – in the more realistic and challenging generalized smooth setting. Our manuscript is not focused on demonstrating the validity of new algorithms.
>
> **We note that finding SOSPs is a significant and difficult problem established in the literature for over a decade**, which carries strong theoretical and practical implications in optimization for machine learning.
> We furthermore note that our work contains simulations. Our simulations show that the dependence on initialization via $\rho_1(F(w_0))$ in the range of working step sizes from our theorems matches behavior on which step sizes successfully optimize generalized smooth functions in practice.
>
> Please let us know if you have further questions; we are very open to discussion. Thank you again for your time and feedback!

---

> > ### Comment · Reviewer_er1q · 2025-08-03
> > **Updating score after rebuttal**
> >
> > Dear Authors,
> >
> > I have carefully considered your rebuttal, and realised that my original feedback was too harsh, given the significant contributions of the paper.
> >
> > Even though the paper does not introduces new algorithms, it analyses several important existing optimisation methods from the angle of finding FOSP and SOSP. Many of the novel bounds obtained here are sharp in the order of the precision parameter $\epsilon$, based on existing lower bounds. The paper has a significant amount of novel mathematical proofs that will be worthy of citation in the future.
> >
> > We agree that Assumption 1.1 is quite general, given that it was already extensively studied in the literature. We appreciate that the authors will further clarify the adaptivity aspect, and show that the bounds are valid under adaptive versions of these algorithms.
> >
> > As a result of these considerations, I have decided to raise my score to 5 (Accept).

---

> > > ### Author Response · Authors · 2025-08-05
> > > **Thank you**
> > >
> > > Dear reviewer, thank you very much for kindly reconsidering your evaluation and for raising your score. We sincerely appreciate your thoughtful feedback on our manuscript and your recognition of the strengths of our work. We will be sure to add discussion on adaptivity in the revised manuscript and again thank the reviewer for the helpful suggestion.

---

### Official Review · Reviewer_LCyX · 2025-07-10

**Clarity:** 3
**Significance:** 3
**Originality:** 3
**Rating:** 5
**Confidence:** 1

**Summary:**

This purely theoretical paper addresses the problem of finding first- and second-order stationary points (FOSP and SOSP) using first-order optimization methods under weak regularity assumptions: second and third order self-bounding regularity. To that end, a decrease procedure is introduced showing when first order methods decrease the value function, leading to convergence guarantees. It is shown that many common first order algorithms (GD, Adaptive GD, SGD, Perturbed GD and Restarted SGD) are covered by the framework and be analyzed accordingly.

**Questions:**

- How common are the self-bounding regularity conditions in modern ML, eg in deep learning?
- You notice that the choice of the step size \eta should depend on suboptimality at initialization, how sensitive is the method in practice ?

**Ethical Concerns:**

["NO or VERY MINOR ethics concerns only"]

**Limitations:**

yes

**Quality:**

3

**Strengths And Weaknesses:**

Strengths

- The problem seems important for the ML community, especially for non-convex optimziation
- Authors provide a thorough and unified convergence analysis for a wide variety of first-order algorithms
- The high level introduction of the decreasing procedure is clear and very helpful, improving the paper's readability.

Weaknesses

- Experimental results could have been introduced in the main paper, even briefly.
- Generality of the self-bounding regularity is not extensively discussed, it is a generalization of (L_0, l_1)-smoothness but are there many known functions satisfying self-bounding regularity and not (L_0, L_1)-smoothness? How frequently do such functions arise in deep learning?

---

> ### Author Rebuttal · Authors · 2025-07-31
>
> Dear reviewer, we thank you for your review and comments. We sincerely appreciate your supportive review, thoughtful questions, and belief that we are solving an important problem in machine learning and that we “provide a thorough and unified convergence analysis for a wide variety of first-order algorithms”. We will be sure to add further discussion on our experiments/simulations in our manuscript as you suggested.
>
> We would just like to respond to the questions and weaknesses you raised as follows. All the sources here are the same as in our manuscript.
>
> **Generality of self-bounding regularity:** This is a good question, and we thank the reviewer for raising it. As discussed in Section 1.1 (see Proposition A.1), second-order self-bounding regularity is strictly **more general than** the $(L_0, L_1)$-smooth conditions which arise frequently in machine learning.
>
> -The $(L_0, L_1)$ smoothness condition arises frequently in machine learning, and therefore by the following discussion, self-bounding regularity arises in the exact same situations as well. Namely $(L_0, L_1)$ smoothness was experimentally shown to hold for LSTMs in Zhang 2019 et al and transformers in Crawshaw et al 2022, and has been subject to extensive study in recent years as a more realistic set of regularity assumptions for optimization in machine learning.
>
> -We demonstrate in Proposition A.1 that $(L_0, L_1)$-smoothness **implies** self-bounding regularity with $\rho_1(x) = \frac32 L_0 + 4 L_1^2 x$. Li et al. 2023a generalized $(L_0, L_1)$-smoothness and once more, our self-bounding regularity conditions subsumes the condition of Li et al 2023a; see our Proposition A.1. We also demonstrate in Example 1 that there are natural univariate functions which do not satisfy $(L_0, L_1)$ smoothness and do not satisfy the condition of Li et al 2023a, but do satisfy self-bounding regularity.
>
> Thus we believe self-bounding regularity is quite general and arises frequently in modern machine learning. Moreover, there is growing evidence that conditions analogous to self-bounding regularity imposing quantitative control of the Hessian in terms of gradient or function value are necessary for optimization to succeed via first-order methods (even for finding stationary points). Namely, see Kornowski et al 2024 Theorem 2.1 and De Sa et al 2022 Theorem 3. That said, we emphasize that we work with self-bounding regularity not just because it is more general than $(L_0, L_1)$ smoothness, but because we believe it is a more natural way of analyzing ubiquitous first-order optimization algorithms under generalizations of smoothness.
>
> For third-order self-bounding regularity, we demonstrate several theoretical examples in optimization for machine learning, namely phase retrieval and matrix PCA, satisfy this condition and also satisfy second-order self-bounding regularity. We also demonstrate in Example 2 that this condition covers several natural growth rates of interest. Moreover, Xie et al 2024 very interestingly demonstrates that under mild assumptions, the objective of Distributionally Robust Optimization (DRO) satisfies their Assumption 3 (see Theorem 3 in their paper), which is subsumed by our third-order self-bounding regularity condition as per our Example 2. Thus our results apply to DRO. DRO is a general optimization problem that has significant applications in fairness in machine learning and in learning under distribution shifts; see Xie et al 2024 for more discussion. We will be sure to add this discussion to our manuscript.
>
> **Dependence on initialization:** It is true that the step size will depend on initialization. This holds true also for many works studying optimization with GD and SGD under generalizations of smoothness, such as Li et al 2023a. We also note that the step size provided is an upper bound on possible working step sizes. Thus, one can find a working step size with cross validation in the learning setting. We will add this point to our manuscript at the reviewer’s suggestion.
>
> In fact, this dependence on initialization carries important practical implications, as we describe in Section 3.6 and detail fully in our experiments/simulations in Section 9. There, we empirically verify that when optimizing natural generalized smooth functions with GD and SGD, the larger the loss $F(w_0)$ is at initialization, the smaller the largest working step size is (in contrast to optimizing smooth functions). These results show that the dependence on initialization via $\rho_1(F(w_0))$ in the range of working step sizes from our theorems matches behavior in practice. Thus our experiments/simulations validate the practical implications of our theoretical results on which step sizes successfully optimize generalized smooth functions.
>
> Please let us know if you have further questions; we are very open to discussion. Thank you again for your time and feedback!

---

### Comment · Reviewer_y3hi · 2025-08-06

First of all, I was invited to this discussion late, and I apologize for joining during the rebuttal phase. I don’t intend to burden the authors at this stage, but I would like to offer a few brief high-level comments that I hope are constructive.

**Motivation and related work**: Two important prior works—(Lee et al., COLT 2016) and (Lee et al., MP 2019)—are not cited. These papers show that first-order methods almost surely avoid strict saddle points and converge to SOSPs without assuming Lipschitz continuity of the Hessian.

In contrast, this paper builds on the perturbed gradient descent as introduced by Jin et al. (ICML 2017), which achieves dimension-free convergence rates but requires the Lipschitz-Hessian assumption. Later work (Du et al., NeurIPS 2017) showed that the plain gradient descent can take exponential time to escape saddle points, which motivates the use of perturbations.

However, since earlier work already established convergence to SOSP without the Lipschitz-Hessian assumption (though without efficiency guarantees), I believe the motivation for generalizing the Lipschitz-Hessian setting should be more carefully articulated. For example, lines 79-80 claim that the Lipschitz-Hessian assumption is ubiquitous in the literature, but in my view, it is typically imposed to enable efficient rates. I encourage the authors to clarify.

Lastly, regarding (Lee et al., COLT 2016) and (Lee et al., MP 2019), a brief discussion on whether plain gradient descent can almost surely converge to SOSPs under the generalized Lipschitz-gradient assumption would also be valuable.

**Assumptions and technical contributions**: Assumptions 1.1 and 1.2 appear to be natural generalizations of gradient and Hessian Lipschitzness, and the subsequent analyses seem novel. However, the explanation of decrease procedure is quite dense, even though the core idea appears to be a generalization of the descent lemma. The exposition could benefit from additional clarification and motivation—especially since many standard methods fall into this framework.

**Step size clarification**: Could the authors clarify under Assumption 1.1 which step size becomes vanishingly small? Also, how does this compare to the step size $1/L_1(w_0)$ in (4)?

**Overall impression**: This paper is technically dense, and due to the limited time, I was not able to digest all the details. That said, I believe a good theory paper should also offer conceptual insight. It seems to me that the core insight boils down to selecting an appropriate step size depending on the initial point $w_0$ and the generalized regularity conditions. I would encourage the authors to emphasize this perspective more clearly, rather than focusing primarily on cataloging the range of methods that fall within the framework.

References:

[1] Lee, Simchowitz, Jordan, and Recht, Gradient Descent Only Converges to Minimizers, COLT 2016.

[2] Lee, Panageas, Piliouras, Simchowitz, Jordan, and Recht, First-order Methods Almost Always Avoid Strict Saddle Points, Mathematical Programming, 2019.

[3] Du, Jin, Lee, Jordan, Póczos, and Singh, Gradient Descent Can Take Exponential Time to Escape Saddle Points, NeurIPS 2017.

---

> ### Author Response · Authors · 2025-08-07
> **Thank you for the comments, here's our response to the questions**
>
> Dear reviewer, thank you for your constructive comments. We sincerely appreciate your reading and feedback, and your engagement with our manuscript. We will be sure to revise our manuscript to improve our exposition of a decrease procedure and to further emphasize central conceptual ideas in our work, particularly in Section 2, taking your feedback fully into account.
>
> We would like to answer your questions as follows. All sources in the following are from our manuscript or your comments.
>
> **Motivation and related work:** This is a good question, and we thank the reviewer for raising it. The results of Lee et al. 2016, 2019 show **asymptotic** guarantees. **However, we are focused on nonasymptotic guarantees/efficient rates for the convergence of first order methods,** which is very important in theory and practice where iterations are limited. (Indeed as reviewer er1q points out in their comment, many of our results are sharp in order of precision $\epsilon$.) For such **nonasymptotic** guarantees of finding SOSPs, the Hessian Lipschitz assumption appears ubiquitously in the literature (except for Xie et al. 2024, whose assumption is generalized by our work as per Example 2). This motivates our generalization of the Hessian Lipschitz assumption.
> At your suggestion, we will be sure to cite and add discussion r.e. Lee et al. 2016, 2019 to our manuscript, including on asymptotic almost sure guarantees under our regularity assumptions. We will also be sure to further emphasize our focus on efficient rates/nonasymptotic guarantees. Again, we thank you for bringing up this point.
>
> We believe our generalization of the Hessian Lipschitz assumption, Assumption 1.2, is quite general: it is satisfied by phase retrieval and matrix PCA (Subsection 3.6), covers many growth rates of interest (Example 2), and furthermore under mild conditions, it is satisfied by Distributionally Robust Optimization (see our response to reviewer SgYJ, which we will add to our manuscript). Moreover, Kornowski et al. 2024 showed quantitative control of the Hessian is in fact necessary for nonasymptotic guarantees of finding first order stationary points. We also note our Assumption 1.1 generalizes the $(L_0, L_1)$ smoothness assumption which is a widely studied relaxation of gradient Lipschitzness, and our Assumption 1.2 generalizes the regularity assumptions of Xie et al. 2024; see our response to reviewer SgYJ and er1q for more details on this.
>
> **Step size clarification:** Thank you for raising the question. Under Assumption 1.1, the standard choice of step size $\eta \le 1/L$ in smooth optimization where $L$ is a global upper bound on $||\nabla^2 F||$ does not work, since no finite bound $L<\infty$ exists (hence one would have to set $\eta=0$). This is what we meant by $\eta$ becoming vanishingly small. However, $1/L(w_0)$ in (4) is not vanishingly small as $L(w_0) < \infty$ is the ‘effective smoothness constant’ in the $F(w_0)$-sublevel set, where the iterates are guaranteed to lie (with high probability). Also note that any step size $\eta \le \frac1{L(w_0)}$ would work, see our response to reviewer er1q.
>
> **Assumptions and technical contributions, and main points of our paper:** We wish to reiterate the core idea in our paper, which builds upon selecting an appropriate step size and generalizing the descent lemma to encompass the following. As discussed in Subsection 2.1 of our manuscript, the crucial idea to ‘chain together’ the descent lemma; to formalize this idea, we introduce the notion of a decrease procedure. In particular, the algorithm being a decrease procedure in the $F(w_0)$-sublevel set guarantees decrease (with high probability) when the step size is chosen appropriately in terms of initialization $w_0$, which guarantees the step size defined in terms of $w_0$ works at the next iterate, which guarantees decrease again, and so forth. Thus all the iterates lie in the $F(w_0)$-sublevel set (with high probability), and furthermore this decrease cannot occur too many times as $F$ is lower bounded, yielding bounds on iteration complexity. We believe this gives a systematic view of how these convergence proofs are structured under generalized smoothness, as reviewer SgYJ also points out. We will be sure to further emphasize this discussion and perspective in our manuscript, by emphasizing it further in Section 2 and adding it to the summary of our contributions in Section 1 and in the exposition in Sections 3 and 4. We thank the reviewer for the helpful suggestion.
>
>
> Please let us know if you have further questions; we are very open to discussion. Thank you again for your time and feedback!

---

> > ### Comment · Reviewer_y3hi · 2025-08-08
> >
> > I appreciate the authors’ detailed response. I would like to note that, since I have not submitted an official review, I do not have access to the discussion except for my own comments. All of my high-level questions have been well addressed, and I have no further questions.

---

### Note · Authors · 2025-08-15

Dear ACs and reviewers, we would like to thank you all for your time and effort, feedback, and service. We sincerely appreciate the constructive comments and engagement with our manuscript. We would just like to reiterate a few points in our final remark; more details are in our earlier replies.

**Regularity assumptions:** On this topic, we believe our regularity assumptions Assumption 1.1 and 1.2 are both quite general and grounded in the literature:

-Assumption 1.1 (second order self-bounding regularity) generalizes the $(L_0, L_1)$-smoothness assumption pioneered in Zhang et al. 2020 and followed up in many subsequent works (Proposition A.1).

-Assumption 1.2 (third order self-bounding regularity) is satisfied by phase retrieval and matrix PCA (Subsection 3.6), covers many growth rates of interest and generalizes the regularity assumptions of Xie et al. 2024 (Example 2), and under mild conditions is satisfied by Distributionally Robust Optimization (see response to reviewer SgYJ).

**Practicality of our algorithms and adaptivity:** We note that to implement our algorithms, one only needs a fixed step size $\eta$. In turn $\eta$ only depends on an **upper bound** on a fixed value dependent on only initialization $F(w_0)$ and the regularity assumptions. Thus our algorithms can be implemented in practice via cross-validation or binary search (which can also be thought of as adaptive algorithms in their own right). See our response to reviewer er1q for more details.

**Presentation and related work:** We will be sure to improve presentation from all the reviewers’ comments, taking all the feedback fully into account. This includes further discussion on the above two points (regularity assumptions, and practicality of our algorithms and adaptivity). We will also cite and add discussion r.e. Lee et al. 2016, 2019 to our manuscript as discussed in our response to the official comment by reviewer y3hi. We will simplify notation in the framework, and first present in Subsection 2.2 a simpler example of our framework with less generality to convey intuition. For more on our concrete plan to this end, see our last comment in the discussion with reviewer SgYJ. Moreover, we will implement all the feedback from the reviewers; this is not a complete list.

Thank you again for all the time, feedback, and service!

---

### Decision · Program_Chairs · 2025-09-17

**Decision:**

Accept (poster)

**Comment:**

The paper provides new (theoretical) results on escaping strict saddle points in nonconvex optimization. While this is an extensively studied topic in the OPT/ML community, the existing nonasymptotic results strongly rely on traditional smoothness assumptions (Lipschitzness imposed on both the gradient and the Hessian). The paper instead provides nonasymptotic results under generalized smoothness assumptions, which can capture problems like matrix PCA, phase retrieval, and certain examples of distributionally robust optimization problems. Despite initial concerns, all reviewers were in agreement that the paper presents strong technical contributions and should thus be accepted. The authors have committed to improving the presentation clarity and I hope will make good on this commitment.